A specimen-level phylogenetic analysis and taxonomic revision of Diplodocidae (Dinosauria, Sauropoda)

Tschopp Emanuel 1 2 3 tschopp.e@gmail.com
Mateus Octávio 1 2
Benson Roger B.J. 4
1 GeoBioTec, Faculdade de Ciência e Tecnologia, Universidade Nova de Lisboa , Monte de Caparica , Portugal
2 Museu da Lourinhã , Lourinhã , Portugal
3 Dipartimento di Scienze della Terra, Università di Torino , Italy
4 Department of Earth Sciences, University of Oxford , Oxford , UK
Farke Andrew
Electronic publication date: 2015 Apr 7
Publication date: 2015
Volume: 3
Electronic Location ID: e857
Received 2014 Jan 30; Accepted 2015 Mar 5
Copyright: © 2015 Tschopp et al.
Copyright year: 2015
Copyright holder: Tschopp et al.
License: This is an open access article distributed under the terms of the Creative Commons Attribution License, which permits unrestricted use, distribution, reproduction and adaptation in any medium and for any purpose provided that it is properly attributed. For attribution, the original author(s), title, publication source (PeerJ) and either DOI or URL of the article must be cited.
License URL: https://creativecommons.org/licenses/by/4.0/

Keywords: Sauropod dinosaurs, Diplodocidae, Specimen-based phylogeny, Numerical taxonomy, New genus

Funding: Fundação para a Ciência e a Tecnologia of the Ministério de Educação e Ciência, Portugal SFRH/BD/66209/2009 Synthesys DE-TAF-1150 Being part of Emanuel Tschopp’s PhD dissertation, this study would not have been possible without the financial support through the doctoral fellowship from the Fundação para a Ciência e a Tecnologia of the Ministério de Educação e Ciência, Portugal (SFRH/BD/66209/2009). Travel subsidies were kindly provided by Synthesys (DE-TAF-1150) for a collection visit at Museum für Naturkunde, Berlin. The funders had no role in study design, data collection and analysis, decision to publish, or preparation of the manuscript.

==============================
Diplodocidae are among the best known sauropod dinosaurs. Several species were described in the late 1800s or early 1900s from the Morrison Formation of North America. Since then, numerous additional specimens were recovered in the USA, Tanzania, Portugal, and Argentina, as well as possibly Spain, England, Georgia, Zimbabwe, and Asia. To date, the clade includes about 12 to 15 nominal species, some of them with questionable taxonomic status (e.g., ‘Diplodocus’ hayi or Dyslocosaurus polyonychius), and ranging in age from Late Jurassic to Early Cretaceous. However, intrageneric relationships of the iconic, multi-species genera Apatosaurus and Diplodocus are still poorly known. The way to resolve this issue is a specimen-based phylogenetic analysis, which has been previously implemented for Apatosaurus, but is here performed for the first time for the entire clade of Diplodocidae.

The analysis includes 81 operational taxonomic units, 49 of which belong to Diplodocidae. The set of OTUs includes all name-bearing type specimens previously proposed to belong to Diplodocidae, alongside a set of relatively complete referred specimens, which increase the amount of anatomically overlapping material. Non-diplodocid outgroups were selected to test the affinities of potential diplodocid specimens that have subsequently been suggested to belong outside the clade. The specimens were scored for 477 morphological characters, representing one of the most extensive phylogenetic analyses of sauropod dinosaurs. Character states were figured and tables given in the case of numerical characters.

The resulting cladogram recovers the classical arrangement of diplodocid relationships. Two numerical approaches were used to increase reproducibility in our taxonomic delimitation of species and genera. This resulted in the proposal that some species previously included in well-known genera like Apatosaurus and Diplodocus are generically distinct. Of particular note is that the famous genus Brontosaurus is considered valid by our quantitative approach. Furthermore, “Diplodocus” hayi represents a unique genus, which will herein be called Galeamopus gen. nov. On the other hand, these numerical approaches imply synonymization of “Dinheirosaurus” from the Late Jurassic of Portugal with the Morrison Formation genus Supersaurus. Our use of a specimen-, rather than species-based approach increases knowledge of intraspecific and intrageneric variation in diplodocids, and the study demonstrates how specimen-based phylogenetic analysis is a valuable tool in sauropod taxonomy, and potentially in paleontology and taxonomy as a whole.

Introduction

Overview of diplodocid sauropods

The dinosaur clade Diplodocidae includes some of the most iconic sauropods. With their greatly elongated necks and tails, diplodocids constitute one of the typical popular images of sauropods. The clade is historically important, having provided the first published reconstruction of an entire sauropod skeleton (‘Brontosaurus’ excelsus; Marsh, 1883), the first complete sauropod skull to be described (Diplodocus; Marsh, 1884), and the first mounted sauropod specimen (Apatosaurus AMNH 460; Matthew, 1905). Diplodocids range from relatively small to gigantic species (Kaatedocus siberi Tschopp & Mateus, 2012, 12–14 m, to Supersaurus vivianae Jensen, 1985, 35–40 m, respectively) with a wide range of body masses (Tornieria africana (Fraas, 1908)), 12 t, to Apatosaurus louisae Holland, 1915a, 41.3 t; Campione & Evans, 2012; Benson et al., 2014). The clade includes the well-known genera Apatosaurus Marsh, 1877a, Diplodocus Marsh, 1878, and Barosaurus Marsh, 1890. Their possible first occurrence dates to the Middle Jurassic of England (Cetiosauriscus stewarti Charig, 1980; but see Heathcote & Upchurch, 2003; Rauhut et al., 2005, for an alternative identification of Cetiosauriscus). Diplodocidae reached a peak in diversity in the Late Jurassic, with finds from North America, Tanzania, Zimbabwe, Portugal and Spain, as well as possibly England and Georgia (Mannion et al., 2012). To date, only one convincing report exists for their presence in the Cretaceous, which is furthermore the only occurrence of the clade in South America (Whitlock, D’Emic & Wilson, 2011; Gallina et al., 2014).

In recent phylogenetic trees, Diplodocidae consistently forms the sister group to the clade Dicraeosauridae, with which they form Flagellicaudata. Flagellicaudata in turn is included with Rebbachisauridae in Diplodocoidea (e.g., Upchurch, 1998; Wilson, 2002; Wilson, 2005; Harris & Dodson, 2004; Upchurch, Barrett & Dodson, 2004; Rauhut et al., 2005; Harris, 2006c; Sereno et al., 2007; Whitlock, 2011a; Carballido et al., 2012b; Mannion et al., 2012; Tschopp & Mateus, 2013b). The taxonomy of these clades was historically somewhat confused, with “Diplodocidae” being used in the same way as Diplodocoidea today (see e.g., McIntosh, 1990a; McIntosh, 1990b). In the following, we use the taxonomy and definitions as clarified by Taylor & Naish (2005).

Although new taxa continue to be discovered (Table 1), the vast majority of diplodocid species were described in the late 1800s and early 1900s. The high rate of early descriptions, particularly during the so-called ‘Bone Wars’ of the late 1800s, resulted also in a large number of species that are now considered invalid, questionable, or synonymous (Taylor, 2010). Species identification is furthermore hampered by the fact that many holotype specimens are incomplete and fragmentary (e.g., Diplodocus longus YPM 1920), or appear to include bones from more than one individual (e.g., Apatosaurus ajax YPM 1860). Due to the absence of field notes or quarry maps for many of these early discoveries, it is often difficult or impossible to confidently assign bones to particular individuals or taxa. Given that most sites in the Upper Jurassic Morrison Formation are multi-taxon assemblages, and that the Morrison Formation has yielded about three-quarters of the diplodocid genera reported so far, it is possible that at least some holotype specimens include material from multiple species. This renders meaningful diagnoses for the species, and thus the identification of new specimens, highly difficult. Nevertheless, detailed studies of original material and their corresponding field notes by McIntosh & Berman (1975), Berman & McIntosh (1978), McIntosh (1981), McIntosh (1990a), McIntosh (1995), McIntosh (2005) and McIntosh & Carpenter (1998) have provided a wealth of important information concerning the composition of diplodocid holotype specimens. This valuable research allows recognition of diagnostic autapomorphies and character combinations for many taxa. However, only one study so far has tested the referral of individual specimens to diplodocid species using phylogenetic methods, focusing on the genus Apatosaurus alone (Upchurch, Tomida & Barrett, 2004). By using individual specimens as operational taxonomic units (OTUs), Upchurch, Tomida & Barrett (2004) generally supported the traditional view of Apatosaurus intrarelationships, which included the species A. ajax, A. excelsus, A. louisae and A. parvus.

Table 1 Species historically described as belonging to Diplodocidae.

Species	Most recent taxonomic opinion	Reference	Occurrence	Comments	
Dystrophaeus viaemalae Cope, 1877b	Sauropoda incertae sedis	Upchurch, Barrett & Dodson, 2004	USA	type species of Dystrophaeus	
Amphicoelias altus Cope, 1877a	Diplodocoidea incertae sedis	Tschopp & Mateus, 2013b	USA	type species of Amphicoelias	
Amphicoelias latus Cope, 1877a	synonym of Camarasaurus supremus	Osborn & Mook, 1921	USA		
Apatosaurus ajax Marsh, 1877a	Apatosaurinae	Upchurch, Tomida & Barrett, 2004	USA	type species of Apatosaurus	
Apatosaurus grandis Marsh, 1877a	Misassigned, ⟹ Camarasaurus grandis	Marsh, 1878; Upchurch, Tomida & Barrett, 2004	USA		
Amphicoelias fragillimus Cope, 1878	synonym of A. altus	Osborn & Mook, 1921	USA		
Atlantosaurus immanis Marsh, 1878	synonym of A. ajax	McIntosh, 1995; Upchurch, Tomida & Barrett, 2004	USA		
Diplodocus longus Marsh, 1878	Diplodocinae	McIntosh & Carpenter, 1998	USA	type species of Diplodocus	
Brontosaurus excelsus Marsh, 1879	Brontosaurus = Apatosaurus; species referred to Apatosaurus (A. Excelsus)	Riggs, 1903; Upchurch, Tomida & Barrett, 2004	USA	type species of Brontosaurus	
Apatosaurus laticollis Marsh, 1879	synonym of A. ajax	McIntosh & Berman, 1975; Upchurch, Tomida & Barrett, 2004	USA		
Brontosaurus amplus Marsh, 1881	synonym of A. excelsus	McIntosh & Berman, 1975; Upchurch, Tomida & Barrett, 2004	USA		
Diplodocus lacustris Marsh, 1884	nomen dubium	McIntosh, 1990a	USA	originally described as Stegosaurus armatus teeth (Marsh, 1877b; McIntosh, 1990a)	
Barosaurus lentus Marsh, 1890	Diplodocinae	Tschopp & Mateus, 2013b	USA	type species of Barosaurus	
Barosaurus affinis Marsh, 1899	synonym of B. lentus	McIntosh, 1990a	USA		
Diplodocus carnegii Hatcher, 1901	unambiguous differential diagnosis from D. longus not yet demonstrated	Gilmore, 1932; McIntosh, 1990a	USA	sometimes misspelled D. carnegiei (e.g., Lull, 1919)	
Elosaurus parvus Peterson & Gilmore, 1902	Elosaurus = Apatosaurus; ⟹ A. parvus	Upchurch, Tomida & Barrett, 2004	USA	type species of Elosaurus	
Gigantosaurus africanus Fraas, 1908	Gigantosaurus preoccupied, ⟹ Tornieria africana; included into Barosaurus (Barosaurus africanus); generic distinction proved valid, ⟹ Tornieria africana	Sternfeld, 1911; Janensch, 1922; Remes, 2006	Tanzania	type species of Tornieria	
Apatosaurus louisae Holland, 1915a	Apatosaurinae	Upchurch, Tomida & Barrett, 2004	USA		
Apatosaurus minimus Mook, 1917	misassigned, Macronaria incertae sedis	McIntosh, 1990a; Mannion et al., 2012	USA		
Diplodocus hayi Holland, 1924	possibly new genus	Holland, 1924; McIntosh, 1990a	USA		
Apatosaurus alenquerensis Lapparent & Zbyszewski, 1957	Misassigned, ⟹ Camarasaurus alenquerensis; later new genus erected: Lourinhasaurus alenquerensis (Macronaria)	McIntosh, 1990b; Dantas et al., 1998; Mocho, Royo-Torres & Ortega, 2014	Portugal	type species of Lourinhasaurus	
Barosaurus gracilis Russell, Béland & McIntosh, 1980	nomen nudum	Remes, 2006	Tanzania	initially described as B. africanus var. gracilis (Janensch, 1961)	
Cetiosauriscus stewarti Charig, 1980	Non-neosauropod Eusauropoda; originally described as Cetiosaurus leedsi	Rauhut et al., 2005	United Kingdom	type species of Cetiosauriscus	
Supersaurus vivianae Jensen, 1985	Diplodocidae	Tschopp & Mateus, 2013b	USA	type species of Supersaurus	
Dystylosaurus edwini Jensen, 1985	synonym of S. vivianae	Curtice & Stadtman, 2001	USA	type species of Dystylosaurus	
Seismosaurus halli Gillette, 1991	Seismosaurus = Diplodocus, possibly D. longus, or D. hallorum	Lucas et al., 2006; Lovelace, Hartman & Wahl, 2007	USA	type species of Seismosaurus; should be called S. hallorum (Gillette, 1994, after a personal comment of G Olshevsky)	
Dyslocosaurus polyonychius McIntosh, Coombs & Russell, 1992	Diplodocoidea incertae sedis	Upchurch, Barrett & Dodson, 2004	USA	type species of Dyslocosaurus	
Apatosaurus yahnahpin Filla & Redman, 1994	new genus: Eobrontosaurus (Diplodocidae)	Bakker, 1998	USA	type species of Eobrontosaurus	
Dinheirosaurus lourinhanensis Bonaparte & Mateus, 1999	Diplodocidae	Tschopp & Mateus, 2013b	Portugal	type species of Dinheirosaurus	
Losillasaurus giganteus Casanovas, Santafé & Sanz, 2001	Turiasauria, sister taxon to Turiasaurus	Royo-Torres & Upchurch, 2012	Spain	type species of Losillasaurus	
Suuwassea emilieae Harris & Dodson, 2004	Dicraeosauridae	Tschopp & Mateus, 2013b	USA	type species of Suuwassea	
Australodocus bohetii Remes, 2007	Titanosauria incertae sedis	Mannion et al., 2013	Tanzania	type species of Australodocus	
Kaatedocus siberi Tschopp & Mateus, 2012	Diplodocinae	Tschopp & Mateus, 2013b	USA	type species of Kaatedocus; published online in 2012, print version is the 2013b paper	
Leinkupal laticauda Gallina et al., 2014	Diplodocinae	Gallina et al., 2014	Argentina	type species of Leinkupal	

The specimen-based phylogenetic analysis is herein extended to the entire clade of Diplodocidae and combined with the most recent analyses of diplodocoid interrelationships (Whitlock, 2011a; Mannion et al., 2012; Tschopp & Mateus, 2013b). Our analysis includes all holotype specimens of every putative diplodocid species yet described (see Table 2). Furthermore, we included many additional, reasonably complete and articulated specimens from various sites in the Morrison Formation, to test their species-level affinities (e.g., Diplodocus sp. AMNH 223, Osborn, 1899; or Barosaurus sp. AMNH 6341, McIntosh, 2005). Among the additional OTUs are also eight specimens from the Howe Ranch in the vicinity of Shell (Bighorn Basin, Wyoming), which are housed at the SMA.

Table 2 Type specimens and localities of diplodocid species, ordered according to date of description.

Species	Holotype	Comments holotype	Type locality	Stratigraphic age	Other type material	
Dystrophaeus viaemalae Cope, 1877b	USNM 2364		East Canyon Quarry, San Juan County, UT, USA	Oxfordian; low in Morrison Form.		
Amphicoelias altus Cope, 1877a	AMNH 5764		Cope Quarry 12, Garden Park, Fremont County, CO, USA	Kimmeridgian/Tithonian; Brushy Basin Member, Morrison Form.		
‘Amphicoelias’ latus Cope, 1877a	AMNH 5765		Cope Quarry 15, Oil Tract, Garden Park, Fremont County, CO, USA	Kimmeridgian; Salt Wash Member, Morrison Form.		
Apatosaurus ajax Marsh, 1877a	YPM 1860	braincase might be from another specimen (YPM 1840)	Lakes Quarry 10, Morrison, Gunnison County, CO, USA	Kimmeridgian/Tithonian, Upper Brushy Basin Member, Morrison Form.		
Apatosaurus grandis Marsh, 1877a	YPM 1901		Reed’s Quarry 1, Como Bluff, Albany County, WY USA	Kimmeridgian/Tithonian; Brushy Basin Member, Morrison Form.	YPM 1905 (paratype)	
Amphicoelias fragillimus Cope, 1878	AMNH 5777	lost, not included into phylogenetic analysis	Cope Quarry 3, Garden Park, Fremont County, CO, USA	Tithonian; Morrison Form.		
Atlantosaurus immanis Marsh, 1878	YPM 1840		Lakes Quarry 10, Morrison, Gunnison County, CO, USA	Kimmeridgian/Tithonian, Upper Brushy Basin Member, Morrison Form.		
Diplodocus longus Marsh, 1878	YPM 1920		Felch Quarry 1, Garden Park, Fremont County , CO, USA	Kimmeridgian/Tithonian; Lower Middle part of Morrison Form.		
Brontosaurus excelsus Marsh, 1879	YPM 1980		Reed’s Quarry 10, Albany County, WY, USA	Kimmeridgian/Tithonian; Brushy Basin Member, Morrison Form.		
Apatosaurus laticollis Marsh, 1879	YPM 1861		Lakes Quarry 10, Morrison, Gunnison County, CO, USA	Kimmeridgian/Tithonian, Upper Brushy Basin Member, Morrison Form.		
Brontosaurus amplus Marsh, 1881	YPM 1981		Reed’s Quarry 10, Albany County, WY, USA	Kimmeridgian/Tithonian; Brushy Basin Member, Morrison Form.		
Diplodocus lacustris Marsh, 1884	YPM 1922		Lakes Quarry 5, Morrison, Gunnison County, CO, USA	Kimmeridgian/Tithonian; Upper Middle part of Morrison Form.		
Barosaurus lentus Marsh, 1890	YPM 429		Hatch Ranch, Piedmont Butte, Meade County, SD, USA	Kimmeridgian/Tithonian; Morrison Form.		
Barosaurus affinis Marsh, 1899	YPM 419		Hatch Ranch, Piedmont Butte, Meade County, SD, USA	Kimmeridgian/Tithonian; Morrison Form.		
Diplodocus carnegii Hatcher, 1901	CM 84		Sheep Creek Quarry D(3), Albany County, WY, USA	Kimmeridgian/Tithonian; Middle part of Morrison Form.	CM 94 (cotype)	
Elosaurus parvus Peterson & Gilmore, 1902	CM 566	young juvenile	Sheep Creek Quarry 4, Albany County, WY, USA	Kimmeridgian; Morrison Form.		
Gigantosaurus africanus Fraas, 1908	SMNS 12141a, 12145a, 12143, 12140, 12142	individual also contains: SMNS 12145c, MB.R.2728, MB.R.2672, MB.R.2713	Tendaguru Quarry A, Tanzania	Tithonian; Upper Dinosaur Member, Tendaguru Form.		
Apatosaurus louisae Holland, 1915a	CM 3018	might include skull CM 11162	Dinosaur National Monument Quarry, Uintah County, UT, USA	Kimmeridgian/Tithonian; Morrison Form.		
Apatosaurus minimus Mook, 1917	AMNH 675		Bone Cabin Quarry, Albany County, WY, USA	Tithonian; Morrison Form.		
Diplodocus hayi Holland, 1924	HMNS 175	previously CM 662, ic and some other bones still housed at CM	Red Fork Powder River Quarry A, Johnson County, WY, USA	Kimmeridgian/Tithonian; Morrison Form.		
Apatosaurus alenquerensis Lapparent & Zbyszewski, 1957	no holotype assigned		Moinho do Carmo, Alenquer, Lourinhã, Portugal	Kimmeridgian/Tithonian; Sobral Member, Lourinhã Form.	MIGM 2, 4931, 4956-57, 4970, 4975, 4979-80, 4983-84, 5780-81, 30370-88 (lectotype)	
Barosaurus gracilis Russell, Béland & McIntosh, 1980	no type	initially used to distinguish two morphotypes of ’B.’ africanus (Janensch, 1961)				
Cetiosauriscus stewarti Charig, 1980	NHMUK R.3078		Peterborough brick-pit, England	Callovian; Oxford Clay Form.		
Supersaurus vivianae Jensen, 1985	BYU 12962		Dry Mesa Quarry, Mesa County, CO, USA	Kimmeridgian/Tithonian; Brushy Basin Member, Morrison Form.		
Dystylosaurus edwini Jensen, 1985	BYU 4503	old specimen number: BYU 5750	Dry Mesa Quarry, Mesa County, CO, USA	Kimmeridgian/Tithonian; Brushy Basin Member, Morrison Form.		
Seismosaurus halli Gillette, 1991	NMMNH 3690		NMMNH locality L-344, Sandoval Countdown, NM, USA	Kimmeridgian; Brushy Basin Member, Morrison Form.		
Dyslocosaurus polyonychius McIntosh, Coombs & Russell, 1992	AC 663	not sure if same individual, or even same locality	unknown, probably close to Lance Creek, Eastern WY, USA	Morrison, or Lance Form.		
Apatosaurus yahnahpin Filla & Redman, 1994	Tate-001		Bertha Quarry, Albany County, WY, USA	Kimmeridgian/Tithonian; low in Morrison Form.		
Dinheirosaurus lourinhanensis Bonaparte & Mateus, 1999	ML 414		Praia de Porto Dinheiro, Lourinhã, Portugal	Late Kimmeridgian; Amoreira-Porto Novo Member, Lourinhã Form.		
Losillasaurus giganteus Casanovas, Santafé & Sanz, 2001	MCNV Lo-5	individual contains MCNV Lo-1 to Lo-26	La Cañada, Barranco de Escáiz, Valencia, Spain	Tithonian/Barresian; Villar del Arzobispo Form.	MCNV Lo-10 and Lo-23 (paratypes)	
Suuwassea emilieae Harris & Dodson, 2004	ANS 21122		Rattlesnake Ridge Quarry, Carbon County, MT, USA	Late Kimmeridgian; Lower Morrison Form.		
Australodocus bohetii Remes, 2007	MB.R.2455	individual also contains MB.R.2454	Tendaguru Quarry G, Tanzania	Tithonian; Upper Dinosaur Member, Tendaguru Form.	MB.R.2454 (paratype)	
Kaatedocus siberi Tschopp & Mateus, 2012	SMA 0004		Howe Quarry, Bighorn County, WY, USA	Kimmeridgian/Tithonian; Brushy Basin Member, Morrison Form.		
Leinkupal laticauda Gallina et al., 2014	MMCH-Pv 63-1		national route 237, 40 km S of Picún Leufú, Neuquén, Argentina	Lower Cretaceous, Bajada Colorada Formation	MMCH-Pv 63-2 to 63-8 (paratypes)	

Due to the good preservation of the SMA material, the addition of these specimens to a specimen-based phylogenetic analysis as attempted herein is of great importance. By doing so, the anatomical overlap among different OTUs is greatly increased—a very welcome fact, when many of the holotypes are fragmentary and only include few bones, as is the case in Diplodocidae. In particular, two specimens with articulated and almost complete skulls and postcrania (SMA 0004 and 0011) yield important new data. Although the clade Diplodocidae has produced the most skulls within sauropods (Whitlock, Wilson & Lamanna, 2010), only two diplodocine (CM 3452, HMNS 175) and three apatosaurine specimens (CM 3018/11162, CMC 7180, YPM 1860) with possibly articulated skull and postcranial material were reported to date (Holland, 1906; Holland, 1924; McIntosh & Berman, 1975; Berman & McIntosh, 1978; Barrett et al., 2011). Other than CM 11162, which is probably the skull of CM 3018 (Berman & McIntosh, 1978), none of them has yet been described in detail. This renders the identification of disarticulated skull material extremely difficult, and impedes specimen-based phylogenetic analyses. The specimens added herein thus allow detailed reassessments of fragmentary material, including type skeletons and disarticulated skulls.

Material

Our phylogenetic analysis is based on a dataset including characters from Whitlock (2011a), with changes introduced by Mannion et al. (2012) and Tschopp & Mateus (2013b), and combined with the specimen-based analysis of Apatosaurus by Upchurch, Tomida & Barrett (2004), and numerous new characters from various sources (both literature and personal observations, see below). The taxon list was extended to include all holotypes of putative diplodocid taxa, as well as reasonably complete specimens previously assigned to any diplodocid taxon (Table S1). The OTUs representing diplodocid genera and species in previously published analyses were therefore substituted by single specimens representing those taxa.

Terminology

The traditional use of anterior and posterior was preferred over cranial and caudal as common in the description of bird osteology. We applied the nomenclature for vertebral laminae of Wilson (1999) and Wilson (2012), with the changes proposed by Tschopp & Mateus (2013b), and the one for fossae of Wilson et al. (2011).

Positional terms for vertebrae

Serial variation within the vertebral column is highly developed in sauropods and is of taxonomic importance (Wilson, 2002; Wilson, 2012). The high level of observed variability requires detailed character descriptions restricted not only to cervical, dorsal or caudal vertebrae, but even to areas within these respective portions of the column. It is thus common for phylogenetic analyses of sauropod dinosaurs to include characters that are restricted to anterior cervical vertebrae, or mid- and posterior caudal vertebrae, for example (e.g., Wilson, 2002; Upchurch, Barrett & Dodson, 2004; Upchurch, Tomida & Barrett, 2004; Whitlock, 2011a; Mannion et al., 2012; Tschopp & Mateus, 2013b). However, few papers include definitions of these subdivisions. The definitions used in the present analysis mostly follow the ones proposed by Mannion et al. (2013), and are summarized in Table 3.

Table 3 Definitions of positional terms for vertebrae.

Vertebrae	Subdivision	Definition	Example Apatosaurus louisae	
Cervical	Anterior	The division is made numerically	CV 1-5	
	Mid-cervical		CV 6-10	
	Posterior		CV 11-15	
Dorsal	Anterior	Parapophysis still touching centrum	DV 1-2	
	Mid-dorsals	Numerical subdivision	DV 3-6	
	Posterior		DV 7-10	
Caudal	Anterior-most	With transverse processes extending onto neural arch	Cd 1-6	
	Anterior	With normal transverse process	Cd 7-14	
	Mid-caudal	Without transverse processes, but still well-developed neural spine	Cd 15-28	
	Posterior	Postzygapophyses reduced	Cd 29-42	
	Distal	Neural arch reduced	Cd 43-82	

Ingroup specimens phylogenetic analysis

The following individual, presumed diplodocid, specimens were included in the ingroup of the phylogenetic analysis. All of these are reasonably complete specimens of reputed diplodocid species, or constitute the holotypes of taxa, irrespective of completeness, which have been either referred or associated to Diplodocidae. Previous classifications and assignments, as well as comments on the likelihood that they represent singular individuals, are given below, alphabetically ordered. Specimens that were at least partially scored based on personal observations are marked with an asterisk. Outgroups comprise species-, or genus-level taxa from non-neosauropod Eusauropoda, Macronaria, as well as closely related Diplodocoidea, and are not further discussed here.

Amphicoelias altus, AMNH 5764* and AMNH 5764 ext*

The holotype of Amphicoelias altus originally included a tooth, two dorsal vertebrae, a pubis, and a femur (Cope, 1877a). A scapula, coracoid, and an ulna were later provisionally referred to the specimen (Osborn & Mook, 1921). However, the strongly expanded distal end of the scapula, and the relatively deep notch anterior to the glenoid on the coracoid actually resemble more Camarasaurus than any diplodocid (McIntosh, 1990b; E Tschopp, pers. obs., 2011). The same accounts for the single tooth stored at AMNH (Osborn & Mook, 1921). The tooth has already been excluded from scores of A. altus in recent phylogenetic analyses (Whitlock, 2011a; Mannion et al., 2012), which is followed here. Mannion et al. (2012) furthermore excluded the referred forelimb elements. Given that personal observations confirmed the rather camarasaurid than diplodocid morphology of the scapula and coracoid, but not particularly the ulna, two different preliminary phylogenetic analyses were performed with a reduced (excluding the tooth, the scapula and the coracoid, but including the ulna) and the extended holotype Amphicoelias altus OTU (including all referred elements other than the tooth). Because both analyses yielded the same position for the specimens, the reduced holotype was preferred in the final analysis. The risk of adding dubious information from potentially wrongly referred material was thus circumvented. More detailed analysis is needed in order to refine these assignments.

“Amphicoelias” latus, AMNH 5765*

This is a fragmentary specimen comprising four caudal vertebrae and a right femur from the same site as the holotypes of Camarasaurus supremus and Amphicoelias altus (Cope, 1877a; Osborn & Mook, 1921; Carpenter, 2006). Both the vertebrae and the femur show greater resemblance with Camarasaurus than to Amphicoelias, which led Osborn & Mook (1921) to synonymize A. latus with C. supremus.

Apatosaurus ajax, YPM 1860*

The holotype of Apatosaurus ajax also constitutes the genoholotype of Apatosaurus (i.e., A. ajax is the type species of Apatosaurus). During collection and shipping it became intermingled with YPM 1840, the holotype of Atlantosaurus immanis (McIntosh, 1995). As a result, it is currently difficult to distinguish the two individuals, even though they come from different quarries. We follow the suggestions of Berman & McIntosh (1978) and McIntosh (1995) in deciding which elements of the mingled taxa comprise the holotype individual of Apatosaurus ajax. The only material not confidently referable to either specimen is a braincase currently labeled ‘YPM 1860.’ In order to investigate the taxonomic implications of the attribution of this braincase to the types of Apatosaurus ajax or Atlantosaurus immanis, two supplementary analyses were performed with scores of the braincase added to YPM 1840 and 1860, respectively. Adding the information from the braincase to YPM 1840, tree length increases but positions of the two specimens remain the same. An assignment of the braincase to the holotype of Apatosaurus ajax appears thus more parsimonious, supporting the possibility that it was labeled correctly.

Apatosaurus ajax, AMNH 460*

This specimen was recovered as Apatosaurus ajax in the specimen-based phylogenetic analysis of Upchurch, Tomida & Barrett (2004). AMNH 460 is currently mounted with reconstructed portions based on other specimens. Therefore, caution was used, to avoid scoring characters based on material belonging to other individuals (for a list of bones belonging to AMNH 460, see Table S1).

Apatosaurus ajax, NSMT-PV 20375

Described by Upchurch, Tomida & Barrett (2004), this specimen is the only fully described skeleton previously referred to A. ajax. It is relatively complete, although abnormal length ratios of the humerus, radius and metacarpal III suggest that NSMT-PV 20375 might be composed of more than one individual, possibly including bones of the Camarasaurus specimens found intermingled in the quarry (Upchurch, Tomida & Barrett, 2004). These forelimb elements were thus excluded from scores of the OTU in the present analysis.

Apatosaurus laticollis, YPM 1861*

Apatosaurus laticollis is based on a single, fragmentary cervical vertebra (Marsh, 1879). Subsequent studies proposed that this vertebra actually belongs to the same individual as the holotype material of Atlantosaurus immanis (YPM 1840), which were both found in the Lakes Quarry 1 (McIntosh, 1995). Here, the specimens were kept apart in order to evaluate this hypothesis.

Apatosaurus louisae, CM 3018* (holotype) and CM 11162*

The most complete specimen of Apatosaurus is CM 3018, a postcranial skeleton that was preliminarily described as a new species by Holland (1915a) and reassessed in a detailed monograph by Gilmore (1936). An obvious diplodocid skull (CM 11162) was found near it, but the referral of this skull remained confused for a long time (Holland, 1915b; Holland, 1924; Berman & McIntosh, 1978). Because Apatosaurus was thought to have a short, Camarasaurus-like skull at the time, Holland’s proposal that CM 11162 was the actual skull of CM 3018 (Holland, 1915b; Holland, 1924) was generally rejected (e.g., Gilmore, 1936). Only with the detailed description and study of the specimen by Berman & McIntosh (1978) was CM 11162 recognized as the now widely accepted long skull-form of Apatosaurus. Given the small distance between skull and postcrania in the quarry, as well as the perfectly fitting size of the cranial occipital condyle and postcranial atlas, the probability that the two belong to the same individual is very high (Holland, 1915b; Berman & McIntosh, 1978). Accordingly, the OTU representing the holotype of Apatosaurus louisae in the present analysis comprises scoring from both CM 3018 and 11162.

Apatosaurus louisae, CM 3378*

This specimen was identified as Apatosaurus louisae in the analysis of Upchurch, Tomida & Barrett (2004). Although it has never been described in detail, CM 3378 yields important information on the number of vertebrae in Apatosaurus, as this specimen is the only one known with an articulated, uninterrupted vertebral column from the mid-cervical region to the last caudal vertebra (Holland, 1915b; McIntosh, 1981). CM 3378 was found at the Dinosaur National Monument, associated with a diplodocid skull (CM 11161; interpreted as Diplodocus), as well as appendicular elements. However, according to McIntosh (1981), these materials cannot be attributed to the same individual as CM 3378 with certainty, and no scores from them were thus included in this OTU.

Apatosaurus louisae, LACM 52844*

As with other specimens previously identified as A. louisae, LACM 52844 also comes from the Dinosaur National Monument quarry. It was found nearly complete and mostly articulated, just below the holotype CM 3018 and skull CM 11162 (McIntosh & Berman, 1975; Berman & McIntosh, 1978). Originally, LACM 52844 was housed at CM and bore the accession number CM 11990 (McIntosh, 1981). Although it was reported to be nearly complete (McIntosh, 1981), only a limited number of bones were located and scored at LACM during our study (Table S1; E Tschopp, pers. obs., 2013).

“Apatosaurus” minimus, AMNH 675*

Initially described as new species of Apatosaurus (Mook, 1917), AMNH 675 is now generally considered an indeterminate sauropod, with affinities to Macronaria, based on pelvic girdle morphology (McIntosh, 1990a; Upchurch, Barrett & Dodson, 2004; Mannion et al., 2013). In order to test this, Isisaurus colberti was added to the analysis. Isisaurus has the typical titanosaurian sacrum with six vertebrae and the preacetabular lobe oriented perpendicular to the vertebral axis (Jain & Bandyopadhyay, 1997), as is the case in AMNH 675. A diplodocid chevron is also accessioned under AMNH 675. However, AMNH records indicate it was ‘found loose with other Bone Cabin Quarry material.’ We therefore excluded it from the A. minimus OTU.

Apatosaurus parvus, UW 15556

This specimen was found by the Carnegie Museum, intermingled with the holotype specimen of Elosaurus parvus, CM 566 (Hatcher, 1902; Peterson & Gilmore, 1902). It was initially accessioned as CM 563, but was later transferred to the University of Wyoming (McIntosh, 1981). Usually identified as A. excelsus (Gilmore, 1936), a specimen-based phylogenetic analysis supported the retention of the species A. parvus for CM 566 and UW 15556 (Upchurch, Tomida & Barrett, 2004).

Apatosaurus sp., BYU 1252-18531*

Only one mention of this specimen exists, discussing sacral rib anatomy (D’Emic & Wilson, 2011). It was found in Utah, and is nearly complete and largely articulated (E Tschopp, pers. obs., 2013). The specimen is partly on display at BYU, where it is labeled as A. excelsus. No more detailed information can be given because the specimen is currently under study.

Apatosaurus sp., FMNH P25112

Riggs (1903) described this specimen (formerly FMNH 7163) as A. excelsus, which led him to two important conclusions: (1) Brontosaurus is a junior synonym of Apatosaurus, and (2) during ontogeny, additional vertebrae are added from the dorsal and caudal series to the sacrum. Later, the specimen-based phylogenetic analysis of Upchurch, Tomida & Barrett (2004) recovered it on a disparate branch within Apatosaurus, suggesting that FMNH P25112 represents a novel species. The specimen is mounted at FMNH together with the neck and forelimbs of FMNH P27021 (W Simpson, pers. comm., 2013).

Apatosaurus sp., ML 418*

This specimen is very badly preserved. It was identified as a possible Dinheirosaurus, Apatosaurus, or a yet unknown, indeterminate diplodocid (Antunes & Mateus, 2003; Mateus, 2005; Mannion et al., 2012). One dorsal vertebra has been prepared and additional unprepared material includes dorsal rib fragments, and a partial tibia. A mid- or posterior cervical vertebra of the same individual was lost due to the friable preservation, and scores concerning the cervical vertebrae are therefore based on photographs taken prior to their loss.

“Atlantosaurus” immanis, YPM 1840*

This is possibly the same individual as YPM 1861 (Apatosaurus laticollis), and it was mingled with YPM 1860 (Apatosaurus ajax) during shipping (see above). McIntosh (1995) tried to separate them based on their color, and on sparse field notes. In the YPM collections, the specimens are still labeled as they were before McIntosh’s study, therefore it is difficult to reproduce his results. Scores for an ischium of YPM 1840 are based on personal observation, whereas cervical and dorsal vertebral characters are derived from the literature (Marsh, 1896; Ostrom & McIntosh, 1966; Upchurch, Tomida & Barrett, 2004).

Australodocus bohetii, holotype* and paratype*

The holotype and paratype of Australodocus bohetii are two successive mid-cervical vertebrae from the same individual (Remes, 2007). A. bohetii was initially described as a diplodocine (Remes, 2007), but Whitlock (2011a) and Whitlock (2011c) suggested titanosauriform affinities for the species. Subsequently, Mannion et al. (2013) suggested Australodocus to be a non-lithostrotian titanosaur. Accordingly, Ligabuesaurus leanzai was added to the taxon list in order to include a possible closely related derived titanosauriform that has anatomical overlap with A. bohetii.

Barosaurus affinis, YPM 419*

The holotype of B. affinis consists only of pedal material, and has no overlap with the holotype of B. lentus (Marsh, 1890; Marsh, 1899). Because they come from the same quarry, the two species were usually regarded as synonyms (Lull, 1919; McIntosh, 2005). McIntosh (2005) identified the elements as mt I and partial mt II, but the latter is herein interpreted to represent the proximal portion of mt V instead. The bone is widely expanded, and has the typical ‘paddle’-shape of the metatarsal V in sauropods (E Tschopp, pers. obs., 2011).

Barosaurus lentus, YPM 429*

Although this specimen is the genoholotype of Barosaurus (Marsh, 1890; Lull, 1919; i.e., B. lentus is the type species of Barosaurus), most characterization of Barosaurus is based on another, more complete, and articulated specimen (AMNH 6341, see below). YPM 429 as presently available has a high degree of reconstruction, especially in some cervical vertebrae.

Barosaurus sp., AMNH 6341*

This specimen is the most complete individual probably referable to Barosaurus (McIntosh, 2005). It was collected in three parts and subsequently separated among three institutions (USNM, CM, and UUVP), but later brought together by B Brown for the AMNH (Bird, 1985). Some doubts exist concerning the correct attribution of a tibia-fibula pair, which might belong to a Diplodocus specimen found in the vicinity of AMNH 6341 (McIntosh, 2005).

Barosaurus sp., AMNH 7530*

Both the holotype specimen of Kaatedocus siberi (SMA 0004) and AMNH 7530 were found at Howe Quarry (Michelis, 2004; Tschopp & Mateus, 2013b). AMNH 7530 is tagged as cf. Barosaurus on display at AMNH, probably based on a tentative identification made by Brown (1935), but without detailed study. Furthermore, the current display label wrongly identifies the specimens as AMNH 7535 (Michelis, 2004). AMNH 7530 is an important specimen for diplodocid taxonomy because it includes articulated anterior and mid-cervical vertebrae and a partial skull.

Barosaurus sp., AMNH 7535*

This specimen was recovered with Kaatedocus siberi SMA 0004 and AMNH 7530 at Howe Quarry (Michelis, 2004; Tschopp & Mateus, 2013b), and has been simply cataloged as Barosaurus in the collections of the AMNH (likely by B Brown; Brown, 1935). AMNH 7535 largely preserves the same elements as SMA 0004 and AMNH 7530, and appears to be of about the same size. A partial tail is also accessioned under AMNH 7535, but given the chaotic distribution of specimens in the quarry (Tschopp & Mateus, 2013a: Fig. 1), it is impossible to confidently attribute disparate and disarticulated portions to any single common individual. A diplodocid quadrate that was initially cataloged under AMNH 7535 now bears the number AMNH 30070. Because the original attribution of this quadrate to AMNH 7535 was probably based on their vicinity in the quarry, two analyses were performed with and without the information of this bone, yielding the same phylogenetic position in both iterations. In both instances, information from the caudal series was omitted from scores of AMNH 7535. Scores on the quadrate were retained in the final analysis because AMNH 30070 shows some differences with the quadrates known from Kaatedocus (e.g., lack of the small fossa dorsomedially on the quadrate shaft, E Tschopp, pers. obs., 2011), as do also the cervical vertebrae.

Figure 1 Sauropod skulls.

Skulls of Mamenchisaurus youngi (A; modified from Ouyang & Ye, 2002), Camarasaurus sp. USNM 13786 (B) Giraffatitan brancai (C; modified from Janensch, 1935), Diplodocus sp. CM 11161 (D) and Galeamopus sp. SMA 0011 (E) in lateral view, illustrating the states of the characters 1, 5, 13, 14, 15, 19, 20, 21, 37, 38, 39, 45, 46, 47, 55, 113. Not to scale.

Barosaurus sp., CM 11984*

Together with YPM 429 and AMNH 6341, CM 11984 represents a third, relatively complete, likely Barosaurus specimen (McIntosh, 2005). Some of the material of CM 11984 is still unprepared, and further crucial information on Barosaurus can be expected once these are freed from matrix. In addition to the vertebral column, a pes is accessioned under CM 11984, which McIntosh (2005) considered to have a dubious association with the remaining material, given the chaotic quarry situation at Dinosaur National Monument. Therefore, this pes is not considered as part of the scoring of CM 11984.

Barosaurus sp., SMA O25-8*

This specimen is a partial skull from the Howe Quarry. Due to differences both in braincase and endocast morphology compared to the holotype of Kaatedocus siberi SMA 0004, Schmitt et al. (2013) showed that two diplodocine taxa were present at the Howe Quarry. SMA O25-8 was tentatively referred to Barosaurus because the elongate cervical vertebrae of the specimen AMNH 7535 (which is different from K. siberi, see above) are more similar to this genus than to any other North American diplodocine (Schmitt et al., 2013).

Brachiosaurus sp., SMA 0009*

Initially described as a diplodocid (Schwarz et al., 2007), a reassessment of the systematic position of SMA 0009 after further preparation of the mid-cervical vertebrae revealed probable titanosauriform affinities (Carballido et al., 2012a). Carballido et al. (2012a) suggested that SMA 0009 represents an immature Brachiosaurus. Therefore, B. altithorax (Riggs, 1904; Taylor, 2009) was included in our dataset to test this possibility.

Brontosaurus amplus, YPM 1981*

The type of B. amplus (Marsh, 1881) is generally referred to Apatosaurus excelsus (Gilmore, 1936; McIntosh, 1990a; McIntosh, 1995; Upchurch, Tomida & Barrett, 2004), but has never been described in detail.

Brontosaurus excelsus, YPM 1980*

The holotype of Brontosaurus excelsus (now commonly synonymized with Apatosaurus) was the first to be published with a reconstruction of the entire skeleton (Marsh, 1883) and is still one of the best preserved diplodocid specimens worldwide. The skeleton was extensively reconstructed prior to being mounted at the YPM. Therefore, special care was taken when scoring characters from the original specimen.

Camarasaurus grandis, YPM 1901

Marsh (1877a) initially assigned this species to Apatosaurus, but subsequently referred it to Morosaurus (Marsh, 1878; later synonymized with Camarasaurus: Mook, 1914). There is some confusion about the correct assignment of several bones to either the holotype YPM 1901 or the referred specimens YPM 1902 or YPM 1905 from the same quarry (see Ostrom & McIntosh, 1966). Herein, scores are included from all elements potentially belonging to YPM 1901 (according to Ostrom & McIntosh, 1966). Because all three specimens were referred to Camarasaurus, this should have no influence on the ingroup relationships of the current phylogenetic analysis.

Cetiosauriscus stewarti, NHMUK R3078*

The holotype specimen was first described in the early 1900s (Woodward, 1905) as Cetiosaurus leedsi. However, Huene (1927) identified ‘Cetiosaurus’ leedsi as a separate genus, Cetiosauriscus, and highlighted the then referred specimen NHMUK R3078 as exemplifying the new genus. NHMUK R3078 was made the holotype of Cetiosauriscus stewarti (Charig, 1980), which later was instated as the type species of Cetiosauriscus (Charig, 1993). It was included in Diplodocidae by McIntosh (1990b), based on pedal morphology, but subsequent analyses proposed a closer relationship with the non-neosauropod eusauropods Mamenchisaurus or Omeisaurus, as well as with Tehuelchesaurus (Heathcote & Upchurch, 2003). Mamenchisaurus and Omeisaurus were thus included in the present analysis in order to test these competing hypotheses. A detailed restudy of C. stewarti is in preparation by P Upchurch, P Mannion & J Heathcote (pers. comm., 2011, 2012), and will doubtlessly reveal more valid comparisons. Because personal observation of the caudal vertebrae of Spinophorosaurus nigerensis revealed high similarity with Cetiosauriscus, S. nigerensis was added to the matrix, in order to appraise the phylogenetic significance of their morphological similarities.

Dinheirosaurus lourinhanensis, ML 414*

The holotype of Dinheirosaurus lourinhanensis was originally referred to Lourinhasaurus alenquerensis by Dantas et al. (1998), but Bonaparte & Mateus (1999) realized that ML 414 represents a different genus. Contrary to the phylogenetic assignment of L. alenquerensis, which is now thought to be a basal macronarian (see below), the diplodocid affinities of D. lourinhanensis are well supported by four phylogenetic analyses (Rauhut et al., 2005; Whitlock, 2011a; Mannion et al., 2012; Tschopp & Mateus, 2013b).

Diplodocinae indet., SMA 0011*

SMA 0011 has been mentioned by Klein & Sander (2008) as Diplodocinae indet, and its ontogenetic stage identified histologically as HOS 9, corresponding to sexual maturity (Klein & Sander, 2008). The specimen is nearly complete and largely articulated, preserving bones from all skeletal regions except for the tail (E Tschopp, pers. obs., 2011). It thus plays a very important role in increasing character overlap between the more fragmentary OTUs.

Diplodocinae indet., SMA 0087*

This specimen comprises a completely articulated skeleton from mid-dorsal vertebrae to mid-caudal vertebrae, the pelvic girdle and left hindlimb. It was found at the Howe-Scott quarry, about one meter below the specimen SMA 0011 (E Tschopp, pers. obs., 2003). The histology of SMA 0087 was studied by Klein & Sander (2008), who showed that it was an adult individual (HOS 11), and identified it as Diplodocinae indet.

Diplodocus carnegii, CM 84*

The holotype of D. carnegii is one of a few specimens of Diplodocus that includes cervical vertebrae. It is mounted at CM, and has been “completed” with bones from various other specimens: CM 94, 307, 21775, 33985, HMNS 175, USNM 2673, and AMNH 965 (McIntosh, 1981; Curtice, 1996). Scores of the holotype of D. carnegii are based on this mounted specimen, with effort taken to ensure that only material from CM 84 was included. D. carnegii was erected based on comparisons to AMNH 223, which showed some differences in caudal neural spine orientation. If compared with the original type material, the differences are not as clear, and were in fact disputed by Gilmore (1932).

Diplodocus carnegii, CM 94*

This specimen was described as a cotype of D. carnegii by Hatcher (1901). Both holotype and cotype specimens were found in the same quarry, alongside material of other genera (Hatcher, 1901). Oddly, CM 94 includes two pairs of ischia, which casts some doubt on the true attribution of bones to individual specimens (McIntosh, 1981; E Tschopp, pers. obs., 2011). Because both pairs of ischia show the same characteristics, we included the entire material excluding one pair of ischia from the OTU representing CM 94 (including some bones mounted with the holotype of ‘Diplodocus’ hayi HMNS 175, see below). However, further studies are needed in order to definitively assign the various bones among the at-least two individuals present.

Diplodocus cf. carnegii, WDC-FS001A*

This specimen has not been described entirely, but is the most complete specimen referred to Diplodocus that has a manus with associated hindlimb and axial material (Bedell & Trexler, 2005). The specimen was found in two spatial clusters in the quarry, but the lack of duplicated bones, the two similarly sized humeri, and osteological indications of a single ontogenetic stage led Bedell & Trexler (2005) to identify the materials as belonging to a single individual with affinities to D. carnegii.

“Diplodocus” hayi, HMNS 175*

The holotype specimen of ‘D.’ hayi was initially housed at CM (as CM 662), prior to residing in Cleveland for a time (formerly CMNH 10670). Holland (1924) described it as a novel species of Diplodocus, based solely on cranial characters. At that time, Apatosaurus was thought to have a Camarasaurus-like skull (see Berman & McIntosh, 1978), which probably influenced researchers to identify any elongate, diplodocid skull as Diplodocus. McIntosh (1990a), amongst others, later suggested that ‘D.’ hayi might actually not belong to Diplodocus, but to a unique genus, based on various similarities with Apatosaurus in the cranium, forelimb, and tail. Because the specimen is mounted at HMNS (together with reconstructions and original bones from CM 94; McIntosh, 1981), it is only of limited accessibility. Nevertheless, the present phylogenetic analysis corroborates a referral of ‘D.’ hayi to a unique genus (see below).

Diplodocus lacustris, YPM 1922*

The original type material of D. lacustris comprises teeth, a premaxilla, and a maxilla (Marsh, 1884). However, personal observations at YPM reveal that the cranial bones clearly belong to Camarasaurus or a morphologically similar taxon, and that there is no relationship between them and the teeth. Mossbrucker & Bakker (2013) described a newly found putative apatosaur maxilla and two premaxillae from the same quarry, proposing that they might belong to the same individual as the teeth of YPM 1922. However, given the lacking field notes from the first excavations, such a referral will be difficult to prove. Therefore, in the present analysis, only the teeth were scored for D. lacustris.

Diplodocus longus, YPM 1920*

YPM 1920 constitutes the genoholotype of Diplodocus (Marsh, 1878; i.e., D. longus is the type species of Diplodocus) and thus has special taxonomic importance. Unfortunately, it is highly incomplete, with only two nearly complete caudal vertebrae, and few additional fragmentary anterior to mid-caudal vertebrae identifiable in the YPM collections. A chevron was reported as belonging to the same individual (Marsh, 1878; McIntosh & Carpenter, 1998), but it could not be located at YPM in 2011. Other articulated vertebrae were found in the field but discarded due to their friable preservation (McIntosh & Carpenter, 1998). Extraneous materials were once assigned to the same specimen, including a skull, femur, tibia, fibula, astragalus, and five metatarsals (still accessioned under YPM 1920), as well as an ulna, radius, and partial manus assigned to YPM 1906 (McIntosh & Carpenter, 1998). However, only the caudal series and the chevron can be confidently identified as belonging to the holotypic individual (McIntosh & Carpenter, 1998), as scored in the present analysis.

Diplodocus sp., AMNH 223*

This specimen was first described as Diplodocus longus (Osborn, 1899). It was the first reasonably articulated specimen of Diplodocus and thus became an important comparative specimen (see Hatcher, 1901). Three partial cervical neural arches, described and figured by Osborn (1899), were not located at AMNH during the collection visits in 2010 and 2011. Coding of these elements is thus based entirely on Osborn (1899).

Diplodocus sp., AMNH 969*

This skull and associated atlas and axis were identified as D. longus, based on an earlier report of a skull allegedly belonging to the holotype specimen of D. longus, YPM 1920 (Marsh, 1884; Holland, 1906). However, the only reported Diplodocus specimen with an articulated skull and anterior cervical vertebrae is CM 3452, of which only the skull has been described (Holland, 1924). Because no anterior cervical vertebrae are definitely attributable to D. longus, the only comparison that can be made is with the D. carnegii type specimens, of which only CM 84 preserves the axis. Because the two differ in morphology (e.g., of the prespinal lamina), AMNH 969 was herein regarded Diplodocus sp.

Diplodocus sp., CM 3452*

On display at CM, this specimen is the only possible Diplodocus with articulated skull and anterior cervical vertebrae (McIntosh & Berman, 1975). However, the cervical vertebrae have not been described, and no detailed study has been done in order to identify the species affinity for CM 3452. Comparison with other specimens referred to Diplodocus is hampered due to the presence of very little anatomical overlap.

Diplodocus sp., CM 11161*

This specimen is only a skull. It was described as Diplodocus longus by Holland (1924) and McIntosh & Berman (1975), based on comparisons with the earlier reported putative Diplodocus skulls AMNH 969, USNM 2672, and 2673. However, because all of them were disarticulated and found in quarries that also produced other diplodocid genera, care must be taken concerning these identifications. Our knowledge of diplodocid skulls to date suggests that they are extremely similar to each other, and very few distinguishing characters have yet been proposed (Berman & McIntosh, 1978; McIntosh, 2005; Harris, 2006a; Remes, 2006; Whitlock, Wilson & Lamanna, 2010; Whitlock, 2011b; Tschopp & Mateus, 2013b; Whitlock & Lamanna, 2012). Thus, we refrain from referring CM 11161 to any species of Diplodocus until postcranial diagnostic traits are robustly linked to cranial morphologies.

Diplodocus sp., CM 11255*

This skull was found without associated postcranial material, in the same quarry as the skulls CM 11161 and 11162. It was first mentioned and figured by Holland (1924), and completely described by Whitlock, Wilson & Lamanna (2010). The latter authors identified CM 11255 as Diplodocus due to obvious differences with skulls referred to Apatosaurus, Suuwassea, and Tornieria, and closer resemblance to skulls referred to Diplodocus (Whitlock, Wilson & Lamanna, 2010). However, Whitlock, Wilson & Lamanna (2010) also acknowledged that several diplodocine taxa are not known from cranial material, so that a definitive assignment to the genus Diplodocus is currently impossible.

Diplodocus sp., DMNS 1494*

This specimen is a relatively complete, articulated find from the Dinosaur National Monument. The only disarticulated elements are the right scapulacoracoid and the left hindlimb. These elements were not included in the present analysis because DMNS 1494 was found intermingled with other skeletons (V Tidwell, pers. comm., 2010). DMNS 1494 was collected by the Carnegie Museum and later transferred to DMNS for exhibit. A right fibula and astragalus of the same specimen remained at CM (presently CM 21763; McIntosh, 1981). The specimen has never been formally described, but is ascribed to D. longus (e.g., Gillette, 1991). Together with CM 84, DMNS 1494 is the only Diplodocus specimen included here with articulated, and complete cervical vertebrae.

Diplodocus sp., USNM 2672*

Like AMNH 969, USNM 2672 preserves a partial skull and atlas. It was the first diplodocid skull to be reported, and was initially included within the holotype of D. longus, YPM 1920 (Marsh, 1884), although labeled YPM 1921 (Berman & McIntosh, 1978). However, this skull and the holotypic caudal vertebrae were not found in articulation or even close association, so this attribution must be regarded as questionable (McIntosh & Carpenter, 1998), and the two specimens were treated as distinct OTUs in our analyses.

Diplodocus sp., USNM 2673*

This specimen was found in the same quarry as USNM 2672, and initially cataloged as YPM 1922, before it was transferred to USNM (McIntosh & Berman, 1975). Although it bore the same YPM specimen number as the D. lacustris holotype, it cannot be from the same specimen as they were found in different quarries (Marsh, 1884; McIntosh & Berman, 1975).

Diplodocus sp., USNM 10865*

Although USNM 10865 is one of the most complete Diplodocus specimens, it has only been preliminarily described and was tentatively referred to D. longus by Gilmore (1932). USNM 10865 was found close to the articulated Barosaurus AMNH 6341 (‘#340’ in Gilmore, 1932; McIntosh, 2005). According to McIntosh (2005), two sets of left lower legs of different lengths were found associated with USNM 10865. The shorter set was mounted by Gilmore (1932), but McIntosh (2005) suggests that this assignment might have been wrong. For our character 440 relating to the tibia/femur length, the higher ratio was therefore used, following McIntosh (2005).

Dyslocosaurus polyonychius, AC 663*

The only specimen of this putative diplodocid sauropod consists solely of appendicular elements of dubious origin and association (McIntosh, Coombs & Russell, 1992). No field notes exist, but personal observations of differing color and preservation among individual bones led to the conclusion that at least the supposed php III-1 was probably not collected at the same place as the rest of the holotype specimen (E Tschopp, 2011, unpublished data). It is therefore excluded from scores of Dyslocosaurus in this phylogenetic analysis. A more detailed reassessment of this specimen is in progress (E Tschopp & J Nair, 2015, unpublished data), and might reveal additional information on its taxonomic affinities. The phylogenetic position yielded in the present analysis is regarded as preliminary.

Dystrophaeus viaemalae, USNM 2364*

This specimen is highly fragmentary, but was identified as possibly diplodocoid by McIntosh (1990b; his ‘Diplodocidae’ conforms to the current use of the Diplodocoidea). The type material is only partly prepared, which largely impedes the identification of crucial character states. The type locality was relocated in the mid-1990s, and more material of the probable holotypic individual was excavated, of which only a phalanx has been identifiable (Gillette, 1996a; Gillette, 1996b). However, Gillette (1996a) and Gillette (1996b) stated that more material is probably present, such that additional information on Dystrophaeus might be forthcoming. Both in the initial description (Cope, 1877b) and a reassessment (Huene, 1904), several of the bones were misidentified: metacarpal V (according to Huene, 1904) is most probably a metacarpal I, based on the angled distal articular surface (McIntosh, 1997; E Tschopp, pers. obs., 2011). Cope (1877b) correctly identified a partial scapula (contra Huene, 1904, who thought it was a pubis), but misidentified a complete ulna and a partial radius as humerus and ulna, respectively, as already recognized by Huene (1904). The OTU as included here therefore consists of a partial dorsal vertebra, a partial scapula, an ulna, a distal radius, and the metacarpals.

Dystylosaurus edwini, BYU 4503*

The holotype of Dystylosaurus edwini is an anterior dorsal vertebra (Jensen, 1985). There is some doubt concerning its taxonomic affinities: it has been identified as either brachiosaurid (Paul, 1988; McIntosh, 1990b; Upchurch, Barrett & Dodson, 2004; Chure et al., 2006) or diplodocid, possibly even from the same individual as the Supersaurus vivianae holotype scapulacoracoid (Curtice & Stadtman, 2001; Lovelace, Hartman & Wahl, 2007). It was included in a preliminary analysis as an OTU independent from Supersaurus vivianae BYU and WDC DMJ-021 in order to clarify its taxonomic status. The results yielded 102 most parsimonious trees, where Dystylosaurus always grouped with the two Supersaurus OTUs, which sometimes included Dinheirosaurus ML 414, “Diplodocus” hayi HMNS 175, Barosaurus affinis YPM 419, or Diplodocus lacustris YPM 1922 within the same branch. In 31 out of 102 most parsimonious trees Dystylosaurus and the two Supersaurus OTUs were found as sister taxa. This result corroborates the hypothesis of Curtice & Stadtman (2001) and Lovelace, Hartman & Wahl (2007) that the Dystylosaurus holotypic vertebra is Supersaurus, and most probably from the same individual as the Supersaurus holotype. In our definitive analysis, BYU 4503 was thus included as part of the combined OTU representing the BYU specimens of Supersaurus vivianae.

“Elosaurus” parvus, CM 566*

CM 566 is a small juvenile that is generally referred to Apatosaurus excelsus (McIntosh, 1995), or constitutes the independent species Apatosaurus parvus together with an adult specimen (UW 15556; Upchurch, Tomida & Barrett, 2004), with which it was found associated (Peterson & Gilmore, 1902). However, it was initially described as a unique genus (Peterson & Gilmore, 1902).

Eobrontosaurus yahnahpin, Tate-001

Initially described as Apatosaurus yahnahpin (Filla & Redman, 1994), a separate genus was erected for the specimen (Bakker, 1998), partly based on differences in coracoid morphology to Apatosaurus. The specimen has been considered a camarasaurid (Upchurch, Barrett & Dodson, 2004), but more recently, Mannion (2010) suggested diplodocid affinities. The taxon has never been included in any phylogenetic analysis, but a detailed description of the entire material appears to be in preparation (R Bakker, pers. comm., 2008, cited in Mannion, 2010).

Kaatedocus siberi, SMA 0004*

Before its detailed examination, the holotype of Kaatedocus siberi was generally reported as Diplodocus (Ayer, 2000) or Barosaurus (Michelis, 2004). Subsequently, a description and phylogenetic reappraisal of SMA 0004 revealed its generic separation from Diplodocus and Barosaurus (Tschopp & Mateus, 2013b).

Kaatedocus siberi, SMA D16-3*

This additional specimen from the Howe Quarry (a partial skull) was referred to K. siberi by Schmitt et al. (2013). The skull bones were found disarticulated but associated (E Tschopp, pers. obs., 2012), and have not been described in detail yet.

Leinkupal laticauda, MMCH-Pv 63-1

The holotype of Leinkupal laticauda was only recently described (Gallina et al., 2014). It includes only a single caudal vertebrae, although more elements from the same quarry were referred to the species by Gallina et al. (2014). All diplodocid remains were found disarticulated and mingled with dicraeosaur material (Gallina et al., 2014), and it is thus currently too early to include more than the holotypic anterior caudal vertebra in a specimen-level cladistic analysis as attempted herein.

Losillasaurus giganteus, MCNV Lo-1 to 26*

This OTU represents an individual containing the holotypic caudal vertebra, Lo-5, the paratypes Lo-10 and Lo-23, and several additional elements. All the bones of MCNV Lo-1 to 26 were found associated and no duplication of bones occurred (Casanovas, Santafé & Sanz, 2001). Initially regarded as a basal diplodocoid (Casanovas, Santafé & Sanz, 2001), Losillasaurus was soon found to represent a non-diplodocoid, and probably a non-neosauropod eusauropod (Rauhut et al., 2005; Harris, 2006c). With the description of Turiasaurus (Royo-Torres, Cobos & Alcalá, 2006), which has since been consistently recovered as sister genus to Losillasaurus (Royo-Torres, Cobos & Alcalá, 2006; Royo-Torres et al., 2009; Barco, 2009; Carballido et al., 2012b; Royo-Torres & Upchurch, 2012), this more basal position has been generally accepted. Therefore, Turiasaurus was added as an outgroup to test their sister relationship.

Lourinhasaurus alenquerensis, lectotype*

This species was first described by Lapparent & Zbyszewski (1957) as referable to Apatosaurus, but later included in Camarasaurus (McIntosh, 1990a). Subsequently, Dantas et al. (1998) erected a new genus for the species, but only Antunes & Mateus (2003) clearly assigned a specific type specimen to the species. Lourinhasaurus has usually been recovered as a basal macronarian in recent phylogenetic analyses (Royo-Torres & Upchurch, 2012; Mocho, Royo-Torres & Ortega, 2014).

“Seismosaurus” hallorum, NMMNH 3690

The holotype of S. hallorum was initially described as S. halli, and as one of the largest sauropods ever (Gillette, 1991). However, this identification as a unique genus, and its size estimate, were mainly based on an incorrect assignment of the position of some mid-caudal vertebrae (Curtice, 1996; Herne & Lucas, 2006). Subsequent reanalysis of the specimen revealed that it is indistinguishable from Diplodocus and that it probably belongs to the same species as AMNH 223 and USNM 10865 (Lucas et al., 2006; Lovelace, Hartman & Wahl, 2007). Gillette himself (1994) corrected the species name from halli to hallorum, as he did not apply the correct latin ending for the plural in the initial description (Gillette, 1991; Gillette, 1994). Because the corrected name has since been used more widely than the original proposal, it is followed here. Herne & Lucas (2006) added a femur (NMMNH 25079) from the same quarry to the holotype individual, which is also used to score the taxon in the analysis herein.

Supersaurus vivianae, BYU (various specimen numbers)*

Supersaurus vivianae is based on a scapulacoracoid (Jensen, 1985; Curtice, Stadtman & Curtice, 1996; Curtice & Stadtman, 2001; Lovelace, Hartman & Wahl, 2007). It was found at the Dry Mesa Quarry, intermingled with other large bones of diplodocid, brachiosaurid, and camarasaurid affinities (Jensen, 1985; Jensen, 1987; Jensen, 1988; Curtice & Stadtman, 2001). Jensen (1985) described three new taxa based on this material: Supersaurus vivianae, Dystylosaurus edwini, and Ultrasauros macintoshi. Subsequent study of the Dry Mesa specimens indicates that the holotypic dorsal vertebra of Dystylosaurus, as well as a dorsal vertebra referred to Ultrasauros by Jensen (1985) and Jensen (1987) probably belonged to the same individual as the holotypic scapulacoracoid of Supersaurus vivianae (Curtice & Stadtman, 2001). Lovelace, Hartman & Wahl (2007) revised this referral based on a new find from Wyoming, agreeing in large parts with Curtice & Stadtman (2001). The revised composition of the holotypic individual is listed in the Table 4. Since a preliminary analysis of the phylogenetic affinities of Dystylosaurus (see above) further corroborated this referral, a combined OTU was used for the final analysis.

Table 4 Anatomical overlap of the OTUs used in the phylogenetic analysis.

Taxa and specimens are ordered according to their latest higher-level taxon identification, and alphabetically within that taxon (see color code). Taxa marked with an asterisk are joined with more complete specimens (see text). Question marks mark dubious assignments.

Taxon	OTU	Specimen(s)	FS	Bc	LJ	T	aCV	mCV	pCV	CR	aDV	mDV	pDV	DR	SV	aCd	mCd	pCd	Ch	PcG	Fl	Ma	PvG	Hl	Pe	
Cetiosauriscus stewarti	–	NHMUK R3078																								
Dystrophaeus viaemalae	–	USNM 2364																								
Jobaria tiguidensis	–	–																								
Losillasaurus giganteus	type	MCNV Lo-1 to 26																								
Mamenchisaurus	–	–																								
Omeisaurus	–	–																								
Shunosaurus lii	−	-																								
Spinophorosaurus nigerensis	−	-																								
Turiasaurus riodevensis	–	–																								
Amphicoelias latus	–	AMNH 5765																								
Apatosaurus grandis	–	YPM 1901																								
Apatosaurus minimus	–	AMNH 675																								
Camarasaurus	–	–																								
Lourinhasaurus alenquerensis	lectotype	MIGM 2, 4931, 4956-57, 4970, 4975, 4979-80, 4983-84, 5780-81, 30370-88																								
Australodocus bohetii	type	MB.R.2454-55																								
Brachiosaurus altithorax	–	–																								
Brachiosaurus sp.	–	SMA 0009																								
Giraffatitan brancai	–	–																								
Isisaurus colberti	–	–																								
Ligabuesaurus leanzai	–	–																								
Haplocanthosaurus priscus	–	–																								
Cathartesaura anaerobica	–	–																								
Demandasaurus darwini	–	–																								
Limaysaurus tessonei	–	–																								
Nigersaurus taqueti	–	–																								
Zapalasaurus bonapartei	–	–																								
Amphicoelias altus	–	AMNH 5764																								
Amphicoelias altus	type ext	AMNH 5764																								
Amargasaurus cazaui	–	–																								
Brachytrachelopan mesai	–	–																								
Dicraeosaurus hansemanni	–	–																								
Suuwassea emilieae	–	ANS 21122																								
Dyslocosaurus polyonychius	–	AC 663																								
Apatosaurus ajax	–	AMNH 460																								
Apatosaurus ajax	–	NSMT-PV 20375																								
Apatosaurus ajax	–	YPM 1860	?	?																						
Apatosaurus laticollis	–	YPM 1861																								
Apatosaurus louisae	–	CM 3018	?	?		?																				
Apatosaurus louisae	–	CM 3378																								
Apatosaurus louisae*	–	CM 11162																								
Apatosaurus louisae	–	LACM 52844																								
Apatosaurus parvus	–	UW 15556																								
Apatosaurus sp.	–	BYU 1252-18531																								
Apatosaurus sp.	–	FMNH P25112																								
Apatosaurus sp.	–	ML 418																								
Atlantosaurus immanis	–	YPM 1840	?	?																						
Brontosaurus amplus	–	YPM 1981																								
Brontosaurus excelsus	–	YPM 1980																								
Elosaurus parvus	–	CM 566																								
Eobrontosaurus yahnahpin	–	Tate-001																								
Barosaurus affinis	–	YPM 419																								
Barosaurus lentus	–	YPM 429																								
Barosaurus sp.	–	AMNH 6341																								
Barosaurus sp.	–	AMNH 7530																								
Barosaurus sp.	AMNH 7535	AMNH 7535, 30070	?																							
Barosaurus sp.	–	CM 11984																								
Barosaurus sp.	–	SMA O25-8																								
Dinheirosaurus lourinhanensis	–	ML 414																								
Diplodocinae indet.	–	SMA 0087																								
Diplodocus carnegii	–	CM 84																								
Diplodocus carnegii	–	CM 94																								
Diplodocus cf. carnegii	–	WDC-FS001A																								
Diplodocus lacustris	–	YPM 1922																								
Diplodocus longus	–	YPM 1920																								
Diplodocus sp.	–	AMNH 223																								
Diplodocus sp.	–	AMNH 969																								
Diplodocus sp.	–	CM 3452																								
Diplodocus sp.	–	CM 11161																								
Diplodocus sp.	–	CM 11255																								
Diplodocus sp.	DMNS 1494	CM 21763; DMNS 1494																		?						
Diplodocus sp.	–	USNM 2672																								
Diplodocus sp.	–	USNM 2673																								
Diplodocus sp.	–	USNM 10865																								
Dystylosaurus edwini*	–	BYU 4503																								
Galeamopus hayi	–	HMNS 175																								
Galeamopus sp.	–	SMA 0011																								
Kaatedocus siberi	–	SMA 0004																								
Kaatedocus siberi	–	SMA D16-3																								
Leinkupal laticauda	–	MMCH-Pv 63-1																								
Seismosaurus hallorum	–	NMMNH 3690																								
Supersaurus vivianae*	holotype	BYU 12962																								
Supersaurus vivianae	BYU	BYU 4503, 4839, 9024-25, 9044-45, 9085, 10612, 12424, 12555, 12639, 12819, 12861, 12946, 12962, 13016, 13018, 13981, 16679, 17462																								
Supersaurus vivianae	–	WDC DMJ-021																								
Tornieria africana	holotype	MB.R.2672, 2713, 2728; SMNS 12140, 12141a, 12142, 12143, 12145a, c																								
Tornieria africana	skeleton k	MB.R.2386, 2572, 2586, 2669, 2673, 2726, 2730, 2733, 2913, 3816									lost															
Notes.

aCd anterior caudal vertebrae

aCV anterior cervical vertebrae

aDV anterior dorsal vertebrae

Bc braincase

Ch chevrons

CR cervical ribs

DR dorsal ribs

Fl forelimb

FS facial skull

Hl hindlimb

LJ lower jaw

Ma manus

mCd mid-caudal vertebrae

mCV mid-cervical vertebrae

mDV mid-dorsal vertebrae

pCd posterior caudal vertebrae

PcG pectoral girdle

pCV posterior cervical vertebrae

pDV posterior dorsal vertebrae

Pe Pes

PvG pelvic girdle

SV sacral vertebrae

T teeth

Color code:	Eusauropoda	Macronaria	Titanosauriformes		
	Diplodocoidea	Rebbachisauridae	Flagellicaudata	Dicraeosauridae	
	Diplodocidae	Apatosaurinae	Diplodocinae		

Supersaurus vivianae, WDC DMJ-021*

WDC DMJ-021 is a reasonably articulated skeleton and represents the most complete specimen of S. vivianae (Lovelace, Hartman & Wahl, 2007). It is not directly comparable with the holotype, because no scapulacoracoid was found. Nevertheless, based on the overlap with additional material attributed to the holotypic individual (see above; Lovelace, Hartman & Wahl, 2007), the identification of WDC DMJ-021 as S. vivianae has been widely accepted.

Suuwassea emilieae, ANS 21122*

Suuwassea was initially identified as a flagellicaudatan with uncertain affinities to Diplodocidae or Dicraeosauridae (Harris & Dodson, 2004). Further analyses suggest a closer relationship with the Dicraeosauridae (Salgado, Carvalho & Garrido, 2006; Whitlock & Harris, 2010; Whitlock, 2011a), which would mean that Suuwassea is the only North American representative of this taxon.

Tornieria africana, holotype (various specimen numbers)*

The holotype specimen of T. africana was found at the locality “A” at Tendaguru, Tanzania (Fraas, 1908; Remes, 2006). Tornieria was initially described as Gigantosaurus africanus (Fraas, 1908), but Sternfeld (1911) noted that this generic name was preoccupied, proposing the combination T. africana as a replacement. Janensch (1922) suggested synonymy of Tornieria and Barosaurus, resulting in the combination Barosaurus africanus, and later referred much more material from various quarries to the same species (Janensch, 1935; Janensch, 1961). However, in a reassessment of the entire material, which also resurrected the name Tornieria africana, only two or three individuals were positively identified as belonging to Tornieria (Remes, 2006). Remes (2006) furthermore identified additional material from the same quarry as most probably belonging to the same individual as the holotype. We therefore follow Remes (2006) by including all the Tornieria material found at locality “A” in the holotypic OTU (Table 4).

Tornieria africana, skeleton k*

A second specimen of T. africana comes from the “k” quarry at Tendaguru and was the only individual found at that site (Heinrich, 1999; Remes, 2006). Initially relatively complete with semi-articulated vertebral column and numerous appendicular elements, much of it has been lost or was destroyed during World War II (Remes, 2006). For these elements, descriptions and figures in Janensch (1929b) were used to complement the scoring.

Character list

The following character descriptions include references for their first recognition as taxonomically useful, their first use in a phylogenetic analysis including sauropod dinosaurs, and for their modified versions, in case these have been preferred over the original reference. References for previous use in sauropod phylogenies are abbreviated as follows: C05, Curry Rogers, 2005; C08, Canudo, Royo-Torres & Cuenca-Bescós, 2008; C12a, Carballido et al., 2012a; C12b, Carballido et al., 2012b; C95, Calvo & Salgado, 1995; D12, D’Emic, 2012; G03, González Riga, 2003; G05, Gallina & Apesteguía, 2005; G09, González Riga, Previtera & Pirrone, 2009; G86, Gauthier, 1986; L07, Lovelace, Hartman & Wahl, 2007; M12, Mannion et al., 2012; M13, Mannion et al., 2013; N12, Nair & Salisbury, 2012; R05, Rauhut et al., 2005; R09, Remes et al., 2009; R93, Russell & Zheng, 1993; S06, Sander et al., 2006; S07, Sereno et al., 2007; S97, Salgado, Coria & Calvo, 1997; T13, Tschopp & Mateus, 2013b; U04a, Upchurch, Barrett & Dodson, 2004; U04b, Upchurch, Tomida & Barrett, 2004; U07, Upchurch, Barrett & Galton, 2007; U95, Upchurch, 1995; U98, Upchurch, 1998; W02, Wilson, 2002; W11, Whitlock, 2011a; W98, Wilson & Sereno, 1998; Y93, Yu, 1993; Z11, Zaher et al., 2011. Original character numbers are added after a hyphen after the reference number, where provided in the reference.

Skull

C1: Premaxillary anterior margin, shape: without step (0); with marked but short step (1); with marked and long step (2) (U98-10; W98-19; modified by C12b-2; Fig. 1). Ordered.

Comments. The character describes the presence and development of a horizontal portion of the premaxilla, which lies anterior to the nasal process. The step, when present, is best visible in lateral view. It was initially proposed by Upchurch (1998), who scored the Diplodocoidea as unknown or inapplicable, due to a supposed absence of the nasal process. However, some diplodocoids, (e.g., Suuwassea) clearly show a distinction between the anterior main body and the posterior nasal process in dorsal view, where they show an abrupt narrowing (Harris, 2006a; ANS 21122, E Tschopp, pers. obs., 2011). Diplodocoidea should therefore be scored as ‘0.’ A third state was added in order to distinguish Brachiosauridae from other macronarian sauropods (Carballido et al., 2012b). The character is treated as ordered, due to the gradational change in morphology.

C2: Premaxilla, external surface: without anteroventrally orientated vascular grooves originating from an opening in the maxillary contact (0); vascular grooves present (1) (Wilson, 2002; S07-3; Fig. 2).

Figure 2 Anterior portions of sauropod premaxillae.

Anterior portions of premaxillae of Camarasaurus (A; modified from Madsen, McIntosh & Berman, 1995) and Galeamopus sp. SMA 0011 (B) in anterodorsal view, illustrating the states of characters 2 and 3. Not to scale.

Comments. The presence of these grooves was previously found as a synapomorphy of Dicraeosauridae (Whitlock, 2011a; Mannion et al., 2012). However, faint grooves originating at the premaxillary-maxillary contact are also visible in Nigersaurus (Sereno et al., 2007) and in some diplodocid specimens. In the latter, they fade shortly anterior to the suture (e.g., in CM 11161, 11162, SMA 0011, USNM 2672). In the present analysis, all of these specimens are scored as apomorphic.

C3: Premaxilla, shape in dorsal view: main body massive, with proportionally short ascending process distinct (0); single elongate unit, distinction between body and process nearly absent (1) (U98-12; wording modified; Fig. 2).

Comments. Upchurch (1998) formulated this character differently, based on his interpretation that the ascending process of the premaxilla was absent in Diplodocoidea. As stated above, this is not the case. The wording of the derived state was thus changed accordingly.

C4: Premaxilla, angle between lateral and medial margins of premaxilla as seen in dorsal view: >40° (0); 17° −40° (1); <17° (2) (Upchurch, 1999; modified; Table S2). Unordered.

Comments. Upchurch (1999) was the first to note significant differences in these angles between diplodocoids (around 10°), nemegtosaurids (18°), and remaining taxa (e.g., Giraffatitan, 30°; Upchurch, 1999: Fig. 7). He used this character (with two states) as one of several that supported the inclusion of Nemegtosauridae within Diplodocoidea (Upchurch, 1999), a view now falsified by nearly complete finds of new nemegtosaurids that show them to be deeply nested within titanosaurians, but with convergences with Diplodocoidea (Wilson, 2002; Curry Rogers, 2005; Zaher et al., 2011). The OTUs included in this dataset were rescored for this character based on figures or on original material. Because the lateral margin is concave to sinuous in most taxa, a straight line was drawn from the anterior-most point of the premaxillary-maxillary contact to the point where the lateral edge curves medially, at the base of the ascending process. The results (Table S2) indicate that the distribution of the character scores is not as straightforward as previously thought: Shunosaurus, as well as some specimens of Camarasaurus appear to show similarly narrow angles as Dicraeosaurus and Suuwassea. A third state was thus added, such that diplodocid and rebbachisaurid OTUs now score in the narrow-most range, and Mamenchisaurus and Jobaria are classed as significantly wide-angled taxa. Because the derived state is ambiguous, the character is most parsimoniously left unordered.

C5: Premaxilla, posteroventral edge of ascending process in lateral view: concave (0); straight and dorsally oriented (1); straight, and directed posterodorsally (2) (W11-3; wording modified; Fig. 1). Unordered.

Comments. Whitlock (2011a: p.35) described the character as follows: ‘Ascending process of the premaxilla, shape in lateral view: convex (0); concave, with a large dorsal projection (0); sub-rectilinear and directed posterodorsally (1).’ This formulation is misleading, and the states overlap with those of character 1, which describes the premaxillary ‘step.’ Varying morphologies of the ascending process, following the states of Whitlock (2011a), were observed among the included taxa regarding the posteroventral edge of the ascending process—the margin that delimits the nasal opening anteriorly. The description of the character was adapted, reducing the character to only encompass the orientation of the posteroventral edge, thereby avoiding overlap with character 1. The directional terms in the states are meant in relation to a horizontally oriented ventral edge of the maxilla. Because no state is obviously intermediate relative to the other two, the character is left unordered.

C6: Premaxilla, posterolateral process and the lateral process of the maxillary, shape: without midline contact (0); with midline contact forming a marked narial depression, subnarial foramen not visible laterally (1) (W02-1; Fig. 3).

Figure 3 Sauropod skulls.

Skulls (A, C–E) or maxilla (B) of Camarasaurus sp. SMA 0002 (A) Dicraeosaurus hansemanni MB.R.2336 (B) Kaatedocus siberi SMA 0004 (C) Galeamopus sp. SMA 0011 (D) and Diplodocus sp. CM 11161 (E) in anterolateral view, illustrating the states of the characters 6, 9, 10, 11, 12, 16, 17, 48. Not to scale.

Comments. Whitlock (2011a) reversed the polarity of this character, due to a more limited outgroup sampling. With the inclusion of Shunosaurus (Mannion et al., 2012), the most basal OTU again lacks the midline contact, as is the case in Diplodocoidea. The original phrasing of Wilson (2002) is therefore preferred.

C7: Premaxilla, dorsoventral depth of anterior portion: remains the same as posteriorly, or widens gradually (0); widens considerably, and abruptly (1) (Harris, 2006a; Fig. 4).

Figure 4 Sauropod premaxillae.

Premaxillae of Suuwassea emilieae ANS 21122 (A) Dicraeosaurus hansemanni MB.R.2337 (B) and Galeamopus sp. USNM 2673 (C, left element reversed) in lateral view, illustrating the states of character 7. Not to scale.

Comments. Harris (2006a) stated this difference as useful to distinguish Suuwassea (which retains the same depth) from Diplodocus (which widens). A similar, narrow premaxilla is furthermore present in Kaatedocus (Tschopp & Mateus, 2013b). The character is difficult to observe in articulated skulls, but single elements do show a significant difference.

C8: Subnarial foramen and anterior maxillary foramen, position: well distanced from one another (0); separated by narrow bony isthmus (1) (W02-5; Fig. 5).

Figure 5 Sauropod skulls.

Skulls of Camarasaurus (A; modified from Wilson & Sereno, 1998), Limaysaurus tessonei MUCPv-205 (B; photo by J Whitlock), Dicraeosaurus hansemanni MB.R.2379 (C) Kaatedocus siberi SMA 0004 (D) and Diplodocus sp. CM 11161 (E) in dorsal view, illustrating the states of the characters 8, 26, 29, 30, 34, 35, 36, 66. Not to scale.

C9: Maxilla, large foramen anterior to the preantorbital fossa, separated by a narrow bony bridge: absent (0); present (1) (Z11-244; wording modified; Fig. 3).

Comments. Generally, sauropod maxillae are pierced by a number of small foramina anteriorly, probably for innervation and/or blood supply of the replacement teeth. The foramen described by Zaher et al. (2011) in Tapuiasaurus, however, is relatively large, and closely attached to the preantorbital fossa. The same is the case in Dicraeosaurus hansemanni MB.R.2336 (Janensch, 1935), but not in diplodocids.

C10: Maxilla, large foramen posterior to anterior maxillary foramen, dorsal to preantorbital fossa: absent (0); present (1) (New; Fig. 3).

Comments. Few diplodocid specimens show a large foramen posterior to the anterior maxillary foramen (e.g., Kaatedocus SMA 0004). This foramen cannot be the same as the one described in character 9, given that both are present in Dicraeosaurus.

C11: Anterior maxillary foramen, location: detached from maxillary-premaxillary boundary, facing dorsally (0); lies on medial edge of maxilla, opening medially into the premaxillary-maxillary boundary (1) (New; Fig. 3).

Comments. Usually, diplodocids have the subnarial and the anterior maxillary foramina enclosed within a single, elongated fossa at the maxillary-premaxillary boundary (Wilson & Sereno, 1998; Whitlock, 2011b). However, in Kaatedocus, the anterior maxillary foramen is detached and laterally positioned, within a unique, small fossa. It thus resembles the plesiomorphic state present in Jobaria or Camarasaurus (Wilson & Sereno, 1998; Sereno et al., 1999), although it is still much closer to the subnarial foramen. Primitive outgroup taxa (those normally basal to Jobaria) were coded as unknown, as it is unclear if the intermaxillary foramen that is present in these taxa (e.g., He, Li & Cai, 1988; Ouyang & Ye, 2002) is homologous to the anterior maxillary foramen or the subnarial foramen.

C12: Maxilla, canal connecting the antorbital fenestra and the preantorbital fossa: absent (0); present (1) (New; Fig. 3).

Comments. Such a canal is only present in SMA 0011 and USNM 2673. Taxa without a preantorbital fossa were scored as unknown in order to avoid absence coding.

C13: Maxilla, dorsal process, posterior extent: anterior to or even with posterior process (0); extending posterior to posterior process (1) (W11-9; Fig. 1).

Comments. The character is applied to skulls in lateral view, with the ventral edge of the maxilla oriented horizontally.

C14: Maxilla-quadratojugal contact: absent or small (0); broad (1) (Y93-14; Fig. 1).

Comments. Upchurch (1998) reported some difficulties in scoring some taxa for his version of this character, which was defined as a simple absence-presence feature. Reduced, small contacts are present in Camarasaurus, but only diplodocids are known to have developed a broad area where the maxilla contacts the quadratojugal (Upchurch, 1998; Wilson & Sereno, 1998). Therefore, Whitlock (2011a) redefined the states, such that the apomorphic state now describes a synapomorphy of at least Diplodocidae (it is unknown in Dicraeosauridae and Rebbachisauridae). The derived state appears to be a convergence in some nemegtosaurids (Upchurch, 1998; Wilson, 2005).

C15: Preantorbital fossa: absent (0); present (1) (T13-10; Fig. 1).

Comments. Although some flagellicaudatan taxa have reduced to entirely closed preantorbital fenestrae, all show a distinct fossa, which is otherwise only present in some nemegtosaurids (Wilson, 2005).

C16: Preantorbital fossa, if present: with relatively indistinct borders (0); dorsally capped by a thin, distinct crest (1) (Wilson, 2002; W11-12; modified; Fig. 3).

Comments. Wilson (2002) originally proposed that the presence of a dorsally capped preantorbital fenestra is an autapomorphy of Diplodocus. A broader survey of this character shows that within Flagellicaudata, the absence of this dorsal crest is instead only known from a single Apatosaurus skull (CM 11162), and thus might represent an autapomorphy of Apatosaurus louisae.

C17: Preantorbital fenestra: reduced to absent (0); present, occupying at least 50% of the preantorbital fossa (1) (Berman & McIntosh, 1978; Y93-21; modified; Fig. 3).

Comments. Yu (1993) was the first to use this feature in a phylogenetic analysis. Tschopp & Mateus (2013b) modified the character, and included the dorsal crest as well. However, because these two features are not correlated (Kaatedocus has a dorsal crest but a reduced to absent fenestra), the states were adjusted, and a ratio is given to distinguish the small opening in Dicraeosaurus from the large ones in Diplodocus, for example. Large preantorbital fenestrae are convergently present in nemegtosaurids (Wilson, 2005; Zaher et al., 2011).

C18: Antorbital fenestra, maximum diameter: much shorter (<90%) than orbital maximum diameter (0); subequal (>90%) to orbital maximum diameter (1) (Y93-7; modified; Table S3).

Comments. Yu (1993) proposed the character without any clear state boundaries, which were later added by Whitlock (2011a), and changed herein from 85% to 90% in order to include Mamenchisaurus within the plesiomorphic state.

C19: Antorbital fenestra, anterior extension: is restricted posterior to preantorbital fossa (0); reaches dorsal to preantorbital fossa (1) (New; Fig. 1).

Comments. The character has to be scored with the ventral border of the maxilla oriented horizontally. Within flagellicaudatans, the derived state is most developed in Kaatedocus SMA 0004, but nemegtosaurids like Rapetosaurus have extremely elongated antorbital fenestrae that even reach anterior to the entire preantorbital fossa (Curry Rogers & Forster, 2004).

C20: Antorbital fenestra, shape of dorsal margin: straight or convex (0); concave (1) (W11-14; Fig. 1).

Comments. The diplodocine skull AMNH 969 appears to have a convex dorsal margin at first glance. However, the presence of a lateral projection in the upper half of this edge indicates that the convex shape might be due to deformation. The lateral projection in AMNH 969 is at the same location, and has the same shape as the osteological feature producing the concave dorsal edge of the antorbital fenestra in CM 11161. AMNH 969 is thus interpreted to possess the derived state, as in all flagellicaudatans.

C21: External nares, position: retracted to level of orbit, facing laterally (0); retracted to position between orbits, facing dorsally or dorsolaterally (1) (McIntosh, 1989; U95; modified by W11-15; Fig. 1).

Comments. Upchurch (1995) was the first to include this character in a phylogenetic analysis, based on observations made by McIntosh (1989). Whitlock (2011a) adjusted the state description, since the reduced taxon sampling made a third state redundant (anterior to orbit, the plesiomorphic state in Sauropoda; Upchurch, 1995).

C22: External nares, maximum diameter: shorter than orbital maximum diameter (0); longer than orbital maximum diameter (1) (U95; modified by W98-89).

Comments. Upchurch (1995) initially defined the character states in relation to skull length, but later, Wilson & Sereno (1998) changed them to relate to orbital diameter. The latter has since been widely used and is thus retained here.

C23: Prefrontal, medial margin, shape: without distinct anteromedial projection (0); curving distinctly medially anteriorly to embrace the anterolateral corner of the frontal (1) (New; Fig. 6).

Figure 6 Skull roof of Diplodocus sp. CM 11161 (A; based on Wilson & Sereno, 1998) and Limaysaurus tessonei MUCPv-205 (B; based on Calvo & Salgado, 1995) in dorsal view.

Note the anteromedial hook in the prefrontal of CM 11161 (A; C23-1), and the differently shaped frontal-nasal suture (straight to anteriorly bowed in A, C28-0; bowed posteriorly in B, C28-1). Abb.: f, frontal; na, nasal; pf, prefrontal. Scaled to the same skull roof length.

Comments. In some basal sauropods, the prefrontal is located entirely anterior to the frontal. These cases are scored as plesiomorphic.

C24: Prefrontal, posterior process size: small, not projecting far posterior of frontal-nasal suture (0); elongate, approaching parietal (1) (W02-14; Fig. 7).

Figure 7 Left (F, H–K) and right (A–E, G) diplodocoid frontals in dorsal view, anterior to the top.

(A–E) shows elements with an anteriorly restricted posterior process of the prefrontal (C24-0), (F–K) have elongated posterior processes (C24-1). Additional states are illustrated from the characters 25, 31, 33. Frontals figured in strict perpendicular view, and scaled to the same anteroposterior length.

Comments. This character is not as straight forward as it seems. Care has to be taken that one observes the frontal and prefrontal in exactly perpendicular view. In some reconstructed dorsal views of the skull of Diplodocus (Wilson & Sereno, 1998; Whitlock, 2011b), the posterior extension of the prefrontal is remarkable, but this is due to the view in which the reconstruction is drawn. The frontal slants posteriorly, and more posterior distances therefore appear shorter. In direct dorsal view, differences in distance between taxa diminish. However, the character remains informative: in diplodocids like Apatosaurus or Diplodocus, the posterior process of the prefrontal almost reaches or surpasses the midlength of the frontal, whereas in Rebbachisauridae or in Kaatedocus and Tornieria, it remains restricted to about the anterior third (Fig. 7).

C25: Prefrontal, posterior process shape: straight (0); hooked (1) (W02-15; modified; Fig. 7).

Comments. As the posterior elongation of the prefrontal, this character was initially defined in a somewhat ambiguous way (flat/hooked). Nigersaurus does have a posteriorly facing, pointed prefrontal. The description ‘flat’ therefore does not fit very well, and it is replaced by ‘straight.’ Hooked is herein interpreted to describe a medially curving posterior process, such that its posterior end forms the medial-most extension of the prefrontal.

C26: Frontals, midline contact (symphysis): patent suture (0); fused in adult individuals (1) (Salgado & Calvo, 1992; Y93-33; Fig. 5).

Comments. Fusion of skull bones is usually considered an ontogenetic feature (Varricchio, 1997; Whitlock & Harris, 2010). However, the ontogenetic stages when fusion begins might still be different between taxa and thus phylogenetically significant. This appears to be the case here, where the braincases of Dicraeosaurus and Amargasaurus have completely obliterated sutures between the frontals, whereas large-sized diplodocid skulls do not (e.g., CM 11161). Nonetheless, it remains possible that non-dicraeosaurid sauropods fuse their frontals at an old age. In future, it might be helpful to constrict the character to a specific age-range (possibly subadult or early adult), but to date, the exact individual age of the specimens showing the fused frontals remains unknown.

C27: Frontal, anteroposterior length: long, >1.4 times minimum transverse width (0); short, 1.4 or less times minimum transverse width (1) (G86; modified; Table S4).

Comments. This character was widely used in phylogenetic analyses of sauropod dinosaurs (Upchurch, 1998; Wilson, 2002; Whitlock, 2011a; Mannion et al., 2012; Tschopp & Mateus, 2013b), with varying definitions of the state boundaries. In addition, it was often unclear if minimum or maximum transverse width was intended (e.g., Whitlock, 2011a; Tschopp & Mateus, 2013b). As shown in Table S4, there are significant differences in the ratios, with more distinct changes when comparing frontal length and minimum transverse width. Therefore, state boundaries were herein defined numerically, which also led to some differential scorings compared to Tschopp & Mateus (2013b). Kaatedocus, for example, is now well within the ratios for the apomorphic state.

C28: Frontal-nasal suture, shape: flat or slightly bowed anteriorly (0); v-shaped, pointing posteriorly (1) (W11-21; Fig. 6).

Comments. The frontals of ‘Diplodocus’ hayi might have a posteriorly pointing nasal contact as well (Holland, 1906). However, the nasals are not preserved in this specimen, and it seems thus more appropriate to score HMNS 175 as unknown.

C29: Frontals, distinct anterior notch medially between the two elements: absent (0); present (1) (T13-25; modified; Fig. 5).

Comments. The shape description of the notch (Tschopp & Mateus, 2013b) was excluded from the character in order to include also Spinophorosaurus, and SMA 0011 in the apomorphic state. The frontal usually becomes extremely thin in this part, and it is thus easily broken. Because the notch still appears genuine in these three taxa/specimens, the character was retained. Tschopp & Mateus (2013b) mentioned this feature as an autapomorphy of Kaatedocus. Given that a similar notch is present in SMA 0011, this character might actually be more widespread within Diplodocidae. In fact, many specimens (e.g., Apatosaurus CM 11162) show broken anteromedial edges in the frontal, which makes it difficult to evaluate this character. New finds of diplodocid frontals might shed some more light on the distribution of this character.

C30: Frontals, dorsal surface: without paired grooves facing anterodorsally (0); grooves present, extend on to nasal (1) (W11-22; Fig. 5).

Comments. Grooves appear to be present on the frontals of the dicraeosaurid Amargasaurus cazaui (Salgado & Calvo, 1992: Fig. 2B), but these extend onto the prefrontals and not the nasals and do not extend as far posteriorly as in Limaysaurus. Amargasaurus is thus scored as plesiomorphic, following Whitlock (2011a).

C31: Frontal, lateral edge in dorsal view: relatively straight (0); deeply concave (1) (New; Fig. 7).

Comments. This character has a somewhat ambiguous distribution. There is some difference in the shapes taken together in the plesiomorphic state as well: Rebbachisauridae, in contrast with most other taxa, have a weakly convex lateral frontal edge. Diplodocids exhibit varying shapes: Apatosaurus and Diplodocus have concave edges, whereas Kaatedocus or Tornieria have straight margins.

C32: Frontal, contribution to dorsal margin of orbit: less than 1.5 times the contribution of prefrontal (0); at least 1.5 times the contribution of prefrontal (1) (W11-23; modified by M12-20; Table S5).

Comments. The lengths of the frontal and prefrontal are measured in a straight line in lateral view, from the mid-point of the frontal-prefrontal articulation to the anterior-most (prefrontal) or posterior-most (frontal) point. Whitlock (2011a) proposed the character, leaving a gap between plesiomorphic and apomorphic states (subequal, or twice), which was changed by Mannion et al. (2012). A comparative analysis of the included specimens confirms the utility of the boundary proposed by Mannion et al. (2012).

C33: Frontal, free lateral margin: rugose (0); smooth (1) (T13-23; Fig. 7).

Comments. Rugosities are present around the dorsal margin of almost all sauropods, but in some cases, they are shifted onto the prefrontal or the postorbital. Tschopp & Mateus (2013b) hypothesized that the rugosities served for an attachment of a palpebral element.

C34: Frontal, contribution to margin of supratemporal fenestra/fossa: present (0); absent, frontal excluded from anterior margin of fenestra/fossa (1) (W98-65; Fig. 5).

Comments. In the derived state, the frontal is excluded from a contribution to the margin of the supratemporal fenestra by a contact between the medial process of the postorbital and the anterolateral process of the parietal.

C35: Frontal-parietal suture, position of medial portion: closer to anterior extension of supratemporal fenestra (0); closer to posterior extension (1) (T13-26; modified; Fig. 5).

Comments. Tschopp & Mateus (2013b) formulated the character inspired by Remes (2006), who mentioned the position of the fronto-parietal suture as a feature to distinguish Tornieria from Diplodocus. They used a tripartite character, with an intermediate state as closer to the central portion of the supratemporal fenestra (Tschopp & Mateus, 2013b). The position of the suture is difficult to assess in some diplodocid specimens, because it describes a strongly sinuous curve (e.g., CM 11161, Fig. 7). The character is thus restricted to the medial portion of the suture herein. By doing so, it becomes clear that the majority of Diplodocus skulls shifted the suture backwards, whereas all other specimens have it anteriorly located. The posterior dislocation might thus prove to be an autapomorphy of Diplodocus. The intermediate state becomes redundant, and is not included here.

C36: Pineal (parietal) foramen between frontals and parietals: present (0); absent (1) (Y93-27; modified; Fig. 5).

Comments. This character was proposed in combination with the presence of a postparietal foramen (Yu, 1993). The two are herein separated in two characters, because Kaatedocus SMA 0004 has a postparietal but no pineal foramen (Tschopp & Mateus, 2013b). The presence of a pineal foramen is often difficult to assess due to breakage of the area around the fronto-parietal suture (McIntosh, 1990b; Upchurch, Barrett & Dodson, 2004; Harris, 2006a). However, in some specimens, the presence or absence of this feature is genuine, and it thus appears appropriate to include this character. Specimens where the presence of the foramen has been doubted previously are scored as unknown. At the current state of knowledge, the presence seems to be a retained plesiomorphy characterizing the Dicraeosauridae, but in many diplodocid specimens its presence cannot be dismissed yet.

C37: Orbit, anterior-most point: anterior to the anterior extremity of lateral temporal fenestra (0); roughly even with or posterior to anterior extent of lateral temporal fenestra (1) (G86; U95; modified by W11-25; Fig. 1).

Comments. The original character was a multistate character (Upchurch, 1995). Given the limited taxon sampling of Whitlock (2011a) and the herein presented analysis, the third state becomes redundant (infratemporal fenestra restricted posterior to orbit).

C38: Orbital ventral margin, anteroposterior length: broad, with subcircular orbital margin (0); reduced, with acute orbital margin (1) (W98-25; Fig. 1).

Comments. The derived state results in a teardrop-shape of the orbit. With the ventral margin of the maxilla held horizontally, the ‘ventral margin’ would be better described with ‘anteroventral corner.’

C39: Postorbital, posterior process: present (0); absent (1) (W02-17; Fig. 1).

Comments. The postorbital is usually a triradiate bone, with a relatively short posterior process that overlaps the squamosal. The latter is absent in rebbachisaurids (Wilson, 2002; Whitlock, 2011a).

C40: Jugal, contribution to antorbital fenestra: very reduced or absent (0); large, bordering approximately one-third of its perimeter (1) (Berman & McIntosh, 1978; U95; modified by W11-28; Fig. 8).

Figure 8 Left jugal of Diplodocus USNM 2672 in lateral view.

Note the large contribution of the jugal to the antorbital fenestra (C40-1), the narrow and elongate posteroventral process (C42-1), the dorsal process of the jugal (C43-0), and the anterior spur (C44-1). Abb.: aof, antorbital fenestra; j, jugal; la, lacrimal; ltf, laterotemporal fenestra; m, maxilla; o, orbit; po, postorbital; qj, quadratojugal.

Comments. Recognized as distinctive feature of Diplodocoidea by Berman & McIntosh (1978), the contribution of the jugal to the antorbital fenestra was first used as phylogenetic character by Upchurch (1995). Whitlock (2011a) defined the state boundaries quantitatively.

C41: Jugal, contact with ectopterygoid: present (0); absent (1) (U95; Fig. 9).

Figure 9 Eusauropod skulls.

Skulls of Shunosaurus lii ZDM 65430 (A; modified from Chatterjee & Zheng, 2002) and Diplodocus sp. CM 11161 (B) in ventral view. Note the anteriorly displaced position of the ectopterygoid ramus of the pterygoid, and the ectopterygoid itself, in Diplodocus (B; C41-1 and C102-1), as well as the vomer that articulates with the premaxilla in Shunosaurus (A; C103-0), but with the maxilla in Diplodocus (B; C103-1). Abb.: aof, antorbital fenestra; bo, basioccipital; bpr, basipterygoid process; bt, basal tuber; ep, ectopterygoid; er, ectopterygoid ramus; j, jugal; m, maxilla; pa, palate; pm, premaxilla; popr, paroccipital process; pt, pterygoid; qj, quadratojugal; v, vomer. Pictures scaled to the same skull length.

Comments. The development of this character is barely known in sauropods. When preserved, the osteology of the palatal complex is often left obscured by matrix for stability of the specimen. At the current state of knowledge, the ectopterygoid becomes anteriorly dislocated in Neosauropoda, and contacts the maxilla instead of the jugal. Future CT scanning of additional skulls will yield more detailed results.

C42: Jugal, posteroventral process: short and broad (0); narrow and elongate (1) (New; Fig. 8).

Comments. This character shows varying shapes in the skulls traditionally identified as Diplodocus (CM 11161 has a short process, whereas in all other skulls they are elongated). However, too few diplodocid jugals are preserved entirely in order to evaluate the distribution of this character to date.

C43: Jugal, dorsal process: present (0); absent (1) (Y93-24; polarity inverted; Fig. 8).

Comments. Yu (1993) proposed the dorsal process as a synapomorphy for Diplodocidae. However, no jugal is known from dicraeosaurids, and such a process is also present in Shunosaurus, Omeisaurus, and Mamenchisaurus (Janensch, 1935; He, Li & Cai, 1988; Salgado & Calvo, 1992; Chatterjee & Zheng, 2002; Ouyang & Ye, 2002). Because the latter basal taxa show dorsal processes of the jugal, the character polarity was inverted relative to the original version (Yu, 1993). Although they are scored for the plesiomorphic state, Diplodocidae is still distinguishable from Shunosaurus and the other taxa by the strong development of the dorsal process, and its anterior displacement. In Omeisaurus, e.g., the dorsal process is short and located at midlength of the jugal-lacrimal suture (He, Li & Cai, 1988).

C44: Jugal, anterior spur dorsally, which projects into antorbital fenestra: absent (0); present (1) (New; Fig. 8).

Comments. Such a spur is present in many diplodocid specimens, although in USNM 2672, it only occurs on the left side (E Tschopp, pers. obs., 2011). However, the possibility to develop such a spur still appears to be restricted to Diplodocidae, and the character is thus used in the analysis. USNM 2672 is scored as ‘present.’

C45: Quadratojugal, position of anterior terminus: anterior margin of orbit or posteriorly restricted (0); beyond anterior margin of orbit (1) (W11-30; modified; Fig. 1).

Comments. The character is coded with the ventral margin of the maxilla held horizontally. State boundaries by Whitlock (2011a: posterior to middle of orbit, anterior margin or beyond) were adjusted because all diplodocoids show strongly elongated anterior processes that end significantly anterior to the orbit. On the other hand, in Mamenchisaurus or Giraffatitan, the processes reach the anterior margin of the orbit (Janensch, 1935; Ouyang & Ye, 2002), which would require a scoring as apomorphic when following the description of Whitlock (2011a).

C46: Quadratojugal, angle between anterior and dorsal processes: less than or equal to 90°, so that the quadrate shaft is directed dorsally (0); greater than 90°, approaching 130°, so that the quadrate shaft slants posterodorsally (1) (G86; U95; Fig. 1).

Comments. The angle between the quadratojugal processes reaches its maximum in the large skulls CM 11161 and 11162. In smaller skulls (of both ontogenetically younger as well as phylogenetically more basal specimens), the angle is of approximately 110°(e.g., Kaatedocus SMA 0004; Tschopp & Mateus, 2013b), but still clearly in the derived state.

C47: Lacrimal, anterior process: absent (0); present (1) (W02-11; polarity reversed by M13-80; Fig. 1).

Comments. Wilson (2002) initially proposed the character with inverted polarity. This was changed by Mannion et al. (2013), and herein in order to have the chosen outgroups showing the plesiomorphic state. An anterior process is usually interpreted to be absent in diplodocoids. However, SMA 0011 and Dicraeosaurus do have one. On the other hand, it is possible that the feature is more widespread among Diplodocoidea, but that the anterior process is obscured by the posterodorsal process of the maxilla. The latter partly overlaps the anterior process of the lacrimal in SMA 0011. The presence of an anterior process of the lacrimal would otherwise be one of the distinguishing characteristics between diplodocoids and nemegtosaurids (Wilson, 2005).

C48: Lacrimal, dorsal portion of lateral edge: flat (0); bears dorsoventrally elongate, shallow ridge (1); bears a dorsoventrally short laterally projecting spur (2) (T13-34; Fig. 3). Ordered.

Comments. There is some evidence that this character is ontogenetically controlled (Tschopp & Mateus, 2013b): only small skulls show the laterally projecting spur. The character is retained here in order to test its validity. The character is treated as ordered due to intermediate morphologies.

C49: Quadrate, articular surface shape: quadrangular in ventral view, orientated transversely (0); roughly triangular in shape (1); thin, crescent-shaped surface with anteriorly directed medial process (2) (W11-32; Fig. 10). Ordered.

Figure 10 Articular surfaces of neosauropod quadrates.

Quadrate articular surface shapes of Camarasaurus sp. SMA 0002 (A, quadrangular, C49-0), Suuwassea emilieae ANS 21122 (B, roughly triangular, C49-1), and Nigersaurus taqueti MNN GAD512-7 (C, crescent-shaped, C49-2). Figures of Suuwassea and Nigersaurus traced from Harris (2006a) and Sereno et al. (2007), respectively.

C50: Quadrate, short transverse ridge medially on posterior side of ventral ramus, close to the articular surface with the lower jaw: absent (0); present (1) (New; Fig. 11).

Figure 11 Neosauropod quadrates.

Quadrates of Camarasaurus sp. SMA 0002 (A) and Diplodocidae indet. SMA D27-7 (B) in posterior view, illustrating the transverse ridge (B, inlet; C50-1), and the deep (A; C51-0) versus shallow (B; C51-1) quadrate fossa. Not to scale.

Comments. This ridge is a detail which appears to be synapomorphic for Diplodocidae. Most of the diplodocid quadrates could not be studied first hand for this character. Therefore a more detailed evaluation of this character has to be undertaken in order to corroborate the presence or absence of such a ridge, and its taxonomic utility.

C51: Quadrate fossa, depth: shallow (0); deeply invaginated (1) (R93-2; Fig. 11).

C52: Quadrate, shallow, second fossa medial to pterygoid flange on quadrate shaft (not the quadrate fossa): absent (0); present, becoming deeper towards its anterior end (1) (T13-37; wording modified; Fig. 12).

Figure 12 Neosauropod quadrates.

Quadrates of Camarasaurus sp. SMA 0002 (A) and Diplodocidae indet. SMA D27-7 (B) in medial view, illustrating the second medial fossa (B; C52-1), the shape of the dorsal margin (C53, concave versus convex), and the stocky versus slender posterior ramus (C54). Scaled to the same height.

Comments. The medial surface of the pterygoid flange is nearly always concave, but concave dorsoventrally. In SMA 0004, as well as some other diplodocid specimens, the second fossa is transversely concave, lies anteriorly on the posterior shaft, medial to where the pterygoid flange originates. There is a chance that the character might be ontogenetic, given that no large-sized skull has yet been identified to bear this second fossa. The character was slightly reworded from its original version (Tschopp & Mateus, 2013b) in order to describe the location of the fossa better.

C53: Quadrate, dorsal margin: concave, such that pterygoid flange is distinct from quadrate shaft (0); straight, without clear distinction of posterior extension of pterygoid flange (1) (New; Fig. 12).

C54: Quadrate, posterior end (posterior to posterior-most extension of pterygoid ramus): short and robust (0); elongate and slender (1) (New; Fig. 12).

C55: Squamosal, anterior extent: restricted to postorbital region (0); extends well past posterior margin of orbit (1); extends beyond anterior margin of orbit (2) (W11-35; Fig. 1). Ordered.

Comments. The anterior extent of the squamosal is measured with the ventral border of the maxilla oriented horizontally.

C56: Squamosal-quadratojugal contact: present (0); absent (1) (U95; Fig. 13).

Figure 13 Temporal region in eusauropod skulls.

Squamosal and adjacent bones in Mamenchisaurus youngi (A; traced from Ouyang & Ye, 2002), Camarasaurus lentus CM 11338 (B; traced from Madsen, McIntosh & Berman, 1995), Amargasaurus cazaui MACN-N15 (C; traced from Salgado & Bonaparte, 1991), and Diplodocinae indet. CM 3452 (D; traced from a 3D model from L Witmer), in right (A, C) and left (B, D) lateral view; illustrating the states of the characters 56, 57, and 58. Abb.: po, postorbital; q, quadrate; qj, quadratojugal; sq, squamosal. Not to scale.

Comments. In diplodocids, where no contact is present, the distance between the squamosal and the quadratojugal varies (Whitlock, Wilson & Lamanna, 2010; Whitlock & Lamanna, 2012). However, most of the diplodocid specimens do not preserve the entire anterior ramus of the squamosal (E Tschopp, pers. obs., 2011) and it seems thus premature to include the distance as a phylogenetic character.

C57: Squamosal, posteroventral margin: smooth, or with short and blunt ventral projection (0); with prominent, ventrally directed ‘prong’ (1) (W11-37; modified; Fig. 13).

Comments. The original character description of Whitlock (2011a) was modified, and an additional binary character was added (see below) in order to describe better the state in Kaatedocus, where a short ventral projection of the squamosal is present.

C58: Squamosal, posteroventral margin: smooth, without ventral projection (0); ventral projection present (1) (W11-37; modified; Fig. 13).

Comments. A short projection is present in almost all preserved flagellicaudatan skulls. In contrast, most non-flagellicaudatan sauropods have smooth posteroventral margins of the squamosal.

C59: Parietal, contribution to posttemporal fenestra: present (0); absent (1) (W02-22; Fig. 14).

Figure 14 Sauropod skulls in posterior view.

Sauropod skulls of Spinophorosaurus nigerensis GCP-CV-4229 (A; traced from Knoll et al., 2012); Suuwassea emilieae ANS 21122 (B; traced from Harris, 2006a); Limaysaurus tessonei MUCPv-205 (C; after Calvo & Salgado, 1995); Kaatedocus siberi SMA 0004 (D); Apatosaurus louisae CM 11162, (E, reversed); Diplodocus sp. CM 11161 (F) in posterior view. Note the participation (C; C59-0) or exclusion (D; C59-1) of the parietal to the posttemporal fenestra; the straight (A; C62-0) or convex (D; C62-1) dorsal edge of the posterolateral process of the parietal; the outwards curve of the distal end of the posterolateral process of the parietal (B; C64-1); the distally expanded (C; C68-0) or straight paroccipital processes (F; C68-1); the dorsally vaulted supraoccipital (E; C73-0); and the narrow contribution of the basioccipital to the dorsal surface of the condyle (B; C78-1). Skulls scaled to the same occipital condyle width.

Comments. The absence of parietal contribution to the posttemporal fenestra is sometimes difficult to observe due to imperfectly preserved or distorted skulls. All diplodocid skulls have exoccipitals that bear a dorsolateral spur, which forms the dorsomedial end of the posttemporal fenestra (the ‘posttemporal process’ of Harris, 2006a). Additionally, most specimens have dorsally extended distal ends of the paroccipital processes, which curve back towards the exoccipital spur. These two prominences are interconnected by the squamosal in complete diplodocid skulls (CM 11161, E Tschopp, pers. obs., 2011).

C60: Parietal, portion contributing to skull roof, anteroposterior length/transverse width: wide, >50% (0); narrow, 7–50% (1); practically nonexistent, <7% (2) (New; Table S6). Ordered.

Comments. In some taxa, the posterior-most point of the fronto-parietal suture is located posterior to the supratemporal fenestra. The minimum values are compared in this ratio. Minimum anteroposterior length is measured between two parallel, transversely oriented lines intersecting the posterior-most point of the fronto-parietal suture and the anterior-most point of the concavity of the edge separating the dorsal portion of the parietal from the nuchal fossa.

C61: Parietal, distance separating supratemporal fenestrae: less than 1.5 times the width of the long axis of the supratemporal fenestra (0); at least 1.5 times the length of the long axis of the supratemporal fenestra (1) (W02-24; modified by M12-37; Table S7).

Comments. The original character states of Wilson (2002) left a gap (subequal, or double). The distance between the supratemporal fenestrae in many diplodocid specimens does not reach two times the maximum diameter of the fenestra, which led Mannion et al. (2012) to adjust the state boundaries. Specimens were remeasured where possible (Table S7), for others scorings of Wilson (2002) or Mannion et al. (2012) were used. The new measurements show that the ratios are often overestimated and that there seem to be three clusters of taxa (less than one: e.g., Giraffatitan; between one and 1.6 times: e.g., Kaatedocus; more than 1.6 times: e.g., Suuwassea). However, a more inclusive study of this character should be performed in order to recognize the most useful state boundaries for phylogenetic analyses. At the moment it seems wisest to retain the proposed version of Mannion et al. (2012).

C62: Parietal, posterolateral process, dorsal edge in posterior view: straight, and ventrolaterally oriented, so that the supratemporal fenestra is slightly facing posteriorly as well (0); convex, so that the postorbital and thus the supratemporal fenestra are not visible (1) (T13-43; Fig. 14).

Comments. The posterior view of the skull corresponds to the view parallel to the long axis of the occipital condylar neck, which was found to be oriented parallel to the lateral semicircular canal, thus indicating the neutral head position (Schmitt, 2012).

C63: Parietal, occipital process, dorsoventral height: low, subequal to less than the diameter of the foramen magnum (0); high, nearly twice the diameter of the foramen magnum (1) (W02-21; modified; Table S8).

Comments. Measurements are taken in strict posterior view (see above). Height is measured vertically between the dorsal-most and ventral-most extension of the occipital process, and the foramen magnum. In case of the occipital process, the dorsal- and ventral-most points are usually transversely shifted against each other. The measurements are therefore taken between horizontal lines intersecting the extremes. The state boundaries are tentatively set at 1.5, but more inclusive analyses would have to be undertaken in order to score this character adequately.

C64: Parietal, occipital process, distal end: ventrolaterally oriented, such that dorsolateral edge is straight or convex (0); curving laterally, such that dorsolateral edge becomes concave distally (1) (New; Fig. 14).

Comments. The distal end of the posterolateral process of the parietal of non-diplodocine flagellicaudatans curves outwards to meet the squamosal. This is not the case in the diplodocine skulls examined for this analysis.

C65: Parietal, distinct horizontal ridge separating dorsal from posterior portion: absent, transition more or less confluent (0); present, creating a distinct nuchal fossa below the ridge (1) (T13-44; wording modified; Fig. 15).

Figure 15 Skull of Kaatedocus siberi SMA 0004 in posterolateral view.

Note the transverse ridge of the parietal (arrow, C65-1). Abb.: anp, antotic process; bo, basioccipital; f, frontal; p, parietal; po, postorbital; popr, paroccipital process; ppfo, postparietal foramen; pra, proatlas; snc, sagittal nuchal crest; so, supraoccipital; stf, supratemporal fenestra.

Comments. This character is best observed in oblique posterolateral view, if one does not have the specimens at hand. In the derived state, the transverse ridge caps the nuchal fossa dorsally, creating a distinct concavity below it. Given that small skulls appear to have this feature most expressed (AMNH 7530, CM 3452, SMA 0004), there is some possibility that the nuchal fossae become shallower during ontogeny.

C66: Postparietal foramen: absent (0); present (1) (U95; Fig. 5).

Comments. Postparietal foramina have been interpreted to be a dicraeosaurid synapomorphy (Whitlock, 2011a), but were recently shown to be present as well in Diplodocidae (Tschopp & Mateus, 2013b). The opening is located at the posteromedial corner of the two parietals, where they meet the supraoccipital. It might be associated with a vertical groove internally on the supraoccipital (Remes, 2006; see below), but additional CT studies would have to be performed in order to check for the presence or absence of this groove in specimens without the postparietal foramen. Many diplodocid specimens are damaged in this region of the skull, which makes it difficult to verify the presence of the foramen and impedes an evaluation of its distribution among flagellicaudatans. The definitive presence in Kaatedocus, and the unknown state in the two apatosaur skulls CM 11162 and YPM 1860 (due to crushing; E Tschopp, pers. obs., 2011), indicates that it might be plesiomorphic for Flagellicaudata, subsequently lost in Tornieria and Diplodocus.

C67: Paroccipital process (popr), posterior face: smooth/flat (0); with longitudinal ridge along popr body extending from dorsomedial to ventrolateral corners (1) (T13-46; Fig. 16).

Figure 16 Skull of Kaatedocus siberi SMA 0004 in posterior view.

Note the oblique ridge on paroccipital process (arrow, C67-1). Abb.: CV, cervical vertebrae; f, frontal; p, parietal; po, postorbital; popr; paroccipital process; ppfo, postparietal foramen; pra, proatlas; ptf, posttemporal fenestra; q, quadrate; qj, quadratojugal; so, supraoccipital; sq, squamosal; stf, supratemporal fenestra.

Comments. Most of the specimens examined have a slightly convex posterior face of the paroccipital processes. However, few have such a distinct ridge as is present in Kaatedocus. In the latter, this ridge is accompanied by a rugose area at its dorsomedial origin. None of these structures are present in CM 11161, for example.

C68: Paroccipital process distal terminus: expanded vertically (0); not expanded (dorsal and ventral edges are subparallel) (1) (U98-38; modified; Fig. 14).

Comments. Upchurch (1998) included two morphologies in one character: the dorsoventral expansion, and the rounded or straight distal edge. The shape of the distal edge is difficult to assess qualitatively, because many specimens have slightly convex, or somewhat triangular lateral ends of the paroccipital process (e.g., Suuwassea ANS 21122, or Kaatedocus SMA 0004, Fig. 14). Therefore, the character description was limited to the distal expansion.

C69: Paroccipital process, distal end in lateral view: straight (0); curved (1) (New; Fig. 17).

Figure 17 Flagellicaudatan braincases.

Braincase of Suuwassea emilieae ANS 21122 (A) and Tornieria africana MB.R.2386 (B) in right (A) and left (B) lateral view, illustrating the curved lateral end of the paroccipital process (A; C69-1), and the short (A; C79-0) and elongate basioccipital (B; C79-1). Abb.: anp, antotic process; bo, basioccipital; bpr, basipterygoid process; bt, basal tuber; cpr, crista prootica; f, frontal; os, orbitosphenoid; p, parietal; popr, paroccipital process. Scale bar = 5 cm.

Comments. Due to the slight posterior orientation of the paroccipital processes in many sauropod taxa, a strict lateral view of the skull does often not allow for an accurate coding of this character. Also, on pictures of articulated skulls it is often difficult to see the distal end of the paroccipital process well enough, because it is partly obscured by the squamosal. In most cases, a posterolateral instead of lateral view would thus be more helpful. Specimens, where the paroccipital processes were bent posteriorly during diagenesis should not be scored for this character because the pressure resulting in such a distortion likely also affected the curvature.

C70: Supratemporal fenestra: present, relatively large (anteroposterior diameter is at least 5% of occiput width) (0); absent, or greatly reduced (so that anteroposterior diameter is less than 5% of occipital width) (1) (W02-25; modified by M12-40).

Comments. Wilson (2002) proposed this feature as present/absent character, but Mannion et al. (2012) showed that one of Wilson’s (2002) derived taxa (Limaysaurus) actually has a supratemporal fenestra, although an extremely reduced one. Because this is a derived state of Rebbachisauridae, and because all diplodocid skulls show large openings, no additional measuring was done for this analysis.

C71: Supratemporal fenestra, maximum diameter: more than 1.2 times greatest diameter of foramen magnum (0); less than 1.2 times the greatest length of foramen magnum (1) (Y93-32; modified by M12-41).

Comments. Mannion et al. (2012) introduced the quantitative state boundaries to the original description (Yu, 1993). Basically, this character is an extension of the previous one, with the exception that Nigersaurus is impossible to score due to the complete absence of the supratemporal fenestra in this taxon. In addition to Limaysaurus, the quantitative boundaries of Mannion et al. (2012) also include the dicraeosaurids Dicraeosaurus and Amargasaurus, which have reduced supratemporal fenestra as well, but not to the extent shown by Rebbachisauridae. As stated above, the difference in relative size of the supratemporal fenestrae between the mentioned taxa and Diplodocidae is large, and thus no additional measurements were taken in order to test the boundaries proposed by Mannion et al. (2012).

C72: Supraoccipital, anterodorsal margin: internally concave, associated with a channel extending ventrally on the internal face (0); straight (1) (Remes, 2006; Fig. 18).

Figure 18 Braincase of Diplodocus sp. CM 11161 (A) and Tornieria africana MB.R.2386 (B) in dorsal view.

Note the concave anterior margin of the supraoccipital in Diplodocus (A; C72-0), in contrast to the convex edge of Tornieria (B; C72-1). The left frontal of MB.R.2386 is lacking. Abb.: f, frontal; na, nasal; os, orbitosphenoid; p, parietal; pf, prefrontal; po, postorbital; popr, paroccipital process; so, supraoccipital; sq, squamosal; stf, supratemporal fenestra. Not to scale.

Comments. The channel was proposed by Remes (2006) as a distinguishing character between Tornieria and Dicraeosauridae, where the presence of the canal is coupled with the presence of a postparietal fenestra. However, as shown in Kaatedocus, these two features are not necessarily correlated. A separate coding for the two characters is thus justifiable. This is the first analysis to include this character.

C73: Supraoccipital, dorsal extension: high and vaulted, such that the dorsolateral edges are strongly sinuous (0); low, with the dorsolateral edges straight (1) (Remes, 2006; Fig. 14).

Comments. Remes (2006) used this character in order to distinguish Tornieria from Apatosaurus, but did not include it in his phylogenetic analysis. The present analysis is thus the first one to do so.

C74: Supraoccipital: sagittal nuchal crest: broad, weakly developed (0); narrow, sharp, and distinct (1) (W11-45; Fig. 19).

Figure 19 Flagellicaudatan skulls in posterior view.

Skulls of Diplodocus sp. CM 11161 (A) and Dicraeosaurus hansemanni MB.R.2379 (B) in posterior view, illustrating the development of the sagittal nuchal crest (C74), and the supraoccipital foramina (C75). Abb.: bo, basioccipital; ex, exoccipital; fm, foramen magnum; p, parietal; po, postorbital; popr, paroccipital process; ptf, posttemporal fenestra; so, supraoccipital; sq, squamosal. Skulls scaled to the same skull width.

Comments. The nuchal crest lies on the midline of the supraoccipital, extending dorsoventrally. A narrow, sharp crest was previously thought to be a synapomorphy for Dicraeosauridae, but Tschopp & Mateus (2013b) showed that it also occurs in certain diplodocids.

C75: Supraoccipital, foramen close to contact with parietal: absent (0); present (1) (T13-52; Fig. 19).

Comments. This foramen is called an external occipital foramen by Balanoff, Bever & Ikejiri (2010) and is sometimes located entirely on the supraoccipital (Dicraeosaurus hansemanni MB.R.2379, Janensch, 1935), and in other cases on the suture with the parietal (Kaatedocus siberi SMA 0004, E Tschopp, pers. obs., 2010). Only taxa with well visible foramina are coded as apomorpic.

C76: Crista prootica, size: rudimentary (0); expanded laterally into dorsolateral process (1) (Salgado & Calvo, 1992; U95; Fig. 20).

Figure 20 Basal tubera and basisphenoid of Dicraeosaurus hansemanni MB.R.2379 in posteroventral (A), left lateral (B), and anterodorsal view (C).

Note the lateral expansion of the anteroventral end of the crista prootica (C76-1), the narrowly diverging, and elongate basipterygoid processes (C92-2 and C94-2, respectively), the deep slot-like cavity separating the bases of the processes (A, arrowhead; C95-1), and the groove on the dorsal surface of the parasphenoid rostrum (C; C99-1). Abb.: bt, basal tuber; bpr, basipterygoid process; cpr, crista prootica; psr, parasphenoid rostrum. Scale bar = 5 cm.

Comments. Although diplodocids have a laterally protruding crista prootica (e.g., SMA 0011), only dicraeosaurids develop distinct lateral processes at the anteroventral ends of the crista prootica.

C77: Occipital condyle, articular surface: well offset from condylar neck (0); continuously grading into condylar neck (1) (New; Fig. 21).

Figure 21 Neosauropod braincases.

Braincase of Camarasaurus sp. UUVP 4286 (A; modified from Madsen, McIntosh & Berman, 1995) and Tornieria africana MB.R.2386 (B) in a view perpendicular to the dorsal surface of the occipital condyle, illustrating the distinctly offset articular surface (arrow in A; C77-0), in contrast to the derived condition of diplodocoids (B; C77-1). Abb.: ex, exoccipital; f, frontal; fm, foramen magnum; oc, occipital condyle; os, orbitosphenoid; p, parietal; pf, prefrontal; popr, paroccipital process. Skulls scaled to same breadth of occipital condyle.

Comments. Whereas in more basal sauropods the articular surface of the occipital condyle is usually well delimited, and offset from the condylar neck by a distinct ridge, diplodocids generally do not have such a clear distinction. The character states are most easily distinguished in dorsal view.

C78: Basioccipital, contribution to dorsal side of occipital condylar neck: present and broad, around 1/3 of entire dorsal side (0); reduced to absent (1) (Harris & Dodson, 2004; Fig. 14).

Comments. Harris & Dodson (2004) proposed the narrow contribution of the basioccipital to the dorsal face of the occipital condyle as characteristic for Suuwassea. A wider survey of the distribution of this character showed that the contribution of the basioccipital to the dorsal side of the occipital condylar neck is reduced in some diplodocid specimens as well.

C79: Basioccipital, distance from base of occipital condyle to base of basal tubera (best visible in lateral view): short, such that area is gently U-shaped in lateral view (0); elongate, with a flat portion between occipital condyle and basal tubera (1) (T13-54; wording modified; Fig. 17).

Comments. The distance is taken relative to the height of the basal tuber, creating a narrow U-shape or a shallow, wide concavity in lateral view (Fig. 17).

C80: Basioccipital depression between foramen magnum and basal tubera: absent (0); present (1) (W02-50; Fig. 22).

Figure 22 Braincase of Losillasaurus giganteus MCNV Lo-26 in posterolateral (A) and posterior (B) view.

Note the lateral basioccipital depression between the foramen magnum and the basal tubera (A; C80-1); the laterally curving distal ends of the basipterygoid processes (B; C97-1), as well as their distinct transverse expansion (B; 98-1). Abb.: bo, basioccipital; bpr, basipterygoid process; bt, basal tuber; ex, exoccipital; fm, foramen magnum; popr, paroccipital process; psr, parasphenoid rostrum; so, supraoccipital. Scale bar = 10 cm.

Comments. The depression is a concave area on the posterolateral sides of the basioccipital, which is different from the concavity on the posterior face of the basal tubera described in character 85.

C81: Basioccipital, pit between occipital condyle and basal tubera: absent (0); present (1) (M13-98; wording modified; Fig. 23).

Figure 23 Hypothetical diplodocid basioccipital-basisphenoid complex in posteroventral view.

Note the locations of pits sometimes present in diplodocid specimens: between occipital condyle and basal tubera (C81-1), in the notch between basal tubera (C90-1), and on the basisphenoid, between the bases of the basipterygoid processes (termed ‘basipterygoid recess’ by Wilson, 2002; C91-1). Abb.: bo, basioccipital; bpr, basipterygoid process; bs, basisphenoid; bt, basal tuber; cpr, crista prootica; ex, exoccipital; popr, paroccipital process.

Comments. Various pits can mark the area around the basal tubera: YPM 1860 bears one in the notch between the tubera (see below), and a second one on the basioccipital posterior to the tubera (which is the one described here). The basipterygoid recess is also located close by, but anterior to the basal tubera on the basisphenoid, instead of the basioccipital. Mannion et al. (2013) described this pit as a fossa on the posterior surface of the basal tubera, but this wording could be understood in a similar way as the concavity coded for in C85 herein. We therefore reworded the character to better delimit the character to the presence of this apparently blind foramen as seen in Fig. 23.

C82: Basal tubera: globular (0); box-like (1) (Whitlock, Wilson & Lamanna, 2010; Fig. 24).

Figure 24 Neosauropod basal tubera.

Basal tubera of Camarasaurus grandis YPM 1905 (A; modified from Madsen, McIntosh & Berman, 1995), Suuwassea emilieae ANS 21122 (B), and Kaatedocus siberi SMA 0004 (C; photo by J Marinheiro) in posterior view. Note the globose (B; C82-0) compared to the box-like shape (C; C82-1) of the tubera, the transverse ridge on their posterior face (C; C86-1), and the ventrolateral (A; C89-0) in contrast to ventral orientation (C; C89-1). Abb.: bo, basioccipital; bpr, basipterygoid process; bs, basisphenoid; bt, basal tuber; ex, exoccipital; fm, foramen magnum; oc, occipital condyle; popr, paroccipital process. Pictures scaled to same distance between dorsal face of occipital condyle and basal tubera.

Comments. Whitlock, Wilson & Lamanna (2010) used this character as one of the features distinguishing the juvenile diplodocid skull CM 11255 from Apatosaurus. It is herein used for the first time as a phylogenetic character.

C83: Basal tubera, breadth: <1.3 times (0); 1.3-1.85 times (1); >1.85 times occipital condyle width (2) (W02-49; modified; Table S9).

Comments. The character was initially defined without clear state borders, and only with two states (Wilson, 2002). Mannion (2011) suggested further subdivision of the character, based on a wider survey of this ratio among sauropods. Mannion’s (2011) table was here extended and the character state boundaries were modified following higher-level taxonomy and gaps in the distribution of the values.

C84: Basal tubera: distinct from basipterygoid (0); reduced to slight swelling on ventral surface of basipterygoid (1) (W11-53; Fig. 25).

Figure 25 Diplodocimorph skulls in occipital view.

Skulls of Nigersaurus taqueti (A; modified from Schmitt, 2012) and Galeamopus sp. USNM 2673 (B) in occipital view. Note the reduced basal tubera in Nigersaurus (A; C84-1), and the convex (A; C 85-0), or concave (B; C85-2) posterior face of the tubera. Abb.: bpr, basipterygoid process; bt, basal tuber; cpr, crista prootica; fm, foramen magnum; oc, occipital condyle; popr, paroccipital process; so, supraoccipital. Skulls scaled to same occipital condyle height.

Comments. The use of this character and its coding overlaps with an additional character proposed by Wilson (2002): ‘Basal tubera, anteroposterior depth: approximately 33%, or more, of dorsoventral height (0); sheetlike, less than 33% (normally around 20%) dorsoventral height (1).’ Whitlock’s (2011a) character is herein preferred because the directional terms used in Wilson (2002) are sometimes confusing due to varying orientations of the basal tubera of Diplodocoidea and non-diplodocoid sauropods.

C85: Basal tubera, shape of posterior face: convex (0); flat (1); slightly concave (2) (W11-54; modified by T13-63; Fig. 25).

Comments. The ‘posterior face’ of the basal tubera is herein intended to be the side facing the occipital condyle. The concavity described herein is different from the concavity sometimes present on the lateral side of the basioccipital (see above).

C86: Basal tubera, posteroventral face: continuous (0); marked by a distinct transverse ridge (1) (New; Fig. 24).

Comments. The surface of the basal tubera is usually regularly rugose, and without distinct structuring. SMA 0004, however, bears a distinct transverse ridge on the posteroventral face of its basal tubera.

C87: Basal tubera, longest axes: parallel (0); in an angle to each other, pointing towards the occipital condyle (1) (New; Fig. 26).

Figure 26 Diplodocid basioccipital-basispenoid complex.

Basioccipital-basispenoid complex of Apatosaurus louisae CM 11162 (A), Kaatedocus siberi SMA 0004 (B; traced from a photo by J Marinheiro), and Diplodocus sp. CM 11161 (C) in posteroventral view. Note the differing orientations of the longest axes of the basal tubera (B; C87-0; in contrast to C; C87-1), as well as the concave (A; C88-1) versus the straight to slightly convex anterior edge of the tubera (B; C88-0). Abb.: bo, basioccipital; bpr, basipterygoid process; bs, basisphenoid; bt, basal tuber; ex, exoccipital. Drawings not to scale.

Comments. The character is to be coded based on a view perpendicular to the orientation of the basipterygoid processes. It is inspired by the character of Tschopp & Mateus (2013b) describing the anterior margin of the tubera as V- or U-shaped, which included two differing morphologies in the same character (orientation of the tubera and shape of the anterior margin). The two morphologies are here treated as different characters (see below). In some cases (e.g., CM 11162), the outline of the tubera is subtriangular, with a more or less right angle pointing posterolaterally. These cases were treated as apomorphic, because the longest distance follows the obliquely oriented hypotenuse of the triangle.

C88: Basal tubera, anterior edge: straight or convex (0); concave (1) (T13-64; Fig. 26).

Comments. The second of the two characters inspired by Tschopp & Mateus’ (2013b) character about the anterior margin of the basal tubera. The anterior edge is the one facing towards the basipterygoid processes, which in non-diplodocoid sauropods is oriented rather anteroventrally. In specimens with angled basal tubera (see above), the anterior margin is oriented obliquely.

C89: Basal tubera in posterior view: facing ventrolaterally (0); facing straight ventrally, forming a horizontal line (1) (T13-65; wording modified; Fig. 24).

Comments. Some specimens (in particular non-flagellicaudatans) have rounded basal tubera, which extend onto the lateral surface of the basioccipital. These are treated as plesiomorphic, because the line projecting through the medial- and lateral-most points of the tubera is oblique in these cases.

C90: Basal tubera, foramen in notch that separates the two tubera: absent (0); present (1) (T13-66; Fig. 23).

Comments. This foramen is one of three openings that can occur in this area (see above and below). However, the pit described in this character cannot be homologous to the other ones because it occurs together with the basipterygoid recess in HMNS 175 (Holland, 1906) and together with the basioccipital pit in YPM 1860 (E Tschopp, pers. obs., 2011).

C91: Basisphenoid/basipterygoid recess: absent (0); present (1) (W02-51; polarity reversed; Fig. 23)

Comments. The basipterygoid recess is a pit located anterior to the basal tubera, on the basisphenoid. Its absence was considered autapomorphic for Apatosaurus, representing a reversal to the plesiomorphic state in Sauropoda (Wilson, 2002). However, in his phylogenetic analysis, Wilson (2002) scored Apatosaurus as having a recess, sharing this state with basal sauropods like Shunosaurus. The character was organized as a presence/absence character, with the presence being plesiomorphic (Wilson, 2002). Assuming that the discussion of the autapomorphies is right, polarity of the character states was inverted herein. The basipterygoid recess might be confused with the pits located in the notch between the tubera or the one posterior to them (see above), so it is important to state that it lies anterior to the tubera, between the bases of the basipterygoid processes.

C92: Basipterygoid processes: widely diverging (>60°) (0); intermediate, 31°−60°(1); narrowly diverging (<31°) (2) (Y93-29; modified; Fig. 20; Table S10).

Comments. There are several modes to measure the angle between the processes, and no previous analysis defines how this angle should be measured. Here, divergence is measured between lines drawn from the basisphenoid center, where the bases of the basipterygoid processes meet, to the anteromedial-most point of the processes. This is preferably done in posterior or posteroventral view, perpendicular to the longitudinal axis of the processes. The present measuring technique yields slightly different results compared to earlier studies, but general trends are similar.

C93: Basipterygoid processes, orientation: directed more than 75° to skull roof (normally perpendicular) (0); angled less than 75° to skull roof (normally approximately 45°) (1) (McIntosh, 1990b; U98-41; modified; Table S11).

Comments. New numeric state boundaries were established, because a survey of diplodocoid braincases showed that there is more variety than previously recognized (Table S11). However, the difference was already recognized as taxonomically important by McIntosh (1990b). The angle is measured between the skull roof and a line through the center of the proximal and distal ends. This is important, especially because macronarian basipterygoid processes tend to curve backwards at their distal ends, thereby increasing the angle as measured here.

It is possible that this character is correlated with the large angle between the anterior and dorsal quadratojugal processes and the backwards inclination of the ventral ramus of the quadrate. This entire region is interconnected by the pterygoid, and the anterior shifting of the basisphenoid-pterygoid articulation due to the changed orientation of the basipterygoid processes might have been caused by, or the reason for the more anteriorly orientated ventral ramus of the quadrate, and therefore also the widening of the angle between the quadratojugal processes. However, because there is no evidence of correlation and no skulls are known of basal diplodocoid taxa that might show intermediate states, the separate characters are retained.

Furthermore, there is some indication that the character could be ontogenetically controlled: the two relatively small diplodocine skulls CM 3452 and SMA 0004 both have somewhat larger angles compared to larger specimens (Table S11), and lower angles in the quadratojugal. However, further studies are needed to decide if this is really ontogenetic, or if it could be taxonomically significant.

C94: Basipterygoid processes, ratio of length:basal transverse diameter: <4 (0); = or >4.0 (1) (W02-46; modified; Fig. 20; Table S12).

Comments. The character was initially defined as ratio of length to maximum basal diameter (Wilson, 2002). However, maximum basal diameter is often oriented dorsoventrally (at least in diplodocids), which means that one cannot take the measurements in a picture of the processes in ventral view only. Also, dorsoventral height changes considerably, and continuously, towards the base of the processes in some specimens (e.g., Dicraeosaurus hansemanni MB.R.2379; Janensch, 1935; Fig. 20). In lateral view, it is sometimes difficult to decide where exactly the base of the process is situated. Therefore, and because ventral views are obtainable more frequently than lateral views, the ratio length/basal transverse diameter is preferred herein. The dimensions should be measured perpendicular to each other. Wilson (2002) initially left a gap in the definition of the states (2 or less, 4 or more), which was corrected for by Mannion et al. (2012). However, as a more rigorous assessment of these ratios shows (Table S12), the state boundary should rather be set to four, the derived, elongate state resulting as a shared synapomorphy for Diplodocinae and Dicraeosauridae.

Measuring the basipterygoid processes in such a way leads to much higher elongation ratios for the holotype of Kaatedocus siberi (SMA 0004) than reported in its initial description (Tschopp & Mateus, 2013b). The low ratio also served as local autapomorphy for the genus (Tschopp & Mateus, 2013b). Following the results presented herein, this is most probably an artifact based on differing measurement protocols, because Tschopp & Mateus (2013b) compared length with dorsoventral height, which is the maximum basal diameter in SMA 0004 (Tschopp & Mateus, 2013b). The current measurements show that Kaatedocus is actually well in the range of Diplodocinae, which can easily be distinguished from Apatosaurus louisae CM 11162 (Table S12).

C95: Basipterygoid, area between the basipterygoid processes and parasphenoid rostrum: is a mildly concave subtriangular region (0); forms a deep slot-like cavity that passes posteriorly between the bases of the basipterygoid processes (1) (U95; U98-44; Fig. 20).

C96: Basipterygoid processes, orientation of proximal-most portions: same as central portion of shaft (0); parallel to each other, outwards curve of shaft happens only more anteriorly (1) (New; Fig. 27).

Figure 27 Diplodocine basisphenoids.

Basisphenoid of Kaatedocus siberi SMA 0004 (A; traced from a photo by J Marinheiro), and Diplodocus sp. CM 11161 (B) in posteroventral view. Note the parallel proximal portion of the basipterygoid processes and the accompanying outwards curve in Kaatedocus (A; C96-1), in contrast to the straight processes of CM 11161 (B; C96-0). Abb.: bo, basioccipital; bpr, basipterygoid process; bs, basisphenoid; bt, basal tuber. Scaled to the same process length.

Comments. The development of this character is best seen in ventral view. In the derived state, the parallel portion of the basipterygoid processes are often interconnected dorsomedially by a thin sheet of bone. On the other hand, a similar sheet can also be present if the processes are entirely straight.

C97: Basipterygoid processes, distal end in anterior view: straight (0); curving laterally (1) (New; Fig. 22).

Comments. This character compares the distal end of the basipterygoid process with the central portion. It is thus different from the feature described in character 96.

C98: Basipterygoid processes, distal lateral expansion: absent (0); present (1) (New; Fig. 22).

Comments. Only abrupt distal expansions are coded as apomorphic. Gradually extending processes are treated as plesiomorphic.

C99: Parasphenoid rostrum, groove on dorsal edge: absent (0); present (1) (U95; U98-45; modified; Fig. 20).

Comments. Upchurch (1995) and Upchurch (1998) proposed the character combining the presence of a dorsal groove with the lateral shape of the rostrum, thereby implying that the dorsoventrally thin parasphenoid of diplodocoids would not bear dorsal grooves. However, a more detailed study of diplodocoids shows that the groove is actually present in most of them.

C100: Optic foramen: paired (0); unpaired (1) (Berman & McIntosh, 1978; S06-129; Fig. 28).

Figure 28 Flagellicaudatan braincases.

Braincases of Suuwassea emilieae ANS 21122 (A), and Tornieria africana MB.R.2386 (B; traced from Janensch, 1935) in anterior view. Note the unpaired optic foramen of Suuwassea (A; C100-1), in contrast to the paired foramen in Tornieria (B; C100-0). Abb.: anp, antotic process; bs, basisphenoid; can, crista antotica; cpr, crista prootica; ls, laterosphenoid; olf, olfactory foramen; opf, optic foramen; os, orbitosphenoid; popr, paroccipital process; pro, prootic. Scaled to the same width of the orbitosphenoids.

Comments. The optic foramen lies close to the midline, within the orbitosphenoid in most sauropod taxa. Generally, the right and left foramina are separated medially by a narrow bony bridge, which is absent in some diplodocoid specimens (e.g., Suuwassea, Harris, 2006a). Sander et al. (2006) were the first to include the character in a phylogenetic analysis.

C101: Palatobasal contact, shape: pterygoid with small facet (0); dorsomedially orientated hook (1) (W02-36; modified by T13-67; Fig. 29).

Figure 29 Pterygoid of Camarasaurus lentus DNM 28.

Left pterygoid of Camarasaurus lentus DNM 28 in medial view. Note the presence of a hook-like process at the articulation surface for the basipterygoid process (C101-1). Diplodocidae, on the other hand, only have shallow articular facets without hooks. Abb.: ap, anterior process; bph, basipterygoid hook; er, ectopterygoid ramus; qr, quadrate ramus. Picture traced from Madsen, McIntosh & Berman (1995). Scale bar = 10 cm.

Comments. Tschopp & Mateus (2013b) deleted a third state from the original character, which describes the specific rocker-like morphology of this region in nemegtosaurid sauropods (Wilson, 2002). Because no taxon of this clade is included, the additional state is redundant here.

C102: Pterygoid, transverse flange (i.e., ectopterygoid process) position: between orbit and antorbital fenestra (0); anterior to antorbital fenestra (1) (U95; Fig. 9).

Comments. The transverse flange of the pterygoid connects to the maxilla through the ectopterygoid (Upchurch, Barrett & Dodson, 2004).

C103: Vomer, anterior articulation: maxilla (0); premaxilla (1) (W02-42; polarity reversed; Fig. 9).

Comments. Polarity was reversed compared to Wilson’s (2002) character due to the limited taxon sampling.

C104: Dentary, anteroventral margin shape: gently rounded (0); sharply projecting triangular process or ‘chin’ (1) (U98-58, modified by W02-56; Fig. 30).

Figure 30 Neosauropod dentaries.

Left dentary of Camarasaurus lentus DNM 28 (A; traced from Madsen, McIntosh & Berman, 1995), Dicraeosaurus hansemanni MB.R.2372 (B; traced from Janensch, 1935), and Nigersaurus taqueti MNN GAD512-10 (C; traced from Sereno et al., 2007) in lingual view. Note the chin-like ventral process in Dicraeosaurus (B; C104-1), the different shapes of the symphysis (C105-1 to 3), and the high elevation of the coronoid eminence in Camarasaurus (A; C108-0). Abb.: an, angular; d, dentary; sa, surangular; sym, symphysis; t, tooth. Scaled to the same anteromedial height of the dentary.

Comments. Usually considered a flagellicaudatan synapomorphy, some specimens of Camarasaurus also show a weak ventral expansion at the anterior extreme of the lower jaw. However, this never reaches the chin-like state present in Diplodocus, and Camarasaurus is thus included in the plesiomorphic state here.

C105: Dentary, cross-sectional shape of symphysis: oblong or rectangular (0); subtriangular, tapering sharply towards ventral extreme (1); subcircular (2) (W11-60; Fig. 30).

Comments. Diplodocids have ventrally tapering symphyses, but they do not taper to a point as in dicraeosaurids (Whitlock & Harris, 2010) and were thus scored as plesiomorphic.

C106: Dentary, tuberosity on labial surface near symphysis: absent (0); present (1) (Whitlock & Harris, 2010; reworded by W11-57; Fig. 31).

Figure 31 Diplodocimorph dentaries.

Left dentary of Dicraeosaurus hansemanni MB.R.2372 (A), and Nigersaurus taqueti MNN GAD512-10 (B; traced from Sereno et al., 2007) in dorsal view. Note the labial tubercle in Dicraeosaurus (A; C106-1), the dentigerous portion that expands laterally in Nigersaurus (B; C107-1), and the anterolaterally displaced tooth row, compared to the usual curvature in both taxa (C112-1). Abb.: sym, symphysis; t, tooth. Scaled to the same anteroposterior length.

Comments. This character was originally proposed by Whitlock & Harris (2010) to unite Suuwassea and Dicraeosaurus.

C107: Dentary, anterolateral corner: not expanded laterally beyond mandibular ramus (0); expanded beyond lateral mandibular ramus (1) (W11-59; Fig. 31).

Comments. The derived state of this character describes the extreme case of character 112. To date, it is only known in the rebbachisaurid Nigersaurus (Sereno et al., 2007).

C108: Mandible, coronoid eminence: strongly expressed, clearly rising above plane of dentigerous portion (0); absent (1) (W11-62; Fig. 30).

Comments. Some diplodocids have dorsally expanded coronoid areas, but they do not reach above the plane of the dentigerous portion.

C109: Surangular foramen: absent (0); present (1) (New; Fig. 32).

Figure 32 Neosauropod lower jaw.

Left lower jaw of Camarasaurus lentus CM 11338 (A; modified from Madsen, McIntosh & Berman, 1995), Nigersaurus taqueti MNN GAD-512 (B; traced from Sereno et al., 2007), and Galeamopus sp. SMA 0011 (C) in lateral view. Note the surangular foramen in A and B (C109-1), the external mandibular fenestra in Nigersaurus (B; C110-0), the strongly overlapping teeth of Camarasaurus (A; C120-0) in contrast to the more widely spaced teeth of diplodocids (C; C120-1), and the anterior inclination of the diplodocid teeth in respect to the jaw axis (C; C122-1). Abb.: an, angular; d, dentary; emf, external mandibular fenestra; sa, surangular; saf, surangular foramen; t, tooth. Scaled to the same mandibular length.

Comments. The location of the surangular foramen can vary in different taxa. Usually, it is situated in the anterodorsal portion, but in some cases it is shifted posteriorly.

C110: External mandibular fenestra: present (0); absent (1) (McIntosh, 1990b; R93-3; Fig. 32).

Comments. The presence is a retained plesiomorphy, shared with early sauropodomorphs (Wilson, 2002).

C111: Snout shape in dorsal view: premaxilla-maxilla index (PMI; Whitlock, Wilson & Lamanna, 2010) <67% (0); 67-85% (1); >85% (2) (U98-1; W11-64; modified; Table S13). Ordered.

Comments. In order to avoid gaps, an intermediate state was added to Whitlock’s (2011a) version. The state boundaries were chosen following high-level phylogenetic differences. Measurements taken on photographs from slightly different angles of the skulls CM 3452, 11161, 11162, and SMA 0011 show that the orientation of the skull has a relatively high influence on the measured PMI (Table S13). In order to avoid this, the same measurements were taken in more than one picture of the same skulls, where possible. In future, one should check and remeasure this ratio in all diplodocid skulls, making sure that they are always taken in exactly the same orientation. Best results are to be expected with the ventral maxillary edge oriented horizontally.

Whitlock, Wilson & Lamanna (2010) reported that the snout becomes more squared during ontogeny in diplodocids. It might thus be possible that more juvenile specimens become artificially grouped closer to more basal taxa when including this character.

Teeth

C112: Shape of tooth row in occlusal view: follows curvature of dentary (0); anterolateral corner of tooth row displaced labially (1) (Whitlock & Harris, 2010; Fig. 31).

Comments. In dicraeosaurids, the tooth row seems to be the main responsible for the squared appearance of the lower jaw. The ventral portions of the dentary would be much more rounded (Whitlock & Harris, 2010). The diplodocid AMNH 969 has a similar development as Suuwassea.

C113: Tooth rows, length: restricted anterior to orbit (0); restricted anterior to antorbital fenestra (1); restricted anterior to subnarial foramen (2) (G86; modified by W11-65; Fig. 1). Ordered.

Comments. In order to score this character, the skull should be held with the ventral margin of the maxilla oriented horizontally. The tooth row is usually more anteriorly restricted in the lower jaw than in the maxilla. Here, the maxillary tooth row is used as a reference. As for the snout shape, the anterior restriction of the tooth row also was interpreted as juvenile feature (Whitlock, Wilson & Lamanna, 2010).

C114: Dentary teeth, number: greater than 17 (0); 10-17 (1); 9 or fewer (2) (W98-67; modified by C12b-96; Table S14). Unordered.

Comments. Carballido et al. (2012b) added a third state to distinguish Demandasaurus and Suuwassea from other sauropod specimens. Given that the derived state is ambiguous, it is more parsimonious to leave the character unordered.

C115: Replacement teeth per alveolus, number: three or fewer (0); four or more (1) (W02-74, modified by W11-71).

Comments. The number of replacement teeth varies between the tooth-bearing bones of the same individual (D Schwarz, pers. comm., 2012). However, maximum number of replacement teeth is still informative, and therefore the character was retained.

C116: Teeth, crown-to-crown occlusion: present (0); absent (1) (W98-35; polarity reversed by W11-66).

C117: Teeth, wear facets shape: v-shaped (0); planar (1) (W98-36; modified; Figs. 33 and 34).

Figure 33 Eusauropod teeth.

Tooth of Omeisaurus tianfuensis ZDM T5705 (A; traced from He, Li & Cai, 1988), Camarasaurus sp. SMA 0002 (B), and Diplodocinae indet. CM 3452 (C) in lingual view. Note the V-shaped wear facets in Camarasaurus (B; C117-0), in contrast to the single, planar facet in diplodocids (C; C117-1), the longitudinal grooves in Omeisaurus and Camarasaurus (A, B; C123-1), and the marginal tooth denticles in Omeisaurus (A; C125-0). Abb.: ato, anterior tooth; dt, denticles; pto, posterior tooth; tc, tooth crown; tr, tooth root; wf, wear facet. Teeth scaled to the same crown length.

Comments. The initial character (Wilson & Sereno, 1998) was first adapted by Sereno et al. (2007), in order to include the paired planar facets of Nigersaurus. Here, the shape and number of wear facets are considered independent characters (see character 118), because they code for varying morphology or processes of food intake.

C118: Teeth, occlusal pattern: paired wear facets (0); single facet (1) (W98-36; modified; Fig. 34).

Figure 34 Tooth of Nigersaurus in labial (A) and lingual (B) view.

Note the paired, planar wear facets typical for Rebbachisauridae (C117-1; C118-0). Abb.: wf, wear facet. Figure traced from Whitlock (2011b).

Comments. See character 117.

C119: Teeth, SI values for tooth crowns: <3.4 (0); 3.4 or greater (1) (McIntosh, 1989; U98-69; modified; Table S15).

Comments. The SI value describes the slenderness of the teeth. It was defined as crown length/mesiodistal width (Upchurch, 1998). The state borders were changed, following large gaps apparently corresponding to higher-level taxonomy (Table S15).

C120: Tooth crowns, orientation: aligned slightly anterolingually, tooth crowns overlap (0); aligned along jaw axis, crowns do not overlap (1) (W98-34; polarity reversed by W11-68; Fig. 32).

C121: Tooth crowns, cross-sectional shape at midcrown: D-shaped (0); cylindrical (1) (R93-7; modified by W98-32; Fig. 35).

Figure 35 Neosauropod tooth cross-sections.

Tooth cross-section of Camarasaurus sp. AMNH 5764 (A), and Demandasaurus darwini MDS-RVII,438 (B; traced from Torcida Fernández-Baldor et al., 2011). Note the D-shaped crown of Camarasaurus (A; C121-0) in contrast with the rounded cross-section of diplodocoids (B; C121-0), and the asymmetric disposition of the enamel typical for rebbachisaurids (B; C124-1). The camarasaur tooth has the same specimen number as the Amphicoelias altus holotype, but does not belong to the same individual (see text). Abb.: de, dentin; en, enamel. Scaled to the same mesiodistal width.

Comments. Unworn diplodocoid teeth often have ellipsoid cross-sections. However, this is different from the spatulate non-diplodocoid teeth as e.g., typical for Camarasaurus. Teeth of the latter genus have a slightly concave lingual face, unlike the convex surface of diplodocoids. In the absence of nemegtosaurid titanosaurs, which show similarly shaped teeth (Upchurch, 1999; Wilson, 2005), the derived state results as an unambiguous synapomorphy of Diplodocoidea.

C122: Teeth, orientation relative to long axis of jaw: perpendicular (0); oriented anteriorly (procumbent) (1) (G86, U98-72; Fig. 32).

Comments. Tooth orientation is best recognized in the posterior-most teeth in the maxilla and dentary.

C123: Teeth, longitudinal grooves on lingual aspect: absent (0); present (1) (W02-76; Fig. 33).

Comments. Wilson (2002) initially scored only rebbachisaurids with the derived state. However, several non-diplodocoid taxa with spatulate teeth actually have a midline ridge on the lingual face of their teeth, creating two grooves mesially and distally to it (e.g., Osborn & Mook, 1921; Ouyang & Ye, 2002). Consequently, these taxa are scored as derived here as well.

C124: Teeth, thickness of enamel asymmetric labiolingually: absent (0); present (1) (W11-74; Fig. 35).

Comments. This feature can be observed easily in wear facets or cross-sections.

C125: Teeth, marginal denticles: present (0); absent (1) (McIntosh, 1990b; U98-66; Fig. 33).

Comments. There is some morphological variation in the location of the denticles (Carballido et al., 2012b), but because no diplodocid shows denticles, this simplified version of the character is used herein.

Cervical vertebrae

C126: Presacral neural spines, bifurcation: absent (0); present (1) (McIntosh, 1989; W02-85, 89; modified; Table S16).

Comments. Wilson (2002) divided this character into the different regions, where the bifurcation can be present. As a result, taxa with unbifurcated neural spines are coded several times for the same state. In the present analysis, presence of bifurcation and the first bifid element are treated as two different characters (see character 140).

C127: Number of cervical vertebrae: <13 (0); 14–15 (1); 16 or more (2) (McIntosh, 1990b; W98-37; modified; Table S17). Unordered.

Comments. The character is used in various versions in different phylogenetic analyses (Upchurch, 1998; Wilson & Sereno, 1998; Whitlock, 2011a), depending on their specific focus. Herein, the states are adjusted to fit the included taxa, excluding redundancy. Only one diplodocid specimen preserves a complete neck (Apatosaurus louisae CM 3018), and even here, the possibility of missing elements cannot be ruled out entirely, due to gaps between certain cervical vertebrae as they were found (McIntosh, 2005). A second specimen (Diplodocus carnegii CM 84) lacks the atlas, and seems otherwise complete, although the same concerns exist as for CM 3018 (McIntosh, 2005). However, as the more anterior and posterior elements in these cases fit well together, we followed McIntosh (2005) in assuming that no vertebra was lost at the position of these gaps in CM 84 and 3018. McIntosh (2005) suggested that Barosaurus had 16 cervical vertebrae, instead of 15 as Apatosaurus and Diplodocus. The assumption was primarily based on the fact that AMNH 6341 only has nine dorsal vertebrae, and that the neosauropod presacral column generally consists of 25 elements (McIntosh, 2005). Because none of the Barosaurus specimens preserves an entire neck, none of the Barosaurus OTUs can be coded for this character. The inability to code incomplete specimens might be circumvented by using additive binary characters (Upchurch, 1998). However, this would imply that the corresponding multistate character is continuous (Wilson, 2002), which means that the number of cervical vertebrae could not increase directly by more than one element during speciation. Given that the contrary is shown to be possible in dorsal and sacral vertebrae of mice (Wellik & Capecchi, 2003), it seems reasonable to argue that the same accounts for sauropod cervical vertebrae. The character is thus treated as unordered herein. This also indicates that ‘analysis 1’ of Mannion et al. (2012), where these characters are treated as unordered, should be preferred over ‘analysis 2.’

C128: Cervical vertebrae width to height ratio: less than 0.5 (0); 0.5–1.5 (1); more than 1.5 (2) (U04b-1; modified; Table S18). Unordered.

Comments. Upchurch, Tomida & Barrett (2004, p. 105) defined the ratio as follows: “Height is measured from the top of the neural spine to the ventral surface of the centrum. Width is defined as the distance between the distal tips of the diapophyses.” A third state was added (less than 0.5) because derived dicraeosaurids have a distinctly lower ratio compared to other flagellicaudatans. Given that outgroups are scored for state 1, this character is left unordered.

C129: Cervical pneumatopores (pleurocoels): absent (0); present (1) (McIntosh, 1990b; U95; Fig. 36).

Figure 36 Mid-cervical vertebra (CV ?10) of Galeamopus sp. SMA 0011 in right lateral view.

Note the pleurocoel typical for advanced eusauropods (C129-1), but highly subdivided (C171-2), the elongate posteroventral fossa present in diplodocines (C131-1), the anteriorly restricted pcdl (C135-0), in contrast to the more posteriorly reaching pcdl of Apatosaurus, the dorsally excavated parapophysis (C173-0), the large foramen connecting the pocdf and the spof (C191-1), and the accessory laminae connecting the podl and the sprl (C197-1), and the pcdl and the podl (C199-1). Abb.: apf, anterior pneumatic fossa; di, diapophysis; pap, parapophysis; pcdl, posterior centrodiapophyseal lamina; pocdf, postzygapophyseal centrodiapophyseal fossa; podl, postzygodiapophyseal lamina; poz, postzygapophysis; prdl, prezygodiapophyseal lamina; pre, pre-epipophysis; prz, prezygapophysis; pvf, posteroventral flange; sdf, spinodiapophyseal fossa; spof, spinopostzygapophyseal fossa; spol, spinopostzygapophyseal lamina; sprl, spinoprezygapophyseal lamina; tpol, interpostzygapophyseal lamina.

Comments. McIntosh (1990b) already used this character to distinguish advanced sauropods from the most basal forms, but Upchurch (1995) was the first to include it into a phylogenetic analysis.

C130: Cervical centra, internal pneumaticity: absent (0); present with single and wide cavities (1); present, with several small and complex internal cavities (2) (W98-102; modified by C12b-120; Fig. 37).

Figure 37 Cross-section of neosauropod cervical vertebrae.

Mid- to posterior cervical vertebrae cross-section of Supersaurus vivianae WDC DMJ-021 (A; modified from Lovelace, Hartman & Wahl, 2007), and Brachiosaurus sp. BYU 12866 (B; modified from Wedel, 2009). Sections at base of diapophysis. Note the different internal pneumatic structure, with few but large cavities in Supersaurus (A; C130-1), in contrast to the many irregularly small fossa typical for titanosauriforms (B; C130-2). The differences shown here in cervical vertebrae apply as well for dorsal vertebrae (C228). Pictures scaled to the same centrum height. Abb: di, diapophysis; nc, neural canal; ns, neural spine; pl, pleurocoel.

Comments. Introduced as a character by Wilson & Sereno (1998), only Wedel, Cifelli & Sanders (2000) and Wedel (2003) analyzed the distribution of this feature in detail. Carballido et al. (2012b) divided the original character, which did not discriminate between cervical and dorsal vertebrae (Wilson & Sereno, 1998).

C131: Cervical vertebrae, small fossa on posteroventral corner: absent (0); shallow, anteroposteriorly elongate fossa present, posteroventral to pleurocoel (1) (W11-83; Fig. 36).

Comments. Kaatedocus siberi SMA 0004, AMNH 7530, and the apatosaurines YPM 1980 and AMNH 460 have shallow depressions at the same place, but they do not create distinct fossae as in Barosaurus or Diplodocus (see Hatcher, 1901; McIntosh, 2005), and are thus coded as plesiomorphic.

C132: Cervical centra, midline keels on ventral surface: prominent and plate-like (0); reduced to low ridges (1) (U98-83; modified; Fig. 38).

Figure 38 Flagellicaudatan mid- to posterior cervical vertebrae.

Mid- to posterior cervical vertebrae of Dicraeosaurus hansemanni MB.R.4886 (A; photo by J Harris), Kaatedocus siberi SMA 0004 (B), and Barosaurus lentus YPM 429 (C) in ventral view (anterior to the top). Note the different developments of the ventral keels (prominent in Dicraeosaurus, A, C132-0; shallow, single in Kaatedocus, B, C132-1 and 175-0; double in Barosaurus, C, C175-1), the ventral sulcus typical for diplodocines (B, C; C133-1), the pneumatic foramina accompanying the ventral keel in Dicraeosaurus (A; C176-1), the posteroventral flanges (C179-1), and the numerous accessory laminae subdividing the prezygapophyseal centrodiapophyseal fossa in Barosaurus (C; C184-2). Vertebrae scaled to same centrum length. Abb: acdl, anterior centrodiapophyseal lamina; CR, cervical rib; di, diapophysis; pap, parapophysis; pcdl, posterior centrodiapophyseal lamina; podl, postzygodiapophyseal lamina; poz, postzygapophysis, prdl, prezygodiapophyseal lamina; prz, prezygapophysis; pvf, posteroventral flange.

Comments. Because the presence or absence is already coded in subsequent characters, the complete absence is here excluded from the original character description (Upchurch, 1998), and taxa without ventral ridges are scored as unknown.

C133: Cervical vertebrae, longitudinal sulcus on ventral surface: absent (0); present (1) (U95, U98-84; Fig. 38).

Comments. Due to the lateroventral projecting cervical parapophyses of Apatosaurus, cervical vertebrae of this genus have a concave anterior portion of the ventral surface. However, this is the case in almost all sauropod taxa, and therefore only specimens with transversely concave ventral surfaces throughout the entire length of the centrum are herein scored as apomorphic.

C134: Cervical vertebra, posterior projection on transverse processes: present (0); absent (1) (R09-78; polarity reversed; Fig. 39).

Figure 39 Diplodocid mid- to posterior cervical vertebrae.

Mid- to posterior cervical vertebrae of Apatosaurus ajax YPM 1860 (A; traced from a photo by M Taylor), and Kaatedocus siberi SMA 0004 (B; CV 13, traced from Tschopp & Mateus, 2013b) in dorsal view (anterior to the top). Note the triangular posterior projection on the diapophysis in Kaatedocus (B; C134-1), the transversely compressed (B; C142-0) in contrast to rounded (A; C142-1) neural spine summits, the transverse sulcus accompanying the prezygapophyseal facet posteriorly in Kaatedocus (B; C195-1), the anterior bulge of the sprl, just below the spine summit, characterizing most diplodocines (B; C196-1), and the median tubercle visible in Apatosaurus (A; C210-1). Abb.: bns, bifid neural spine; CR, cervical rib; di, diapophysis; epi, epipophysis; pcdl, posterior centrodiapophyseal lamina; podl, postzygodiapophyseal lamina; prdl, prezygodiapophyseal lamina; prz, prezygapophysis; spol, spinopostzygapophyseal lamina; sprl, spinoprezygapophyseal lamina; tpol, interpostzygapophyseal lamina; tprl, interprezygapophyseal lamina. Vertebrae scaled to same total length.

Comments. A distinct, triangular posterior projection marks the transverse process of Spinophorosaurus and many diplodocines. Posteriorly convex transverse processes are not considered projections. Due to reduced taxon sampling, the character polarity of the original version (Remes et al., 2009) was inverted here.

C135: Cervical vertebrae, posterior extension of posterior centrodiapophyseal lamina: is anteriorly restricted (0); reaches below posterior end of neural canal (1) (New; Figs. 36 and 40).

Figure 40 Cervical vertebra 11 of diplodocids.

Cervical vertebra 11 of Apatosaurus louisae CM 3018 (A; modified from Gilmore, 1936) and Diplodocus carnegii CM 84 (B; modified from Hatcher, 1901) in left (A) and right (B) lateral view. Note the posteriorly extending posterior centrodiapophyseal lamina in Apatosaurus (A; C135-1), the anteriorly restricted pneumatic foramen typical for most apatosaurs (A; C172-1), the pre-epipophysis (A; C181-1), the subdivided prezygapophyseal centrodiapophyseal fossa, characterizing A. louisae (A; C184-1), the posteriorly expanded interpostzygapophyseal lamina of Diplodocus (B; C190-1), the posteriorly restricted prezygapophysis of A. louisae (A; C194-1), compared to the state in Diplodocus, where it reaches the anterior edge of the condyle (B; C194-0), the vertical accessory spinal lamina marking Diplodocus (B; C203-1), the different positions of the cervical ribs (ventrally projecting, A, C216-1; or level with centrum, B, C216-0), and the absence (A; C219-1) or presence (B; C219-0) of the anterior process of the cervical rib. Vertebrae scaled to same posterior cotyle height. Abb: apf, anterior pneumatic fossa; CR, cervical ribs; podl, postzygodiapophyseal lamina; poz, postzygapophysis; ppf, posterior pneumatic fossa; prdl, prezygodiapophyseal lamina; pvfo, posteroventral fossa; spol, spinopostzygapophyseal lamina; sprl, spinoprezygapophyseal lamina.

Comments. Apatosaurus specimens appear to have a consistently more developed pcdl compared to Diplodocinae. The only apatosaur specimen with an anteriorly restricted pcdl is the juvenile holotype of Elosaurus parvus, CM 566. However, because the development of vertebral laminae has previously been linked with ontogeny (Schwarz, Frey & Meyer, 2007b; Carballido & Sander, 2014), the anteriorly restricted pcdl in CM 566 might be an ontogenetic feature. Articulated cervical series (e.g., Apatosaurus louisae CM 3018, Diplodocus carnegii CM 84, Kaatedocus siberi SMA 0004) show that this character is stable throughout the column, and can thus be used in all cervical sections.

C136: Cervical vertebrae, short second posterior centrodiapophyseal lamina ventral to the one uniting with the dorsal shelf of the diapophysis: absent (0); present (1) (New; Fig. 41).

Figure 41 Cervical vertebra 6 of neosauropods.

Cervical vertebra 6 of Australodocus bohetii MB.R.2455 (A) and Galeamopus sp. SMA 0011 (B) in left (A) and right (B) lateral view. Note the short second pcdl in Australodocus (A; C136-1), the foramen piercing the podl (A; C137-1), the projection formed by the epipophysis (B; C138-1), the low (A; C164-0), and high (B; C164-1) neural spines, and the cervical rib, which is slightly longer than the centrum in Galeamopus (B; C215-1). Abb.: acdl, anterior centrodiapophyseal lamina; apf, anterior pneumatic fossa; cpol, centropostzygapophyseal lamina; cprl, centroprezygapophyseal lamina; CR, cervical rib; naf, neural arch foramen; pcdl, posterior centrodiapophyseal lamina; podl, postzygodiapophyseal lamina; poz, postzygapophysis; ppf, posterior pneumatic fossa; prz, prezygapophysis; spol, spinopostzygapophyseal lamina; sprl, spinoprezygapophyseal lamina; tpol, interpostzygapophyseal lamina. Vertebrae scaled to the same centrum length.

Comments. A short accessory pcdl appears to be linked with the bifurcation of the pcdl in more posterior elements in SMA 0011. However, a bifurcated pcdl also occurs in some apatosaur specimens, which do not have an additional pcdl in more anterior elements (e.g., UW 15556; Gilmore, 1936), and therefore, these morphologies are treated as independent characters.

C137: Cervical vertebrae, foramen on dorsal side of postzygodiapophyseal lamina, just anterior to base of neural spine process: absent (0); present (1) (Remes, 2007; Fig. 41).

Comments. Distinct foramina in the sdf are usually considered typical for brachiosaurids, and their presence in Australodocus was therefore one of the reasons why Whitlock (2011c) reinterpreted Australodocus bohetii as a titanosauriform, instead of a diplodocine as initially proposed (Remes, 2007). However, Barosaurus sometimes shows small foramina in similar positions (YPM 429, E Tschopp, pers. obs., 2011), but they are usually less prominent. The putative juvenile Brachiosaurus specimen SMA 0009 does not have such foramina, but because the development of pneumatic structures appears to be ontogenetically controlled (Schwarz et al., 2007; Carballido et al., 2012a), this might be explained as such.

C138: Cervical vertebrae, epipophysis: reduced or absent (0); pronounced, forming a distinct projection above the postzygapophysis (1) (R09-80; modified; Fig. 41).

C139: Cervical vertebrae, pneumatized epipophyses: absent (0); present (1) (New; Fig. 42).

Figure 42 Diplodocine mid- to posterior cervical vertebrae.

Mid- to posterior cervical vertebrae of Barosaurus lentus YPM 429 (A) and Diplodocus carnegii CM 84 (B) in left posterolateral (A) and left dorsolateral view (B). Note the differently pneumatized epipophyses (C139-1), the transversely compressed epipophysis (B; C202-1), and the horizontal ridge below the neural spine summit in Diplodocus (B; C205-1). The cervical vertebra of B. lentus is partly covered by matrix and plaster. Abb.: apf, anterior pneumatic fossa; CR, cervical rib; pap, parapophysis; pcdl, posterior centrodiapophyseal lamina; ppf, posterior pneumatic fossa; prdl, prezygodiapophyseal lamina; prz, prezygapophysis; pvf, posteroventral flange; pvfo, posteroventral fossa; spol, spinopostzygapophyseal lamina; sprl, spinoprezygapophyseal lamina. Vertebrae scaled to the same posterior cotyle height.

Comments. The pneumatic foramen can be situated anteriorly as in Diplodocus carnegii (CM 84, 94, E Tschopp, pers. obs., 2011), or posteriorly as in Barosaurus lentus YPM 429 (E Tschopp, pers. obs., 2011).

C140: Cervical neural spines, bifurcation, if present, anterior extension within column includes: CV 3 (0); all mCV (1); posterior mCV (2); only pCV (3) (R93-9; modified; Table S16). Ordered.

Comments. Taxa with unbifurcated neural spines are scored as unknown. The subdivision into anterior, mid-, and posterior cervical vertebrae depends on the number of elements in the column (Table 3). Absolute numbers other than CV 3, which is the first postaxial cervical element, would thus be misleading and are avoided here.

C141: Cervical vertebrae, unbifurcated neural spines in anterior/posterior view: with parallel lateral edges or converging (0); distal end expanded laterally (1) (New; Fig. 43).

Figure 43 Cervical vertebra 5 of flagellicaudatans.

Cervical vertebra 5 of Suuwassea emilieae ANS 21122 (A) and Kaatedocus siberi SMA 0004 (B; modified from Tschopp & Mateus, 2013b) in anterior view. Note the transversely widening (A; C141-1) instead of straight (B; C141-0) neural spine, and the presence of a prespinal lamina in Kaatedocus (B; C161-1). The neural spine of Suuwassea (A) is not bifurcated, but broken (as indicated by the dashed line). Abb.: cprl, centroprezygapophyseal lamina; pap, parapophysis; poz, postzygapophysis; prdl, prezygodiapophyseal lamina; prz, prezygapophysis; sprl, spinoprezygapophyseal lamina. Vertebrae scaled to the same anterior condyle length.

Comments. The real distribution of this character within Diplodocidae is difficult to assess to date, because there are only a few specimens reported that preserve complete neural spines of anterior, unbifurcated neural spines.

C142: Cervical vertebrae, summits of bifid neural spines: are laterally compressed (0); are rounded (1) (U04b-7; Fig. 39).

Comments. The derived state of this character is shared by some apatosaur specimens and Suuwassea. The spine summits in most other taxa with bifurcated spines are generally anteroposteriorly elongate and transversely compressed, resulting in narrow sheets of bone. In Suuwassea as well as in some apatosaur specimens, the lateral edge of the spine summit is distinctly convex, producing a semi-circular outline. Some other taxa (e.g., Kaatedocus; Tschopp & Mateus, 2013b) have medial ridges connecting the summit with the base, but these are always relatively shallow, and do not form rounded outlines. Taxa with unbifurcated neural spines are scored as unknown.

C143: Proatlas, distal end: broadly rounded (0); narrow and elongate, almost pointed (1) (New; Fig. 44).

Figure 44 Diplodocine proatlases.

Proatlas of ?Kaatedocus SMA P29-1 (A) and Galeamopus sp. SMA 0011 (B) in medial view, illustrating the broad (A; C143-0) and narrow distal tips (B; C143-1). Abb.: pas, proximal articular surface. Scaled to the same articular surface height.

C144: Atlantal intercentrum, anteroventral lip: absent, anterior edge of intercentrum straight in lateral view (0); present, anterior edge of intercentrum concave (1) (W02-79; modified; Fig. 45).

Figure 45 Neosauropod atlantes.

Atlas of Camarasaurus sp. UUVP 10070 (A; modified from Madsen, McIntosh & Berman, 1995), and Galeamopus sp. AMNH 969 (B) in posterior (left) and right lateral view (right, A shows left side reversed). Note the distinct anteroventral lip characterizing diplodocids (B; C144-1), and the foramen between the posterior ventrolateral processes in AMNH 969 (B; C145-1). Abb.: ncs, neurocentral synchondrosis; pvlp, posterior ventrolateral process. Scaled to the same centrum height.

Comments. Initially regarded as flagellicaudatan synapomorphy (Wilson, 2002), an anteroventral lip is now known to occur in Mongolosaurus as well (Mannion, 2011). Following the original description of the character states (Wilson, 2002: intercentrum shape in lateral view: rectangular or ventrally longer than dorsally), Camarasaurus and other non-flagellicaudatan taxa also would have to be scored as apomorphic. However, they do not show a distinct anteroventral lip, resulting in a strongly concave anterior edge of the intercentrum, when seen in lateral view.

C145: Atlantal intercentrum, ventral surface, foramen between posterior ventrolateral processes: absent (0); present (1) (New; Fig. 45).

C146: Atlantal neurapophyses, anteromedial process: weakly developed (0); well-developed and distinct from posterior wing (1) (New; Fig. 46).

Figure 46 Diplodocid atlantal neurapophyses.

Neurapophyses of Apatosaurus louisae CM 3018 (A; modified from Gilmore, 1936), Kaatedocus siberi SMA 0004 (B; traced from 3D model provided by G Dzemski), and Galeamopus sp. SMA 0011 (C) in lateral (A; left side reversed), and dorsolateral view (B, C). Note the weak (B; C146-0) in contrast to well-developed medial process (C; C146-1), the subtriangular lateral spur in Galeamopus (C; C147-1), the different shapes of the distal process (tapering, B, C148-0; wide, C, C148-1), and the foramen characterizing A. louisae (A; C149-1). Abb.: dip, distal process; ncs, neurocentral synchondrosis. Scaled to the same anteroposterior length.

Comments. The anteromedial process corresponds to the prezygapophyses of more posterior elements. It articulates with the posterior end of the proatlas. In Kaatedocus, this process is relatively short transversely, and curves gradually into the posterior process, whereas in SMA 0011 and AMNH 969 the anteromedial process is distinct and at least as wide transversely as long anteroposteriorly.

C147: Atlantal neural arch, small subtriangular, laterally projecting spur at base: absent (0); present (1) (New; Fig. 46).

Comments. When present, this spur is located at the base of the neurapophysis, opposite the anteromedial process, and much smaller. It is also present in some, but not all, Camarasaurus specimens (Ikejiri, 2004).

C148: Atlantal neurapophyses, posterior wing: gradually tapering along its length (0); of subequal width along most of its length (1) (New; Fig. 46).

Comments. The posterior wing of the neurapophysis articulates with the prezygapophysis of the axis.

C149: Atlantal neurapophyses, posterior wing: without foramen (0); with foramen (1) (Wilson, 2002; W11-85; wording modified; Fig. 46).

Comments. Wilson (2002) proposed the presence of such a foramen as an autapomorphy of Apatosaurus, and it was included as character in the phylogenetic analysis of Whitlock (2011a). Due to the small number of preserved atlantal neurapophyses, only one specimen can currently be positively assigned to the apomorphic state (Apatosaurus louisae CM 3018). It could thus also represent a species autapomorphy, instead of being valid for the entire genus.

C150: Axial centrum, pneumatic fossae on ventrolateral edges, posterior and adjacent to parapophyses: absent (0); present (1) (New; Fig. 47).

Figure 47 Axis of Diplodocus carnegii CM 84 in posterolateral view.

Note the pneumatic slot-like fossa posterior to the parapophysis (C150-1), and the presence of a postspinal lamina (C152-1). Abb.: at, atlas; CV 3, cervical vertebra 3; pap, parapophysis; pl, pleurocoel; poz, postzygapophysis; prdl, prezygodiapophyseal lamina; prsl, prespinal lamina; spol, spinopostzygapophyseal lamina.

Comments. Many specimens have a well-developed median keel on their ventral surfaces. In lateral view, this sometimes appears as a bifurcation of the ventrolateral edge, although this is not the case. The apomorphic state of the character proposed herein only includes fossae bordered by ridges that originate at the parapophysis anteriorly.

C151: Axis, prespinal lamina: of constant width (0); developing a transversely expanded, knob-like tuberosity at its anterior end (1) (New; Fig. 48).

Figure 48 Axis of Galeamopus sp. AMNH 969 in dorsal (top), right lateral (bottom left), and anterior (bottom right) view.

Note the anteriorly expanded prespinal lamina (C151-1), and the anteriorly restricted neural spine summit (C153-2). Abb.: di, diapophysis; epi, epipophysis; pap, parapophysis; pcdl, posterior centrodiapophyseal lamina; pl, pleurocoel; poz, postzygapophysis; prsl, prespinal lamina; prz, prezygapophysis. Scale bar = 10 cm.

C152: Axis, postspinal lamina: absent (0); present (1) (Harris & Dodson, 2004; Fig. 47).

C153: Axis neural spine: projects beyond posterior border of centrum (0); terminates in front of or at posterior border of centrum (1); is restricted anterior to postzygapophyseal facets (2) (New; Fig. 48). Ordered.

Comments. Due to intermediate morphologies, this character is treated as ordered.

C154: Anterior cervical vertebrae, total height/centrum length ratio: <0.9 (0); 0.9–1.2 (1); >1.2 (usually around 1.5) (1) (W11-87; modified; Table S19). Unordered.

Comments. Total height is herein measured between the ventral-most expansion of the centrum (usually the parapophysis or posterior cotyle) and the highest point of the neural spine. A third state was added in order to distinguish apatosaurs from Diplodocus. Given the high amount of changes in ratios during evolution, as indicated by the analysis, the character is left unordered.

C155: Cervical vertebrae 2 and 3, centrum length: moderate length increase, CV3 <1.3 × CV 2 (0); length increases considerably CV 3 at least 1.3 × CV 2 (1) (Russell & Zheng, 1993; Table S20).

Comments. Even though this does not seem to follow higher-level taxonomy, there are two groups with ratios well separated from each other (Table S20). The state boundaries are therefore set in order to distinguish between these two groups.

C156: Anterior cervical vertebrae, posterior edge of anterior condyle: anteriorly inclined (0); posteriorly inclined (1) (New; Fig. 49).

Figure 49 Cervical vertebra 4 of flagellicaudatans.

Cervical vertebra 4 of Dicraeosaurus hansemanni MB.R.4886 (A) and Galeamopus sp. SMA 0011 (B) in right lateral view. Note the differently inclined posterior border of the anterior condyle (A, C156-0; B, C156-1), the subdivision of the pleurocoel in Galeamopus (B; C157-1), which is absent in anterior cervical vertebrae of Dicraeosaurus (A; C157-0), the anterior pneumatic fossa that extends onto the parapophysis (B; C158-0), the presence of a prespinal lamina in Galeamopus (B; C161-1), and the posteriorly projecting spur on the dorsal edge of the posterior process of the cervical rib of Dicraeosaurus (A; C217-1). Abb.: acdl, anterior centrodiapophyseal lamina; apf, anterior pneumatic fossa; cprl, centroprezygapophyseal lamina; CR 3, cervical rib 3; CV 3, cervical vertebra 3; epi, epipophysis; naf, neural arch foramen; pl, pleurocoel; podl, postzygodiapophyseal lamina; poz, postzygapophysis; ppf, posterior pneumatic fossa; prdl, prezygodiapophyseal lamina; prz, prezygapophysis; spol, spinopostzygapophyseal lamina; sprl, spinoprezygapophyseal lamina. Vertebrae scaled to the same cotyle height.

Comments. This character is strictly applicable to anterior cervical vertebrae. In SMA 0011, which has apomorphic anterior vertebrae, CV 6 and more posterior elements show the usual anteriorly inclined edge.

C157: Anterior cervical centra, pleurocoels: single (0); subdivided (1) (New; Fig. 49).

Comments. The subdivision of the pleurocentral cavity is sometimes regarded as ontogenetically controlled (Schwarz, Frey & Meyer, 2007b; Carballido & Sander, 2014). However, given that the completely mature anterior cervical vertebrae (sensu Carballido & Sander, 2014) of the Kaatedocus siberi holotype SMA 0004 have undivided pleurocoels, in contrast to the still immature vertebrae of other specimens like SMA 0011 (see above), at least some taxonomic differences are likely.

C158: Anterior cervical vertebrae, pleurocoel extending onto dorsal surface of parapophysis: absent (0); present (1) (U98-86; modified by W11-88; polarity reversed; Fig. 49).

Comments. Upchurch (1998) distinguished between continuous extensions or fossae that are separated from the main anterior pneumatic fossa or pleurocoel by a transverse ridge. The latter distinction was abandoned by Whitlock (2011a), who instead divided the character into the different regions (anterior and mid- and posterior cervical vertebrae, see below). Character polarity was herein reversed because basal outgroups used in the present analysis do have expanded pleurocoels.

C159: Anterior cervical vertebrae, longitudinal ridge on ventral surface: present (0); absent (1) (U98-83; modified).

Comments. The ventral ridge (if present) can have various morphologies in diplodocid specimens, which is accounted for in other characters of this analysis. In addition to the original version of Upchurch (1998; character 132 herein), a strict presence–absence character was included for both anterior and mid- and posterior cervical vertebrae in the present analysis. The subdivision is necessary because in some specimens, a ventral keel only occurs in anterior elements (ANS 21122, SMA 0011, Tate-001). This indicates that incomplete necks without ventral keels on posterior cervical vertebrae might still bear midline ridges anteriorly. For the various developments of the keels see Fig. 38, which shows mid- and posterior cervical vertebrae, but the morphology is the same in anterior elements.

C160: Anterior cervical vertebrae, paired pneumatic fossae on ventral surface: absent (0); present (1) (W11-89).

Comments. Like the ventral keel, the paired pneumatic foramina are sometimes restricted to the anterior cervical vertebrae (e.g., in SMA 0011, see above). Whereas the presence of paired pneumatic foramina imply the presence of a ventral keel, this does not apply the other way around, as shown by the anterior cervical vertebrae of Kaatedocus SMA 0004 (Tschopp & Mateus, 2013b). The characters are therefore retained as independent. The morphology of the foramina is equal in anterior and mid- and posterior cervical vertebrae, where present (see Fig. 38). In our analysis, paired pneumatic foramina only occur at the anterior end of the ventral surfaces. However, given that paired fossae in the posterior cervical vertebra of Dinheirosaurus lourinhanensis ML 414 occur at the posterior end of the ventral surface, we refrained from restricting the character definition to anteriorly placed foramina.

C161: Anterior cervical vertebrae, prespinal lamina: absent (0); present (1) (C12b-121; Figs. 43 and 49).

Comments. In some diplodocid specimens, it appears that the prespinal lamina in undivided vertebrae gives rise to the median tubercle in divided, more posterior elements. However, given the presence of a prespinal lamina in Camarasaurus (Madsen, McIntosh & Berman, 1995), which does not have a median tubercle between bifurcated neural spines, these two characters should be treated as independent.

C162: Anterior and mid-cervical centra, pleurocoel pierced by one or two large, rounded foramina around centrum midlength: absent (0); present (1) (New; Fig. 50).

Figure 50 Cervical vertebra 6 of flagellicaudatans.

Cervical vertebra 6 of Dicraeosaurus hansemanni MB.R.4886 (A) and Galeamopus sp. SMA 0011 (B) in right lateral view. Note the large, rounded pneumatic foramen marking the anterior end of the posterior pneumatic fossa in Galeamopus (B; C162-1), the elongate foramen in the neural spine (B; C165-1), the right (A; C170-1), or acute angles (B; C170-0) between the spinopostzygapophyseal and the postzygodiapophyseal laminae, and the vertical (A; C218-0) or posteriorly inclined tuberculum (B; C218-1). Abb.: acdl, anterior centrodiapophyseal lamina; apf, anterior pneumatic fossa; cprl, centroprezygapophyseal lamina; CR, cervical rib; CV 5, cervical vertebra 5; pap, parapophysis; pl, pleurocoel; poz, postzygapophysis; ppf, posterior pneumatic fossa; prz, prezygapophysis; pvf, posteroventral flanges; sprl, spinoprezygapophyseal lamina. Scaled to the same cotyle height.

Comments. Such a foramen is absent in the anterior-most elements, but very distinct in CV 5 or 6 of SMA 0011, whereas it disappears again by CV 8 or 9. In SMA 0011, these foramina are situated at the anterior end of the posterior pneumatic fossa. Taxa where CV 5 to 7 or 8 are not preserved, and other elements do not show such a development, are scored as unknown. Similarly distinct, rounded foramina are only present in Supersaurus (Lovelace, Hartman & Wahl, 2007), and Australodocus (Remes, 2007; Whitlock, 2011c).

C163: Anterior and mid-cervical vertebrae, spinoprezygapophyseal lamina, development at base of prezygapophyseal process: distinct (0); reduced to broad ridge or totally interrupted (1). (T13-103; wording modified; Fig. 51).

Figure 51 Diplodocine mid-cervical vertebrae.

Mid-cervical vertebrae of Kaatedocus siberi SMA 0004 (A; CV 10, modified from Tschopp & Mateus, 2013b) and Diplodocus carnegii CM 84 (B; CV 8) in right lateral (A) and left laterodorsal view (B). Note the reduced spinoprezygapophyseal lamina (B; C163-1), the pre-epipophysis (C181-1), which is anteriorly expanded in K. siberi (A; C167-1), the distinct fossa posterolaterally to the prezygapophysis (A; C183-1), which is absent in CM 84 (B; C183-0), and the short cervical ribs (B; 214-1). Abb.: apf, anterior pneumatic fossa; cprl, centroprezygapophyseal lamina; CR, cervical rib; CV 7, cervical vertebra 7; podl, postzygodiapophyseal lamina; poz, postzygapophysis; ppf, posterior pneumatic fossa; prz, prezygapophysis; pvf, posteroventral flanges; spol, spinopostzygapophyseal lamina; sprl, spinoprezygapophyseal lamina. Not to scale.

Comments. The character was clarified in order to specify that the reduction to a ridge and the interruption of the sprl are restricted to the base of the prezygapophyseal process. Otherwise one could understand that the reduction to a ridge would affect the entire sprl, which is not what was intended to code for with this character initially.

C164: Anterior and mid-cervical neural spines height: high (project well above the level of postzygapophyses) (0); low (terminates level with postzygapophyses) (1) (U04b-8; modified; Fig. 41).

Comments. This character is similar to character 168. It was added because it includes anterior cervical vertebrae, which are different in height among diplodocids and within Diplodocinae, and because it would have differing state boundaries, if it would be treated numerically.

C165: Anterior and mid-cervical neural spines, dorsoventrally elongate coel on lateral surface: absent (0); present (1) (M12-99; modified; Fig. 50).

Comments. The presence of a dorsoventrally elongate fossa in the spinodiapophyseal fossa is usually used as derived character for posterior cervical vertebrae only (Mannion et al., 2012). However, there are differences in anterior and mid-cervical neural arches as well, which appear to be phylogenetically significant.

C166: Mid-cervical centra, anteroposterior length/height of posterior face: 2.5–3.2 (0); 3.3–4.4 (1); 4.5+(2) (U95; modified; Table S21).

Comments. Elongation index as used herein is measured following the protocol of Wilson & Sereno (1998: total centrum length/height posterior cotyle). The mean elongation index is used for this metric. Tornieria specimen k is scored ‘2’ because the centrum length to width ratio is very high (5.4; Remes, 2006), and thus a high EI as used herein can be expected with confidence.

C167: Mid-cervical pre-epipophyses anterior extreme: about the same as prezygapophyseal facet (0); projects considerably anterior to articular facet, forming a distinct spur (1) (Sereno et al., 1999; Fig. 51).

Comments. A distinct anterior extension of the pre-epipophysis was used as an autapomorphy for Australodocus bohetii within Diplodocidae (Remes, 2007). However, it has been shown to be present in Kaatedocus as well as in some non-diplodocid sauropods (Sereno et al., 1999; Ksepka & Norell, 2006; Tschopp & Mateus, 2013b). Taxa without pre-epipophyses are scored as unknown.

C168: Mid-cervical neural spine height: considerably shorter than height of neural arch, <0.45 (0); subequal to height of neural arch, 0.45–1.6 (1); considerably higher than neural arch, >1.6 (2) (R05-69; modified; Table S22). Unordered.

Comments. Neural arch height is measured in a vertical line from the centrum to an imaginary line connecting the dorsal edges of the postzygapophyses, and neural spine height from dorsal edge of the postzygapophyses to the spine summit. The centrum is oriented such that the ventral floor of the neural canal is horizontal. The majority of the ratios were measured from photographs or figures in lateral view. As exemplified by CV 6 of Suuwassea ANS 21122, this approach can yield major differences depending on slight changes in perspective (or left and right lateral views; CV 6 of ANS 21122 has ratios ranging from 0.91–1.27; Table S22). Although such differences are partly avoided by using mean ratios, it would be unwise to use closely spaced numerical state boundaries in this case. Therefore, only two steps were regarded as sufficiently objective and phylogenetically significant. The character was left unordered due to diverging evolutionary trends.

C169: Mid-cervical neural spines, orientation: vertical (0); anteriorly inclined (1) (R05-68; Fig. 52).

Figure 52 Cervical vertebra 8 of flagellicaudatans.

Cervical vertebra 8 of Dicraeosaurus hansemanni MB.R.4886 (A) and Kaatedocus siberi SMA 0004 (B) in right lateral view. Note the different inclinations of the neural spine (C169), and the small tuberosity marking the anterodorsal corner of the centrum in Kaatedocus (B; C178-1). Abb.: cpol, centropostzygapophyseal lamina; CV 7, cervical vertebra 7; epi, epipophysis; mt, median tubercle; podl, postzygodiapophyseal lamina; poz, postzygapophysis; pre, pre-epipophysis; prz, prezygapophysis; pvf, posteroventral flanges; spol, spinopostzygapophyseal lamina; sprl, spinoprezygapophyseal lamina. Scaled to the same cotyle height.

Comments. The neural spine is interpreted to be anteriorly inclined, when the anterior end of the summit reaches further anterior than the posterior-most point of the sprl.

C170: Mid-cervical vertebrae, angle between postzygodiapophyseal and spinopostzygapophyseal laminae: acute (0); right angle (1) (R05-67; Fig. 50).

Comments. Angles are measured between lines connecting the posterior-most point of podl and spol (often the epipophyses) with their opposing ends.

C171: Mid- and posterior cervical centra, pleurocoels: single without division (0) divided by a bone septum, resulting in an anterior and a posterior lateral excavation (1); divided in three or more lateral excavations, resulting in a complex morphology (2) (C12b-115; modified; Fig. 36).

Comments. The original character (Carballido et al., 2012b) includes a fourth character state, which describes the shallow posterior pneumatic fossa. As such, it overlaps with character 172, introduced by Whitlock (2011a). Furthermore, subdivision of the pleurocoel is not correlated with the depth of the single pneumatic fossae in diplodocids. Therefore, the fourth state was omitted here.

C172: Mid- and posterior cervical vertebrae, pneumatization of lateral surface of centra: large, divided pleurocoel over approximately half of centrum (0); reduced, large fossa but sharp-bordered coel, if present, restricted to area above parapophysis (1) (W11-81; Fig. 40).

Comments. Taxa with single pleurocoels are scored as unknown.

C173: Mid- and posterior cervical vertebrae, pleurocoel extending onto dorsal surface of parapophysis: present (0); absent (1) (U98-86; modified by W11-95; Fig. 36).

C174: Mid- and posterior cervical vertebrae, longitudinal ridge on ventral surface: present (0); absent (1) (New).

C175: Mid- and posterior cervical vertebrae, ventral keel: single (0); bifid, connects posterolaterally to the ventrolateral edges of the centrum (1) (New; Fig. 38).

Comments. Taxa without ventral keels are scored as unknown.

C176: Mid- and posterior cervical vertebrae, paired pneumatic fossae on ventral surface, separated by ventral midline keel: absent (0); present (1) (New; Figs. 38 and 53).

Figure 53 Cervical vertebra 14 of “Dinheirosaurus” lourinhanensis ML 414 in lateroventral view.

Note the particular ventral morphology with posteriorly located paired pneumatic foramina (C176-1), lateral grooves posterior to the parapophyses (C177-1), a posteriorly restricted ventral keel (C193-1), and the elongated lateral spinal cavity (C204-1). Abb.: acdl, anterior centrodiapophyseal lamina; CR, cervical rib; pap, parapophysis; poz, postzygapophysis; prz, prezygapophysis; pvf, posteroventral flanges; pvfo, posteroventral fossa.

Comments. Usually, these fossae are situated anteriorly between the parapophyses, separated by a ventral keel. Some apatosaur specimens (e.g., YPM 1861, E Tschopp, pers. obs., 2011) show paired pneumatic fossae located posterior to the parapophyses, facing ventrolaterally, and not separated by a keel. This morphology is considered different, and accounted for in character 177.

C177: Mid- and posterior cervical vertebrae, lateral edge posterior to parapophysis: continuous (0); marked by a deep groove extending anteroposteriorly along the edge (1) (New; Fig. 53).

Comments. This groove results in the presence of two distinct laminae or ridges extending from the parapophysis posteriorly.

C178: Mid- and posterior cervical centra, rugose tuberosity on anterodorsal corner of lateral side: absent (0); present (1) (T13-120; modified; Fig. 52).

Comments. The character description was extended to mid-cervical vertebrae in order to include Suuwassea emilieae. In the latter, the distinct rugose tubercles appear in mid-cervical vertebrae, whereas in Kaatedocus siberi, mid-cervical vertebrae only have very shallow tubercles. An additional character for serial variation is avoided because it could only be scored for these two taxa and would thus not be phylogenetically significant.

C179: Mid- and posterior cervical centra with longitudinal flanges in the lateroventral edge on the posterior part of the centrum: absent (0); present (1) (T13-113; Fig. 38).

Comments. These flanges are mainly responsible for the posterior portion of the ventral sulcus typical for diplodocines. However, some apatosaur specimens also have weak flanges, but no continuous ventral sulcus marking the ventral surface (BYU 1252-18531, NSMT-PV 20375 and UW 15556).

C180: Mid- and posterior cervical prezygapophyses, articular surfaces: flat (0); strongly convex transversely (1) (U95, U98-89; Fig. 54).

Figure 54 Cervical vertebra 11 of eusauropods.

Cervical vertebra 11 of Jobaria tiguidensis MNN TIG (A; traced from photo by J Carballido), Camarasaurus supremus AMNH 5671 (B; based on Osborn & Mook, 1921), and Diplodocus carnegii CM 84 (C; based on Hatcher, 1901) in anterior view. Note the straight (A; C180-0), in contrast to convex prezygapophyseal facet (C; C180-1), and the different morphologies of the centroprezygapophyseal lamina (single, A, C185-0; divided, and connecting to tprl, B, C185-1; divided with both branches connecting to prezygapophysis, C, C185-2). Abb.: di, diapophysis; nc, neural canal; pap, parapophysis; podl, postzygodiapophyseal lamina; poz, postzygapophysis; prdl, prezygodiapophyseal lamina; prz, prezygapophysis; spol, spinopostzygapophyseal lamina; sprl, spinoprezygapophyseal lamina. Scaled to the same condyle height.

C181: Mid- and posterior cervical vertebrae, pre-epipophysis: absent (0); present (1) (Remes, 2007; Figs. 40 and 51).

Comments. The pre-epipophysis is herein defined as a rugose, horizontal ridge laterally below the prezygapophyseal facet, which connects with the prdl anteriorly.

C182: Mid- and posterior cervical vertebrae, spinoprezygapophyseal lamina, anterior end: remains vertical, with the free edge facing dorsally (0); is strongly inclined laterally (sometimes roofing a lateral fossa in the prezygapophyseal process (1) (T13-117; modified; Fig. 55).

Figure 55 Cervical vertebra 12 of Kaatedocus siberi SMA 0004 in lateral anterodorsal view.

Note the laterally tilted anterior portion of the sprl (C182-1), the lateral fossa marking the anterior end of the spinodiapophyseal fossa (C183-1), and the transverse sulcus accompanying the prezygapophyseal facet posteriorly (C195-1). Abb.: cpol, centropostzygapophyseal lamina; CR, cervical rib; epi, epipophysis; poz, postzygapophysis; prdl, prezygodiapophyseal lamina; pre, pre-epipophysis; prz, prezygapophysis; spol, spinopostzygapophyseal lamina; sprl, spinoprezygapophyseal lamina. 3D digital model provided by G Dzemski.

Comments. At a first glance, it appears possible that this character is correlated with the occurrence of transversely convex prezygapophyseal facets. However, this is not the case, as can be seen in the several varying scores for these two characters.

C183: Mid- and posterior cervical neural arches, lateral fossae on the prezygapophysis process: absent (0); present (1) (Harris, 2006b; C12b-124; modified by T13-118; Figs. 51 and 55).

Comments. Where such a lateral fossa is present, it is dorsally roofed by a laterally tilted anterior end of the sprl. However, not all specimens with a laterally tilted lamina also bear these fossae, which justifies the use of two independent characters. The character was first used in a phylogenetic analysis by Carballido et al. (2012b), but was modified by Tschopp & Mateus (2013b) in order to include posterior cervical vertebrae as well.

C184: Mid- and posterior cervical vertebrae, prezygapophyseal centrodiapophyseal fossa: single cavity (0); subdivided into two cavities by a ridge (1); several accessory laminae subdivide the fossa into various smaller partitions (2) (Gilmore, 1936; U04b-2; modified; Figs. 38 and 40). Ordered.

Comments. A third state was added in order to be able to accurately code the holotype specimen of Barosaurus lentus (YPM 429), as well as a few other specimens. Two specimens coded as ‘0’ actually only preserve mid-cervical vertebrae (AMNH 7535, CM 3452, E Tschopp, pers. obs., 2011). It would thus be possible that more posterior elements of these cervical columns had subdivided prcdf. The character is treated as ordered, because an increase in lamination is thought to happen during ontogeny as well (Schwarz et al., 2007).

C185: Mid- and posterior cervical neural arches, centroprezygapophyseal lamina: single (0); dorsally divided, resulting in a lateral and medial lamina, the medial lamina being linked with interprezygapophyseal lamina and not with prezygapophysis (1); divided, resulting in presence of “true” divided centroprezygapophyseal lamina, dorsally connected to prezygapophysis (2) (U95; modified by C12b-127; Fig. 54).

Comments. Usually, taxa with “true” divided cprl also have a lamina connecting from the base of the cprl to the tprl.

C186: Mid- and posterior cervical transverse processes: posterior centrodiapophyseal lamina (pcdl) and postzygodiapophyseal laminae (podl) meet at base of transverse process (0); pcdl and podl do not meet anteriorly, postzygapophyseal centrodiapophyseal fossa extends onto posterior face of transverse process (1) (New; Fig. 56).

Figure 56 Cervical vertebra 12 of diplodocids.

Cervical vertebra 12 of Apatosaurus louisae CM 3018 (A; based on Gilmore, 1936), and Kaatedocus siberi SMA 0004 (B; based on Tschopp & Mateus, 2013b) in posterior view. Note the separated (A; C186-1) or connected pcdl and podl (B; C186-0), the divided (A; C189-1) or single cpol (B; C189-0), the accessory lamina in the postzygapophyseal centrodiapophyseal fossa (B; C198-1), and the tpol that connects directly (B; C201-0) or indirectly with the neural canal roof (A; C201-1). Abb.: CR, cervical rib; pcdl, posterior centrodiapophyseal lamina; podl, postzygodiapophyseal lamina; poz, postzygapophysis; prdl, prezygodiapophyseal lamina; spol, spinopostzygapophyseal lamina; tpol, interpostzygapophyseal lamina. Scaled to the same posterior cotyle height.

C187: Mid- and posterior cervical vertebrae, accessory horizontal lamina in center of spinodiapophyseal fossa, not connected with any surrounding laminae: absent (0); present (1) (New; Fig. 57).

Figure 57 Posterior cervical vertebra of Barosaurus lentus YPM 429 in right lateral view.

Note the short horizontal accessory lamina within the spinodiapophyseal fossa (C187-1), the anteriorly bifurcated posterior centrodiapophyseal lamina (C188-1), and the anteriorly restricted postzygapophyses (C200-1). Abb.: cpol, centropostzygapophyseal lamina; pap, parapophysis; podl, postzygodiapophyseal lamina; prdl, prezygodiapophyseal lamina; prz, prezygapophysis; pvf, posteroventral flanges; spol, spinopostzygapophyseal lamina; sprl, spinoprezygapophyseal lamina. Scale bar = 10 cm.

Comments. This accessory lamina could be a vestigial version of the epipophyseal-prezygapophyseal lamina (sensu Wilson, 2012) or the accessory lamina connecting the podl with the sprl (as used herein, following Carballido et al., 2012b). However, because no connection exists with any surrounding lamina, this cannot be definitely confirmed in the cases included here. The use of an independent character is thus preferred. The lamina is generally situated in the center of the sdf.

C188: Mid- and posterior cervical vertebrae, posterior centrodiapophyseal lamina: is single (0); bifurcates towards its anterior end (1) (U04b-5; wording modified; Fig. 57).

Comments. Evidence from SMA 0011 shows that the presence of anteriorly bifurcated pcdl sometimes are a precursor of entirely double pcdl (see above). However, because in various specimens only bifurcated and not entirely double pcdl exist, the character was retained as independent from the one describing the single or double pcdl (see character 136).

C189: Mid- and posterior cervical vertebrae, centropostzygapophyseal lamina (cpol): single (0); divided, with medial part contacting interpostzygapophyseal lamina (1) (C12b-128; Fig. 56).

C190: Mid- and posterior cervical neural arches, interpostzygapophyseal lamina projects beyond the posterior margin of the neural arch (including the centropostzygapophyseal lamina), forming a prominent subrectangular projection in lateral view: absent (0); present (1) (D12-26; modified by M13-131; Fig. 40).

Comments. A reduced subrectangular projection is present in mid-cervical vertebrae of Supersaurus WDC DMJ-021. Generally, the development of this feature increases in more posterior elements (e.g., in Diplodocus carnegii CM 84; Hatcher, 1901). Supersaurus WDC DMJ-021 was thus scored as apomorphic, although it is not prominent in the preserved vertebrae. On the other hand, Apatosaurus louisae CM 3018, where only CV 13–15 bear weak projections, was coded as plesiomorphic.

C191: Mid- and posterior cervical vertebrae, postzygapophyseal centrodiapophyseal fossa and spinopostzygapophyseal fossa: entirely separated (0); connected by a large foramen (1) (New; Fig. 36).

Comments. The laminae in this area are very thin and might break easily. In fact, many specimens do show an opening here, but most of them also show broken margins around this opening, making it impossible to decide if the feature is genuine or not. Often, possible foramina are also closed with plaster or similar material during preparation, probably for stability reasons, and because the presence of such foramina has never been reported before. In fact, only SMA 0011 can be confidently scored as apomorphic to date.

C192: Posterior cervical vertebrae, Elongation Index (cervical centrum length, excluding condyle, divided by posterior centrum height): less than 2.0 (0); 2.0–2.6 (1); higher than 2.6 (2) (G86; M12-90, 91; modified; Table S23).

Comments. In vertebrae with inclined posterior edges of the anterior condyle, a vertical line is drawn through the posterior-most point of the posterior edge, and the horizontal distance from this vertical line to a second vertical line through the posterior-most extension of the centrum is measured and taken as centrum length in this case. In some cases, only measurements of the complete centrum length were available, and the EI for the centrum length without anterior ball was calculated based on the mean difference between EI with and without condyle. Singular ratios given in Table S23 have to be taken with care, as they differ considerably within posterior cervical centra (decreasing towards posterior). Ratios based only on anterior posterior cervical vertebrae thus have to be corrected to a lower ratio (e.g., in UW 15556, Table S23). A simple EI is preferred over an average EI (centrum length divided by the mean of posterior centrum height and width; Chure et al., 2010) because many specimens could not be measured directly and lack published measurements. Therefore, many OTUs included herein had to be scored based on figures. Given that the lateral view is often the only one provided, reasonable comparisons could only be made when using the simple version of the EI.

C193: Posterior cervical vertebrae, ventral keel: anteriorly placed (0); restricted to posterior portion of centrum (1) (New; Fig. 53).

Comments. Taxa without ventral ridges are scored as unknown. The posterior restriction of the keel was proposed as an autapomorphy of Dinheirosaurus lourinhanensis by Mannion et al. (2012).

C194: Posterior cervical prezygapophyses: terminate with or in front of articular ball of centrum (0); terminate well behind articular ball (1) (U04b-3; modified; Fig. 40).

Comments. The neural canal should be held horizontally, in order to accurately assess the expansion of the prezygapophysis.

C195: Posterior cervical vertebrae, prezygapophysis articular facet posterior margin: confluent with prezygapophyseal process (0); bordered posteriorly by conspicuous transverse sulcus (1) (T13-121; Figs. 39 and 55).

Comments. The distribution of this character is dubious, because it is difficult to observe in photographs and drawings. To date, only the holotype specimen of Kaatedocus siberi (SMA 0004) was reported to bear such a sulcus. The character in its present state thus does not contribute to the resolution of the tree. It was retained because more work on actual specimens has to be performed in order to confirm or discard this character as an unambiguous autapomorphy of K. siberi.

C196: Posterior cervical vertebrae, spinoprezygapophyseal lamina: continuous (0); developing an anterior projection (just beneath but independent from the spine summit) (1) (T13-124; Fig. 39).

Comments. Sometimes the spine summit projects anteriorly (in particular in dicraeosaurs), which is not what this character describes. Diplodocines often have an anterior projection below the summit, which forms the most anterior point of the spine.

C197: Posterior cervical vertebrae, accessory lateral lamina connecting postzygodiapophyseal and spinoprezygapophyseal laminae: absent (0); present (1) (G05-25; Fig. 36).

Comments. This lamina was termed epipophyseal-prezygapophyseal lamina by Wilson & Upchurch (2009), but there are different ways of how to unite the epipophysis with the prezygapophysis (Carballido et al., 2012b; Wilson, 2012). Therefore, the description of Carballido et al. (2012b) was preferred herein.

C198: Posterior cervical vertebrae, accessory, subvertical lamina in the postzygapophyseal centrodiapophyseal fossa, with free edge facing laterally: absent (0); present (1) (New; Fig. 56).

Comments. Two types of accessory laminae occur in the pocdf of certain sauropod taxa: (1) laterally facing, relatively broad laminae, which are mostly located posteriorly, marking the lateral wall of the neural canal, and (2) more distinct, posteriorly facing laminae connecting the pcdl and podl anteriorly, at the base of the transverse process. The present character describes the presence of the first type, and the second type is accounted for in character 199.

C199: Posterior cervical vertebrae, accessory, subvertical lamina in the postzygapophyseal centrodiapophyseal fossa, with free edge facing posteriorly: absent (0); present (1) (Gilmore, 1936; U04b-6; modified; Fig. 36).

Comments. This accessory lamina is the one character 95 of Mannion et al. (2012) codes for. Rarely, posteriorly facing accessory laminae appear as a parallel pair (e.g., SMA 0011; Fig. 36). Jobaria has posteriorly facing laminae in the posterior portion of the pocdf, connecting to the postzygapophyses. They are herein interpreted as lateral cpol, which are somewhat anteriorly shifted. Jobaria is thus scored as plesiomorphic in this character.

C200: Posterior cervical postzygapophyses: terminate at or beyond posterior edge of centrum (0); terminate in front of posterior edge (1) (U04b-4; modified by T13-129; Fig. 57).

C201: Posterior cervical neural arch, interpostzygapophyseal lamina (tpol): connects directly with roof of neural canal (0); vertical lamina connects tpol with neural canal roof (1) (New; Fig. 56).

Comments. Carballido & Sander (2014) termed this vertical lamina ‘single intrapostzygapophyseal lamina’ (stpol).

C202: Posterior cervical neural arches, epipophyses: transversely compressed (0); dorsoventrally compressed (1) (New; Fig. 42).

Comments. Two different morphologies of the epipophyses occur in diplodocids: (1) dorsoventrally compressed, usually forming a horizontal, rugose ridge above the postzygapophyseal facet, on the lateral side of the spol, and (2) transversely compressed, such that it is formed by a dorsal expansion of the posterior end of the spol, in some cases (e.g., Diplodocus carnegii CM 84) forming a rugose, vertical plate above the zygapophyseal facet, but never accompanied by a horizontal ridge. Taxa without epipophyses are scored as unknown.

C203: Posterior cervical neural arches, accessory spinal lamina: absent (0); present, running vertically just posterior to spinoprezygapophyseal lamina (1) (W11-98; Fig. 40).

Comments. This lamina could represent a reduced spdl. The presence of a distinct lamina is restricted to advanced diplodocines, but a reduced lamina is present in Spinophorosaurus as well (NMB-1699-R, E Tschopp, pers. obs., 2011).

C204: Posterior cervical neural spines, dorsoventrally elongate coel on lateral surface: absent (0); present (1) (M12-99; Fig. 53).

C205: Posterior cervical neural spines, horizontal, rugose ridge right below spine summit on lateral surface: absent (0); present, serves as distinct dorsal edge of the spinodiapophyseal fossa (1) (T13-127; Fig. 42).

Comments. The ridge is slightly curved in some specimens (e.g., SMA 0011). When absent (plesiomorphic state), the sdf fades dorsally.

C206: Posterior bifid, cervical neural spines, medial surface: marked by distinct, dorsoventral ridge from base to spine summit (0); smooth (1) (New; Fig. 58).

Figure 58 Diplodocine posterior cervical vertebrae.

Posterior cervical vertebrae of Kaatedocus siberi SMA 0004 (A), and Barosaurus lentus YPM 429 (B) in dorsal view. Note the dorsoventral ridge on the medial side of the metapophysis (A; C206-1) and the anterior projection lateral to the prezygapophyseal facet (B; C213-1). Abb.: di, diapophysis; epi, epipophysis; podl, postzygodiapophyseal lamina; prdl, prezygodiapophyseal lamina; prz, prezygapophysis; spol, spinopostzygapophyseal lamina; sprl, spinoprezygapophyseal lamina. Scaled to the same total length.

C207: Posterior cervical neural and/or anterior-most dorsal neural spines: vertical (0); anteriorly inclined (1) (R05-71).

Comments. See comments in character 169 for definition of inclined.

C208: Posterior cervical and anterior dorsal vertebrae, roughened lateral aspect of prezygodiapophyseal lamina: absent (0); present (1) (W11-102; Fig. 59).

Figure 59 Dorsal vertebra 1 of diplodocids.

Dorsal vertebra 1 of Apatosaurus louisae CM 3018 (A; modified from Gilmore, 1936), and Diplodocus carnegii CM 84 (B; modified from Hatcher, 1901) in left and right lateral view, respectively. Note the roughened prdl (B; C208-1), and the different location of the pleurocoels (C240). Abb.: cpol, centropostzygapophyseal lamina; di, diapophysis; pap, parapophysis; pcdl; posterior centrodiapophyseal lamina; pl, pleurocoel; poz, postzygapophysis; spol, spinopostzygapophyseal lamina. Scaled to same posterior cotyle height.

Comments. The rugose area in the derived taxa lies ventrolateral to the pre-epipophysis, when present.

C209: Posterior cervical and anterior dorsal vertebrae, prespinal lamina: absent (0), present (1) (S97-14, modified; Fig. 60).

Figure 60 Flagellicaudatan anterior dorsal vertebrae.

Anterior dorsal vertebrae of Suuwassea emilieae ANS 21122 (A), Brontosaurus parvus UW 15556 (B; modified from Gilmore, 1936), and Apatosaurus ajax YPM 1860 (C) in anterior view. Note the prespinal lamina (A and C; C209-1), the diverging (B; C211-0) or parallel neural spines (A; C211-1), the wide (C; C212-0) or narrow (A; C212-1) distance between the spine tops, and the ridge on the medial side of the neural spine (C; C245-1). Abb.: di, diapophysis; nc, neural canal; pap, parapophysis; pcdl, posterior centrodiapophyseal lamina; prdl, prezygodiapophyseal lamina; prpl, prezygoparapophyseal lamina; prz, prezygapophysis; sprl, spinoprezygapophyseal lamina; tprl, interprezygapophyseal lamina. Scaled to same anterior condyle height.

Comments. The presence of a prespinal lamina does not imply the presence of a median tubercle or vice versa. However, a dorsally expanded prespinal lamina can form a median tubercle (see below). In anterior dorsal vertebrae of Diplodocus carnegii CM 94, the median tubercle leans anteriorly, but no lamina connects it with the base of the notch between the metapophyses (E Tschopp, pers. obs., 2011).

C210: Posterior cervical and anterior dorsal bifid neural spines, median tubercle: absent (0); present (1) (McIntosh, 1990b; U95; Fig. 39).

Comments. The median tubercle can be either an independent structure in the trough between the metapophyses, or a dorsal projection of the prespinal lamina.

C211: Posterior cervical and anterior dorsal bifid neural spines, orientation: diverging (0); parallel to converging (1) (R05-74; modified; Fig. 60).

Comments. Some taxa have diverging neural spines, with only their summits approaching an almost parallel orientation (e.g., CM 11984 or USNM 10865). They are scored as plesiomorphic herein. The character was initially proposed including the rate of divergence (Rauhut et al., 2005). The character was divided because the dorsal portions of the metapophyses can be parallel, but still widely separated from each other, as is the case in Camarasaurus.

C212: Posterior cervical and anterior dorsal bifid neural spines, divergence: wide (0); narrow, distance between spine summits subequal to neural canal width (1) (R05-74; modified; Fig. 60).

Comments. This is the second part of the character proposed by Rauhut et al. (2005; see character 211).

C213: Posterior cervical, and anterior and mid-dorsal vertebrae, anterior projection of diapophysis laterally adjacent to prezygapophyseal facet: absent (0); present (1) (New; Fig. 58).

Comments. The projection described herein is not to be confused with the projection sometimes formed by the pre-epipophysis, which is posteriorly accompanied by a horizontal, rugose ridge.

C214: Cervical ribs, length: long, reaching posterior to posterior end of centrum (0); short, not reaching posterior end of centrum (1) (R93-12; modified; Fig. 51).

Comments. An additive binary version describing cervical rib length is preferred herein over the multistate character of Whitlock (2011a).

C215: Cervical ribs, length: overlapping several centra posterior (0); overlapping no more than the next cervical vertebra in sequence (1) (R93-12; modified; Fig. 41).

C216: Cervical ribs, position relative to centrum: not projecting far beneath centrum (0); projecting well beneath centrum, such that length of posterior process is subequal in length to fused diapophysis/tuberculum (1) (Wilson, 2002; W11-153; modified; Fig. 40).

Comments. Whitlock (2011a) included two characters describing the length of the ventral projection (from Wilson, 2002) and comparing the length of the posterior process with the length of the fused diapophysis/tuberculum. However, the length of the fused diapophysis and tuberculum depends on how far the cervical ribs project ventrally, and the length of the posterior process is accounted for in the characters defining cervical rib length. Wilson (2002) defined the ventral projection as strong when it leads to a vertebral height that exceeds its length. Such a ratio is also present in dicraeosaurids, but because of their highly elevated neural spines. The ventral projection of the cervical rib of dicraeosaurids is minimal as in all taxa other than apatosaurs. Therefore, the two characters of Wilson (2002) and Whitlock (2011a) are herein combined, in order to define ventral projection compared to the length of the posterior process of the cervical rib.

C217: Cervical ribs, posteriorly projecting spur on dorsolateral edge of posterior shaft: absent (0); present (1) (New; Fig. 49).

Comments. The spur was proposed as autapomorphic for Turiasaurus (Royo-Torres, Cobos & Alcalá, 2006), but it is also present in some apatosaurs and Dicraeosaurus (E Tschopp, pers. obs., 2011; E Tschopp, pers. obs., 2012).

C218: Anterior and mid-cervical ribs, tuberculum in lateral view: is directed nearly vertically (0); is directed upwards and backwards (1) (U04b-12; modified; Fig. 50).

Comments. The orientation of the tuberculum tends to become more vertical in more posterior elements. Some apatosaurs scored as plesiomorphic here actually do not have any anterior cervical vertebrae preserved, which means that they could still have inclined tubercula in the anterior elements. However, because others have distinctly inclined tubercula in mid-cervical ribs as well, a differential coding is still justifiable. Taxa that do not preserve cervical ribs were coded based on the relative positions of diapophysis and parapophysis.

C219: Posterior cervical ribs, anterior process: present (0); absent (1) (U04b-9; modified; Fig. 40).

C220: Posterior cervical ribs, anterior process: distinct, much longer anteroposteriorly than high dorsoventrally (0); reduced to a short bump-like process or absent (1) (New; Fig. 61).

Figure 61 Apatosaurine posterior cervical ribs.

Posterior cervical ribs of Brontosaurus parvus UW 15556 (A; after Gilmore, 1936) and Apatosaurus louisae CM 3018 (B; after Gilmore, 1936) in right lateral view (B inverted). Note the short, reduced anterior projection (A; C220-1), the pointed anterior process (A; C221-1), the ventrolateral process (B; C222-1), and the downwards curving posterior process (A; C223-1). Abb.: cap, capitulum; tub, tuberculum. Scaled to same length.

Comments. The last two characters serve as additive binary characters describing the reduction of the anterior process in apatosaurs in general and its complete absence in some apatosaur specimens (e.g., CM 3018; Gilmore, 1936; Wedel & Sanders, 2002).

C221: Posterior cervical ribs, anterior process: rounded in lateral view (0); has an acute pointed tip in lateral view (1) (U04b-10; modified; Fig. 61).

Comments. The anterior processes of cervical ribs can be rounded in dorsal view, but dorsoventrally compressed (as in SMA 0011, see above). Therefore, they are still pointed in lateral view.

C222: Posterior cervical ribs, rounded sub-triangular process in lateral view, posteroventral to tuberculum: absent (0); present (1) (Wedel & Sanders, 2002; U04b-11; wording modified; Fig. 61).

Comments. Upchurch, Tomida & Barrett (2004) scored the holotypic cervical vertebra of Apatosaurus laticollis YPM 1861 as plesiomorphic. However, as Wedel & Sanders (2002) showed, a process is clearly present in this specimen.

C223: Posterior cervical rib shafts: nearly straight and directed backward and a little upwards (0); initially directed in same direction but turn to run a little downwards toward distal tip (1) (U04b-13; Fig. 61).

Dorsal vertebrae

C224: Number of dorsal vertebrae: 13 or more (0); 12 (1); 10 (2); 9 (3) (McIntosh, 1990b; R93-14; modified; Table S24).

Comments. Amargasaurus was initially described to have 9 dorsal vertebrae (Salgado & Bonaparte, 1991), but the putative first dorsal has the parapophysis positioned dorsally to the pleurocoel, which is highly unusual in sauropods (Carballido et al., 2012a). Generally, this position marks the second or third dorsal vertebrae, which means that there would be at least ten dorsal elements, which was the coding used by Mannion et al. (2012). Herein, a coding as unknown is preferred, following Carballido et al. (2012b).

C225: Dorsal centrum length (excluding articular ‘ball’), remains approximately the same along the sequence (0); shortens from anterior to posterior dorsal vertebrae (1) (M12-106; Table S25).

Comments. The exclusion of the articular ball for measuring centrum length for this character is crucial, because anterior dorsal vertebrae often have considerably larger anterior condyles than posterior elements. In taxa lacking measurements or good figures to compare between anterior and posterior elements, scores of Mannion et al. (2012) were used (e.g., Omeisaurus).

C226: Dorsal vertebrae, opisthocoely (including a prominent anterior articular ‘ball’) disappears: between DV2 and DV3 (0); between DV3 and DV4 or more posteriorly (1) (Holland, 1915a; Gilmore, 1936; U04b-15; Table S26).

Comments. The definition of ‘prominent anterior ball’ is somewhat ambiguous. However, a new definition is not given here, because the character is interpreted to describe a significant change within the same vertebral column. These changes can be of different absolute size if one compares between specimens, but are relatively obvious within the same individual. The decrease is thus relative to its development in more anterior elements, but can be low in an absolute sense.

C227: Dorsal pneumatopores (pleurocoels): present (0); absent (1) (G86; McIntosh, 1990b; U95; polarity reversed; Fig. 62).

Figure 62 Dorsal vertebra 3 of eusauropods.

Dorsal vertebra 3 of Shunosaurus lii ZDM T5401 (A; modified from Zhang, 1988), and Brontosaurus parvus UW 15556 (B; modified from Gilmore, 1936) in left (A) and right (B) lateral view. Note the slightly concave lateral surface of the centrum in Shunosaurus (A; C227-0), in contrast to the well-defined pneumatopore in UW 15556 (B; C227-1), and the different locations of the parapophyses (C246). Abb.: cpol, centropostzygapophyseal lamina; di, diapophysis; pcdl, posterior centrodiapophyseal lamina; poz, postzygapophysis. Scaled to the same total vertebral height.

Comments. The dorsal centra of all included sauropod taxa have pleurocoel-like depressions on their lateral side, but in some taxa they do not bear a foramen.

C228: Dorsal centra, pneumatic structures: absent, dorsal centra with solid internal structure (0); present, dorsal centra with simple and big air spaces (1); present, dorsal centra with small and complex air spaces (2) (W02-77; modified by C12b-139; Fig. 37).

C229: Dorsal neural arches, paired, subdivided pneumatic chambers dorsolateral to neural canal: absent (0), present (1) (Sereno et al., 1999; W11-106; Fig. 63).

Figure 63 Diplodocoid posterior dorsal vertebrae.

Posterior dorsal vertebrae of Haplocanthosaurus priscus CM 572 (A; modified from Hatcher, 1903), Demandasaurus darwini MDS-RVII 798 (B; modified from Torcida Fernández-Baldor et al., 2011), and Apatosaurus louisae CM 3018 (C; modified from Gilmore, 1936) in posterior view. Note the paired pneumatic foramen dorsolateral to the neural canal in Demandasaurus (B; C229-1), the different orientations of the diapophyses in Haplocanthosaurus (A; C230-1) and Apatosaurus (C; C230-0), the single lamina that supports the hyposphene from below (C; C238-0), the dorsal spur on the tip of the transverse process (A; C264-1), the small triangular lateral projections at the spine top (A; C267-1), or their absence (C; C267-0), the rhomboid (C; C276-0) in contrast to laminar (B; C276-1) hyposphene, and the ventrally forked spol (B; C277-1). Abb.: cpol, centropostzygapophyseal lamina; nc, neural canal; pap, parapophysis; posl, postspinal lamina; poz, postzygapophysis; spdl, spinodiapophyseal lamina; spol, spinopostzygapophyseal lamina. Scaled to same posterior centrum height.

Comments. Paired pneumatic foramina occur in some diplodocids (e.g., UW 15556, YPM 1840), but they are not subdivided and are far less deep than in Nigersaurus or Demandasaurus. The latter are thus the only taxa with the apomorphic state.

C230: Dorsal transverse processes, orientation: horizontal or only slightly inclined dorsally (0); more than 30° inclined dorsally from the horizontal (1) (Y93-58; modified by U98-102; Fig. 63).

Comments. The angle of the transverse processes is easily affected by diagenetic distortion, as can be seen in DV 3 of Suuwassea ANS 21122, which most probably would actually have horizontal transverse processes.

C231: Dorsal vertebrae, single (not bifid) neural spines, spinoprezygapophyseal laminae: separate along entire length (0); joined distally, forming single prespinal lamina (1) (U95; modified by W11-107; Fig. 64).

Figure 64 Dorsal vertebra 8 of neosauropods.

Dorsal vertebra 8 of Camarasaurus supremus AMNH 5760 (A; traced from Osborn & Mook, 1921) and Apatosaurus louisae CM 3018 (B; traced from Gilmore, 1936) in anterior view. Note the separated (A; C231-0) or dorsally united spinoprezygapophyseal laminae (B; C231-1), the fossa between them (B; C233-0), and the triangular processes of the neural spine, that project further than the zygapophyses (A; C267-2). Abb.: acpl, anterior centroparapophyseal lamina; cprl, centroprezygapophyseal lamina; nc, neural canal; pap, parapophysis; prsl, prespinal lamina; prz, prezygapophysis; spdl, spinodiapophyseal lamina; tp, transverse process. Scaled to same anterior condyle height.

Comments. In some taxa (e.g., Losillasaurus or Camarasaurus), the sprl unite dorsally with the prsl, but remain separate up to that point. Here, only taxa where the prsl is formed by the junction of the two sprl are scored as apomorphic.

C232: Dorsal vertebrae, spinodiapophyseal webbing: laminae follow curvature of neural spine and diapophysis in anterior view (0); laminae ‘festooned’ from spine, dorsal margin does not closely follow shape of neural spine and diapophysis (1) (S07-43; Fig. 65).

Figure 65 Diplodocimorph dorsal neural arches.

Dorsal neural arches of Diplodocus carnegii CM 84 (A; traced from Hatcher, 1901) and Nopcsaspondylus alarconensis holotype specimen (B; traced from Nopcsa, 1902) in anterior view. Note the festooned spdl typical for rebbachisaurids (B; C232-1), in contrast to the plesiomorphic state (A; C232-0), and the notched (A; C281-1), or straight to convex spine summits (B; C281-0). Abb.: cprl, centroprezygapophyseal lamina; prdl, prezygodiapophyseal lamina; prsl, prespinal lamina; tp, transverse process. Not to scale.

C233: Dorsal vertebrae with single neural spines, middle single fossa projected through midline of neural spine: present (0); absent (1) (C12b-144; Fig. 64).

Comments. The fossa described herein is a distinctly confined area within the sprf, restricted to the anterior edge of the neural spine process.

C234: Dorsal (single) neural spines, postspinal lamina, dorsal end: flat to convex transversely (0); concave transversely (1) (New; Fig. 66).

Figure 66 Eusauropod posterior dorsal vertebrae.

Posterior dorsal vertebrae of Losillasaurus giganteus MCNV Lo-11 (A), and Apatosaurus louisae CM 3018 (B; modified from Gilmore, 1936) in posterior view. Note the concave dorsal end of the posl (A; C234-1), the horizontal (A; C275-0), instead of angled (B; C275-1) postzygapophyseal facets, and the medial spinopostzygapophyseal lamina (B; C278-1). Abb.: cprl, centroprezygapophyseal lamina; hys, hyposphene; lspol, lateral spinopostzygapophyseal lamina; posl, postspinal lamina; poz, postzygapophysis; spdl, spinodiapophyseal lamina. Scaled to same posterior cotyle height.

C235: Dorsal vertebrae, transition from bifid to single neural spines: gradual (0); abrupt (1) (New).

Comments. Gradual transitions go from deeply bifid, to shallowly bifid, to notched, to unsplit, as defined by Wedel & Taylor (2013). If one of the intermediate states is lacking, the taxon is scored as derived. Obviously, only specimens with articulated dorsal vertebrae can be scored for this character. Taxa without spine bifurcation are scored as unknown.

C236: Dorsal neural arches, hyposphene-hypantrum articulations: present (0); absent (1) (G86; S97-25; Table S27).

C237: Dorsal vertebrae, hyposphene first appears: on DV3 (0); on DV4 or more posteriorly (1) (U04b-19; modified; Table S27).

Comments. Both in Apatosaurus and Camarasaurus there are differences in the appearance of the hyposphene (Ikejiri, 2004; Upchurch, Tomida & Barrett, 2004). Because the type species, C. supremus, appears to show the plesiomorphic state, the genus was scored as such as well. Ikejiri (2004) suggests that the development of the hyposphene might depend on ontogeny, based on observations in the juvenile specimen CM 11338. However, the latter specimen is articulated and the region with the hyposphene is obliterated, such that its presence or absence is difficult to assess (McIntosh et al., 1996a).

C238: Dorsal vertebrae, single vertical lamina supporting the hyposphene from below: absent (0); present (1) (Gilmore, 1936; U04b-20; modified; Fig. 63).

Comments. The original character description (Upchurch, Tomida & Barrett, 2004) interfered with the character proposed by Wilson (2002) distinguishing between single and double cpol in mid- and posterior dorsal vertebrae (see character 261). The character of Upchurch, Tomida & Barrett (2004) was thus simplified, and polarity was reversed due to the differential taxon sampling. The lamina described herein corresponds to the stpol (Carballido & Sander, 2014). Taxa without hyposphene are scored as unknown.

C239: Dorsal vertebrae 1 and 2, centrum length: DV 1 > DV 2 (0); DV 2 > DV 1 (1) (U04b-14; modified; Table S28).

Comments. The character was originally defined implying that either DV 1 or 2 were the longest in the series (Upchurch, Tomida & Barrett, 2004), which is not always the case (see Table S28).

C240: First dorsal vertebrae, pleurocoel location: occupy the anterior and middle part of the centrum (0); occupy the posterior part of the centrum (1) (Holland, 1915a; Gilmore, 1936; U04b-17; modified; Fig. 59).

Comments. The character was restricted to the first dorsal, as also in Apatosaurus louisae, for which this character was proposed as a species autapomorphy (Holland, 1915a; Gilmore, 1936; Upchurch, Tomida & Barrett, 2004). In this taxon, DV 2 and 3 already have a centrally placed pleurocoel (CM 3018, E Tschopp, pers. obs., 2011).

C241: Anterior dorsal vertebrae, pleurocoels in first few centra: become larger along the series (0); become smaller (1) (Gilmore, 1936; U04b-16; wording modified; Table S29).

Comments. Taxa without dorsal pleurocoels are scored as unknown.

C242: Anterior dorsal vertebrae, ventral keel: absent (0); present (1) (M12-110; Fig. 67).

Figure 67 Dorsal vertebra 4 of “Dinheirosaurus” lourinhanensis ML 414 in ventral view.

Note the ventral keel (C242-1) in anterior dorsal vertebrae. Abb.: DV, dorsal vertebra; pcdl, posterior centrodiapophyseal lamina; pl, pleurocoel; podl, postzygodiapophyseal lamina; tp, transverse process.

C243: Anterior dorsal transverse process position: high, considerably above dorsal edge of posterior cotyle (0); low, ventral edge about level to dorsal edge of posterior cotyle (1) (Gilmore, 1936; Fig. 68).

Figure 68 Dorsal vertebra 1 of apatosaurines.

Dorsal vertebra 1 of Apatosaurus louisae CM 3018 (A) and Brontosaurus parvus UW 15556 (B; both traced from Gilmore, 1936) in posterior view. Note the different positions of the transverse processes (high, A, C243-0; low, B, C243-1), and the varying width of the base of the bifurcated spines (wide, A, C244-0; narrow, B, C244-1). Abb.: di, diapophysis; pcdl, posterior centrodiapophyseal lamina; podl, postzygodiapophyseal lamina; poz, postzygapophysis; prz, prezygapophysis. Scaled to same posterior cotyle height.

Comments. The differing dorsoventral extension of the transverse processes in the anterior-most dorsal vertebrae was proposed as character to distinguish Apatosaurus louisae CM 3018 from the supposed Apatosaurus excelsus UW 15556 (Gilmore, 1936). It is here applied for the first time in a phylogenetic analysis. In most taxa, position of the transverse process rises considerably dorsally in the first few dorsal vertebrae. Therefore, this description applies best for the first element in the series.

C244: Anterior, bifid dorsal vertebrae, base of notch between metapophyses: wide and rounded (0); narrow, V-shaped (1) (Gilmore, 1936; Fig. 68).

Comments. As observed in Apatosaurus, Camarasaurus also appears to show intrageneric variation: C. lewisi has narrow troughs throughout its bifurcated presacral vertebrae, whereas other Camarasaurus species have wide bases (Jensen, 1988; McIntosh et al., 1996b). Herein, Camarasaurus was scored as plesiomorphic, scoring the type species C. supremus.

C245: Anterior dorsal, bifid neural spines, medial surface: gently rounded transversely (0); subtriangular (1) (New; Fig. 60).

Comments. Some diplodocid specimens bear a dorsoventral ridge on the medial surface of the anterior dorsal neural spines, similar to the ridge present in some diplodocid posterior cervical neural spines. The ridge results in a subtriangular shape of the medial surface.

C246: Dorsal vertebra 3, parapophysis: lies at the top of the centrum (0); lies mid-way between the top of the centrum and the level of the prezygapophyses (1) (Gilmore, 1936; U04b-18; modified; Fig. 62).

C247: Anterior and mid-dorsal centra, pleurocoels: situated entirely on centrum (0); invade neural arch pedicels (1) (Holland, 1915a; Fig. 69).

Figure 69 Diplodocine mid-dorsal vertebrae.

Mid-dorsal vertebrae of “Dinheirosaurus” lourinhanensis ML 414 (A) and Galeamopus sp. SMA 0011 (B) in lateral view. Note the pleurocoels that are entirely situated on the centrum (A; C247-0), or invade the neural arch (B; C247-1), the accessory spinal lamina connecting to the junction of spol and spdl (A; C251-1), the vertical lamina subdividing the pleurocoel (A; C253-1), the anteriorly displaced parapophysis (A; C256-1) in contrast to its usual position above the anterior edge (B; C256-0), and the horizontal accessory lamina connecting the hyposphene with the pcdl (A; C260-1). Abb.: cpol, centropostzygapophyseal lamina; pcdl, posterior centrodiapophyseal lamina; pcpl, posterior centroparapophyseal lamina; podl, postzygodiapophyseal lamina; spol, spinopostzygapophyseal lamina. Scaled to same vertebral height.

Comments. Holland (1915a) proposed this morphology as diagnostic for Apatosaurus louisae. It is included in a phylogenetic analysis for the first time. Taxa without dorsal pleurocoels are scored as unknown.

C248: Anterior and mid-dorsal neural arch, hyposphene shape: rhomboid (0); laminar (1) (New; Table S27).

Comments. Hyposphene shape can change considerably from front to back, as is seen in specimens of Camarasaurus (Osborn & Mook, 1921; McIntosh et al., 1996b). In the present analysis, two different characters thus code for the anterior and mid-dorsal vertebrae, as well as for the posterior elements, which are often less developed (see character 276). See Fig. 63 for an example of a laminar hyposphene.

C249: Mid-dorsal neural arches, height above postzygapophyses (neural spine) to height below (pedicel): 2.1 or greater (0); <2.1 (1) (W11-114; modified; Table S30).

Comments. Pedicel height is measured from the neural canal floor to the ventral-most point of the postzygapophyseal facets, neural spine height from there to the spine top. Both measurements are taken vertically, ignoring spine inclination. The ratio changes considerably between mid- and posterior dorsal vertebrae, therefore the original character of Whitlock (2011a) was divided in two (see character 272). Furthermore, a numerical boundary was introduced.

C250: Mid-dorsal neural spines, form: single, bifid form (if present) does not extend past second or third dorsal (0); bifid, inclusive of at least fifth dorsal vertebrae (1) (W11-108; Table S31).

Comments. Notched and unsplit neural spines (sensu Wedel & Taylor, 2013) are counted as single; shallowly and deeply bifurcated spines as bifid. An additional character is used to account for the notched spines. The taxon scores are thus slightly different from the ones in Whitlock (2011a).

C251: Mid-dorsal neural spines, oblique accessory lamina connecting postspinal lamina with spinopostzygapophyseal lamina: absent (0); present (1) (New; Fig. 69).

Comments. In Supersaurus and Dinheirosaurus, this accessory lamina extends posterodorsally-anteroventrally from near the dorsal end of the posl to the junction of the spol with the spdl.

C252: Mid- and posterior dorsal vertebrae, lateral pleurocoels present in centra: absent (0); present (1) (G86; McIntosh, 1990b; U95; modified by W11-111).

C253: Mid- and posterior dorsal vertebrae, vertically oriented rod-like struts divide the lateral pneumatic foramina: absent (0); present (1) (M12-115; Fig. 69).

Comments. Mannion et al. (2012) proposed the presence of such a strut as a synapomorphy for the clade uniting Supersaurus and Dinheirosaurus. However, similar struts occur in some apatosaurs. The pleurocoel is often not completely liberated from matrix during preparation, potentially obscuring the presence or absence of this structure.

C254: Mid- and posterior dorsal vertebrae, height of neural arch below postzygapophyses (pedicel) divided by posterior cotyle height: <0.8 (0); 0.8 or greater (1) (G05-36; modified; Table S32).

Comments. Neural arch height is measured from the neural canal floor to where the postzygapophyseal facets meet medially, above the hyposphene, where present.

C255: Mid- and posterior dorsal neural arches, prezygoparapophyseal lamina: present (0); absent (1) (W02-97; Fig. 70).

Figure 70 Diplodocid posterior dorsal vertebrae.

Posterior dorsal vertebrae of Apatosaurus louisae CM 3018 (A; traced from Gilmore, 1936) and Supersaurus vivianae BYU 9044 (B; traced from Jensen, 1985) in left (A) and right (B) lateral view. Note the prpl (A; C255-1), the anteriorly displaced parapophysis (B; C256-1), the acpl (A; C257-1), the pcpl (B; C258-1), the lateral branch of the cpol (B; C261-1), the pronounced opisthocoely (B; C270-2), and the anteriorly inclined base of the neural spine (A; C280-1). Abb.: cpol, centropostzygapophyseal lamina; hys, hyposphene; pl, pleurocoel; posl, postspinal lamina; poz, postzygapophysis; spdl, spinodiapophyseal lamina. Scaled to same posterior cotyle height.

C256: Mid- and posterior dorsal parapophyses, location: above centrum, posterior to anterior edge of centrum (0); straight above anterior edge of centrum, or anteriorly displaced (1) (New; Figs. 69 and 70).

Comments. The anterior edge of the centrum corresponds to the rim of the anterior condyle in opisthocoelous elements. In some taxa, the position of the parapophysis changes from front to back. These taxa are scored for the majority of the elements in the series (e.g., Haplocanthosaurus, where DV 10 has a posteriorly placed parapophysis, but the majority of the mid- and posterior dorsal vertebrae have anteriorly displaced parapophyses; Hatcher, 1903).

C257: Mid- and posterior dorsal neural arches, anterior centroparapophyseal lamina: absent (0); present (1) (U04a-133; modified; Fig. 70).

Comments. The character was herein adapted to restrict the positions to mid- and posterior caudal vertebrae, instead of including all dorsal vertebrae as in Upchurch, Barrett & Dodson (2004).

C258: Mid- and posterior dorsal neural arches, posterior centroparapophyseal lamina: absent (0); present as single lamina (1); present, double (2) (S97-22; modified after M13-148, based on D12-36; Figs. 70 and 71). Ordered.

Figure 71 Neosauropod posterior dorsal vertebrae.

Posterior dorsal vertebrae of Giraffatitan brancai MB.R.3822 (A), Apatosaurus louisae CM 3018 (B; traced from Gilmore, 1936), and Diplodocus carnegii CM 84 (C; traced from Hatcher, 1901) in right lateral view. Note the double pcpl (C; C258-2), the accessory lamina in the parapophyseal centrodiapophyseal fossa (C; C259-1), the accessory lamina connecting the hyposphene with the pcdl (C; C260-1), the infradiapophyseal pneumatic foramen (A; C262-1), the dorsally tapering neural spine (A; C265-1), the different shapes of the pleurocoels (C271), and the ventrally open parapophyseal, centrodiapophyseal fossa (B; C273-0). Abb.: cprl, centroprezygapophyseal lamina; lspol, lateral spinopostzygapophyseal lamina; posl, postspinal lamina; poz, postzygapophysis; prsl, prespinal lamina; prz, prezygapophysis; spdl, spinodiapophyseal lamina; tp, transverse process. Scaled to same total height.

Comments. In taxa, where the pcpl is double, the more dorsal branch often connects to the pcdl. Mannion et al. (2013) defined the third state as ‘two parallel laminae,’ but in certain specimens (e.g., Diplodocus carnegii CM 84), the dorsal branch becomes more horizontal (Hatcher, 1901). Mannion et al. (2013) based their character modification on character 36 of D’Emic (2012), which cites the occurrence of a single versus a double posterior centrodiapophyseal lamina (pcdl). However, this character should have referred to the posterior centroparapophyseal lamina (pcpl) rather than the pcdl (M D’Emic, pers. comm., 2015). Among the apomorphic features, D’Emic (2012) listed this character correctly as a double posterior centroparapophyseal lamina twice, referring to character 36 (D’Emic, 2012: appendices 3 and 4). Thus, character 36 of D’Emic (2012) is the same character as 148 of Mannion et al. (2013), and is included and slightly modified in our analysis. The character is treated as ordered, because it codes for both presence/absence and morphology.

C259: Mid- and posterior dorsal vertebrae, accessory laminae in region between posterior centrodiapophyseal lamina and posterior centroparapophyseal lamina: absent (0); present (1) (M12-116; Fig. 71).

Comments. This character is somewhat ambiguous. Some of these accessory laminae might actually represent dorsal branches of the pcpl (see character 258) or dislocated ppdl. Here, only laminae not directly connecting to any specifying landmark (see Wilson, 1999) are considered accessory. More studies are needed to see if these are homologous to the above mentioned laminae.

C260: Mid- and posterior dorsal vertebrae, accessory lamina linking hyposphene with base of posterior centrodiapophyseal lamina: absent (0); present (1) (New; Figs. 69 and 71).

Comments. The presence of such an accessory lamina was proposed as autapomorphic for Dinheirosaurus (Bonaparte & Mateus, 1999; Mannion et al., 2012), but is herein interpreted to occur in other diplodocids as well. The accessory lamina can easily be confused with the lateral branch of the cpol, but the latter connects directly with the postzygapophyseal facet and not with the hyposphene. The accessory lamina described herein is thus situated between the two branches of the cpol.

C261: Mid- and posterior dorsal neural arches, centropostzygapophyseal lamina: single (0); divided, lateral branch connecting to posterior centrodiapophyseal lamina (1) (W02-95; wording modified; Fig. 70).

Comments. The lateral branch is often only visible in lateral view.

C262: Mid- and posterior dorsal neural arches, infradiapophyseal pneumatopore between anterior and posterior centrodiapophyseal laminae: absent (0); present (1) (W02-103; Fig. 71).

Comments. Even though the development of pneumatic structures has been shown to depend on the ontogenetic stage (Wedel, 2003; Schwarz et al., 2007), the early juvenile brachiosaur SMA 0009 already has this pneumatopore.

C263: Mid- and posterior dorsal transverse processes, length: short (0); long (projecting <1.3 times posterior cotyle width) (1) (C12b-153; modified; Table S33).

Comments. The length of a single transverse process is compared to the maximum width of the posterior cotyle. Transverse process length is measured in a horizontal plane. Measurements taken from figures in posterior view generally underestimate the ratio, which has to be accounted for when scoring the taxa. In the case of Brachiosaurus altithorax FMNH P25107, true ratios based on the measurements by Riggs (1904) are about 120% of the ratios taken from published figures (Taylor, 2009), whereas in Apatosaurus NSMT-PV 20375 or Diplodocus CM 84, they are only 103% higher. This percentage depends on the relative position of the transverse processes above the centrum. Ratios generally decrease from anterior to posterior dorsal vertebrae. Taxa or specimens that preserve only posterior elements (e.g., Amphicoelias altus AMNH 5764) should thus have higher actual ratios than shown in Table S33.

C264: Mid- and posterior dorsal transverse processes, dorsal edge: straight, or curving downwards at distal end (0); developing a distinct dorsal bump or spur (1) (New; Fig. 63).

Comments. Spurs are usually situated at the distal tip, whereas bumps are located more medially.

C265: Mid- and posterior dorsal neural spines, anteroposterior width: approximately constant along height of spine, with subparallel anterior and posterior margins (0); narrows dorsally to form triangular shape in lateral view, with base being approximately twice the width of dorsal tip (1) (Taylor, 2009; M13-159; modified; Fig. 71).

Comments. Mannion et al. (2013) were the first to include this character in a phylogenetic analysis, based on observations by Taylor (2009), and encompassing the entire dorsal column. Herein, we restricted the character to mid- and posterior dorsal neural spines.

C266: Middle and posterior dorsal neural spines, breadth at summit: much narrower (0); equal to or broader (1) transversely than anteroposteriorly (W02-92; modified).

Comments. Neural spine width can change considerably from the spine bottom to the top. The original character was thus divided in two (see character 265).

C267: Mid- and posterior dorsal neural spines, triangular aliform processes: absent (0); present, do not project as far laterally as postzygapophyses (1); present, project at least as far laterally as postzygapophyses (2) (U98-116; modified after C12b-163; Figs. 63 and 64). Ordered.

C268: Posterior dorsal centra, total length/height of posterior articular surface: 1.0 or greater (0); short, <1.0 (1) (New; Table S34).

C269: Posterior dorsal centra, posterior articular surface width to height: 1.0 or less (0); >1.0 (1) (Gilmore, 1936; Table S34).

Comments. The boundary is set between 1.0 and 1.1 in the present study, because it was suggested by Gilmore (1936) to distinguish Apatosaurus louisae from A. ajax and A. excelsus.

C270: Posterior dorsal centra, articular face shape: amphicoelous (0); slightly opisthocoelous (1); strongly opisthocoelous (2) (Y93-40; wording modified by C12b-174; Fig. 70).

Comments. Slightly opisthocoelous means that the condyle is either ventrally or dorsally restricted, but still visible in lateral view. Strongly opisthocoelous vertebrae have anterior balls that reach from the dorsal to the ventral edge of the centrum. In Apatosaurus ajax YPM 1860, no anterior articulation surface of a posterior dorsal vertebrae is observable, but the posterior articulation surface of a posterior element has a small, but distinct fossa marking its upper half. This indicates a slightly opisthocoelous centrum in the following element.

C271: Posterior dorsal vertebrae, pleurocoel shape: oval to circular (0); subtriangular with apex dorsally (1) (New; Fig. 71).

Comments. Taxa without dorsal pleurocoels are scored as unknown.

C272: Posterior dorsal neural arches, height above postzygapophyses (neural spine) to height below (pedicel): <3.1 (0); 3.1 or greater (1) (W11-114; modified; Table S30).

Comments. See character 249.

C273: Posterior dorsal neural arches, parapophyseal centrodiapophyseal fossa: ventrally open, relatively shallow (0); deep, triangular (1) (G05-41; Fig. 71).

Comments. The apomorphic state is applied to specimens with the pcpl connecting to the pcdl or acdl, thus creating a ventrally closed, triangular fossa between them and the ppdl or prdl. In plesiomorphic taxa, the pcpl fades out posteroventrally or connects to the centrum anterior to the ventral end of the pcdl.

C274: Posterior dorsal vertebrae, spinoprezygapophyseal lamina: absent or greatly reduced (0); present (1) (U07-131; modified; Fig. 72).

Figure 72 Posterior dorsal vertebra of “Elosaurus” parvus CM 566 in lateral anterodorsal view.

Note the greatly reduced spinoprezygapophyseal lamina, which does not reach the prezygapophysis (C274-0). Only the base of the neural arch is preserved (see Peterson & Gilmore, 1902). Abb.: lspol, lateral spinopostzygapophyseal lamina; podl, postzygodiapophyseal lamina; posl, postspinal lamina; poz, postzygapophysis; prdl, prezygodiapophyseal lamina.

Comments. Reduced sprl fade out anteroventrally and/or join the prsl at a very ventral level.

C275: Posterior dorsal postzygapophyses: almost horizontal, such that the two articular facets include a wide angle (0); articular facets oblique, including an almost 90° angle (1) (New; Fig. 66).

Comments. Some diplodocine taxa have curved facets. These are interpreted as horizontal because their lateral halfs are oriented horizontally.

C276: Posterior dorsal vertebrae, hyposphene-hypantrum system: well developed, rhomboid shape up to last element (0); weakly developed, mainly as a laminar articulation (1) (C12b-152; modified; Fig. 63; Table S27).

Comments. Taxa without hyposphenes are scored as unknown.

C277: Posterior dorsal neural arches, spinopostzygapophyseal laminae: single (0); divided near postzygapophyses (1) (W02-100; Fig. 63).

Comments. The spol can bifurcate in two ways in different taxa: rebbachisaurids have ventrally forked laminae, whereas in some diplodocids the spol bifurcates dorsally, creating a medial and a lateral branch. The presence of a medial spol is accounted for in character 278, the present one describes the ventral bifurcation.

C278: Posterior dorsal vertebrae, medial spinopostzygapophyseal lamina: absent (0); present and forms part of median posterior lamina (1) (C12b-172; Fig. 66).

Comments. The mspol can either be connected with the lspol ventrally or they can remain separated.

C279: Posterior dorsal vertebrae, base of neural spines just above transverse processes: longer than wide (0); subequal in width and length (1) (New).

Comments. This is the second character about spine width to length, inspired by a character from Wilson (2002) (see character 266).

C280: Posterior dorsal neural spines, orientation at its base: vertical (0); anteriorly inclined (1) (New; Fig. 70).

Comments. Anterior inclination can be restricted to the very base of the neural spine, as is the case in Apatosaurus louisae CM 3018 (Fig. 70A). The best indication for the inclination is the prsl in lateral view.

C281: Posterior dorsal neural spines, midline cleft along the dorsal surface: absent (0); present (1) (M12-121; modified; Fig. 65; Table S31).

Comments. The midline cleft described herein corresponds to the notched spines of Wedel & Taylor (2013). Not all posterior dorsal spines have to be notched in order to be scored as apomorphic.

C282: Posterior dorsal and/or sacral neural spines (not including arch), height: less than 2 times centrum length (0); 2–3 times centrum length (1); more than 3 times centrum length (2) (M12-123; modified; Table S35). Ordered.

Comments. Neural spine height is measured from the top of the postzygapophyses to the highest point of the spine, vertically. Centrum length does not include the anterior ball. The original version (Mannion et al., 2012) was restricted here to posterior dorsal and sacral vertebrae only, because mid-dorsal elements of diplodocids considerably lower the mean ratio in some cases (Table S35). Also, state boundaries are adapted.

C283: Dorsal ribs, rib head: area between capitulum and tuberculum flat (0); oblique ridge present that connects medial and lateral edge at the base of the rib head (1) (New; Fig. 73).

Figure 73 Flagellicaudatan dorsal rib heads.

Dorsal rib heads of Suuwassea emilieae ANS 21122 (A; modified from Harris, 2006b), Apatosaurus louisae CM 3018 (B; modified from Gilmore, 1936) and Barosaurus lentus AMNH 6341 (C, fragment) in anterior (A, B) and posterior (C) view. Note the transverse ridge (C; C283-1), the pneumatic foramen (B; C284-1), and two of three different orientations of the tuberculum in respect to the rib shaft (C285). Grey lines in C indicate the continuation of the rib if complete. Abb.: cap, capitulum; tub, tuberculum. Not to scale.

Comments. The ridge marks the posterior surface of the rib head of advanced diplodocines.

C284: Dorsal ribs, proximal pneumatopores: absent (0); present (1) (W02-141; Fig. 73).

Comments. In some taxa, only one rib of the entire series bears a pneumatopore. However, the ability to develop pneumatized ribs appears to be restricted to certain diplodocid groups, therefore the character was included in this analysis.

C285: Mid-dorsal ribs, orientation of tuberculum: spreading outside from rib shaft (0); following straight direction of rib shaft (1); following medial bend of rib shaft (2) (G05-39; Fig. 73).

Sacral vertebrae

C286: Sacral vertebrae, number: 4 (0); 5 (1); 6 (2) (S97-2; modified; Table S36).

Comments. Some Camarasaurus specimens appear to have six sacral vertebrae, which is usually considered a synapomorphy of advanced titanosauriforms (Tidwell, Stadtman & Shaw, 2005). The addition of a sacral vertebra was suggested to be a sign of very old age (Tidwell, Stadtman & Shaw, 2005). The unusual six sacral vertebrae in the holotype of ‘Apatosaurus’ minimus AMNH 675 (Mook, 1917) might thus also be ontogenetic.

C287: Sacral vertebral centra, pleurocoels: absent (0); present (1) (U04a-165; wording modified).

C288: Sacral rib III, ventral surface: smooth (0); with oblique ridge (1) (Mook, 1917; Fig. 74).

Figure 74 Sacrum of Brontosaurus amplus YPM 1981 in ventral view (modified from Ostrom & McIntosh, 1966).

Note the oblique ridge on sacral rib III (C288-1). Abb.: DV, dorsal vertebra; SV, sacral vertebra; sy, sacricostal yoke. Scale bar = 20 cm.

Comments. The presence of an oblique ridge was proposed as synapomorphy of Apatosaurus by Mook (1917), but later regarded as ambiguous and thus of little use to diagnose the genus (McIntosh, 1995). The presence of this ridge is herein used for the first time as a phylogenetic character, in order to test its utility. According to Mook (1917), the ridge marks the ventral face of sacral rib II. However, as shown in the holotype specimen of Brontosaurus amplus YPM 1981 (Ostrom & McIntosh, 1966), among others, the ridge actually lies on sacral rib III. Some Camarasaurus specimens bear oblique ridges on their sacral ribs (e.g., AMNH 690; Osborn, 1904), but not the genotype specimen AMNH 5761. In the present analysis, Camarasaurus was thus scored as plesiomorphic.

C289: Sacral neural spines, lateral side, towards summit: flat, with only spinodiapophyseal lamina (spdl) well-developed (0); with distinct horizontal accessory laminae that connect spdl to pre- and/or postspinal lamina (1) (New; Fig. 75).

Figure 75 Diplodocid sacra.

Sacra of Brontosaurus parvus UW 15556 (A; modified from Hatcher, 1903) and Diplodocus hallorum AMNH 223 (B; modified from Osborn, 1899) in left lateral view. Note the flat (A; C289-0) instead of ornamented sacral neural spine top (B; C289-1), the spdl that extends ventrally to the diapophysis (A; C290-1), and the parallel (A; C291-0) in contrast to converging neural spines (B; C291-1). Abb.: DR, dorsal rib; il, ilium; SV, sacral vertebra. Scaled to the same height.

C290: Sacral neural spines, lateral view, spinodiapophyseal lamina: reduced to absent, does not connect summit and diapophysis (0); present and distinct, connects spine summit with diapophysis (1) (New; Fig. 75).

C291: Sacral neural spines, lateral view, spinodiapophyseal laminae (spdl): remain vertical and thus parallel to each other (0); spdl of neighboring spines converge (1) (New; Fig. 75).

Comments. Diplodocinae develop a very distinct dorsal widening of the sacral spdl. Together with the inclination of the spines towards the central portion of the sacrum, this often leads to a fusion of these anteroposteriorly widened dorsal ends of the spdl.

Caudal vertebrae

C292: Caudal neural spines, elliptical depression between lateral spinal lamina and postspinal lamina on dorsolateral surface: absent (0); present (1) (S07-75; modified; Fig. 76).

Figure 76 Anterior caudal vertebra of Diplodocus carnegii CM 94 in left lateral view.

Note various characters typical for the genus: a depression between the lateral spinal lamina and the postspinal lamina (C292-1), the large pleurocoel (C297-1), an additional pneumatic foramen posterodorsally in the caudal centrum (C298-1), the accessory lamina between pre- and postzygapophysis (C301-1), a dorsally widened lateral spinal lamina (C303-1), a pre-epipophysis (C311-1), the double anterior centrodiapophyseal lamina (C314-1), the distinct spinoprezygapophyseal lamina that extends onto the lateral surface of the spine (C318-1) and contacts the spinopostzygapophyseal lamina (C319-1), the presence of a prespinal lamina (C320-1) with a thickened anterior rim (C321-1), and the presence of a postspinal lamina (C323-1). Scale bar = 10 cm.

Comments. Sereno et al. (2007) initially defined the character as follows: ‘elliptical depression between spinodiapophyseal lamina and postspinal lamina on lateral neural spine.’ However, the spinal lamina they were most probably referring to (herein called lateral spinal lamina) is usually the united spol and sprl (at least in diplodocids). The character description has thus been reworded in order to clarify this. Sereno et al. (2007) recovered the presence of such a depression as a synapomorphy of Nigersaurinae, but actually it is present in any taxon with transversely widened posl, and spol that either fuse with the spdl or the posl. Anterior caudal vertebrae of Diplodocus are a good example for this, although they were scored as plesiomorphic by Sereno et al. (2007). Taxa without spdl or posl are scored as unknown.

C293: Caudal neural spines with triangular lateral processes: absent (0); present (1) (S07-76; Fig. 77).

Figure 77 Diplodocimorph anterior caudal vertebrae.

Anterior caudal vertebrae of Demandasaurus darwini MDS-RVII,610 (A; traced from Torcida Fernández-Baldor et al., 2011), Brontosaurus excelsus YPM 1980 (B; traced from Ostrom & McIntosh, 1966), and Diplodocus carnegii CM 84 (C; traced from Hatcher, 1901) in anterior view. Note the lateral triangular processes (B; C293-1), the mostly rectangular outline of the spine (B; C294-0), the wing-like transverse processes (A; C299-1), the convex prezygapophyses (B; C310-1), the laterally (C; C312-0) or dorsally directed ventral surface of the transverse process (A; C312-1), the notched neural spine top (C; C326-1), the gradual (C; C328-0) or abrupt distal expansion of the spine (B; C328-1), and the foramen piercing the transverse process (B; C350-0). Abb.: prsl, prespinal lamina; prz, prezygapophysis; sprl, spinoprezygapophyseal lamina; tp, transverse process. Scaled to same total height.

Comments. These processes correspond to the triangular lateral processes of dorsal neural spines, but do not appear to be correlated. They are restricted to anterior caudal vertebrae in the OTUs with the derived state included here, but because this is a simple presence–absence character, restriction to anterior caudal vertebrae is not necessary in the character definition.

C294: Posterior dorsal, sacral and anterior caudal neural spines, shape in anterior/posterior view: rectangular through most of length (0); ‘petal’ shaped, expanding transversely through 75% of its length and then tapering (1) (Calvo & Salgado, 1995; U98-117; Fig. 77).

Comments. Plesiomorphic caudal neural spines can still be transversely expanded at their ends. Also, taxa with gradually expanding neural spines that do not taper dorsally are herein scored as plesiomorphic, because without the tapering, the spines do not develop the ‘petal’ shape typical for rebbachisaurs and dicraeosaurs.

C295: First caudal centrum, articular face shape: flat (0); procoelous (1); opisthocoelous (2) (W02-116; modified).

Comments. The fourth state (biconvex) of Wilson (2002) was deleted because no OTU in this analysis has a biconvex first caudal vertebra. The probable brachiosaurid SMA 0009 and Demandasaurus have platycoel first caudal vertebrae (Torcida Fernández-Baldor et al., 2011; Carballido et al., 2012a), and are herein scored as opisthocoelous rather than flat.

C296: Anterior-most caudal centra, transverse cross-section: sub-circular with rounded ventral margin (0); ‘heart’-shaped with an acute ventral ridge (1) (Gilmore, 1936; U04b-22; wording modified; Fig. 78).

Figure 78 Diplodocid anterior caudal vertebrae.

Anterior caudal vertebrae of Apatosaurus ajax YPM 1860 (A) and Diplodocus sp. DMNS 462 (B) in ventral view. Note the ventral keel (A; C296-1), the ventral foramen (B; C305-1) within the ventral longitudinal hollow (B; C330-1), and the anteroposteriorly expanded distal end of the transverse process (A; C316-1). Abb.: ns, neural spine. Scaled to same centrum length.

Comments. Taxa with ventral hollows in their anterior caudal centra are scored as plesiomorphic, because the presence of the ventral ridge is regarded as the crucial trait for which this character codes.

C297: Anterior-most caudal centra, pneumatic fossae: reduced to absent (0); large pleurocoels (1) (New; Fig. 76).

Comments. Some apatosaur specimens and Supersaurus have distinct pleurocoels in their anterior-most caudal centra, whereas in anterior centra (as defined in Table 3), pleurocoels are reduced to foramina in these taxa (see e.g., Riggs, 1903). The current character is thus added to the usual one coding for pleurocoels in anterior caudal vertebrae in general.

C298: Anterior-most caudal vertebrae, additional pneumatic fossa on posterodorsal corner of centrum: absent (0); present (1) (New; Fig. 76).

Comments. In lateral views, these additional pneumatic foramina are often obscured by the transverse process.

C299: Anterior-most caudal transverse processes, shape: triangular, tapering distally (0); wing-like (1) (McIntosh, 1990b; Y93-44; modified; Fig. 77).

Comments. A transverse process is herein interpreted as wing-like if it has a distinct shoulder, i.e., an angled bump on its dorsolateral edge.

C300: Anterior-most caudal vertebrae, transition from ‘fan’-shaped to ‘normal’ caudal ribs: between Cd 1 and 2 (0); Cd4 and Cd5 (1); Cd5 and Cd6 (2); Cd6 and Cd7 (3); Cd7 and Cd8 or more posteriorly (4) (U04b-23; modified; Table S37).

C301: Anterior-most caudal neural arches, accessory lamina connecting pre- and postzygapophyses: absent (0); present (1) (New; Fig. 76).

Comments. This accessory lamina usually connects the postzygapophysis with the sprl.

C302: Anterior-most caudal neural spine (not including arch), height: less than 1.5 times centrum height (0); 1.5 times centrum height or more (1) (Y93-59; modified after W11-126; Table S38).

Comments. Neural spine height is measured from the dorsal edge of the postzygapophyses to the spine top, vertically. Centrum height is measured at the posterior articular surface. Yu (1993) used the entire neural arch height for the ratio and formulated it as a multistate character, restricted to the first two caudal vertebrae. The ratio is herein adapted following Upchurch & Mannion (2009), but keeping the restriction to the anterior-most elements, instead of including all anterior caudal vertebrae as implemented by Upchurch & Mannion (2009).

C303: Anterior-most caudal neural spines, lateral spinal lamina: has the same anteroposterior width ventrally and dorsally (0); expands anteroposteriorly towards its distal end, and becomes rugose (1) (Upchurch, Barrett & Dodson, 2004; Fig. 76).

Comments. SMA 0087 appears to show the plesiomorphic state. However, due to the bad preservation of the bones, the true morphology of the lateral spinal lamina is difficult to assess, and it might actually turn out to be widened as in apatosaurines, once all of the material is prepared.

C304: Anterior caudal centra (excluding the first), articular surface shape: amphiplatyan or amphicoelous (0); procoelous/distoplatyan (1); slightly procoelous (2); procoelous (3) (McIntosh, 1990b; R93-17; modified after G09-52; Table S37).

Comments. The definition of “slightly procoelous” in this character is the same as for the “slightly opisthocoelous” in posterior dorsal centra (see character 270). In diplodocids, the centra change their shape in anterior to middle caudal vertebrae from slightly procoelous to procoelous/distoplatyan to amphicoelous/amphiplatyan. This change occurs more posteriorly in Diplodocus than in Apatosaurus, for example. Therefore, specimens of the former genus have to be scored as slightly procoelous for this character, whereas Apatosaurus specimens are scored as procoelous/distoplatyan. However, more detailed studies about this transition are needed in order to score this character appropriately, because the specimens used herein generally show some correlation (within Flagellicaudata) of the development of procoely and the presence of wing-like transverse processes, which also mark more caudal vertebrae in Diplodocus than in less derived Flagellicaudata.

C305: Anterior caudal centra, ventral surface: without irregularly placed foramina (0); irregular foramina present on some anterior caudal vertebrae (1) (W11-133; Fig. 78).

Comments. Foramina can also be present in anterior caudal vertebrae without concave ventral surfaces (see Suuwassea emilieae ANS 21122; Harris, 2006b).

C306: Anterior caudal centra, pneumatopores (pleurocoels): absent (0); present (1) (McIntosh, 1990b; Y93-32).

Comments. Small pneumatopores also mark the lateral surfaces of the centra in non-diplodocine sauropods (e.g., Lourinhasaurus alenquerensis MIGM specimen, E Tschopp, pers. obs., 2012). The development of the pneumatopores as foramina or deep coels is described in character 307.

C307: Anterior caudal centra, pneumatopores: restricted to foramina (0); large coels present (1) (T13-173; modified; Fig. 79).

Figure 79 Flagellicaudatan anterior caudal vertebrae.

Anterior caudal vertebrae of Dicraeosaurus hansemanni MB.R.3774 (A), the indeterminate apatosaurine NHMUK R.3211 (B), and Barosaurus lentus YPM 429 (C) in left lateral view. Note the reduced (B; C307-0) or large pneumatopores (C; C307-1), the distinct posterior centrodiapophyseal and postzygodiapophyseal laminae (C; C315-1), and the postspinal lamina that projects dorsally (A; C324-1). Abb.: prz, prezygapophysis; sprl, spinoprezygapophyseal lamina; tp, transverse process. Scaled to same posterior centrum height.

Comments. This character only codes for the anterior caudal vertebrae, excluding the anterior-most elements with wing-like transverse processes. The presence of a large coel in the latter is coded for in character 297. Taxa without pneumatopores are scored as unknown.

C308: Anterior caudal centra, pneumatopores: disappear by caudal 15 (0); present until caudal 16 or more (1) (McIntosh, 2005; Table S37).

Comments. McIntosh (2005) recognized this as character distinguishing between Diplodocus and Barosaurus, but it is applied for the first time as a phylogenetic character.

C309: Anterior caudal centra, length: subequal amongst first 20 (0); more or less doubling over first 20 (1) (U98-133; modified; Table S39).

Comments. Lengths were compared between the shortest element among the first three, and the longest preserved vertebrae within Cd 17 and 22 (or if this part of the tail is lacking, the longest element preserved). Taxa with a ratio of 1.5 or more are scored as derived.

C310: Anterior caudal vertebrae, concavo-convex zygapophyseal articulation: absent (0); present (1) (Wilson, 2002; W11-143; Fig. 77).

Comments. This character is similar to the one for cervical vertebrae, which describes the flat versus convex prezygapophyses of diplodocine cervical vertebrae. Wilson (2002) suggested that convex prezygapophyses and concave postzygapophyses are diagnostic for Diplodocus, but Whitlock (2011a) showed that Barosaurus also showed the derived state. During the current study, some apatosaur specimens also were observed to have the apomorphic condition (BYU 1252-18531, UW 15556, YPM 1860, YPM 1980, YPM 1981).

C311: Anterior caudal prezygapophyses, pre-epipophysis laterally below articular facet: absent (0); present (1) (New; Fig. 76).

Comments. A rugose horizontal ridge marks the lateral surface of the prezygapophysis of Diplodocus and very few other taxa, below the articular facet. The position corresponds to where the pre-epipophysis of cervical vertebrae is located and is thus termed equally here.

C312: Anterior caudal vertebrae, transverse processes: ventral surface directed laterally or slightly ventrally (0); directed dorsally (1) (W11-125; Fig. 77).

Comments. This character describes the orientation of the ventral edge of the transverse process in anterior or posterior view.

C313: Anterior caudal transverse processes, anterior diapophyseal laminae (acdl, prdl): reduced or absent (0); present, well defined (1) (W02-129; modified; see Fig. 79C, 315-1 for equivalent in posterior diapophyseal laminae).

Comments. The original character (Wilson, 2002) was split in two, because the development of the posterior centrodiapophyseal and the postzygodiapophyseal laminae differs between Apatosaurus and Diplodocus.

C314: Anterior caudal transverse processes, anterior centrodiapophyseal lamina, shape: single (0); divided (1) (W02-130; Fig. 76).

Comments. In contrast to dicraeosaurids or more basal diplodocoids, diplodocids have wing-like transverse processes, which are anteriorly supported by two independent laminae, which both originate on the centrum and thus classify as acdl (and the latter thus as divided or double). In advanced diplodocines, the lower of the two acdl is furthermore branching in two towards the transverse process.

C315: Anterior caudal transverse processes, posterior diapophyseal laminae (pcdl, podl): reduced or absent (0); present, well defined (1) (W02-129; modified; Fig. 79).

C316: Anterior caudal transverse processes, anteroposteriorly expanded lateral extremities: absent (0); present (1) (New; Fig. 78).

Comments. Backwards curving transverse processes are not necessarily anteroposteriorly expanded.

C317: Anterior caudal neural spines, maximum mediolateral width to anteroposterior length ratio: <1.0 (0); 1.0 or greater (1) (U98-141; modified by M13-32; Table S38).

Comments. The anteroposterior length of the spine is measured at the same level as the maximum mediolateral width, perpendicular to the inclination of the neural spine. The unusual plesiomorphic state of SMA 0087 within the apatosaur specimens might be due to diagenetic transverse compression.

C318: Anterior caudal neural spines, spinoprezygapophyseal lamina: absent, or present as small short ridges that rapidly fade out into the anterolateral margin of the spine (0); present, extending onto lateral aspect of neural spine (1) (W02-121; modified by M12-145; Fig. 76).

C319: Anterior caudal neural spines, spinopre- and spinopostzygapophyseal laminae contact: absent (0); present (1) (W02-122; Fig. 76).

C320: Anterior caudal neural arches, prespinal lamina: absent (0); present (1) (U95; Fig. 76).

Comments. Sauropod anterior caudal neural spines are generally rugose anteriorly and posteriorly, but only derived eusauropods develop distinct ridges or laminae.

C321: Anterior caudal neural spines, thickened anterior rim of prespinal lamina: absent (0); present (1) (G05-54; Fig. 76).

Comments. Specimens without prespinal lamina are scored as unknown.

C322: Anterior caudal neural spines, prespinal lamina or rugosity: terminate at or beneath dorsal margin of neural spine (0); project dorsally above neural spine (1) (W11-131; modified; see Fig. 79A, 324-1 for equivalent in postspinal lamina).

Comments. The original character (Whitlock, 2011a) was split in two, because in the anterior caudal vertebrae of Cetiosauriscus stewarti NHMUK R.3078 only the postspinal rugosity expands dorsally above the spine summit (Woodward, 1905). The character description was slightly changed in order to include taxa without distinct prsl.

C323: Anterior caudal neural arches, postspinal lamina: absent (0); present (1) (U95; Fig. 76).

Comments. See character 320. The two characters coding for the presence of pre- or postspinal laminae, are scored equally in the present analysis, as also in Wilson (2002), and might thus prove correlated in future. They were both retained herein as they distinguish between basal and derived non-neosauropod eusauropods and should thus have no influence on the relationships between ingroup diplodocids.

C324: Anterior caudal neural spines, postspinal lamina or rugosity: terminate at or beneath dorsal margin of neural spine (0); project dorsally above neural spine (1) (W11-131; modified; Fig. 79).

Comments. See character 322.

C325: Anterior caudal neural arches; hyposphenal ridge on posterior face of neural arch; present (0); absent (1) (U95; polarity reversed by M12-142; Fig. 80).

Figure 80 Anterior caudal vertebra of Dicraeosaurus hansemanni MB.R.3774 in posterior view.

Note the hyposphenal ridge (C325-0). Abb.: cpol, centropostzygapophyseal lamina; posl, postspinal lamina; poz, postzygapophysis; tp, transverse process. Scale bar = 10 cm.

C326: Anterior caudal neural spines, shape: single (0); slightly bifurcate anteriorly (1) (W11-139; Fig. 77).

Comments. Anterior caudal neural spines can be bifid in two ways: anteroposteriorly and transversely. The former is coded for in characters 322 and 324, whereas the latter is described in the present character.

C327: Anterior caudal neural spines, maximum mediolateral width to minimum mediolateral width ratio: <2.0 (0); 2.0 or greater (1) (C08-239; Taylor, 2009; modified by M13-34; Table S38).

C328: Anterior caudal neural spines, lateral expansion at distal end: gradual, expanding through the last third of the neural spine (0); abrupt, restricted to distal fourth of neural spine (1) (New; Fig. 77).

C329: Anterior and mid-caudal vertebrae, ventrolateral ridges: absent (0); present (1) (U04a-183; Fig. 81).

Figure 81 Diplodocid mid-caudal vertebrae.

Mid-caudal vertebra of SMA 0087 (A) and Diplodocus hallorum AMNH 223 (B) in right (A) and left (B) lateral view. Note the ventrolateral (A; C329-1) and lateral ridges (A; C333-1), the flat ventral border of the centrum (B; 335-1), the anteriorly shifted neural arch (B; C337-1), the differing inclinations of the neural spine (C340), which overhang the postzygapophyses (A; C343-0), or not (B; C343-1). Abb.: ns, neural spine; prz, prezygapophysis. Scaled to the same anterior articular surface height.

Comments. Two horizontal ridges mark some diplodocid caudal centra: a lateral ridge and a ventrolateral ridge. Usually, only one of the two is present, which is interpreted as the lateral ridge, given its often rather dorsal position. The ventrolateral ridge as used herein does not describe the borders of the ventral longitudinal hollow of advanced diplodocines.

C330: Anterior and mid-caudal centra, ventral longitudinal hollow: absent (0); present (1) (McIntosh, 1990b; Y93-63; Fig. 78).

Comments. A ventral hollow is herein interpreted to be a longitudinal concavity occupying the entire ventral surface. Various taxa have very distinct posterior chevron facets with distinct ridges leading to them, thus creating a posteriorly concave ventral surface. However, these ridges often fade anteriorly. In some anterior diplodocine caudal centra, longitudinal struts subdivide the ventral hollow (e.g., Tornieria africana SMNS 12141a; Remes, 2006).

C331: Anterior- and mid-caudal vertebrae, ventral hollow depth: shallow, 10 mm or less (0); deep, >10 mm (1) (Curtice, 1996; Table S39).

Comments. Ventral hollow depth is used as a character distinguishing between Diplodocus and Barosaurus (Curtice, 1996; McIntosh, 2005). Curtice (1996) showed that a caudal centra with a ventral hollow depth of more than 10 mm can be confidently identified as Diplodocus, whereas shallower centra are typical for less derived diplodocines. Only very limited measurements were available, and the scoring was mainly based on descriptions and thus the subjective opinion of the respective authors. An interesting case is present in Tornieria, where the only preserved caudal vertebra of the holotype specimen (SMNS 12141a, Cd 2) has a deep ventral hollow, whereas the medial caudal vertebra of skeleton k (MB.R.2913) is only shallowly excavated (Remes, 2006). More detailed research is needed in order to sort this out.

C332: Mid-caudal vertebrae, ratio of centrum length to posterior height: <1,7 (0); 1,7 or greater (1) (Y93-45; modified; Table S39).

Comments. Usually, this character is included in analyses with its state boundary set at 2. In the present analysis, it was regarded more useful to put the boundary at 1.7, because some diplodocine taxa have ratios between 1.7 and 2. Generally, the ratio increases in more posterior elements, therefore specimens with only anterior mid-caudal vertebrae preserved (e.g., Diplodocus longus YPM 1920, see McIntosh & Carpenter, 1998) most probably would have higher ratios than indicated in the table.

C333: Mid-caudal vertebrae, lateral surface of centra: without longitudinal ridge at midheight (0); longitudinal ridge present, centra hexagonal in anterior/posterior view (1) (Upchurch & Martin, 2002; U04a-186; modified by W11-146; Fig. 81).

Comments. This ridge is not the same as the ventrolateral ridge described above, which is located below midheight.

C334: Mid-caudal centra, articular surface shape: cylindrical (0); quadrangular (1); trapezoidal (2); with flat ventral margin but rounded lateral edges (3) (W02-131; modified; Fig. 82).

Figure 82 Eusauropod mid-caudal vertebrae.

Mid-caudal vertebrae of Losillasaurus giganteus MCNV Lo-32 (A), Isisaurus colberti ISIR335/42 (B; traced from Jain & Bandyopadhyay, 1997), Diplodocus sp. AMNH 655 (C), and Barosaurus lentus AMNH 6341 (D) in anterior view, illustrating the four states of character 334 (A, circular; B, quadrangular; C, trapezoidal; D, flat ventral margin with rounded lateral edges). Abb.: nc, neural canal; ns, neural spine; prz, prezygapophysis. Scaled to same anterior surface height.

Comments. The character was modified in order to be able to code for the various intermediate states between cylindrical, quadrangular, and triangular as described by earlier workers (Gallina & Apesteguía, 2005; Carballido et al., 2012b). Articular surfaces of a rather hexagonal shape are scored as cylindrical, because the hexagonal shape is created by the lateral ridge described in character 333.

C335: Mid-caudal centra ventral surface in lateral view: gently curved (0); greater portion straight, with expansions on both ends to form the chevron facets restricted to about last fourth of centrum length (1) (New; Fig. 81).

Comments. This description applies especially for anterior mid-caudal elements, more posterior vertebrae of derived specimens tend to develop a more gentle curvature. This can create problems in taxa preserving only posterior mid-caudal vertebrae. For instance, Tornieria specimen k is herein scored as plesiomorphic for this character. Caudal vertebrae from trench dd, however, indicate that Tornieria actually might show the derived state, but these have not been found in articulation, and because anatomical overlap with the referred specimens included herein is minimal, their attribution to the species should be regarded as doubtful.

C336: Mid-caudal posterior articular surface: concave (0); flat (1); convex (2) (New; Table S37).

C337: Mid-caudal neural arches: over the midpoint of the centrum with approximately subequal amounts of the centrum exposed at either end (0); on the anterior half of the centrum (1) (Huene, 1929; S97-15; Fig. 81).

Comments. For this character, the distance between pre- and postzygapophyses and their location above the vertebral centrum is regarded as reference. The pedicels can still be dislocated anteriorly in plesiomorphic taxa. This character is generally used as a titanosauriform synapomorphy (Salgado, Coria & Calvo, 1997; Wilson, 2002), but also is convergently present in some Diplodocus specimens (e.g., AMNH 223, or USNM 10865).

C338: Mid-caudal prezygapophyses: free (0); posteriorly interconnected by a transverse ridge, creating a triangular fossa together with the spinoprezygapophyseal laminae (1) (New; Fig. 83).

Figure 83 Mid-caudal vertebra of Diplodocus longus YPM 1920 in dorsal view.

Note the transverse ridge connecting the prezygapophyses posteriorly (C338-1). Abb.: poz, postzygapophysis; prz, prezygapophysis; sprl, spinoprezygapophyseal lamina. Scale bar = 5 cm.

Comments. This transverse lamina marks the caudal vertebrae of Diplodocus longus YPM 1920 and might prove a valid autapomorphy for the species in the future.

C339: Mid-caudal prezygapophyses position: terminate at or behind anterior edge of centrum (0); project considerably beyond anterior edge of centrum (1) (New).

Comments. Only taxa where the prezygapophyses clearly overhang the centrum (i.e., recognizable without any need of measuring) are scored as derived.

C340: Mid-caudal neural spines, orientation: directed posteriorly (0); vertical (1) (McIntosh, 1990a; S97-10; modified; Fig. 81).

C341: Mid-caudal neural arch, anterior extreme of spine summit: smooth (0); developing a short anterior or anterodorsal projection, such that anterior edge of spine becomes slightly concave (1) (New; Fig. 84).

Figure 84 Eusauropod mid-caudal vertebrae.

Mid-caudal vertebrae of Cetiosauriscus stewarti NHMUK R.3078 (A; traced from Woodward, 1905) and Supersaurus vivianae WDC DMJ-021 (B; traced from a photo by D Lovelace) in left lateral view, illustrating the anterodorsal projection on the spine top (B; C341-1), and the posteriorly elongated neural spine (A; C344-0). Abb.: lr, lateral ridge; poz, postzygapophysis; prz, prezygapophysis. Scaled to same total vertebral height.

Comments. Such a spur might also be interpreted as pathologic or ontogenetic. However, its presence in the juvenile to subadult Apatosaurus (= Camarasaurus) grandis YPM 1901 suggests that ontogeny can probably be excluded as a cause. More studies are needed in order to confirm or refute pathology, in the meanwhile the character is kept in the analysis.

C342: Mid- and posterior caudal vertebral centra, articular surfaces: subequal in width and height or higher than wide (0); considerably wider than high (1) (S97-34; modified; Table S39).

Comments. A ratio of 1.2 or greater is regarded as considerably wider than high.

C343: Mid- and posterior caudal neural spines: spine summit overhangs postzygapophyses considerably posteriorly (0); posterior end of spine summit more or less straight above postzygapophyses (1) (New; Fig. 81).

C344: Mid- and posterior caudal spines: elongate and strongly caudally directed, extending over more than 50% of length of succeeding vertebral centrum (0); short, not extending far beyond caudal articular facet of centrum (1) (R09-132; polarity reversed; Fig. 84).

C345: Posterior caudal prezygapophyses position: terminate at or behind anterior edge of centrum (0); project beyond anterior edge of centrum (1) (New).

C346: Distal-most caudal centra, articular face shape: platycoelous (0); biconvex (1) (Wilson & Carrano, 1999; W02-136; Table S37).

Comments. Taxa without distal caudal vertebrae are scored as unknown.

C347: Distal-most caudal centra, length-to-height ratio: <4.0 (0); 4.0–6.5 (1); >6.5 (2) (U98-134; modified after Wilson, Martinez & Alcober, 1999; Table S39).

C348: Distal-most caudal centra, number: ten or fewer (0); more than 30 (1) (W02-138; modified).

Comments. The character was modified such that it was not restricted to distal-most ‘biconvex’ caudal centra as in Wilson (2002).

C349: Caudal ribs, last occurs on: Cd 12 or more anteriorly (0); Cd 13 (1); Cd 14 (2); Cd 15–17 (3); Cd 18 or more posteriorly (4) (Holland, 1915a; Gilmore, 1936; U04b-24; modified; Table S37). Unordered.

Comments. Upchurch, Tomida & Barrett (2004), who were the first to include this positional character in a phylogenetic analysis, only distinguished between two states: Cd 14 and/or Cd 12. However, enlarging the taxon list, a greater variety becomes evident (Table S37). The state description was thus adapted accordingly. The character is left unordered because no obvious step-like evolution is recognizable.

C350: Anterior, ‘fan’-shaped caudal ribs, foramen: present (0); absent (1) (Gilmore, 1936; U04b-25; polarity reversed; Fig. 77).

Comments. Polarity was reversed herein given the different taxon sampling compared to Upchurch, Tomida & Barrett (2004).

Chevrons

C351: Chevrons, ‘crus’ bridging haemal canal: absent in some (0); present in all (1) (Y93-47; modified after M12-162).

Comments. Additive binary coding is preferred here in order to be able to code incomplete tails (following Mannion et al., 2012).

C352: Chevrons, ‘crus’ bridging haemal canal: present in some (0); absent in all (1) (Y93-47; modified after M12-163; Fig. 85).

Figure 85 Anterior chevron of Apatosaurus ajax YPM 1860 in anterior, right lateral, and posterior view (left to right).

Note the crus bridging the haemal canal dorsally (broken here; C352-0), the anterior, longitudinal median ridge (C354-1), and the step-like posterior expansion of the distal blade (C355-1). Abb.: db, distal blade; hc, haemal canal. Scale bar = 10 cm.

Comments. See character 351.

C353: Chevrons with anterior and posterior projections: present (0); absent (1) (McIntosh, 1989; R93-18; modified; Fig. 86).

Figure 86 Mid-chevron of Diplodocus hallorum AMNH 223 in dorsal, left lateral, and ventral view (top-bottom).

Note the anterior and posterior projections (C353-1), the rugose horizontal ridge (C356-1), and the medial fossa (C357-1). Abb.: pas, proximal articular surface. Scale bar = 5 cm.

Comments. This character describes the oft-termed ‘forked chevrons’ that inspired Marsh (1878) to name the specimen YPM 1920 Diplodocus (= double beam).

C354: Anterior chevrons, longitudinal median ridge on anterior surface: absent (0); present (1) (New; Fig. 85).

Comments. The ridge extends proximodistally.

C355: Anterior chevrons, posterior edge of distal blade in lateral view: continuous (0); posteriorly expanded in a step-like fashion (1) (New; Fig. 85).

C356: Anterior mid-chevrons, lateral surface: smooth (0); marked by a horizontal ridge right below articulation surfaces (1) (New; Fig. 86).

Comments. The ridge can be quite broad, but it is always rugose. Anterior mid-chevrons are meant to be the first elements with anterior projections on the distal blade.

C357: Middle chevrons, distinct fossae on medial surfaces of proximal branches: absent (0); present (1) (New; Fig. 86).

C358: Forked chevrons, anteroposterior length: short, about 50% of relative vertebral centrum length (0); elongate, approaching corresponding vertebral centrum length (1) (McIntosh, 1995).

Comments. The increased relative length of the chevron compared to its corresponding caudal vertebra was proposed as a useful character to distinguish Diplodocus from Apatosaurus by McIntosh (1995), and is herein used for the first time in a phylogenetic analysis.

Pectoral girdle

C359: Scapular length/minimum blade breadth: >5.5 (0); 5.5 or less (1) (C12b-236; polarity reversed; Table S40).

Comments. Measurements are taken from figures in lateral view, ignoring the proximodistal curve of the scapula. Greatest length follows the long axis of the scapula, such that orientation within the articulated skeleton is not taken into account, because this is still debated (see Schwarz, Frey & Meyer, 2007a; Remes, 2008; Hohn, 2011). Minimum blade breadth is measured perpendicular to the long axis.

C360: Scapular acromion length/scapular length: >0.54 (0); 0.46–0.54 (1); <0.46 (2) (G05-68; modified; Table S40). Ordered.

Comments. Measurements were taken from figures in lateral view. Acromion length is measured perpendicular to scapular length, between horizontal lines extending through the ventral- and dorsal-most points of the acromion, with the distal blade oriented horizontally.

C361: Scapula, orientation of scapular, angle with coracoid articulation: >80° (0); 80° or less (1) (W02-151; modified; Table S40).

Comments. The angle is measured from figures or photographs in lateral view.

C362: Scapula, angle between acromial ridge and distal blade: <70° (0); 70° −81° (1); >81° (2) (Riggs, 1903; Carpenter & McIntosh, 1994; U04b-26; modified; Table S40). Unordered.

Comments. The angle to be measured lies between the dorsal half of the acromial ridge and the long axis of the scapular blade. An additional state was added to the original version (Upchurch, Tomida & Barrett, 2004), in order to be able to score specimens with intermediate ratios. The character is left unordered because no obvious evolutionary trend is observable.

C363: Scapular acromion process, dorsal part of posterior margin: convex or straight (0); U-shaped concavity (1) (Sereno et al., 1999; S07-88; Fig. 87).

Figure 87 Scapula outlines of Diplodocoidea.

Scapula outlines of Haplocanthosaurus priscus CM 879 (A), Limaysaurus tessonei MUCPv-205 (B), Apatosaurus louisae CM 3018 (C; all traced from Mannion, 2009), and Diplodocus hallorum AMNH 223 (D; traced from Osborn, 1899). Note the concave dorsal border of the acromion process (B; C363-1), the acromion process that reaches almost half the scapular length (D; C364-1), the different shapes of the acromial edge (straight, C, C367-0; with rounded expansion distally, A, C367-1; raquet-shaped, B, C367-2), the ventrally curving ventral margin (A; C368-1), and the subtriangular process (D; C370-1). Abb.: acm, acromion; ca, coracoid articulation; db, distal blade. Scaled to same scapular length.

C364: Scapular, acromion process position: lies near the glenoid level (0); lies nearly at midpoint of scapular body (1) (C12b-238; Fig. 87).

Comments. The position of the acromion process relative to the glenoid has to be checked with the long axis of the distal blade oriented horizontally.

C365: Scapula, area posterior to acromial ridge and distal blade: is excavated (0); is flat or slightly convex (1) (U04b-27; Fig. 88).

Figure 88 Apatosaurine scapulae.

Right scapulae of “Elosaurus” parvus CM 566 (A) and Brontosaurus excelsus YPM 1980 (B) in lateral view. Note the excavated area between the acromial edge and the distal blade (A; C365-0) and the flat muscle scar at the base of the distal blade (B; C369-1). Abb.: acr, acromial ridge; db, distal blade. Scaled to same length.

Comments. This character describes the area posterior to the acromial ridge and dorsal to the distal blade, where the two meet.

C366: Scapular glenoid, orientation: relatively flat or laterally facing (0); strongly beveled medially (1) (W98-104).

Comments. The medially beveled glenoid surface was proposed as autapomorphic for Apatosaurus (Wilson, 2002), but Upchurch, Tomida & Barrett (2004) showed that the orientation was actually variable within Apatosaurus specimens, which is confirmed herein.

C367: Scapular blade, acromial edge: straight (0); rounded expansion at distal end (1); racquet-shaped (2) (Marsh, 1896; W98-109; modified after W02-152; Fig. 87).

C368: Scapular blade, ventral edge in lateral view: is straight (0); curves ventrally towards its distal end (1) (Marsh, 1896; U04b-28; wording modified; Fig. 87).

Comments. Whereas the original character (Upchurch, Tomida & Barrett, 2004) described the entire blade, the derived ventral curving is here restricted to the ventral edge of the blade.

C369: Scapula: without semi-ovate, flat muscle scar just distal to glenoid on scapular shaft (0); scar present (1) (W11-158; Fig. 88).

Comments. The scar described herein lies on the lateral side of the blade.

C370: Scapular blade, subtriangular projection on anterior portion of ventral edge: absent (0); present (1) (G05-66; Fig. 87).

Comments. In Diplodocus sp. AMNH 223, there are two eminences close to each other (E Tschopp, pers. obs., 2011). They are considered equivalent to the single subtriangular projection of this character.

C371: Scapular blade, expansion of distal end: wide (at least 2 times narrowest width of shaft in lateral view) (0); narrow (<2 times narrowest width of shaft) (1) (Y93-48; modified; Table S40).

Comments. Measurements are taken perpendicular to the long axis of the blade.

C372: Coracoid, anteroventral margin shape: rounded (0); rectangular (1) (Marsh, 1896; W02-156; Fig. 89).

Figure 89 Neosauropod coracoids.

Left coracoids of Amphicoelias altus AMNH 5764? (A) and Apatosaurus ajax YPM 1860 (B; traced from Bakker, 1998) in anterolateral view. Note the rounded (A; C372-0) instead of rectangular shape (B; C372-1), and the deep (A; C373-1) in contrast to shallow infraglenoid groove (B; C373-0). Abb.: CF, coracoid foramen. Scaled to the same height.

C373: Coracoid, infraglenoid groove: reduced to absent (0); present and distinct (1) (C12b-245; modified; Fig. 89).

C374: Sternal plates, shape: subcircular or oval (0); subtriangular with widened posterior border (1); elliptical to crescentic, with concave lateral margin (2) (C95-39; modified; Fig. 90). Unordered.

Figure 90 Neosauropod sternal plates.

Right (A, B) and left (C) sternal plates of Giraffatitan brancai MB.R.2181 (A; modified from Janensch, 1961), Brontosaurus amplus YPM 1981 (B), and Tornieria africana MB.R.2726 (C) in ventral view. Note the different shapes (oval, B, C374-0; triangular, C, C374-1; crescentic, A, C374-2), the longitudinal ridge (A; C375-1), the anterior dorsoventral thickening (C; C376-1), and the straight posterior border (C; C377-1). Scaled to same length.

Comments. The subtriangular shape was added to the original version of Calvo & Salgado (1995) in order to better describe the difference between typical basal neosauropod or macronarian, and diplodocid shape. The character is treated as unordered, because none of the states can convincingly be interpreted as intermediate.

C375: Sternal plate, ridge on the ventral surface: absent (0); broad and shallow, or elongate and prominent (1) (U04a-213; wording modified; Fig. 90).

C376: Sternal plate, anterior end: expanded dorsoventrally (0); flat, not expanded (1) (Tschopp & Mateus, 2012; Fig. 90).

C377: Sternal plate, posterior border: convex (0); straight (1) (G03-29; modified; Fig. 90).

Comments. The true shape of the posterior border can sometimes be obscured due to the presence of fused sternal ribs (Tschopp & Mateus, 2012).

Forelimb

C378: Forelimb: hindlimb length ratio: 0.76 or greater (0); less than 0.76 (1) (U95; U98-158; modified; Table S41).

Comments. Forelimb length is the sum of the lengths of the humerus, radius, and metacarpal III; hindlimb length the sum of the lengths of femur, tibia, and metatarsal III.

C379: Humerus-to-femur ratio: <0.7 (0); 0.7–0.76 (1); 0.77–0.89 (2); = or >0.90 (3) (McIntosh, 1990a; modified; Table S42). Ordered.

Comments. State boundaries are chosen such that the generally accepted genera Apatosaurus and Diplodocus can be distinguished from Tornieria and Barosaurus.

C380: Humerus, RI (sensu Wilson & Upchurch, 2003): gracile (less than 0.27) (0); medium (0.28–0.32) (1); robust (more than 0.33) (2) (C12b-256; Table S43). Ordered.

Comments. The humerus RI was defined as the mean between proximal, distal, and midshaft transverse widths, divided by humerus length (Wilson & Upchurch, 2003). Scores for taxa where no measurements were available were taken from Carballido et al. (2012b).

C381: Humerus, shaft twist: minor to absent (0); high, distal articular surface twisted by at least 30° compared to proximal articular surface (1) (Gilmore, 1932; Table S43).

Comments. This angle is difficult to measure due to lacking references. It was proposed as a distinguishing feature of Diplodocus (Gilmore, 1932) and is here included into a phylogenetic analysis for the first time.

C382: Humerus, midshaft cross-section, shape: circular, transverse diameter: anteroposterior diameter ratio is 1.5 or lower (usually close to 1.3) (0); elliptical, transverse diameter: anteroposterior diameter ratio is greater than 1.5 (usually close to 1.8) (1) (W02-162; modified by M12-170; Table S43).

C383: Humerus, pronounced proximolateral corner: absent (0); present (1) (U98-160; Fig. 91).

Figure 91 Eusauropod humeri.

Humeri of Turiasaurus riodevensis CPT 1195 (A; traced from Royo-Torres, Cobos & Alcalá, 2006) and Suuwassea emilieae ANS 21122 (B; traced from Harris, 2007) in anterior view. Note the pronounced proximolateral corner (B; C383-1), the symmetrical proximal transverse expansion (B; C384-1), the unexpanded (A; C385-1) or expanded lateral edges (B; C385-0), and the tubercle marking the center of the proximal concavity (B; C386-1). Abb.: dpc, deltopectoral crest. Scaled to same length.

Comments. A pronounced proximolateral corner forms a weak hump in anterior or posterior view.

C384: Humerus, proximal expansion: more or less symmetrical (0); asymmetrical, proximomedial corner much more pronounced than proximolateral one (1) (Wilhite, 2005; Fig. 91).

Comments. The differing expansions were found to be taxonomically significant (Wilhite, 2005), but have not been previously included in any phylogenetic analysis. This character forms an additive binary character together with character 385.

C385: Humerus, proximal end expanded laterally in anterior/proximal view: expanded, lateral margin concave in anterior/posterior view (0); not expanded (1) (C05-266; polarity reversed; Fig. 91).

Comments. Polarity was reversed compared to the original description (Curry Rogers, 2005), due to the differing taxon sampling.

C386: Humerus, shallow, but distinct rugose tubercle at the center of the concave proximal portion of the anterior surface: absent (0); present (1) (New; Fig. 91).

C387: Ulna to humerus length: <0.65 (0); 0.66–0.76 (1); >0.76 (2) (Janensch, 1929b; Table S44). Ordered.

Comments. The states were defined in order to include the majority of diplodocids in the same state.

C388: Ulna, proximal condylar processes: subequal in length (0); anterior arm longer (1) (W02-166; Table S45).

Comments. The state boundary is here set at 1.1, as this follows best higher-level taxonomy.

C389: Ulna, proximal articular surface, angle between anterior and lateral branch: 90° (0); acute (1) (New; Table S45).

Comments. Taxa with angles greater than 83° were scored as plesiomorphic.

C390: Ulna, distal transverse expansion: slight, <1.3 times minimum shaft width (min sw) (0); wide, 1.3 times min sw or greater (1) (New; Table S45).

Comments. Some width measurements published do not state explicitly if they are taken transversely or anteroposteriorly; they just report maximum distal width. Anteroposterior width is often much greater than transverse width in distal surfaces of the sauropod ulnae. This leads to exaggerated ratios, if erroneously included here. Also, particularly disarticulated ulnae, where both proximal processes are equally long, are difficult to orient properly. Nonetheless, the differences in these ratios still appear significant.

C391: Radius, maximum diameter of the proximal end divided by greatest length: <0.3 (0); 0.3 or greater (1) (McIntosh, 1990a; U95; modified by M13-45; Table S46).

Comments. Maximum diameter can be width or depth.

C392: Radius, distal articular surface for ulna: reduced and relatively smooth (0); well developed with one or two distinct longitudinal ridges (1) (New; Fig. 92).

Figure 92 Distal half of radius of Dyslocosaurus polyonychius AC 663.

Note the very weak ridges for the articulation with the ulna (C392-0). Scale bar = 10 cm.

C393: Radius, distal condyle orientation in anterior view: perpendicular or beveled less than 15° to long axis of shaft (0); beveled at least 15° to long axis of shaft (1) (Curry Rogers & Forster, 2001; W02-171; modified; Table S46).

Comments. As stated by Mannion et al. (2013), the beveling of the distal surface often only affects the lateral half of the distal end. Given the different scope of the phylogenetic analysis, character state boundaries are different herein compared to Mannion et al. (2013).

C394: Radius, distal breadth: <1.8 times larger than midshaft breadth (0); at least 1.8 times midshaft breadth (1) (W02-170; modified).

Comments. Breadth is measured mediolaterally.

C395: Carpus, number of carpal bones: 3 or more (0); 2 (1); 1 or less (2) (McIntosh, 1990b; U98-163 to 165; modified). Ordered.

Comments. The character was initially proposed as three additional binary characters (Upchurch, 1998). These were combined here to a single three-state character. Even though SMA 0011 was found with only one carpal preserved, its articulated position directly below the radius and articulation with the first two to three metacarpals suggest that a second element was present. Such a presence is also indicated by the proximodistal width of the preserved element, which in articulation would create a large gap between the ulna and the lateral metacarpals. A similar case can be seen in the putative Diplodocus manus described by Bedell & Trexler (2005). The opposite can be seen in apatosaurs, where the only carpal lies above mc II–IV, is proximodistally flattened, and metacarpals I and V are proximally dislocated in respect to the inner elements (CM 3018, UW 15556; Hatcher, 1902; Gilmore, 1936; Bonnan, 2003). Due to the probable gradual decrease in the number of carpal bones the character is treated as ordered.

C396: Carpals: block-like (0); proximodistally compressed discs (1) (New; Fig. 93).

Figure 93 Diplodocid carpals.

Carpal elements of Galeamopus sp. SMA 0011 (A) and Brontosaurus parvus UW 15556 (B; traced from Bonnan, 2003) in anterior view, illustrating the two different shapes described in C396: (0) block-like (A), and (1) disc-like (B). Scaled to the same transverse width.

C397: Metacarpus, shape: spreading (0); bound, with subparallel shafts and articular surfaces that extend half their length (1) (W02-175).

C398: Metacarpals, shape of proximal surface in articulation: gently curving, forming a 90° arc (0); U-shaped, subtending a 270° arc (1) (W02-176).

C399: Metacarpus, ratio of longest metacarpal to radius: <0.40 (0); 0.40 or greater (1) (C95-49; modified by M13-52; Table S47).

Comments. The longest metacarpal is usually mc II or mc III.

C400: Metacarpal I, length: shorter than IV (0); longer than IV (1) (W98-94; Table S47).

Comments. The state boundary applied herein lies at 1.0.

C401: Metacarpal I, proximal end dorsoventral height to mediolateral width ratio: <1.8 (0); 1.8 or greater (1) (Apesteguía, 2005; Mannion & Calvo, 2011; M13-53; Table S47).

Comments. Mannion et al. (2013) were the first to include this ratio in a phylogenetic analysis.

C402: Metacarpal III, robustness (length/distal transverse width): robust, <2.9 (0); intermediate, 2.9–3.5 (1); slender, >3.5 (2) (Bedell & Trexler, 2005; Table S47). Ordered.

Comments. Suggested as a distinguishing character between Diplodocus and Apatosaurus, and especially between WDC-FS001A and HMNS 175 (Bedell & Trexler, 2005), which are both probably not Diplodocus (see below), metacarpal robustness is herein used for the first time as a character in a phylogenetic analysis.

C403: Metacarpal V, proximal articular surface: subequal to smaller than (0); or significantly larger than proximal articular surface of mc III and IV (1) (Janensch, 1929b; Fig. 94).

Figure 94 Metacarpals III–V of Apatosaurus louisae CM 3018.

Articulated metacarpals III–V of Apatosaurus louisae CM 3018 in proximal view (traced from Gilmore, 1936), showing the greatly enlarged mc V, in comparison to mc III and IV (C403-1).

Comments. An enlarged proximal articular surface of mc V can be seen in Apatosaurus louisae CM 3018 (Gilmore, 1936). However, this does not seem to be the case in another apatosaur specimen (NSMT-PV 20375; Upchurch, Tomida & Barrett, 2004), such that the derived state might prove an autapomorphy of the species A. louisae. A similar development can be seen in the manus of Janenschia robusta (Janensch, 1922).

C404: Manual phalanx I-1, flange-like sheet of bone projecting from the proximoventral margin: absent (0); present (1) (Hatcher, 1902; Gilmore, 1936; U04b-31; Fig. 95).

Figure 95 Manual phalanx phm I-1 of Apatosaurinae indet. NSMT-PV 20375 in medial view (traced from Upchurch, Tomida & Barrett, 2004).

Note the proximoventral lip-like projection (C404-1).

Pelvic girdle

C405: Ilium, ratio of blade height above pubic peduncle to anteroposterior length: <0.40 (0); 0.40 or more (1) (New; Table S48).

Comments. Blade height is measured vertically above the base of the pubic pedicel, with the ischiadic tubercle and the anteroventral-most point of the preacetabular process oriented on a horizontal line.

C406: Iliac preacetabular process, shape: sharply pointed (0); blunt to semicircular anterior margin (1) (S97-17; Fig. 96).

Figure 96 Neosauropod ilia.

Right (A) and left (B) ilium of Brachiosaurus altithorax FMNH P25107 (A; modified from Riggs, 1904) and Diplodocus hallorum DMNS 1494 (B) in lateral view. Note the pointed (B; C406-0) or semicircular preacetabular process (A; C406-1), the straight (A; C409-0) or strongly convex dorsal edge (B; C409-1), the location of the highest point (anterior to pubic peduncle, A, C410-1; posterior to pubis peduncle, B, C410-1), the triangular fossa on the pubic peduncle base (B; C412-1), and the tubercle in the postacetabular region (A; C413-1). Abb.: prap, preacetabular process; pup, pubic peduncle. Scaled to same height.

Comments. A strict lateral view of the ilium is often misleading, given the anterolateral to lateral orientation of the preacetabular lobe. A posterolateral view would be preferable.

C407: Ilium, preacetabular process, orientation of anterior tip in dorsal view: pointing anterolaterally (0); pointing laterally (1) (W02-187; wording modified).

Comments. The perpendicular orientation of the preacetabular process was found as synapomorphic for derived titanosauriforms (Wilson, 2002), but they also occur in the holotype of ‘Apatosaurus’ minimus AMNH 675 (Mook, 1917).

C408: Ilium, angle between the ventral edge of anterior iliac lobe and the anterior surface of the pubis process: is ∼90° (0); is acute (1) (Gilmore, 1936; S97-18; polarity reversed by U04b-32).

C409: Ilium, dorsal margin shape: flat to slightly convex (0); semicircular (1) (W02-186; modified; Fig. 96).

Comments. Taxa with the derived state have uniformly convex dorsal margins, whereas taxa with the apomorphic state generally have a large straight portion.

C410: Ilium, highest point on dorsal margin: lies posterior to base of pubic process (0); lies anterior to base of pubic process (1) (U04a-245; Fig. 96).

Comments. The position of the highest point in respect to the pubic peduncle is assessed with the ischiadic tubercle and the anteroventral-most point of the preacetabular process lying on a horizontal line.

C411: Ilium, pubic peduncle (measured at the articular surface), anteroposterior to mediolateral width ratio: >0.80 (0); 0.80 or less (1) (Taylor, 2009; M13-57; modified; Table S48).

Comments. Mannion et al. (2013) was the first to include this character in a phylogenetic analysis, based on observations made by Taylor (2009). State boundaries are adapted herein from 0.5 to 0.8, given the different scope and thus taxon sampling of the present analysis.

C412: Ilium, triangular fossa laterally at base of pubic peduncle: absent (0); present (1) (New; Fig. 96).

Comments. The apex of this fossa points ventrally. Its presence was figured as well in the titanosaur Rocasaurus (Salgado & Azpilicueta, 2000), and described in Cetiosaurus (Upchurch & Martin, 2003) and Lirainosaurus (Díez Díaz, Pereda Suberbiola & Sanz, 2013), but it has never been included in a phylogenetic analysis.

C413: Ilium, distinct tubercle in the postacetabular region: absent (0); present (1) (C12a-334; Fig. 96).

Comments. The herein described tubercle is not the transverse widening of the dorsal edge towards its posterior end, but a second rugose area laterally on the blade (see Schwarz et al., 2007; Carballido et al., 2012a).

C414: Pubis, ambiens process development: small, confluent, not differentiated from anterior border of the pubis (0); evident, but not especially developed (1); prominent, hook-like (2) (McIntosh, 1990b; Y93-49; wording modified; Fig. 97). Ordered.

Figure 97 Neosauropod pubes.

Left (A, C) and right (B, reversed) pubis of Camarasaurus supremus AMNH 5761 (A; modified from Osborn & Mook, 1921), Dicraeosaurus hansemanni MB.R.4886 (B; modified from Janensch, 1961), and Brontosaurus excelsus YPM 1980 (C; modified from Ostrom & McIntosh, 1966) in lateral view. Note the different sizes of the ambiens process (C414, arrowheads: absent, A; hook-like, B; incipient, C). Abb.: ac, acetabular surface; ip, iliac peduncle; isa, ischial articular surface; of, obturator foramen. Scaled to same length.

Comments. The hook-like ambiens process is interpreted to represent an increased development of the incipient shape. This character is thus treated as ordered.

C415: Pubis, length of puboischial contact: less than 0.41 total length of pubis (0); 0.41 or more of total length of pubis (1) (C95-41; modified; Table S49).

Comments. Mannion et al. (2012) used a ratio of 0.45 as state boundary, but as shown in Table S49, 0.41 appears more appropriate for the present set of taxa.

C416: Pubis, participation in acetabulum: subequal to larger, compared to ischium (0); significantly smaller (1) (Janensch, 1961; Table S50).

Comments. A state boundary of 0.8 was used herein because the included OTUs show a large step from ratios below 0.75 to ratios greater than 0.83. The character was proposed as potentially useful to distinguish taxa by Janensch (1961). It is included in a phylogenetic analysis for the first time.

C417: Ischium, acetabular articular surface: maintains approximately the same transverse width throughout its length (0); is transversely narrower in its central portion and strongly expanded as it approaches the iliac and pubic articulations (1) (M12-180).

Comments. The narrow acetabular surface is only present in some rebbachisaurids (Mannion et al., 2012).

C418: Ischium, acetabular margin, in lateral view: flat or mildly concave (0); strongly concave, pubic articular surface forms an anterodorsal projection (1) (D12-104; modified by M13-252; Fig. 98).

Figure 98 Diplodocoid ischia.

Left ischium of Haplocanthosaurus priscus CM 572 (A; modified from Hatcher, 1903), Demandasaurus darwini MPS-RVII,18 (B; modified from Pereda Suberbiola et al., 2003), and Brontosaurus excelsus YPM 1980 (C; modified from Ostrom & McIntosh, 1966) in lateral (left) and distal (right) view. Note the flat (C; C418-0) in contrast to strongly concave acetabular margin (B; C418-1), the constricted neck of the iliac tubercle (B; C419-1), the elongate muscle scar on the proximal shaft (A; C421-1), the lateral fossa at the base of the blade (C; C422-1), the blade-like (B; C423-0) or medially expanded distal ends (C; C423-1), which form a more or less straight line (B; C424-1) or a V (C; C424-0), and can be straight (A; C426-0) or expanded dorsoventrally as well as transversely (C; C425-1). The light gray line in B indicates the distal view of the right ischium. Scaled to same length.

Comments. In some diplodocids (e.g., Apatosaurus excelsus YPM 1980, see Fig. 98), the lateroventral edge of the acetabular surface is strongly concave, whereas the mediodorsal margin forms a bony sheet extending straight from the iliac to the pubic articular surfaces. In lateral view, this configuration appears straight and was thus scored as plesiomorphic herein.

C419: Ischium, iliac peduncle: iliac peduncle straight or widening in smooth curve distally (0); narrow, with distinct ‘neck’ (1) (S07-98; Fig. 98).

C420: Ischia pubic articulation/anteroposterior length of pubic pedicel: <1.5 (0); 1.5 or greater (1) (S97-13; modified; Table S51).

Comments. Anteroposterior length of the pubic pedicel is measured perpendicular to the articular surface, from its ventral-most point to the point where it intersects with a line following the ventral edge of the distal shaft. A numerical state boundary was added to the original version of Salgado, Coria & Calvo (1997), which separates Macronaria from basal Eusauropoda, and most diplodocines from most apatosaurs (Table S51).

C421: Ischium, elongate muscle scar on proximal end: absent (0); present (1) (S07-99; Fig. 98).

Comments. We follow Mannion et al. (2012), in that the presence of a distinct ridge on the dorsolateral edge qualifies for the apomorphic state.

C422: Ischium, lateral fossa at base of shaft: absent (0); present (1) (Wilson, 2002; W11-176; Fig. 98).

Comments. The presence of such a fossa was interpreted as autapomorphic for Dicraeosaurus by Wilson (2002), and first included into a phylogenetic analysis by Whitlock (2011a). As interpreted herein, the fossa is longitudinally oriented and marks the dorsolateral edge of the shaft.

C423: Ischial distal shaft, shape: blade-like, medial and lateral depths subequal (0); triangular, depth of ischial shaft increases medially (1) (W98-9; polarity reversed by W11-171; Fig. 98).

C424: Ischial distal shafts, cross-sectional shape: V-shaped, forming an angle of nearly 50° with each other (0); flat, nearly coplanar (1) (U98-181; W98-88; Fig. 98).

C425: Ischial shaft, transverse distal expansion: absent (0); present (1) (W11-175; Fig. 98).

Comments. Due to the V-shaped distal end of the ischia, ‘transverse’ and ‘posterodorsal’ do not apply very well to the ingroup specimens. However, given the twist of the ischial shaft in the taxa with coplanar distal shafts, which results in almost horizontally oriented distal ends, the main expansion of diplodocid ischia should be regarded as transverse, even though in lateral view it would appear rather dorsoventral.

C426: Ischium, posterodorsal expansion of distal end: absent (0); present (1) (L07-235; Fig. 98).

Comments. See comment on transverse expansion in character 425.

Hindlimb

C427: Femur, robustness index (sensu Wilson & Upchurch, 2003): gracile, <0.22 (0); intermediate, 0.22–0.25 (1); robust, >0.25 (2) (Janensch, 1961; Table S52). Ordered.

Comments. Due to the gradual increase in the ratio across the sauropods included in our analysis, this character is treated as ordered.

C428: Femur, lateral bulge (marked by the lateral expansion and a dorsomedial orientation of the laterodorsal margin of the femur, which starts below the femur head ventral margin): absent (0); present (1) (S97-19; modified; Fig. 99).

Figure 99 Neosauropod femora.

Right femur of Giraffatitan brancai MB.R.2694 (A), Dicraeosaurus hansemanni MB.R.4886 (B; both modified from Janensch, 1961), and Tornieria africana SMNS 12140 (C; modified from Fraas, 1908) in anterior view. Note the lateral bulge (A; C428-1), the medial deflection of the femoral head (A; C429-1), the different positions of the highest point of the femoral head (C431), the stepped ventral margin of the head (B; C432-1), the nutrient foramen (B; C434-1), the fourth trochanter, which is visible in anterior view (A; C436-0), and the anteriorly extended distal articular surface of the condyle (C; C439-1). Scaled to same length.

Comments. The definition of this character changed in different phylogenetic analyses (e.g., Salgado, Coria & Calvo, 1997; Mannion et al., 2012). Here, we follow Mannion et al. (2012) in that we also score incipient lateral bulges as apomorphic.

C429: Femoral shaft, lateral margin shape: straight (0); proximal one-third deflected medially (1) (W02-199; Fig. 99).

Comments. The fact that the probable brachiosaurid juvenile SMA 0009 (in contrast to other brachisaurids) does not show any medial deflection might indicate that this character changes during ontogeny. This might be correlated with the weak development of the articular surface in juvenile specimens (Ikejiri, Tidwell & Trexler, 2005; Schwarz et al., 2007).

C430: Femur, cross-sectional shape: subequal to anteroposterior diameter (0); 125–150% anteroposterior diameter (1); at least 185% anteroposterior diameter (2) (Wilson & Smith, 1996; W02-198; Table S52). Ordered.

Comments. The character was added in order to distinguish between titanosauriforms, but it is also useful for the distinction of Amphicoelias altus AMNH 5764. Taxa scored but without entries in Table S52 are taken from Carballido et al. (2012b).

C431: Femoral head, position of highest point in anterior view: above point of maximum curvature of ventral edge of femoral head (0); laterally shifted, above main portion of shaft (1) (New; Fig. 99).

C432: Femur, ventral surface of head: confluent with shaft (0); stepped (1) (New; Fig. 99).

C433: Femur, greatest anteroposterior thickness of shaft: less than or approximately equal to half anteroposterior depth of distal articular condyles (0); much greater than half anteroposterior depth of distal articular condyles (1) (W11-179; Table S52).

Comments. The state boundary used herein is 0.6. Taxa scored for this character, but not having any values in Table S52, are taken from Whitlock (2011a).

C434: Femur, large nutrient foramen opening midshaft anteriorly on femur: absent (0); present (1) (Wilson, 2002; W11-182; Fig. 99).

Comments. Initially proposed as autapomorphy of Dicraeosaurus (Wilson, 2002), such a foramen is also present in some diplodocids (e.g., Diplodocus carnegii CM 94). Whitlock (2011a) was the first to include this character in a phylogenetic analysis.

C435: Femur, pronounced ridge on posterior surface between greater trochanter and head: absent (0); present (1) (S07-101).

Comments. The derived state is a synapomorphy for Nigersaurinae, convergently present in Rapetosaurus (Sereno et al., 2007; Curry Rogers, 2009).

C436: Femur, fourth trochanter: not visible in anterior view (0); prominent, visible in anterior view (1) (G05-76; modified by W11-178; Fig. 99).

Comments. In certain taxa, a small bulge is visible on the medial edge in anterior view, which represents the medially positioned, and prominent fourth trochanter.

C437: Femoral fourth trochanter, present as low rounded ridge (0); greatly reduced so that it is virtually absent (1) (M12-187).

Comments. A reduced fourth trochanter is synapomorphic for rebbachisaurs and some titanosauriforms (Torcida Fernández-Baldor et al., 2011; Mannion et al., 2012). The reduced fourth trochanter of the juvenile Elosaurus parvus CM 566 indicates that the reduction of this structure in rebbachisaurs and some titanosauriforms represents a paedomorphic feature.

C438: Femur, fourth trochanter, position: distally displaced (0); on proximal half of shaft (1) (Schwarz-Wings & Böhm, 2014; Table S52).

Comments. Distance between femoral head and fourth trochanter is measured to the distal end of the trochanter. Taxa with ratios of 0.4 or less are scored as apomorphic.

C439: Femur, shape of distal condyles: articular surface restricted to distal portion of femur (0); expanded onto anterior portion of femoral shaft (1) (Wilson & Carrano, 1999; W02-202; Fig. 99).

C440: Tibia to femur length: <0.68 (0); 0.68 or greater (1) (Gauthier, 1986; U98-192; modified; Table S53).

C441: Tibia, proximal articulation surface, shape: subcircular to transversely compressed (0); anteroposteriorly compressed (1) (W02-203; modified; Fig. 100).

Figure 100 Eusauropod tibiae.

Tibia of Omeisaurus tianfuensis ZDM T5701 (A; traced from He, Li & Cai, 1988), Dyslocosaurus polyonychius AC 663 (B), and Apatosaurus louisae CM 3018 (C; traced from Gilmore, 1936) in proximal view. Note the different outlines (anteroposteriorly compressed, A, C441-1; subtriangular, B, C442-1; subrectangular, C, C442-0), and the projection posterior to the cnemial crest (B; C446-0). Abb.: cc, cnemial crest. Scaled to same anteroposterior length.

Comments. Character descriptions was slightly changed such that subcircular surfaces are now scored together with the transversely compressed ones, instead of the anteroposteriorly compressed ones as in Wilson (2002). Like this, distribution of character states follow better higher-level taxa used in our analysis.

C442: Tibia, proximal articular surface, shape: subrectangular (0); subtriangular (1) (Harris & Dodson, 2004; Fig. 100).

Comments. Rhomboid or suboval outlines are scored as plesiomorphic.

C443: Tibia, short transverse ridge on anteromedial surface of distal end: absent (0); present (1) (New; Fig. 101).

Figure 101 Distal end of tibia of Dyslocosaurus polyonychius AC 663 in medial view.

Note the transverse ridge on the anteromedial surface, close to the distal end (C443-1). Scale bar = 10 cm.

C444: Tibia, cnemial crest in anterior view: widely rounded (0); subtriangular (1) (New; Fig. 102).

Figure 102 Diplodocoid tibiae.

Tibia of Zapalasaurus sp. MOZ-Pv 1244 (A; traced from Salgado et al., 2012) and Tornieria africana MB.R.2572 (B; traced from Remes, 2006) in anterolateral view, illustrating the different shapes of the cnemial crest (widely rounded, A, C444-0; triangular, B, C444-1). Scaled to same length.

C445: Tibia, posterior surface of cnemial crest: smooth (0); bears a distinct fibular trochanter (1) (Harris, 2007; Fig. 103).

Figure 103 Proximal end of the tibia of Suuwassea emilieae ANS 21122.

Proximal end of the tibia of Suuwassea emilieae ANS 21122 in posterolateral view, showing the distinct fibular trochanter on the posterior surface of the cnemial crest (C445-1). Scale bar = 10 cm.

Comments. A distinct fibular trochanter marks the posterior face of the cnemial crest of Suuwassea (Harris, 2007). The character is herein included in a phylogenetic analysis for the first time.

C446: Tibia, lateral edge of proximal end forms a pinched out projection, posterior to cnemial crest (the ‘second cnemial crest’ of Bonaparte, Heinrich & Wild, 2000): present (0); absent (1) (M13-261; Fig. 100).

C447: Fibula, proximal end with anteromedially directed crest extending into a notch behind the cnemial crest of the tibia: absent (0); present (1) (Wilson & Upchurch, 2009; D12-111; modified by M13-262).

Comments. Most sauropods have ellipsoid proximal articular surfaces of the fibula. However, some diplodocid specimens (as well as some titanosauriforms; Wilson & Upchurch, 2009; D’Emic, 2012; Mannion et al., 2013) develop a distinct, narrow, anteromedial crest.

C448: Fibula, insertion of the M. iliofibularis: located approximately at mid-shaft (0); proximal, located above midshaft (1) (W11-183; Table S54).

Comments. Distance from the proximal articular surface to the center of the tubercle was measured and compared to greatest length. Values of 0.4 or lower were scored as derived.

C449: Astragalus, morphology in anterior view: rectangular (0); wedge-shaped, narrowing medially (1) (N12-300; Fig. 104).

Figure 104 Flagellicaudatan astragali.

Astragalus of SMA 0087 (A) and Dyslocosaurus polyonychius AC 663 (B) in dorsal (top) and posterior (bottom) view. Note the triangular shape in both views (B; C449-1, C450-1), the ascending process that reaches the posterior border (A; C453-1), the anterior border of the fibular facet, which is visible in posterior view (B; C454-1), the presence (B; C455-0) or absence (A; 455-1) of a sheet underlying the fibula, and the blunt (A; C456-0) in contrast to elongate medial end (B; C456-1). Scaled to the same proximodistal height.

C450: Astragalus, anteroposterior dimension as seen in dorsal view: widens medially or does not change in width (0); narrows medially (1) (Cooper, 1984; U98-195; Fig. 104).

Comments. The taxonomic significance of this character was recognized by Cooper (1984), but included in a phylogenetic analysis for the first time by Upchurch (1998).

C451: Astragalus, proximodistal length/transverse breadth: <0.55 (0); 0.55 or greater (1) (McIntosh, Coombs & Russell, 1992; Table S55).

Comments. This ratio was used by McIntosh, Coombs & Russell (1992) to distinguish Dyslocosaurus from Diplodocus, here included in a phylogenetic analysis for the first time.

C452: Astragalus, mediolateral width to maximum anteroposterior length ratio: 1.6 or greater (0); <1.6 (1) (S06-127; modified; Table S55).

C453: Astragalus, ascending process length: limited to anterior two-thirds of astragalus anteroposterior width (0); extends beyond two-thirds of astragalus anteroposterior width (normally to posterior margin of astragalus) (1) (W98-84; modified by M12-193; Fig. 104).

C454: Astragalus, fibular facet: faces laterally (0); faces posterolaterally, anterior margin visible in posterior view (1) (W11-186; Fig. 104).

C455: Astragalus, laterally directed ventral shelf underlies distal end of fibula: present (0); absent (1) (Wilson & Upchurch, 2009; M13-267; Fig. 104).

Comments. The ventral shelf only underlies a part of the fibula.

C456: Astragalus, anteromedial corner in posterior view: short and blunt (0); elongate and narrow (1) (New; Fig. 104).

Comments. This character described the development of the anteromedial process. The short and blunt shape is a somewhat intermediate state between triangular and rectangular outlines, as described in character 449. A second character was preferred over a merged version in order to avoid a combination of a character coding for the presence of the anteromedial process and a character describing its shape.

C457: Calcaneum: proximodistally compressed (0); globular (1) (Harris & Dodson, 2004).

Comments. Suuwassea has a globular calcaneum, whereas most other sauropods that preserve calcanea have dorsoventrally compressed elements. These bones are very rarely preserved and were even proposed to be absent in diplodocids (McIntosh, 1990b; Upchurch, 1998). However, Bonnan (2000) reported a probable calcaneum from Diplodocus, and also an apatosaur specimen from Como Bluff, Wyoming (NHMUK R.3215) appears to show such an element (E Tschopp, pers. obs., 2011).

C458: Metatarsals, metatarsal I to metatarsal V proximodistal length ratio: 1.0 or greater (0); <1.0 (1) (M13-72; polarity reversed; Table S56).

Comments. Length is measured between parallel lines through the proximal- and distal-most points of the metatarsals.

C459: Metatarsal I, dorsal/anterior surface: without foramina (0); several foramina present (1) (New; Fig. 105).

Figure 105 Metatarsal I of Cetiosauriscus stewarti NHMUK R3078 in dorsal/anterior view.

Note the foramina (C459-1), the angled proximal (C460-0) and distal articular surfaces (C462-0), and the distinct posterolateral process on the distal articular surface (C463-1, C464-1). Scale bar = 5 cm.

C460: Metatarsal I proximal articular surface, transverse axis orientation: angled ventromedially approximately 15° to (0); perpendicular to axis of shaft (1) (W02-218; polarity reversed; Fig. 105).

Comments. Polarity was reversed due to the different taxon sampling.

C461: Metatarsal I, robustness (proximal transverse width/greatest length): relatively gracile, <0.8 (0); robust, 0.8 or more (1) (U04a-292; modified; Table S57).

C462: Metatarsal I distal articular surface, transverse axis orientation: angled dorsomedially to (0); perpendicular to axis of shaft (1) (W02-219; polarity reversed; Fig. 105).

C463: Metatarsal I distal condyle, posterolateral projection: absent (0); present (1) (Berman & McIntosh, 1978; Y93-54; Fig. 105).

Comments. All taxa where the posterolateral corner of the distal articular surface can be seen in anterior view are scored as apomorphic.

C464: Metatarsal I, distolateral projection, if present: small and blunt, not projecting considerably lateral to dorsal edge of distal articular surface (0); prominent and pointed, reaching significantly more laterally than dorsal edge of distal articular surface (1) (McIntosh, 1990b; Fig. 105).

Comments. Usually, a prominent posterolateral or distolateral projection exceeds the lateral expansion of the proximal articular surface in anterior view.

C465: Metatarsals I–III, rugosities on dorsolateral margins near distal ends: absent (0); present (1) (U95; Fig. 106).

Figure 106 Flagellicaudatan metatarsals II.

Right (A) and left (B) metatarsal II of SMA 0087 (A) and Dyslocosaurus polyonychius AC 663 (B) in dorsal/anterior view. Note the dorsolateral rugosity (C465-1) with its different developments (reduced, laterally, A, C468-0; prominent, reaching center or shaft, B, C468-1), or the posterolateral process (absent, A, C469-0; present, B, C469-1). Scaled to same proximodistal length.

Comments. A second character (C468) accounts for the strength of the rugosity on mt II.

C466: Metatarsal II, robustness (mean proximal and distal transverse breadth /maximum length): slender, <0.53 (0); intermediate, 0.53–0.65 (1); robust, >0.65 (2) (McIntosh, Coombs & Russell, 1992; Table S57). Ordered.

Comments. The robustness of metatarsal II was used by McIntosh, Coombs & Russell (1992) to distinguish between diplodocids, but has never been included in a phylogenetic analysis.

C467: Metatarsal II, lateral margin in proximal view: concave (0); straight (1) (M13-273; Fig. 107).

Figure 107 Dicraeosaurid metatarsals II.

Right (A) and left (B) metatarsal II of Suuwassea emilieae ANS 21122 (A) and Dyslocosaurus polyonychius AC 663 (B) in proximal view, illustrating the concave (A) and straight (B) lateral margins (arrows; C467). Scaled to the same dorsoventral height.

Comments. The medial margin is usually concave. With the lateral margin being concave as well, the outline of the proximal articular surface of mt II becomes somewhat hourglass-shaped.

C468: Metatarsal II, rugosity on dorsolateral margin near distal end (if present): shallow (0); well-developed, extending to center of shaft (1) (New; Fig. 106).

Comments. The development of the rugosities in mt I to III differs within the pes (mt II bearing the most prominent ridge), but more so between taxa. This is exemplified by the well-developed, rugose ridge of metatarsal II in Dyslocosaurus polyonychius AC 663, which extends almost to the center of the shaft. Taxa without any rugosities are scored as unknown.

C469: Metatarsal II distal condyle, posterolateral projection: absent (0); present (1) (New; Fig. 106).

Comments. The distribution of the posterolateral projection in mt II was discussed by Nair & Salisbury (2012).

C470: Metatarsal IV, proximal articular surface, outline: L- to V-shaped, with distinctly concave posterolateral edge (0); subtriangular (1) (New; Fig. 108).

Figure 108 Eusauropod metatarsals IV.

Right (A) and left (B) metatarsal IV of Suuwassea emilieae ANS 21122 (A) and Cetiosauriscus stewarti NHMUK R3078 (B) in proximal view, illustrating the curved (A; C470-0) and subtriangular outlines (B; C470-1). Scaled to the same dorsoventral height.

C471: Metatarsal V, proximal articular surface, shape: triangular (0); rhomboid (1) (New; Fig. 109).

Figure 109 Diplodocid metatarsal V.

Metatarsal V of Barosaurus affinis YPM 419 (A) and SMA 0087 (B) in proximal view, illustrating the rhomboid (A; C471-1) or triangular outline of the articular surface (B; C471-0). Scaled to the same transverse width.

C472: Metatarsal V proximal end to distal end maximum mediolateral width ratio: 1.6 or greater (0); <1.6 (M13-74; Table S57).

Comments. Transverse width was measured between parallel vertical lines through the medial- and lateral-most points of the articular surfaces.

C473: Pes, phalanx I-1: proximal and ventral surfaces meet at approximately 90° (0); proximoventral corner drawn out into thin plate underlying metatarsal I (1) (McIntosh, Coombs & Russell, 1992; Fig. 110).

Figure 110 Pedal phalanx I-1 of the indeterminate apatosaurine NHMUK R3215 in medial view.

Note the ventral shelf (C473-1). Scale bar = 2 cm.

C474: Pes, phalanx I-1, distal articular surface shape: wide, maximum transverse width >1.1 times anteroposterior height (0); narrow, maximum transverse width 1.1 times anteroposterior height or less (1) (New; Table S58).

C475: Pes, phalanx II-2: well developed and subrectangular in dorsal view (0); reduced, with an irregular D-shaped outline and proximal and distal articular surfaces that meet virtually along dorsal and plantar margins (1) (McIntosh, Coombs & Russell, 1992).

C476: Pes, phalanges III-1 and IV-1: equal to longer than wide (0); wider than long (1) (McIntosh, Coombs & Russell, 1992; Table S58).

Comments. The greatly elongate php IV-1 of the early juvenile SMA 0009 indicates that phalanges grow allometrically during early ontogeny.

C477: Pedal unguals, groove on lateral surface: follows curvature of claw (0); straight horizontally (1) (New; Fig. 111).

Figure 111 Eusauropod pedal ungual I.

Pedal ungual I of Cetiosauriscus stewarti NHMUK R3078 (A) and Dyslocosaurus polyonychius AC 663 (B) in lateral view, illustrating the two different courses of the canals (curved, A, C477-0; straight, B, C477-1). Dotted lines indicates the broken tip. Scaled to same proximal articular surface height.

Methods

Phylogenetic analysis

In the present analysis, 243 characters were added to the analysis published by Tschopp & Mateus (2013b), based on earlier publications or personal observations of the ingroup specimens. Changes and character deletions proposed by Tschopp, Russo & Dzemski (2013) were applied. Operational taxonomic units were scored based on personal observations where possible, on published descriptions where existing, or on photographs from fellow researchers (Table S1).

Phylogenetic analysis was performed with the software TNT (version 1.1 for Windows, no taxon limit; Goloboff, Farris & Nixon, 2008), using the New Technology Search tool and enabling all options (Sect. Search, Ratchet, Drift, and Tree Fusing). Of the 53 multi-state characters, 23 were treated as ordered (characters 1, 48, 49, 55, 60, 111, 113, 140, 153, 184, 258, 267, 282, 360, 379, 380, 387, 395, 402, 414, 427, 430, and 466; see character descriptions). The consensus tree was stabilized five times with factor 75.

Main analyses

Several preliminary analyses were run in order to test previous hypotheses that unified several specimens into one individual (see above). By doing so, the data set was reduced from 86 operational taxonomic units to 81. This decreased the percentage of highly incomplete taxa and increased taxon overlap, which would otherwise have been very low (Table 4). The final, reduced data set was then analyzed again, with the settings stated above. Additionally, in order to find all possible shortest trees, the TNT script ‘bbreak’ was used with tree bisection and reconnection (command: bbreak = tbr safe). A reduced consensus tree was produced using the heuristic method (Trees > Comparison > Agreement subtrees). Specimens not represented in the reduced consensus were added one by one to check their possible phylogenetic positions. Subsequently, pruned trees were generated (Trees > Comparison > Pruned Trees), with the parameters different from the default set as follows: up to four taxa, list as text. Finally, a pruned strict consensus tree was generated excluding the four most unstable OTUs a posteriori. These analyses thus produced an equally weighted complete, a pruned, and a reduced consensus tree, which will be called S_ew, P_ew, and R_ew in the following.

Given the low consistency index (CI) and thus high number of homoplasies in the dataset, an additional analysis with the same settings was conducted using implied weighting (iw). Implied weighting iteratively calculates the weights of the characters during analysis, based on the consistency index of each character on the topology recovered at each step (Goloboff, 1993). Because characters with a high number of homoplasies in a specimen-based analysis are possibly coding for individual, intraspecific or ontogenetic variation, and are thus not phylogenetically significant, down-weighting of these characters, as implemented by implied weighting, should yield more accurate results. Down-weighting of the homoplastic characters was preferred over character deletion because certain characters were only homoplastic in one part of the tree. Traits that are variable within one clade can thus still be diagnostic for another group. In short, four main trees were generated by a posteriori deletion of certain OTUs, which are both a pruned (deleting the four most unstable taxa) and a reduced (deleting all unstable taxa) consensus tree per weighting strategy.

Overall CI and RI were calculated for the most parsimonious trees using the stats.run script. For both analyses, symmetric resampling was preferred over bootstrapping or jackknifing for quantifying node support (Analyze > Resample; using the default settings). Symmetric resampling is not affected by differential weighting of the characters, and is therefore more meaningful for analyses using implied weights (Goloboff et al., 2003), thus allowing fair comparison between support values for trees generated both using and not using implied weights.

Influence of ontogeny

Juvenile individuals have been sometimes shown to group with more basal taxa in a phylogenetic tree, instead of being nested within the taxon they belong to (e.g., Campione et al., 2013; Carballido & Sander, 2014). Given that the dataset includes several putative juvenile to subadult specimens (YPM 1901, SMA 0009, CM 566, and possibly ANS 21122, SMA 0004, CM 3452, SMA 0011, AMNH 7530, AMNH 7535, SMA O25-8, SMA D16-3), it was important to address this issue. Implied weighting addresses this problem at least partially: because ontogenetic changes should generally occur in a similar way among closely related taxa, characters describing them are probably more homoplastic than others and thus should be down-weighted as well.

In order to decrease the influence of ontogeny in the final tree, scoring juvenile or subadult individuals as unknown for ontogenetically changing features can be an additional approach to down-weighting. However, in many cases ontogenetic variability of characters is ambiguous (e.g., for the development of bifurcation of neural spines; Woodruff & Fowler, 2012; Wedel & Taylor, 2013). Ambiguity also occurs in the identification of the ontogenetic stage of certain specimens, sometimes even where histological information from longbones is available, but in conflict with open neurocentral sutures (e.g., Suuwassea; Harris, 2006b; Woodruff & Fowler, 2012; Hedrick, Tumarkin-Deratzian & Dodson, 2014). Given that earlier studies including small, juvenile sauropods (Upchurch, Tomida & Barrett, 2004; Carballido et al., 2012a), as well as our study, found the smallest juvenile specimens CM 566 and SMA 0009 nested within well-defined clades (see below), the influence of ontogenetically variable characters appears minimal. Furthermore, small juvenile individuals included herein generally group with large specimens, that are generally considered adult, instead of grouping together in a basal clade, as recovered by Campione et al. (2013). A similar result was obtained by a specimen-based phylogenetic analysis of the ceratopsian species Auroraceratops rugosus (Morschhauser, You & Dodson, 2014). This indicates that ontogenetically variable traits were outweighed by taxonomically informative characters in our analysis, to some extent. In order to evaluate the influence of ontogenetically variable traits, also small juvenile specimens were scored completely in our analysis. However, when assessing their position in the tree, and applying our quantitative approaches for taxonomic implications (see below), we took possible ontogenetic variability of the recovered potential apomorphies into account.

The low influence of ontogenetically variable characters might be a positive side-effect of large sets of characters, where these characters are more easily outweighed by taxonomically valid ones, although Upchurch, Tomida & Barrett (2004), with their very reduced character list of 32 character statements also obtained a promising result for the juvenile holotype of “Elosaurus” parvus CM 566 (which was corroborated by our analysis, see below). More methodological studies would be needed to address this particular issue.

Anatomical overlap

A major challenge of a specimen-based phylogenetic analysis is the limited anatomical overlap between specimens compared to that between species or genera (which can be composites of multiple specimens, and therefore more anatomically complete), most importantly between incomplete historic holotype specimens, as is the case in most diplodocid type specimens described by Marsh and Cope (Cope, 1877a; Cope, 1877b; Marsh, 1877a; Marsh, 1878; Marsh, 1879; Marsh, 1881; Marsh, 1884; Marsh, 1890; Marsh, 1899). New species were rushed into press without detailed description, sometimes even lacking illustrations (e.g., Marsh, 1881; Marsh, 1899). In certain cases, subsequent studies proposed that multiple species were erected based on different bones of possibly the same individual skeleton (‘Atlantosaurus’ immanis YPM 1840 and Apatosaurus laticollis YPM 1861; Marsh, 1877a; Marsh, 1879; McIntosh, 1995). More complete skeletons were later recovered, but many of these are still undescribed and were identified as a particular genus or species without any detailed study (e.g., ‘Diplodocus longus’ DMNS 1494). Lately, more and more nearly complete specimens have become available for study (e.g., Harris & Dodson, 2004; Upchurch, Tomida & Barrett, 2004; Barrett et al., 2011; Tschopp & Mateus, 2013b). Complete, articulated specimens, or parts of skeletons preserving portions underrepresented in earlier finds (e.g., skulls attached to their necks, transitions from cervical to dorsal vertebrae, articulated manus or pedes), are crucial for a specimen-based phylogenetic analysis. They provide the anchorage with which fragmentary specimens can be compared, thereby allowing for indirect comparisons. Care has to be taken to include articulated specimens and exclude information from portions of the skeleton for which an unambiguous association with the specimen to be studied cannot be ascertained. The most valuable documents to assure genuine association of skeletal parts to one individual are detailed quarry maps and field notes, but these are often absent for historical type specimens. Efforts were made lately to unravel excavation stories and bone associations of the most important holotype specimens (e.g., McIntosh, 1990a; McIntosh, 1995; McIntosh & Carpenter, 1998). The present study heavily relies on these earlier studies to confirm or discard bone associations. However, the circumstances of collection for some specimens still require detailed investigation, so their phylogenetic positions should be regarded as provisional (see below).

Two overlap indices quantify character overlap within individual clades, and were created using Microsoft Excel® in collaboration with F Tschopp (Jona, Switzerland; Table S59). These indices quantify (1) how many characters of the total 477 are available for analysis of the ingroup species (the “all chars” overlap index), and (2) how many overlaps are present in the characters for which a specific group of specimens actually shows overlaps (the “comparable chars” overlap index). Overlaps were defined as the number of specimens for which a character was able to be scored, minus one (because if only one specimen of the group preserves a certain bone, no anatomical overlap is present).

The all chars overlap index sums the mean number of overlaps present in every single character, and divides them by the maximum number of possible overlaps. Therefore, it increases when more characters are scored in at least two specimens, or when the number of specimens scorable for the same character is enlarged. Thus, it combines a measure for the completeness of the matrix with the comparability of single characters within specimens of a single group. Thereby, it gives an idea of the strength of the matrix to recover certain clades. By calculating the overlap index for the sister group arrangements including a questionable taxon, researchers get an idea of how well the arrangement is supported based on overlapping skeletal material. The all chars overlap index is thus useful to evaluate the phylogenetic position of unstable taxa. However, it does not provide a measure for the significance of phylogenetic results, because incomplete specimens with few characters scored in common might still bear taxonomically important characters, allowing robust identification to genus or species level.

The comparable chars overlap index calculates the mean of the overlaps present in the characters that actually show anatomical overlap in the group under question, instead of including all characters. This index is thus always higher than the all chars overlap index. For groups in which only two specimens are present, the comparable chars overlap index always reaches 100%. It is thus more informative for larger groups of specimens, where it gives a value of how many specimens are scorable for characters with anatomical overlap in the group. More detailed descriptions and assessments of the implications of these indices will be provided elsewhere.

Deformation

An additional problem, for quantitative characters in particular, is specimen deformation. Whereas brittle deformation can be readily identified due to the introduced cracks, plastic deformation results in unfractured but distorted fossils (Tschopp, Russo & Dzemski, 2013). Plastic deformation, if it occurs symmetrically, is almost impossible to identify and least of all to quantify. Retrodeformation can yield some information on how bones were deformed, but only in bilaterally symmetrical elements (Arbour & Currie, 2012; Tschopp, Russo & Dzemski, 2013). For species- or genus-level phylogenetic analyses, mean ratios can be taken from different individuals of the same taxon, thereby approaching more closely the ratios generally typical for that taxon. In specimen-based analyses, such an approach is not possible. However, if a specimen is deformed in such a way that it would be scored differently from closely related species, or specimens from the same species, it increases homoplasy of this single character, and decreases its consistency index. By using implied weighting, as was done in the second analysis herein, deformation can thus be partly accounted for.

Morphological details

During the study of single specimens, one usually records and describes morphological details unique to the animal, which might or might not be taxonomically significant. If the phylogenetic analysis accompanying the description recovers the new specimen on a separate branch, and thus as a new taxon, these traits are generally interpreted as autapomorphic for the new taxon. The confirmation of such an interpretation can only be made with the discovery of additional specimens of the same species, preserving the same portions of the skeleton. Before that, variation due to any pre- or post-mortem processes (ontogeny, individual variation, sexual dimorphism, or taphonomic deformation) cannot be excluded with certainty as a cause for the morphological disparity found in the fossil. Specimen-based phylogenetic analyses are the only way to test for such variation. As mentioned above, highly homoplastic characters are the most likely to encompass variation seen between individuals in specimen-based phylogenetic analyses. These characters should either be deleted or down-weighted compared to the less variable characters, as is done by implied weighting (Goloboff, 1993). Finally, by scoring single specimens of a species, and thereby detecting individual variation in some characters, researchers create a firmer base for how to score species- or even genus-level OTUs.

Quantitative taxonomy

One of the problems raised in a specimen-based phylogenetic analysis is where to draw the line between morphological variation among individuals within species, and variation that allows distinction between species or genera. The decision for specific versus generic separation is somewhat arbitrary, in particular in paleontology, where no tests exist for the biological species concept (Carpenter, 2010), and where specimens are sampled on a temporal axis, that can blur the distinction between reproductively isolated populations. If qualitatively assessing the validity and significance of single characters, subjectivity of these interpretations is especially great. Therefore, a quantitative approach was developed to limit this subjectivity. With a numerical approach, personal influence can be minimized, and the taxonomic decisions about generic separation can be rendered more repeatable and thus scientifically sound. Two approaches are used herein: pairwise dissimilarity (Benson, Evans & Druckenmiller, 2012), and apomorphy counts as mapped on a phylogeny.

Pairwise dissimilarity

Pairwise phenetic dissimilarity between taxa in our data matrix was calculated by dividing the number of character scores that differed between each taxon pair by the number of informative comparisons (i.e., not “?”, inapplicable, or polymorphic/uncertain in either taxon; Sneath & Sokal, 1973; Foote, 1994; Wagner, 1997) using a custom script written in R version 3.1.0 (R Development Core Team, 2014). For comparisons among sets of taxa, weighted mean pairwise disparity was used, with individual pairwise values weighted according to the number of informative comparisons that could be made between the taxon pair. Dissimilarity values were used as a second quantitative criterion to guide our taxonomic decisions (a similar approach to plesiosaur taxonomy was taken by Benson, Evans & Druckenmiller, 2012), and to illustrate the distribution of taxa in character spaces constructed by applying principal co-ordinates analysis (PCo) to the inter-taxon dissimilarity matrix (e.g., Foote, 1994; Wills, Briggs & Fortey, 1994; Wagner, 1997; Benson, Evans & Druckenmiller, 2012; Butler et al., 2012), using the R package labdsv version 1.6-1 (Roberts, 2013). To avoid the presence of inapplicable comparisons between OTUs for which no overlapping character scores were known, the dissimilarity matrix was pre-processed prior to all PCo analyses, iteratively removing taxa with at least one inapplicable comparison, and beginning with those taxa with the greatest amount of missing data. The first three PCo axes are plotted for all such character spaces and the proportions of variance explained by each axis are given in Fig. 112.

Figure 112 Diplodocid morphospace.

First two principal coordinate axes of dissimilarity among Diplodocidae. The third axis is indicated by the size of the points. Note the intermediate, but rather diplodocine position of Amphicoelias altus AMNH 5764 and the rather apatosaurine position of FMNH P25112 (white circles).

One distinct break was found in the calculated dissimilarity values within Diplodocinae, whereas the situation within Apatosaurinae appears a bit more complicated (Data S1). Within Diplodocinae, specimens considered to belong to the same genus exhibit values below 0.181, whereas different genera show values of 0.222 and higher. Two generally accepted species within a single genus (Diplodocus carnegii and Diplodocus hallorum) have a value of 0.1195. Well-defined species (e.g., Diplodocus carnegii or Supersaurus vivianae) have mean pairwise dissimilarity rates of less than 0.08. The values within apatosaurines will be discussed below together with the validity of the recovered clades.

Apomorphy counts

This method is based on the number and quality of ‘synapomorphies’ and ‘autapomorphies,’ as found by the software TNT. Because the analysis is specimen-based, these do not universally conform to real species or genus autapomorphies or synapomorphies, but describe unique or shared morphological features of specimens and groups of specimens, only some of which correspond to formal taxonomic units such as genera or species. These ‘false’ apomorphies are given in quotation marks in the following. However, qualitative assessment of the apomorphies, as outlined below, include counts of both real and ‘false’ apomorphies.

Synapomorphies are separated into four qualitatively different categories. Unambiguous synapomorphies are shared by all ingroup members of the respective clade, and only by them. Exclusive synapomorphies only mark ingroup members, but not all of them. Shared synapomorphies are present in all ingroup members, but also occasionally occur in taxa outside the clade in question. Ambiguous synapomorphies are neither exclusive nor shared by all ingroup members, but are still recovered as synapomorphies by at least one analysis with equal weighting and one with implied weighting. Ambiguous synapomorphies recovered by only one type of analysis (equal or implied weighting) are not considered reliable.

Specimen ‘autapomorphies’ are divided into unambiguous, or ambiguous (also occurring in other taxa). Ambiguous ‘autapomorphies’ of apatosaurine specimens, which are shared with other apatosaurine specimens or clades (or diplodocine with diplodocine) are interpreted as inappropriate for species diagnosis.

‘Synapomorphies’ of diplodocid genera and species generally considered valid (including ambiguous, shared, exclusive, and unambiguous apomorphies) were counted and summed between sister taxa (specimens or clades, in this case). A minimum number of synapomorphies was defined for justifying specific or generic separation. The minimum number of required differences for generic separation was chosen based on the count obtained from the well-established species of Apatosaurus (A. ajax and A. louisae) and Diplodocus (D. carnegii and D. hallorum). These species are all represented by reasonably complete specimens, allowing for good comparison, have been generally accepted as species within their respective genera in the past, and were recovered as sister taxa in our analysis. Character changes amount to 12 between A. ajax and A. louisae, and eleven between D. carnegii and D. hallorum. Therefore, a minimum of 13 changes is herein considered necessary for generic separation.

A second count of changes was made between specimens generally referred to the same species, and recovered within the same clade in our analysis (Diplodocus carnegii CM 84 and CM 94, Apatosaurus louisae CM 3018 and CM 3378, Brontosaurus excelsus YPM 1980 and B. ‘amplus’ YPM 1981). The sum of changes between these specimens amounts to one in D. carnegii and A. louisae, and five in B. excelsus. A minimum of six differences is thus considered enough for species-level separation, thereby accounting for individual variation (which is already accounted for by the evaluation of the validity of the autapomorphies, but a wider margin is preferred herein in order to be more cautious). Given that we included juvenile specimens as well as OTUs, apomorphic features recovered for clades and branches including such specimens were checked for potential ontogenetic variability, and the count adapted where necessary.

The precise numbers established here (six and 13 changes) cannot be applied to any other analysis, even of the same clade, because the recovery of ‘autapomorphies’ and ‘synapomorphies’ depends on the number of characters and OTUs included in the analysis and also on the software used. However, the general approach can be used in other analyses as well.

Results

Equal weighting

The first iteration of the equally weighted analysis yielded 164 most parsimonious trees with a score of 1,976 steps. The second iteration using the command ‘bbreak’ increased this number to 60,000 (more was not possible due to computer limitations). CI and RI under equal weights are equal to 27.3 and 58.8, respectively. The strict consensus tree had only twelve resolved nodes, which are exclusively located outside Diplodocidae, meaning that all ingroup specimens formed one large polytomy (Fig. 113).

Figure 113 Strict consensus tree of the complete analysis with equal weighting.

OTUs with species names and specimen numbers are type specimens. Tree length is 1,976 steps.

The single most unstable taxon as recovered by the pruned trees approach was Diplodocus lacustris YPM 1922. By excluding this taxon from the strict consensus tree, 31 additional nodes were resolved. Diplodocus lacustris YPM 1922 was shown to group with a large number of OTUs within Flagellicaudata, as exemplified by the large polytomy of the reduced consensus tree including the specimen. Because YPM 1922 includes only teeth, the result mentioned above indicates that flagellicaudatan teeth cannot be distinguished to lower taxonomic levels at the present state of knowledge. Besides D. lacustris YPM 1922, the following three OTUs were recovered as highly unstable: ‘Apatosaurus’ minimus AMNH 675, Australodocus bohetii type, and Dystrophaeus viaemalae USNM 2364. Deleting these four most unstable taxa a posteriori resulted in resolution of higher-level clades within Flagellicaudata (Dicraeosauridae, Apatosaurinae, and Diplodocinae), as well as several lower-level clades (Fig. 114).

Figure 114 Pruned strict consensus tree obtained by equal weighting.

The following OTUs were pruned a posteriori: ‘Apatosaurus’ minimus AMNH 675, ‘Diplodocus’ lacustris YPM 1922, Dystrophaeus viaemalae USNM 2364, and the type individual of Australodocus bohetii. Note the dicraeosaurid affinities of Dyslocosaurus and Suuwassea, the inclusion in Diplodocinae of FMNH P25112, and the close association of Apatosaurus ajax with Apatosaurus louisae (in red).

The equally weighted reduced consensus tree includes 66 of the original 81 taxa. The classical diplodocid genera as used in earlier phylogenetic analyses (e.g., Whitlock, 2011a; Mannion et al., 2012; Tschopp & Mateus, 2013b; Gallina et al., 2014) are all recovered (Fig. 115).

Figure 115 Reduced consensus tree obtained by equal weighting.

Fifteen OTUs were deleted a posteriori. Numbers at the nodes indicate the number of changes between the two branches departing from the node (for the apomorphy count).

Implied weighting

The analysis done under implied weighting yielded 115 most parsimonious trees of a length of 194.21603, but the number of trees was increased by the second iteration of tree bisection and reconnection to 60,000. CI and RI under implied weights correspond to 27 and 58.3, respectively, and are thus slightly lower than under equal weights. The strict consensus tree included 24 resolved nodes (Fig. 116), double that for our equal-weights analysis.

Figure 116 Strict consensus tree of the complete analysis with implied weighting.

OTUs with species names and specimen numbers are type specimens. Tree length is 194.21603 steps. Note the basal position of Barosaurus affinis, Cetiosauriscus stewarti, the somphospondylian affinities of ‘Apatosaurus’ minimus, the diplodocine affinities of Australodocus bohetii, as well as the contrasting positions of Apatosaurus ajax YPM 1860 and FMNH P25112 when compared with the result under equal weights (in red).

The pruned tree analysis under implied weights confirmed that the Diplodocus lacustris holotype specimen (YPM 1922) is the least stable. Deletion of YPM 1922 resulted in the resolution of an additional 39 nodes compared to the original strict consensus tree. Omission of the four least stable taxa (D. lacustris YPM 1922, the diplodocine skulls CM 11161 and USNM 2672, and the genoholotype specimen of Diplodocus, D. longus YPM 1920) resulted in a pruned consensus tree with 44 additional resolved nodes compared to the complete strict consensus tree, and 12 additional resolved nodes compared to the pruned tree with equal weighting (Fig. 117). The reduced consensus tree with implied weights includes 73 taxa, seven more than the equally weighted reduced consensus tree (Fig. 118).

Figure 117 Pruned strict consensus tree obtained by implied weighting.

The following OTUs were deleted a posteriori: Diplodocus lacustris YPM 1922, CM 11161, USNM 2672, and Diplodocus longus YPM 1920. Note the position of Apatosaurus ajax as most derived apatosaurine, Dystrophaeus viaemalae within Dicraeosauridae, and Australodocus bohetii as a close relative of Dinheirosaurus and Supersaurus (arrowheads).

Figure 118 Reduced consensus tree obtained by implied weighting.

Eight OTUs were deleted a posteriori. Numbers at the nodes indicate the number of changes between the two branches departing from the node (for the apomorphy count), where they differ from the trees under equal weights.

Support values

Symmetric resampling did not find strong support for diplodocid ingroup clades (Table S59), most probably due to the limited anatomical overlap between OTUs. Values range from zero to 32 within Diplodocidae. The following clades were only found by symmetric resampling with either equal or implied weighting: UW 15556 + more derived Apatosaurinae (mdA), and UW 15556 + Apatosaurus ajax YPM 1860 (ew); NSMT-PV 20375 + Atlantosaurus immanis YPM 1840, Brontosaurus excelsus YPM 1980 + B. amplus YPM 1981, BYU 1252-18531 + UW 15556, Dinheirosaurus + Supersaurus, and Diplodocus carnegii + more derived (md) Diplodocus (iw). Symmetric resampling also found support for seven clades that were not recovered in any of the six main trees: (1) Apatosaurus laticollis YPM 1861 + (Apatosaurus louisae CM 3018 + CM 3378) (resampling values 4 (ew)/28 (iw)); (2) Apatosaurus louisae CM 3018 + CM 3378 (11/5); (3) Eobrontosaurus yahnahpin Tate-001 + Amphicoelias altus AMNH 5764 (3, iw only); (4) USNM 2673 + SMA 0011 (2/2); (5) Diplodocus longus YPM 1920 + (AMNH 223 + md Diplodocus) (4, iw only); (6) SMA O25-8 + (Kaatedocus siberi SMA 0004 + md Kaatedocus) (2/2); (7) USNM 2672 and CM 11161 (3/8). The grouping of the two skulls CM 11161 and USNM 2672 indicates that they are more similar to each other than to any other diplodocine skull.

Tree topology

Diplodocoidea was found as the sister-taxon of Titanosauriformes, with Camarasaurus and Turiasauria forming a more basal clade. This result contradicts most of the recent analyses of sauropods, and in particular studies of early macronarian relationships (Carballido et al., 2012b; D’Emic, 2012; Mannion et al., 2013). Our results therefore appear to corroborate preliminary results from Upchurch (2009) and Mateus et al. (2011), which recovered Macronaria as polyphyletic. However, many important taxa and characters relevant to defining Macronaria are missing from the present matrix, due to our focus on Diplodocoidea. Because diplodocoid synapomorphies are often shared with derived titanosauriforms, preferential sampling of these characters is probably responsible for the recovery of a clade comprising Diplodocoidea + Titanosauriformes, and excluding the non-titanosauriform macronarian Camarasaurus in our analysis.

Within Diplodocoidea, Rebbachisauridae is the sister taxon of a clade comprising Dicraeosauridae (including Suuwassea emilieae) and Diplodocidae. Diplodocidae is divided into Apatosaurinae and Diplodocinae. The taxonomically significant holotype specimen of Diplodocus longus, YPM 1920, is lacking from both reduced consensus trees as well as from the pruned tree using implied weights. This is important because D. longus is the type species for the genus Diplodocus.

The differences in the recovered tree topologies under equal and implied weights concern only few OTUs. The most important for the present analysis is the placement of the holotype of the type species for Apatosaurus, A. ajax (YPM 1860). Equal weighting found the specimen as sister taxon to a clade including the type specimens of A. louisae (CM 3018) and A. laticollis (YPM 1861). In contrast, the analysis under implied weights recovered A. ajax YPM 1860 separated from the A. louisae specimens, as the sister taxon to the specimen BYU 1252-18531. Australodocus bohetii, which was excluded from both the pruned and reduced consensus trees under equal weights, was found as sister taxon to Supersaurus vivianae under implied weights. The specimen FMNH P25112, found as a diplodocine with equal weighting, was recovered within Apatosaurinae under implied weights. In the outgroup, implied weighting led to an exclusion of Cetiosauriscus stewarti + Barosaurus affinis from Diplodocoidea (equal weighting found it as sister taxon to Flagellicaudata), and even from Neosauropoda. ‘Apatosaurus’ minimus, which was excluded from the equally weighted trees, was recovered within Somphospondyli under implied weights. Dystrophaeus viaemalae—deleted under equal weights as well—was found as sister taxon to the dicraeosaurid Suuwassea emilieae with implied weighting.

Validity of recovered diplodocoid subclades

The following discussion includes only the clades recovered within Diplodocoidea, because the present analysis was designed for the study of diplodocid intrarelationships, and is not suitable for inferring the phylogeny of clades outside Diplodocoidea. Definitions of the clade names follow Taylor & Naish (2005) and Whitlock (2011a).

The discussion of the various clades recovered is done following a bottom-up approach, starting with dichotomies between single specimens. This is preferred over a top-down approach, because it is the specimens that define the taxa, not the taxa that determine the affiliation of the specimen. Based on the validity of the recovered dichotomies between single specimens, the evaluation of species, genera and higher-level taxa can be performed more accurately.

Barosaurus lentus YPM 429 + AMNH 6341

These two specimens were recovered as sister taxa in both reduced trees. This clade has the highest resampling support value of all clades within Diplodocidae and is supported by four shared ‘synapomorphies’ found by both reduced trees (137-1, 183-1, 188-1, 200-1; Tables S59 and S60). Whereas two ‘synapomorphies’ are only shared with taxa outside Diplodocoidea (137-1, 200-1; with the possible exception of Australodocus bohetii, see below), the other two are also shared with various specimens within Diplodocidae, or even Diplodocinae. The two specimens are separated by one change only, indicating that they belong to a single species.

CM 11984 + (Barosaurus lentus YPM 429 + AMNH 6341)

Both reduced trees recovered this clade and found one shared (172-1) and one ambiguous ‘synapomorphies’ (184-2) defining it (Tables S59 and S61). The ambiguous ‘synapomorphy’ (prezygapophyseal centrodiapophyseal fossa of mid- and posterior cervical vertebrae subdivided into various smaller partitions by several accessory laminae) is absent in a mid-cervical vertebra of AMNH 6341 held in storage at AMNH. However, the determination of presence or absence of accessory laminae was not possible for more posterior cervical vertebrae of this specimen that are currently on public display. Therefore, further studies are needed to clarify this character state. Both ‘synapomorphies’ are shared with other diplodocine specimens, and therefore do not qualify as species autapomorphies.

The apomorphy count of changes is four, which indicates that all three specimens belong to a single species. On the other hand, mean pairwise dissimilarity within this triplet is 0.1217, which is higher than that found between the two species of Diplodocus (0.1195). The identification of CM 11984 will be discussed in more detail below.

AMNH 7535 + (CM 11984 + (Barosaurus lentus YPM 429 + AMNH 6341))

As for its two subclades (discussed above), this clade of four specimens was recovered by both reduced trees. However, statistical support for this clade is lower, and only one shared ‘synapomorphy’ is found (166-2; Tables S59 and S62). This ‘synapomorphy’ (very elongate mid-cervical vertebrae, EI > 4.5) is the best known and most widely used trait to distinguish Barosaurus from Diplodocus (e.g., McIntosh, 2005). The lack of other synapomorphies is probably due to the very restricted anatomical overlap among the four specimens of this clade, and also with their closest sister group (Kaatedocus siberi SMA 0004 + (SMA D16-3 + AMNH 7530)), which is only known from neck and skull material. It is likely that more synapomorphies will be recovered when more specimens preserving overlapping material are included in phylogenetic analyses.

Two ‘autapomorphies’ of AMNH 7535 were found. However, the total of four changes between AMNH 7535 and its sister-clade does not allow the erection of a distinct species for this specimen. However, as in the subclade discussed above, mean pairwise dissimilarity of 0.1236 among the four specimens is above the 0.1195 found between the two species of Diplodocus. Thus, while accepting a referral of all four specimens to a single genus, the results of the numerical approaches are ambiguous concerning referral of all specimens to a single species. Therefore, and because this clade includes the genoholotype specimen of Barosaurus (YPM 429), AMNH 7535, CM 11984 and AMNH 6341 are herein referred to Barosaurus.

SMA D16-3 + AMNH 7530

This clade is recovered in all four trees. One shared ‘synapomorphy’ is found by the pruned trees (87-1; Table S63). No ‘autapomorphies’ were found for the specimens within the clade, so that they can be assigned to a single species with confidence.

Kaatedocus siberi SMA 0004 + (SMA D16-3 + AMNH 7530)

This triplet constitutes the sister group to the Barosaurus lentus clade discussed above. It is found in all four analyses and supported by resampling values of seven (ew) and 16 (iw; Table S59). Nineteen shared ‘synapomorphies’ are recovered (17-0, 27-0, 32-1, 35-0, 80-0, 85-1, 131-0, 166-1, 178-1, 183-1, 187-0, 199-1, 202-1, 203-0, 205-0, 211-1, 212-1, 213-0, 214-0; Table S64). Six of these are unique within Diplodocinae and thus qualify as species autapomorphies (27-0, 32-1, 178-1, 202-1, 211-1, 212-1). One additional unambiguous autapomorphy of Kaatedocus was proposed by Tschopp & Mateus (2013b), but is not recovered as such by the present analyses: a transverse sulcus bordering the prezygapophyseal facets of posterior cervical vertebrae posteriorly. This feature was impossible to code in the other two specimens of Kaatedocus siberi, which is probably the reason why it was not found as a synapomorphy or autapomorphy herein. However, SMA 0004 is the only specimen positively scored for the presence of this feature in the current analysis, indicating that one more synapomorphy, possibly unambiguous for this clade, might be present.

Only two changes separate SMA 0004 from the other two specimens. Mean pairwise dissimilarity among these specimens is very low (0.0435) as well. Therefore, SMA 0004, SMA D16-3 and AMNH 7530 are referred to K. siberi, the type and only species of Kaatedocus.

Kaatedocus + Barosaurus

The sister arrangement of Barosaurus and Kaatedocus is recovered by all analyses herein, supported by two shared synapomorphies (157-0, 164-1; Table S65). These traits are somewhat problematical, as they concern anterior and mid-cervical vertebrae. Many specimens within Diplodocidae are not represented by anterior cervical vertebrae, and within Barosaurus, AMNH 7535 is the only specimen preserving them. Furthermore, anatomical overlap between Kaatedocus and Barosaurus is low. However, differences in the heights of anterior neural spines are very pronounced when comparing Kaatedocus SMA 0004 with Diplodocus CM 84 or ‘Diplodocus’ hayi HMNS 175, members of the two clades most closely related to Kaatedocus + Barosaurus within Diplodocidae. Dorsoventrally elongate coels on the lateral surfaces of the neural spines are typical for posterior cervical vertebrae of Diplodocus, among others, but in Diplodocus, these coels are not present in anterior elements. In Kaatedocus and Barosaurus AMNH 7535, the serial pattern is inverted, and the coels only mark anterior cervical neural spines. Additional synapomorphies, in particular from appendicular bones, might be found once a more complete specimen of Kaatedocus siberi is described.

The nineteen shared ‘synapomorphies’ of K. siberi plus the single ‘synapomorphy’ of its sister clade Barosaurus lentus sum to twenty changes, which is deemed enough for generic separation. The retention of two genera is also supported by mean pairwise dissimilarity, which finds a value of 0.2515 between specimens from the two clades.

CM 3452 + (Kaatedocus + Barosaurus)

The specimen CM 3452 is one of very few diplodocids preserving an almost complete skull in articulation with postcranial material. Although generally identified as Diplodocus (Holland, 1924; McIntosh & Berman, 1975; Whitlock, Wilson & Lamanna, 2010), CM 3452 is recovered as sister taxon to Barosaurus + Kaatedocus in all four trees found here. The affiliation of CM 3452 with this group is supported by one unambiguous (48-2), nine shared (2-0, 10-1, 19-1, 65-1, 67-1, 113-1, 134-0, 140-2, 182-1), and two ambiguous synapomorphies (184-0, 187-1; Table S66). One of the ambiguous synapomorphies is present in a specimen recovered within the Diplodocus clade (187-1: presence of an accessory horizontal lamina in the sdf) and is thus dubious. The lateral lacrimal spur recovered as an unambiguous synapomorphy of this clade was proposed as an autapomorphy of Kaatedocus (Tschopp & Mateus, 2013b), but is actually not unambiguous among sauropods: Tschopp & Mateus (2013b) reported a camarasaurid specimen (SMA 0002), which shows a similar trait, as do some other camarasaurid lacrimals (Madsen, McIntosh & Berman, 1995). However, within Diplodocidae, of the few skulls known, only CM 3452, SMA 0004, and CM 11255 bear such a spur (Tschopp & Mateus, 2013b). Indeed, CM 11255 is also recovered within this clade in the pruned consensus trees, although its exact position is impossible to determine. Although tree topologies suggest that CM 3452 constitutes its own genus, the low number of three changes between the specimen and the Kaatedocus + Barosaurus clade does not support the erection of a new genus nor a species. The affinities of CM 3452 will be discussed in more detail below.

DMNS 1494 + USNM 10865

These two specimens traditionally referred to Diplodocus (Gilmore, 1932; McIntosh, 2005) are recovered in both trees obtained with implied weighting, as well as in the reduced consensus tree with equal weighting. The equally weighted pruned consensus tree shows a polytomy formed by all putative Diplodocus specimens and the clade CM 3452 + (Kaatedocus + Barosaurus). This is probably a consequence of the incompleteness of important specimens like D. longus YPM 1920, or the skulls CM 11161 and USNM 2672, all of which were deleted from the other trees a posteriori during implementation of reduced and pruned consensus approaches. The clade DMNS 1494 + USNM 10865 is supported by a resampling value of 15 (ew) or 19 (iw), and one shared ‘synapomorphy’ (258-1; Tables S59 and S67). However, the ‘synapomorphy’ is shared with other diplodocine specimens and would thus not be a valid species autapomorphy. Because only one change separates DMNS 1494 from USNM 10865, the two specimens are referred to a single species.

‘Seismosaurus’ hallorum NMMNH 3690 + (DMNS 1494 + USNM 10865)

This triplet is found in the equally weighted reduced consensus tree, as well as in both pruned and reduced consensus trees when applying implied weights. It has a resampling value of seven (ew) or 12 (iw; Table S59), and is supported by one shared ‘synapomorphy’ (426-1; Table S68). This ‘synapomorphy’ is shared with other diplodocine specimens, and would thus not be a good species autapomorphy. The four changes separating ‘S.’ hallorum from the clade DMNS 1494 + USNM 10865 are not enough to justify the erection of two different species, therefore the entire triplet is referred to a single species.

AMNH 223 + (‘Seismosaurus’ hallorum NMMNH 3690 + (DMNS 1494 + USNM 10865))

As with its two subclades discussed above, this quartet of specimens is recovered in all trees except for the equally weighted pruned tree. It has a resampling value of 12 (ew) or eight (iw; Table S59), and is supported by one unambiguous (340-1) and four shared ‘synapomorphies’ (234-1, 337-1, 343-1, 357-1), which distinguish it from the other Diplodocus specimens (Table S69). One of these ‘synapomorphies’ (357-1: a subtriangular process on the scapular blade) also occurs in other diplodocines.

Three changes are recovered between AMNH 223 and the remaining triplet, indicating that they belong to a single species, as was already suggested by McIntosh (2005). The mean pairwise dissimilarity value recovered for comparison among these four specimens (0.0534) also support a referral to a single species.

Diplodocus longus YPM 1920 + (AMNH 223 + (‘Seismosaurus’ hallorum NMMNH 3690 + (DMNS 1494 + USNM 10865)))

Although not recovered in the four main trees discussed here, symmetric resampling yielded a value of four for this clade (using implied weighting; Table S59). Such a grouping, however, is not supported by any ‘synapomorphy.’ In fact, when adding the holotype specimen of D. longus to the reduced consensus trees, a polytomy is recovered between CM 84, CM 94, YPM 1920 and the clade including ‘Seismosaurus’ hallorum. Because the holotype of D. longus was excluded from all main trees except for the strict consensus trees, no autapomorphies were recovered for the specimen.

Diplodocus carnegii CM 84 + CM 94

In all but the equally weighted pruned tree, this clade forms the sister group to the clade including ‘Seismosaurus’ hallorum, as discussed above. The clade is supported by six shared ‘synapomorphies’ (139-1, 199-0, 247-1, 277-1, 295-0, 421-0). However, five of these are present in other diplodocine specimens (139-1, 199-0, 247-1, 295-0, 421-0; Table S70).

The two specimens are separated from each other by a single change, confirming Hatcher’s (1901) assignment of CM 94 as the paratype of the species D. carnegii. Both specimens were found in the same stratigraphic level of the same quarry. The mean pairwise dissimilarity value between the two specimens is 0.0638, and thus corroborates this referral to a single species as well.

Diplodocus carnegii + ‘Seismosaurus’ hallorum (=Diplodocus hallorum)

The grouping of these two species as sister taxa occurs in all trees that exclude the skull specimens CM 11161 and USNM 2672. The clade is united by one unambiguous (300-4) and eight shared ‘synapomorphies’ (182-0, 265-0, 280-1, 308-1, 331-1, 336-1, 414-2, 468-1; Table S71). Six of the shared ‘synapomorphies’ also occur in some other diplodocines (182-0, 265-0, 331-1, 336-1, 414-2, 468-1).

Eleven changes lie between the D. carnegii pair and the ‘Seismosaurus’ hallorum clade, whereas only six changes are recovered between the ‘Seismosaurus’ hallorum clade and D. longus. Mean pairwise dissimilarity between specimens in the two clades (0.1195) is higher than what is usually found within a species (<0.08), but substantially lower than values recovered between genera (>0.222). Both the apomorphy count and pairwise dissimilarity therefore suggest a distinction between D. carnegii and ‘S.’ hallorum at the species level, but they are not sufficient for genus-level separation. Seismosaurus is therefore here considered a synonym of Diplodocus, but as its own species D. hallorum, including the specimens AMNH 223, DMNS 1494, NMMNH 3690, and USNM 10865.

When adding Diplodocus longus YPM 1920 to our analyses, the grouping of CM 84 and 94 is lost, and a polytomy is formed as explained above. An inclusion of YPM 1920 in this diplodocine subclade is also supported by the mean pairwise dissimilarity value calculated for a group including both specimens of D. carnegii, the four specimens of D. hallorum, and YPM 1920. The value (0.0951) is lower than what is found in most other diplodocine genera (e.g., Barosaurus, 0.1236; Supersaurus, 0.1423). Because D. longus is the type species of Diplodocus (see below for a more detailed assessment of YPM 1920), the specimens included in its clade are herein referred to Diplodocus.

Diplodocus + md Diplodocinae (mdD)

Diplodocus is recovered as sister taxon to the clade of Kaatedocus + Barosaurus in all four principal trees discussed here. It is diagnosed by 16 synapomorphies, of which one is unambiguous (335-1), ten are shared (69-0, 154-0, 160-0, 196-1, 333-0, 381-1, 405-1, 416-1, 440-1, 461-0), and five are ambiguous (238-1, 258-2, 269-1, 281-1, 367-0; Table S72). Twelve synapomorphies are unique within Diplodocinae (69-0, 154-0, 160-0, 196-1, 238-1, 269-1, 335-1, 367-0, 405-1, 416-1, 440-1, 461-0).

The Diplodocus clade is separated from its sister clade CM 3452 + (Kaatedocus + Barosaurus) by 21 changes, and both Diplodocus and its sister clade are diagnosed with an unambiguous synapomorphy. Seven synapomorphies of the clade CM 3452 + mdD are based on cranial material, none of which is definitely attributable to the Diplodocus clade (2-0, 10-1, 19-1, 48-2, 52-1, 65-1, 67-1). All of these traits are different from the two included skulls CM 11161 and USNM 2672, which probably belong to the genus Diplodocus (see below for a discussion of their taxonomic affinities). The synapomorphies are thus tentatively retained in the count of changes between the clades, confidently justifying generic separation of CM 3452 + (Kaatedocus + Barosaurus) from Diplodocus. Mean pairwise dissimilarity values between specimens in these genera indicate that Diplodocus is morphologically most similar to Barosaurus (0.2048). This value is actually intermediate to what is generally found between different genera (>0.222) and species of the same genus (<0.181), however, because in the tree topology, Kaatedocus is found as sister taxon to Barosaurus, and because the mean pairwise dissimilarity value between these two genera (0.3029) is clearly above the threshold established for generic distinction, we prefer to keep all three genera as valid.

‘Diplodocus’ hayi HMNS 175 + ML 418

All four principal trees show this clade, which is supported by one shared ‘synapomorphy’ (165-0; Table S73). The ‘synapomorphy’ does not occur in any other diplodocine, and would therefore count as species autapomorphy. There are no valid distinguishing characters between these two specimens. However, they were found on different continents. Furthermore, given the high incompleteness of ML 418, and the very low overlap index of 4% when including all characters (Table S59), an assignment of ML 418 to the same species as HMNS 175 seems to be supported by very little positive data. The affinities of ML 418 will be discussed in more detail below.

SMA 0011 + (‘Diplodocus’ hayi HMNS 175 + ML 418)

This clade is found by all major trees. Six shared ‘synapomorphies’ are recovered to support this triplet (35-1, 60-2, 67-1, 72-1, 79-0, 90-1), but only one is unique within Diplodocinae (60-2; Table S74). All ‘synapomorphies’ describe skull features, and thus do not confirm the inclusion of ML 418, which only includes axial and appendicular elements.

Nine changes occur between SMA 0011 and its sister-clade. None of these characters are shared with other diplodocines. Although this would allow specific separation of SMA 0011 from its sister-clade, following the quantitative guidelines established above, it is herein refrained from naming a new species without providing a detailed description.

USNM 2673 + (SMA 0011 + (‘Diplodocus’ hayi HMNS 175 + ML 418))

Both reduced consensus trees and the pruned tree found by implied weighting recover this clade (Table S59). One unambiguous synapomorphy is recovered to diagnose this clade (12-1; Table S75), but can only be scored in half of the included specimens.

Eight changes lie between USNM 2673 and the other specimens. Three of them are unique within Diplodocinae. Whereas this would allow specific separation, a detailed assessment of the specimens included in this clade is needed before we can assess specific diversity. Furthermore, attribution of disarticulated skulls to diplodocine species is still a difficult task, given the small number of specimens preserving cranial and postcranial material together. A more detailed assessment of the affinities of USNM 2673 will follow below.

AMNH 969 + (USNM 2673 + (SMA 0011 + (‘Diplodocus’ hayi HMNS 175 + ML 418)))

This group is recovered in all main trees, except for the reduced consensus tree obtained with equal weighting, where AMNH 969 was pruned a posteriori (Table S59). It is supported by one ambiguous (62-0), two shared (47-1, 146-1), and two unambiguous ‘synapomorphies’ (148-1, 151-1), which all describe morphological features of the skull, or the atlas-axis complex (Table S76). Due to the rare finds of atlantes and axes, these synapomorphies are somewhat dubious, and will have to be assessed in more detail once more complete specimens become available for study. However, the consistent recovery of this clade in the same phylogenetic position, as well as the fact that this clade is separated from its sister clade Diplodocus + mdD by 21 differing apomorphic features, 15 of which are unique within Diplodocinae, indicates that this grouping forms a unique genus. The genus will be called Galeamopus gen. nov., typified by its type species Galeamopus hayi comb. nov. (see systematic paleontology below).

Two changes lie between AMNH 969 and the rest of the clade, therefore not allowing the erection of a separate species. The affinities of AMNH 969 will be discussed in more detail below. Mean pairwise dissimilarity within Galeamopus (not including the dubious specimen ML 418) is highest when compared to other diplodocine genera (0.1805 compared to otherwise maximum 0.1423 in Supersaurus), indicating that a presence of two species within this clade could be supported also by morphological disparity.

Galeamopus + mdD

All four trees show the new genus Galeamopus as sister taxon to the clade of Diplodocus + (Kaatedocus + Barosaurus). One unambiguous synapomorphy diagnoses this group (298-1; Table S77). This low number results from the fragmentary state of the closest outgroup to Galeamopus + mdD, which is the holotype specimen of Leinkupal laticauda, a single anterior caudal vertebra. The recovery of apomorphies for both Leinkupal and Galeamopus + mdD is thus limited to characters of the anterior caudal vertebrae.

Leinkupal + mdD

The position of Leinkupal as a sister-taxon to the clade Galeamopus + mdD is found by all of our principal analyses. It is supported by one unambiguous synapomorphy (315-1; Table S78).

Within this clade, Leinkupal is only separated from Galeamopus + mdD by two changes. Although this would not typically be seen as evidence for generic or even specific separation, it is clear that Leinkupal is a unique genus, based on its geographic and temporal isolation compared to all other diplodocids (Gallina et al., 2014). Also, mean pairwise dissimilarity between Leinkupal and Galeamopus shows a relatively high value (0.25). Finally, because the current paper was already in review when Gallina et al. (2014) was published, we refrained including the apomorphic features proposed by Gallina et al. (2014) in the present character list, thereby further limiting the apomorphy count. The autapomorphies of Leinkupal proposed by Gallina et al. (2014) will be discussed below.

Supersaurus vivianae BYU + WDC DMJ-021

This clade comprising the two Supersaurus specimens included in the present analysis is well supported. All four trees show this arrangement, and resampling yielded support values of 26 (ew) or 24 (iw), which are among the three highest support values recovered within Diplodocidae (Table S59). Eight shared ‘synapomorphies’ define this clade (131-0, 136-1, 172-1, 184-2, 231-0, 254-0, 296-1, 307-0; Table S79). Recovery of these ‘synapomorphies’ highly depends on tree topology, and thus the type of analysis performed. In the main trees obtained through implied weighting, where Supersaurus was found as the sister-group to Australodocus, only one ‘synapomorphy’ was found to unite the two Supersaurus specimens (184-2). On the other hand, from the other seven ‘synapomorphies,’ three are shared with Australodocus bohetii (131-0, 136-1, 172-1), and are found as synapomorphies of the clade uniting Supersaurus and Australodocus as recovered by the main implied weights trees (see below). In any case, attribution of the two specimens to Supersaurus appears to be well supported, and the absence of any valid differences between the specimens confirms the referral of WDC DMJ-021 to the type species S. vivianae, also corroborating the assignment of the various bones in the BYU collection to a single individual, as suggested by Lovelace, Hartman & Wahl (2007). A referral of the two specimens to a single species is also supported by the low pairwise dissimilarity value of 0.0738 between the BYU individual and WDC DMJ-021.

Australodocus bohetii type + Supersaurus

This group was only recovered in the main trees of the analysis with implied weighting. In these trees, Australodocus is nested within the clade uniting Dinheirosaurus and Supersaurus, which is in contrast to the latest identifications of Australodocus as a titanosauriform (Whitlock, 2011c; Mannion et al., 2013). Resampling does not support the clade of Australodocus + Supersaurus (Table S59). However, three shared synapomorphies are found, all of which do also occur in other diplodocines (131-0, 136-1, 172-1; Table S80). Two of them are shared with titanosauriforms (131-0, 172-1).

Australodocus bohetii and Supersaurus vivianae are separated by twelve changes, supporting specific, but not generic separation. However, pairwise dissimilarity values between the A. bohetii holotypic individual and the two Supersaurus vivianae OTUs yield values close to or greater than 0.222 (0.2188 with the BYU specimen; 0.3571 with WDC DMJ-021), indicating generic distinction. Given the weak morphological support for a position close to Supersaurus, and the fact that Australodocus comes from Tanzania, the relatively small number of changes in the apomorphy count is herein interpreted as a result of the incompleteness of the Australodocus remains, and possible convergent features in Australodocus and Supersaurus cervical vertebrae. It thus seems more prudent to retain Australodocus as a valid genus. The phylogenetic position of Australodocus will be discussed in more detail below.

Dinheirosaurus lourinhanensis ML 414 + Supersaurus

A sister taxon relationship of these two taxa to the exclusion of others is only recovered by using equal weights. When applying implied weights, this clade also includes Australodocus bohetii. Interestingly, only the latter arrangement including Australodocus is supported by resampling, although by a low value (Table S59). When excluding Australodocus, one unambiguous (251-1) and five shared synapomorphies (176-1, 177-1, 250-0, 272-1, 284-1) diagnose this clade, but this count is reduced to one unambiguous (251-1) and three shared synapomorphies (176-1, 250-0, 284-1) in the clade including Australodocus (Table S81).

Dinheirosaurus lourinhanensis ML 414 is separated from Supersaurus by eleven, and from the Australodocus + Supersaurus clade by six changes. The low number would thus not allow for generic separation. Mean pairwise dissimilarity values also appear to reject generic distinction. The value between D. lourinhanensis ML 414 and the two specimens of S. vivianae (0.2) is considerably higher than the difference between Diplodocus carnegii and Diplodocus hallorum (0.1195), and approaches the threshold for generic distinction (0.222). However, the value within the group including the two specimens of Supersaurus and the holotype of Dinheirosaurus lourinhanensis (0.1423) is lower than the value for the genus Galeamopus (0.1805), and considerably below the threshold for generic separation (0.222). Given that both the apomorphy count as well as the pairwise dissimilarity values seem to be quite consistent within Diplodocinae, and that genera of other dinosaurs were already reported to be present in both the Morrison Formation and the Lourinhã Formation, where Supersaurus and Dinheirosaurus come from, respectively, it would thus be best supported to synonymize Dinheirosaurus with Supersaurus, creating the new combination Supersaurus lourinhanensis.

Supersaurus + mdD

This clade is found in all four trees, with one single difference: the exclusion or inclusion of Australodocus bohetii (see above; Table S59). Two shared (339-1, 420-0) and two ambiguous synapomorphies (412-1, 448-1) were recovered for the clade of Supersaurus + mdD, regardless of the inclusion of Australodocus (Table S82). The ambiguous synapomorphies are absent in some single specimens of Diplodocus (412-0: AMNH 223 and CM 84; 448-0: DMNS 1494), but otherwise present in all other diplodocine specimens included in the analysis, indicating that these features might be individually variable in Diplodocus, but not in other diplodocines.

Within the clade Supersaurus + mdD, the Supersaurus clade is separated from Leinkupal + mdD by seven (excluding Australodocus) or five (including Australodocus) changes. This low number is mostly due to the fact that, because of the incompleteness of the type of Leinkupal, only one recognizable synapomorphy was found to diagnose Leinkupal + mdD. We therefore retain the generic separation of Leinkupal and other taxa at this phylogenetic split, which is furthermore supported by the tree topology, as well as by mean pairwise dissimilarity, which finds a value of 0.2564 for comparisons between Supersaurus and Leinkupal.

Tornieria africana holotype + skeleton k

The earlier referral of these two specimens to Tornieria (Remes, 2006; Remes, 2009) is confirmed by all analyses performed herein. They show a resampling value of two (ew) or ten (iw; Table S59), and four shared synapomorphies, which all describe appendicular morphology, and all also occur in other diplodocine species (362-0, 379-1, 418-1, 426-1; Table S83). The apparent lack of vertebral characters is due to the destruction of most putative Tornieria vertebrae during World War II (Remes, 2006; Whitlock, 2011a). A series of caudal vertebrae from trench “dd” from Tendaguru (MB.R.2956), referred to Tornieria by Remes (2006), was not included in our analysis, due to concerns of their attribution to the same individual raised by Remes (2006).

No valid autapomorphies are recovered for either Tornieria specimen, and mean pairwise dissimilarity between the two specimens shows the lowest value for any clade (0.0333). The referral of skeleton k to the species T. africana is therefore well-supported.

Tornieria + mdD

A clade with Tornieria and more derived Diplodocoidea to the exclusion of other diplodocine specimens was recovered in both analyses (Table S59). One shared (332-1) and two ambiguous ‘synapomorphies’ were found for this clade (307-1, 329-0; Table S84).

Eight changes are recovered between Tornieria and the more derived clade Supersaurus + mdD. This would not typically be considered as supporting the maintenance of a distinct genus for Tornieria, but generic distinction is supported by tree topology, geographical separation, and mean pairwise dissimilarity between specimens of Tornieria and those of other genera, which range from 0.2222 (Leinkupal) to 0.3333 (Kaatedocus). A high value is also found between Tornieria and Supersaurus (0.2987), which form two successive clades within Diplodocinae.

WDC-FS001A + SMA 0087

The clustering of these two specimens is found in all principal trees. They have a very low anatomical overlap, indicated by the “all chars” index of 11% (Table S59). Four shared ‘synapomorphies’ characterize the clade (324-1, 444-0, 445-1, 455-1; Table S85). A single change separates the two specimens, indicating that they might belong to a single species. Support for such a referral by pairwise dissimilarity is ambiguous, given the value of 0.1132, which is lower, but very close to the 0.1195 found between the two species of Diplodocus. Therefore, more detailed study of the material will be needed in order to definitely assess the systematic position of these two specimens.

WDC-FS001A + mdD

This clade includes all diplodocines (under implied weighting) or all diplodocines other than FMNH P25112 (under equal weights). Therefore, in the analysis using implied weighting, this clade is equivalent to Diplodocinae (Table S59), and will be discussed below.

Four ambiguous synapomorphies support this clade to the exclusion of FMNH P25112 (273-1, 355-0, 421-1, 422-0; Table S86). However, none of these are shared by both members of the clade WDC-FS001A + SMA 0087. This arrangement, with FMNH P25112 as most basal member of Diplodocinae is therefore not strongly supported by synapomorphies.

WDC-FS001A + SMA 0087 are separated from the more derived diplodocines by seven changes, which would allow specific separation, but not erection of a distinct genus. On the other hand, mean pairwise dissimilarity values between these specimens and those within distinct genera are relatively high, the lowest being 0.2988 (Supersaurus), and would thus support generic separation. Given that the two specimens are not fully prepared, more detailed studies have to be awaited to establish their systematic position.

Diplodocinae

As mentioned above, the composition of Diplodocinae changes depending on the weighting strategy applied. Equal weighting finds the specimen FMNH P25112 as most basal diplodocine taxon, whereas implied weighting recovers the same specimen as an apatosaurine. Another difference between the two weighting strategies is the position recovered for Australodocus. Although deleted from the main pruned and reduced consensus trees in the equally weighted analysis, a pruned consensus tree with Australodocus added to the OTUs retained in the reduced consensus tree finds Australodocus in a polytomy with FMNH P25112, SMA 0087, WDC-FS001A, and Tornieria + mdD. Thus, although it was recovered as diplodocine regardless, the position of Australodocus is shifted basally when applying equal weighting, in comparison to implied weighting.

Applying the guidelines for assessing the significance of synapomorphies, implied weighting finds one shared (442-1) and five ambiguous synapomorphies for Diplodocinae as recovered here (267-1, 273-1, 300-3, 421-1, 459-1; Table S87). One of these is shared with the Diplodocinae clade as found by the equally weighted analysis (267-1), which found two additional ambiguous synapomorphies for Diplodocinae including FMNH P25112 (293-1, 416-0; Table S87). Of the latter clade, only one synapomorphy is not shared with any apatosaurine specimen (416-0), whereas the synapomorphies found by using implied weighting include three features that are absent in Apatosaurinae (421-1, 442-1, 459-1).

In trees depicting FMNH P25112 as the most basal diplodocine, this specimen is separated from the more derived group by 13 changes, which would allow the erection of a new genus. However, given that its position changes in the two analyses, we refrain from erecting a new taxon based on this single specimen. Its affinities are discussed in more detail below.

Apatosaurus ajax YPM 1860 + BYU 1252-18531

This clade is only found by the trees recovered when using implied weighting, and is supported by a resampling value of two (Table S59). It is characterized by one unambiguous (206-1) and nine shared ‘synapomorphies’ (253-1, 260-1, 270-1, 293-0, 328-0, 329-0, 361-0, 365-1, 369-1), but only the unambiguous and one shared ‘apomorphies’ (260-1) are unique within Apatosaurinae (Table S88), and would thus qualify as species autapomorphies. These two specimens would be separated by five changes, if this position of Apatosaurus ajax YPM 1860 within Apatosaurinae were confirmed. A more detailed assessment will be given below.

UW 15556 + BYU 1252-18531

A clade comprising these two specimens to the exclusion of all others is only found by the equally weighted analysis, and is supported by a resampling value of ten, which is considerably higher than the value found for a clade uniting Apatosaurus ajax YPM 1860 and BYU 1252-18531 (Table S59). However, only five shared ‘synapomorphies’ are found to unite this clade (163-0, 179-1, 188-1, 259-1, 264-1), and only one of these is unique within Apatosaurinae (264-1; Table S89). The two specimens are separated from each other by eight changes, which would be enough for specific, but not generic separation. However, a detailed description of BYU 1252-18531 is in preparation (R Scheetz, pers. comm., 2014), and we therefore refrain from naming a new taxon at this time.

UW 15556 + mdA

This clade is equivalent to the grouping discussed above in the equally weighted analyses, but includes Apatosaurus ajax YPM 1860 when applying implied weighting (Table S59). The latter topology is supported by four shared synapomorphies (163-0, 259-1, 264-1, 390-1), of which one is unique among apatosaurines (264-1), and two could not be scored in A. ajax YPM 1860 (259-1, 264-1; Table S89). If A. ajax YPM 1860 does belong to this clade then 14 changes are present between UW 15556 and (BYU 1252-18531 + YPM 1860). This would allow for generic separation and restrict Apatosaurus to a single species.

Elosaurus parvus CM 566 + mdA

This grouping is recovered by all principal trees, with the sole difference of the inclusion or exclusion of Apatosaurus ajax YPM 1860 (see above; Table S59). Depending on the weighting strategy, and thus the inclusion of A. ajax YPM 1860, this clade is supported by three (ew, without A. ajax; 238-0, 274-0, 388-1; Table S90), or only one ‘synapomorphy’ (iw, including A. ajax; 274-0). Given that A. ajax YPM 1860 could not be scored for C274, the inclusion of A. ajax in this clade is not supported by synapomorphies.

Ten (ew) or nine (iw) changes are found between Elosaurus parvus CM 566 and its sister-clade. Although the number of changes would allow for specific distinction, a closer look at the distinguishing features reveals that all five ‘autapomorphies’ found for CM 566 are morphologies that were reported to change during ontogeny in the past (reduced cervical lamination, 135-0; lack of dorsal pneumatopores, 227-0, and 252-0; reduced muscle attachments in humerus, 386-0, or femur, 437-1; Varricchio, 1997; Schwarz et al., 2007; Carballido & Sander, 2014). Subtracting them from the count, support for specific separation is lost. Mean pairwise dissimilarity among specimens in this group, excluding Apatosaurus ajax YPM 1860, (0.2255) is higher than that within other clades representing single species (ranging from 0.1204, A. louisae to 0.2, NSMT-PV 20375 + “Atlantosaurus” immanis YPM 1840), and also exceeds some values between species found in different subclades within Apatosaurinae. On the other hand, it is considerably lower than the difference between genus-level clades (>0.26, see below). This indicates the presence of multiple distinct species within the triplet CM 566 + (UW 15556 + BYU 1252-18531), as is also shown by the apomorphy count between BYU 1252-18531 and UW 15556. However, given the unclear relationships within the clade, it seems most reasonable to refer the three specimens CM 566, UW 15556, and BYU 1252-18531 to a single species at this stage.

Eobrontosaurus yahnahpin Tate-001 + mdA

Such a clade is recovered in all our primary analyses. As for the Elosaurus + mdA clade discussed above, the results of the two analyses differ in the inclusion or exclusion of Apatosaurus ajax YPM 1860. Combining the information of the main trees, three shared synapomorphies are found (267-1, 271-1, 273-1; Table S91). One of these (271-1) is absent in A. ajax YPM 1860, and shared with other apatosaurine specimens, which is the reason why it was not recovered as a synapomorphy in the analysis with implied weighting. Another synapomorphy recovered under equal weighting is shared by FMNH P25112 (267-1). Given that FMNH P25112 is the sister-taxon to Eobrontosaurus + mdA in the analysis with implied weights, this synapomorphy is here found to characterize the clade FMNH P25112 + mdA instead (n.b. FMNH P25112 is found as the most basal diplodocine under equal weights).

In the equally weighted reduced consensus tree (excluding Apatosaurus ajax YPM 1860), Eobrontosaurus yahnahpin Tate-001 is separated from its sister-clade by seven changes. The trees obtained by implied weighting yield a distance of five changes from E. yahnahpin Tate-001 to Elosaurus parvus CM 566 + mdA. Whereas this is not enough for generic separation, it is sufficient for specific distinction. Specific separation, at the least, is also supported by mean pairwise dissimilarity between Tate-001 and the specimens in the ‘Elosaurus’ parvus clade, which shows a value of 0.2298.

FMNH P25112 + mdA

This clade is only present in the trees recovered with implied weighting (Table S59; FMNH P25112 is a diplodocine under equal weights). It is supported by two synapomorphies (267-1, 438-1; Table S92), one of which characterizes the less inclusive clade Eobrontosaurus + mdA in the equally weighted analysis (267-1). Five changes occur between FMNH P25112 and Eobrontosaurus yahnahpin Tate-001 + mdA. This low number does not allow specific separation. The affinities of FMNH P25112 are discussed in more detail below.

Amphicoelias altus AMNH 5764 + mdA

Although this clade was found in all our principal trees, its composition changes depending on the weighting strategies applied: implied weighting includes the specimens FMNH P25112 and Apatosaurus ajax YPM 1860, whereas equal weighting excludes these specimens (Table S59). In both cases, the clade Amphicoelias altus AMNH 5764 + mdA is supported by two synapomorphies (280-0, 430-0; Table S93), which are both shared in the equally weighted analysis, whereas one of them (430-0, shape of the cross-section of the femur) is ambiguous in the trees recovered with implied weighting, because of the much more elliptical femur midshaft section of the specimen FMNH P25112 compared to Amphicoelias altus AMNH 5764 or Elosaurus parvus CM 566.

Amphicoelias altus AMNH 5764 is separated from its sister clade by six (ew) or five (iw) changes. However, given its insecure position (see below) and incomplete preservation, Amphicoelias altus should be kept as a separate genus and species from the remaining taxa in the clade A. altus + mdA. It is probable that more complete finds of A. altus will clarify its position in future. A more detailed assessment of its phylogenetic position will follow below.

Brontosaurus excelsus YPM 1980 + Brontosaurus amplus YPM 1981

These two specimens were found to form a clade in all four principal trees, and are supported by a resampling value of three when applying implied weighting (Table S59). The clade is characterized by three shared synapomorphies, all of which also occur in other apatosaurines (284-1, 310-1, 427-2; Table S94). Two of them are found with equal weighting, and one with implied weighting.

The single specimens are separated from each other by five changes, which does not allow for specific distinction. Furthermore, pairwise dissimilarity is relatively low for apatosaurines (0.1429). The generally applied synonymization of B. amplus with B. excelsus (e.g., McIntosh, 1995; Upchurch, Tomida & Barrett, 2004) is therefore supported by our analysis.

Brontosaurus + mdA

This clade is found in all four principal trees, and is supported by four shared (237-1, 288-1, 350-1, 451-1) and two ambiguous synapomorphies (293-1, 184-0; Table S95). Two shared synapomorphies only occur in one analysis each (237-1 in ew, and 288-1 in iw) and are considered invalid in the other analysis. In both cases, the sum of synapomorphies is thus five.

The changes found between this clade including the two type specimens of the proposed Brontosaurus species and its sister-clade are five. In case Amphicoelias altus should not be an apatosaurine (see below), five changes lie between Brontosaurus excelsus + B. amplus and FMNH P25112 (using implied weights), and six changes separate B. excelsus + B. amplus from Eobrontosaurus yahnahpin + mdA (using equal weights). Specific distinction is thus probable, but generic separation is not warranted. Mean pairwise dissimilarity values between B. excelsus and other apatosaurine taxa range from 0.1826 (Apatosaurus ajax) to 0.239 (Apatosaurus louisae), which are all around the boundary recognized between species and genera in Diplodocinae (0.2). However, when calculating the distance for higher-level groups, a genus including the type specimens of Brontosaurus, Elosaurus, and Eobrontosaurus has an internal mean pairwise dissimilarity value of 0.2149, whereas the differences between this group and a clade with Apatosaurus ajax, Apatosaurus louisae, and Apatosaurus laticollis (which is the sisterclade in the equally weighted analysis) results in a considerably higher dissimilarity (0.2606). Generic distinction between these two clades therefore seems supported, whereas the clade Brontosaurus + mdA appears to include only specific variation. Given that Brontosaurus is the earliest genus named from the ones included in this clade (when excluding Amphicoelias, see below), Elosaurus and Eobrontosaurus should be treated as junior synonyms of Brontosaurus, which would include the species B. excelsus, B. yahnahpin, and B. parvus.

Apatosaurus laticollis YPM 1861 + Apatosaurus louisae type

This grouping is only found in the reduced consensus trees, when excluding CM 3378 and LACM 52844, which create polytomies in the pruned consensus trees (Table S59). Three shared synapomorphies are recovered to support A. laticollis YPM 1861 + A. louisae type, but all of them are shared with other apatosaurine specimens (199-0, 219-1, 222-1; Table S96), and would thus not qualify as species autapomorphies following our protocol. The two specimens are separated from each other by only two changes, thereby not supporting specific separation between them.

Apatosaurus laticollis YPM 1861 + Apatosaurus louisae type + CM 3378

A clade only including these three specimens is recovered in the pruned tree using implied weighting, and supported by a relatively high resampling value of 28 (or four with equal weighting; Table S59). Two shared ‘synapomorphies’ are considered reliable (222-1, 329-0; Table S97), although one of these ‘synapomorphies’ (329-0) also occurs in other apatosaurines.

One change separates CM 3378 from the other two specimens. Mean pairwise dissimilarity is lowest within Apatosaurinae (0.1204), supporting a referral of all three specimens to a single species.

LACM 52844 + (Apatosaurus laticollis YPM 1861 + Apatosaurus louisae type + CM 3378)

A clade comprising LACM 52844 as the sister taxon of a group including A. laticollis YPM 1861 and A. louisae type occurs in both trees obtained with implied weighting (Table S59), but CM 3378 was deleted during the calculation of the reduced consensus tree. The pruned tree resulting from the equally weighted analysis shows a polytomy of the four specimens. Depending on the position of A. ajax YPM 1860, which is found as the sister-taxon to the present clade in the equally weighted analysis, but not when applying implied weights, eight (ew; 194-1, 199-0, 217-1, 218-1, 219-1, 222-1, 240-1, 283-1) or two (iw; 208-0, 218-1) synapomorphies are found to support the clade including LACM 52844 + (A. laticollis YPM 1861 + A. louisae type + CM 3378; Table S98). Only one of these synapomorphies is found by both analyses (218-1). Three are recovered as ‘synapomorphies’ for the clade A. laticollis YPM 1861 + A. louisae type by the equally weighted reduced consensus tree (199-0, 219-1, 222-1), which excludes both CM 3378 and LACM 52844, whereas one (222-1) is found to characterize the clade excluding LACM 52844 in the pruned tree with implied weighting. The latter interpretation is the most parsimonious, because LACM 52844 indeed shows the plesiomorphic state for character 222. The derived state (222-1) was only found to be a synapomorphy of the clade discussed in this paragraph by the equally weighted pruned tree, which recovers all four specimens in a polytomy, and thus the synapomorphy as an exclusive trait within this quartet (Table S98).

Seven changes separate LACM 52844 from the remaining triplet, which would allow specific separation, according to the guidelines established above. However, mean pairwise dissimilarity among the four specimens amounts to 0.1944, which is below the value found for Brontosaurus parvus (0.2255). A more detailed assessment of the affinities of LACM 52844 will follow below.

Apatosaurus ajax YPM 1860 + (LACM 52844 + mdA)

A. ajax YPM 1860 is only found as sister taxon to LACM 52844 + mdA in the equally weighted analysis (Table S59; under implied weights, YPM 1860 is found as the sister taxon of BYU 1252-18531). Six shared synapomorphies characterize the clade including YPM 1860 + (LACM 52844 + mdA) (169-1, 187-1, 208-0, 253-1, 328-0, 368-0; Table S99). One of these synapomorphies (208-0) was found to unite the less inclusive clade LACM 52844 + mdA by the analysis with implied weights (in which A. ajax YPM 1860 was recovered elsewhere). Only two synapomorphies found for A. ajax YPM 1860 + (LACM 52844 + mdA) are unique within Apatosaurinae (187-1, 368-0).

The distance between Apatosaurus ajax YPM 1860 and its sister clade is 12 changes. Because A. ajax and A. louisae are generally considered two species of the same genus, and were recovered as such in our equally weighted analysis, this taxon pair was taken as one of the main pairs on which quantitative thresholds of our numerical taxonomic approach were based. They are therefore two distinct species of a single genus by default. Mean pairwise dissimilarity between specimens of these two species equals 0.1831, which is also lower than the 0.222 found significant enough to distinguish genera in Diplodocinae. A referral of the specimens CM 3018, CM 3378, LACM 52844, YPM 1860 and YPM 1861 to a single genus is thus supported by morphology. Given that this clade includes the genoholotype specimen of Apatosaurus (A. ajax YPM 1860), Apatosaurus is the preferred name for this genus.

Apatosaurus + mdA

This clade is found with both weighting methods. As mentioned previously, the two weighting strategies yield different positions for A. ajax YPM 1860 and the specimen FMNH P25112, but otherwise the composition of this clade is invariant. One unambiguous (223-1) and one ambiguous synapomorphies (297-0) support this clade (Table S100). None of these synapomorphies occur in any other apatosaurine specimen.

In the equally weighted trees, the clade comprising A. ajax YPM 1860 + A. louisae type is separated from its sister clade Brontosaurus excelsus YPM 1980 + mdA by eleven changes, whereas in the analysis with implied weights, 14 changes are counted between the A. louisae and the B. excelsus + UW 15556 clades. The difference lies in the position of A. ajax YPM 1860, which influences the number of ‘synapomorphies’ found in these two groups. Therefore, the analysis with implied weights suggests the presence of two different genera, whereas only specific separation is supported with equal weighting. As mentioned above, also mean pairwise dissimilarity between specimens of Apatosaurus and those of Brontosaurus (0.2606) supports generic distinction: intrageneric mean pairwise dissimilarity is lower (0.1831 for Apatosaurus, and 0.2149 for Brontosaurus) than what is found between the two groups. Both Brontosaurus and Apatosaurus should therefore be considered valid genera within Apatosaurinae.

AMNH 460 + (Apatosaurus + Brontosaurus)

This clade is recovered by all analyses (Table S59). A single shared (174-1) and eight ambiguous synapomorphies (138-1, 159-1, 179-0, 225-0, 238-1, 250-0, 254-0, 296-1) are found to support this arrangement (Table S101). Only two changes separate AMNH 460 from the more derived clade. Neither specific nor generic separation of AMNH 460 from its sister groups is thus warranted. The taxonomic affinities of AMNH 460 will be addressed below.

‘Atlantosaurus’ immanis YPM 1840 + NSMT-PV 20375

The grouping of these two specimens is recovered under both weighting strategies. Both specimens are usually interpreted as belonging to Apatosaurus ajax (McIntosh, 1995; Upchurch, Tomida & Barrett, 2004), but are here found as the most basal apatosaurines. Anatomical overlap is low, as indicated by the “all chars” index of 15%, but resampling by using implied weighting finds a support value of eleven for this clade, which is relatively high compared to other groups (Table S59). Five shared ‘synapomorphies’ are found (128-2, 168-0, 188-1, 237-1, 426-1; Table S102). All of them are shared with other apatosaurines and would thus not qualify as species autapomorphies. Two traits also occur in Apatosaurus ajax YPM 1860 (168-0, 426-1), which supports the earlier identifications, and casts additional doubt on the position recovered herein. Assuming that ‘Atlantosaurus’ immanis YPM 1840 and NSMT-PV 20375 do form a clade, no changes are found to separate the two specimens. Pairwise dissimilarity is relatively high between the two specimens (0.2), compared to othe apatosaurine species, but still lower than the value found for Brontosaurus parvus (0.2255). We therefore refer both specimens to a single species.

Apatosaurinae

The phylogenetic definition of Apatosaurinae specifies all taxa more closely related to Apatosaurus ajax than to Diplodocus longus. Therefore, an apatosaurine clade must be recovered by our analysis, which includes both species, although the composition might change. However, other than the differences in tree topology within Apatosaurinae discussed above, none occurs here (Table S59). One ambiguous (372-1) and five shared synapomorphies (160-0, 186-1, 216-1, 220-1, 324-0) of Apatosaurinae are found under equal weighting, of which one shared synapomorphy becomes unambiguous (216-1), and a second invalid (324-0) when applying implied weighting, due to the different position of FMNH P25112 (Table S103).

The most basal apatosaurine recovered by our analyses, ‘Atlantosaurus’ immanis YPM 1840 + NSMT-PV 20375 is separated from more derived apatosaurines by 14 changes, which would be enough to maintain a distinct genus. The same conclusion can be drawn from pairwise dissimilarity values between ‘Atlantosaurus’ immanis YPM 1840 + NSMT-PV 20375 and the genera Apatosaurus and Brontosaurus as defined above. The values are both higher (0.2704 with Apatosaurus; 0.2609 with Brontosaurus) than the difference between Apatosaurus and Brontosaurus (0.2545), thus indicating that a third apatosaurine genus might be present.

Diplodocidae

Twenty-two synapomorphies support this clade, two unambiguous (25-1, 127-1), six exclusive (17-1, 23-1, 224-2, 259-1, 314-1, 329-1), five shared (263-0, 316-1, 319-1, 383-0, 428-1), and nine ambiguous (50-1, 199-1, 208-1, 256-0, 275-1, 297-1, 321-1, 392-1, 461-1; Table S104). Diagnostic synapomorphies occur in all regions of the skeleton, including cranial, axial, and appendicular domains. Depending on the position of FMNH P25112, 16 (ew) or eight (iw) changes separate Apatosaurinae from Diplodocinae.

Flagellicaudata

The node-based taxon Flagellicaudata includes Diplodocidae and Dicraeosauridae. It is supported by eight unambiguous (8-1, 15-1, 54-1, 56-1, 59-1, 104-1, 122-1, 423-1), two exclusive (87-1, 303-1), nine shared (51-0, 123-0, 276-0, 313-1, 318-1, 352-0, 424-0, 425-1, 463-1), and eleven ambiguous synapomorphies (58-1, 126-1, 179-1, 202-0, 250-1, 261-1, 304-1, 305-1, 355-1, 371-1, 465-1; Table S105). One of the above mentioned synapomorphies was recovered as instead diagnosing Diplodocimorpha in the implied weight trees (318-1), because the sprl also extends onto the lateral aspect of the caudal neural spines in rebbachisaurids. Because Cetiosauriscus and Haplocanthosaurus were recovered as diplodocoid sauropods more derived than rebbachisaurids in the equally weighted analysis, but have a reduced caudal sprl, a well developed sprl is a shared synapomorphy of rebbachisaurids and flagellicaudatans under equal weights. However, if, as in the trees found by using implied weighting, Cetiosauriscus and Haplocanthosaurus are found to be more basal to rebbachisaurids, the well-developed caudal sprl becomes a synapomorphy of Diplodocimorpha as defined by Taylor & Naish (2005).

Proximally closed haemal arches (352-0) are also present in Cetiosauriscus stewarti NHMUK R3078. In the equally weighted pruned tree, where C. stewarti is recovered as diplodocoid more than Rebbachisauridae, this feature thus appears synapomorphic for the clade C. stewarti + mdD.

Cetiosauriscus + mdD

Such a clade is only found under equal weighting, where Cetiosauriscus stewarti NHMUK R3078 is recovered in a position between Rebbachisauridae and Flagellicaudata (implied weighting finds C. stewarti as a non-neosauropod eusauropod). Two shared synapomorphies support the placement of C. stewarti within Diplodocoidea (290-1, 352-0; Table S106). All of these synapomorphies are shared with more basal taxa, close to the position where Cetiosauriscus is recovered in the implied weights trees, and are thus not conclusive evidence for diplodocoid affinities of Cetiosauriscus.

Haplocanthosaurus + mdD

This clade corresponds to Diplodocoidea in the implied weights trees (i.e., Haplocanthosaurus is the most basal diplodocoid under implied weights), but is more restricted when applying equal weighting. In the latter analysis, Haplocanthosaurus is recovered as being more derived than Rebbachisauridae. Such an arrangement is supported by one exclusive (324-1) and four ambiguous synapomorphies (160-1, 181-1, 368-1, 412-1; Table S107). Therefore, under equal weights, no synapomorphy of Haplocanthosaurus + mdD is shared by all ingroup members.

Diplodocimorpha

This clade is often used in the same way as Diplodocoidea, but in fact has a node-based definition, whereas Diplodocoidea is stem-based (Taylor & Naish, 2005). In the present analyses, Diplodocimorpha is only different from Diplodocoidea when using implied weighting, where Haplocanthosaurus is recovered as being more basal than Rebbachisauridae. Under implied weights, even the complete strict consensus tree recovered a distinct Diplodocimorpha excluding Haplocanthosaurus (Table S59). One unambiguous synapomorphy (318-1) and one exclusive synapomorphy of Diplodocimorpha (300-1) are found to be reliable in the implied weights trees (Table S108). Even though there are few features supporting a diplodocimorph clade to the exclusion of Haplocanthosaurus, the fact that one of the apomorphies is unambiguous (318-1) indicates tangible support for such an arrangement.

Diplodocoidea

The clade Diplodocoidea is represented in all consensus trees except for the complete strict consensus tree obtained under equal weighting (Table S59). Due to the more derived position of Haplocanthosaurus priscus in the equally weighted analyses compared to the analysis with implied weights, Diplodocoidea is equivalent to Diplodocimorpha in the former analysis. Synapomorphies of Diplodocoidea include 13 unambiguous (3-1, 5-2, 13-1, 21-1, 40-1, 45-1, 46-1, 93-1, 102-1, 115-1, 117-1, 121-1), six exclusive (2-1, 11-1, 18-1, 49-1, 119-1, 214-1), four shared (6-0, 22-0, 215-1, 384-0), and six ambiguous traits (64-1, 77-1, 379-0, 416-1, 428-0, 455-0; Table S109). Twenty-one of these synapomorphies describe cranial features, which are rarely preserved, and unknown in Haplocanthosaurus, which does not preserve a skull. Therefore, the analysis using implied weighting, in which Haplocanthosaurus is the most basal diplodocoid, was not able to recover any cranial apomorphies for Diplodocoidea. The assignment of cranial synapomorphies of Diplodocoidea should thus be regarded provisional.

Validity and taxonomic assessment of the holotype specimens

Discussion of the taxonomic affinities of the holotype specimens is ordered based on date of description. By doing so, possible synonymy of the species and genera can be assessed in a more intuitive way. The specimens are listed with the initially proposed name. The species referrals of the specimens proposed herein are summarized in Table 5.

Table 5 Summary of the taxonomic referrals of the specimen-level OTUs included in our analysis.

Specimens are ordered alphabetically. Diplodocids are marked with bold font, diplodocines are highlighted in green, and apatosaurines in blue. Where an assignation to a higher-level taxon is ambiguous (Australodocus), only parts of the row are highlighted.

Specimen-level OTU	Proposed species identification	Higher-level taxonomy	Type of	
AC 663	Dyslocosaurus polyonychius	Dicraeosauridae	Dyslocosaurus polyonychius	
AMNH 223	Diplodocus hallorum	Diplodocinae		
AMNH 460	Apatosaurinae indet.	Apatosaurinae		
AMNH 675	Macronaria indet.	Macronaria	‘Apatosaurus’ minimus	
AMNH 969	Galeamopus sp.	Diplodocinae		
AMNH 5764	Amphicoelias altus	Diplodocidae	Amphicoelias altus	
AMNH 5765	Camarasaurus supremus	Macronaria	‘Amphicoelias’ latus	
AMNH 6341	Barosaurus lentus	Diplodocinae		
AMNH 7530	Kaatedocus siberi	Diplodocinae		
AMNH 7535	Barosaurus sp.	Diplodocinae		
ANS 21122	Suuwassea emilieae	Dicraeosauridae	Suuwassea emilieae	
BYU 1252-18531	Brontosaurus parvus	Apatosaurinae		
BYU 4503	Supersaurus vivianae	Diplodocinae	Dystylosaurus edwini	
BYU 4503, 4839, 9024-25, 9044-45, 9085, 10612, 12424, 12555, 12639, 12819, 12861, 12946, 12962, 13016, 13018, 13981, 16679, 17462	Supersaurus vivianae	Diplodocinae	Supersaurus vivianae (in parts)	
CM 84	Diplodocus carnegii	Diplodocinae	Diplodocus carnegii	
CM 94	Diplodocus carnegii	Diplodocinae	Diplodocus carnegii (cotype)	
CM 3018, 11162	Apatosaurus louisae	Apatosaurinae	Apatosaurus louisae (in parts)	
CM 3378	Apatosaurus louisae	Apatosaurinae		
CM 3452	Diplodocinae indet.	Diplodocinae		
CM 566	Brontosaurus parvus	Apatosaurinae	“Elosaurus” parvus	
CM 11161	Diplodocinae indet.	Diplodocinae		
CM 11255	Diplodocinae indet.	Diplodocinae		
CM 11984	Barosaurus sp.	Diplodocinae		
DMNS 1494	Diplodocus hallorum	Diplodocinae		
FMNH P25112	Diplodocidae indet.	Diplodocidae		
HMNS 175	Galeamopus hayi	Diplodocinae	“Diplodocus” hayi	
LACM 52844	Apatosaurus sp.	Apatosaurinae		
MB.R.2386, 2572, 2586, 2669, 2673, 2726, 2730, 2733, 2913, 3816	Tornieria africana	Diplodocinae		
MB.R.2454-55	Australodocus bohetii	Titanosauriformes or Diplodocinae	Australodocus bohetii (in parts)	
MB.R.2672, 2713, 2728; SMNS 12140, 12141a, 12142, 12143, 12145a, c	Tornieria africana	Diplodocinae	Tornieria africana (in parts)	
MCNV Lo1-26	Losillasaurus giganteus	Turiasauria	Losillasaurus giganteus (in parts)	
MIGM 2, 4931, 4956-57, 4970, 4975, 4979-80, 4983-84, 5780-81, 30370-88	Lourinhasaurus alenquerensis	Macronaria	“Apatosaurus” alenquerensis (lectotype)	
ML 414	Supersaurus lourinhanensis	Diplodocinae	“Dinheirosaurus” lourinhanensis	
ML 418	Diplodocinae indet.	Diplodocinae		
MMCH-Pv 63-1	Leinkupal laticauda	Diplodocinae	Leinkupal laticauda	
NHMUK R3078	Cetiosauriscus stewarti	Eusauropoda	Cetiosauriscus stewarti	
NMMNH 3690	Diplodocus hallorum	Diplodocinae	“Seismosaurus halli”	
NSMT-PV 20375	new genus and species	Apatosaurinae		
SMA 0004	Kaatedocus siberi	Diplodocinae	Kaatedocus siberi	
SMA 0009	Brachiosaurus sp.	Titanosauriformes		
SMA 0011	Galeamopus sp.	Diplodocinae		
SMA 0087	new genus and species	Diplodocinae		
SMA D16-3	Kaatedocus siberi	Diplodocinae		
SMA O25-8	Diplodocinae indet.	Diplodocinae		
Tate-001	Brontosaurus yahnahpin	Apatosaurinae	“Eobrontosaurus” yahnahpin	
USNM 2364	Dystrophaeus viaemalae	Eusauropoda	Dystrophaeus viaemalae	
USNM 2672	Diplodocinae indet.	Diplodocinae		
USNM 2673	Galeamopus sp.	Diplodocinae		
USNM 10865	Diplodocus hallorum	Diplodocinae		
UW 15556	Brontosaurus parvus	Apatosaurinae		
WDC DMJ-021	Supersaurus vivianae	Diplodocinae		
WDC-FS001A	new genus and species	Diplodocinae		
YPM 419	Sauropoda indet.	Sauropoda	“Barosaurus” affinis	
YPM 429	Barosaurus lentus	Diplodocinae	Barosaurus lentus	
YPM 1840	new genus and species	Apatosaurinae	‘Atlantosaurus’ immanis	
YPM 1860	Apatosaurus ajax	Apatosaurinae	Apatosaurus ajax	
YPM 1861	Apatosaurus louisae	Apatosaurinae	Apatosaurus laticollis	
YPM 1901	Camarasaurus grandis	Macronaria	“Apatosaurus” grandis	
YPM 1920	Diplodocus sp.	Diplodocinae	Diplodocus longus	
YPM 1922	Flagellicaudata indet.	Flagellicaudata	“Diplodocus” lacustris	
YPM 1980	Brontosaurus excelsus	Apatosaurinae	Brontosaurus excelsus	
YPM 1981	Brontosaurus excelsus	Apatosaurinae	Brontosaurus amplus	

Dystrophaeus viaemalae USNM 2364

The phylogenetic position of Dystrophaeus viaemalae is dubious, mostly due to its fragmentary remains. In our analysis, the holotype USNM 2364 was among the four most unstable taxa, and thus was pruned in the equally weighted trees. Implied weighting recovered it consistently within Dicraeosauridae, as sister taxon to Suuwassea emilieae. The validity and phylogenetic position of Dystrophaeus viaemalae is particularly important because it was the first sauropod to be described from North America, and would thus have priority over any possibly synonymous taxon. The present study is the first to include the specimen in a phylogenetic analysis. Earlier studies proposed diplodocid affinities (McIntosh, 1997), but that was mainly based on the plesiomorphically short and robust metacarpals (Upchurch, Barrett & Dodson, 2004). The latter did not find any diagnostic feature in the fragmentary material, but refrained to classify Dystrophaeus as nomen dubium because it was found very low in stratigraphy, possibly even below the Morrison Formation.

One single, ambiguous autapomorphy was recovered for USNM 2364 (370-1; Table S110), describing the presence of a subtriangular projection on the ventral edge of the scapular blade. As recovered herein, this projection occurs in specimens from all major taxonomic groups included in the analysis. A single character ties Dystrophaeus viaemalae to Suuwassea emilieae (365-1). This trait is shared with Shunosaurus, Omeisaurus, Cetiosauriscus stewarti and several apatosaurine specimens. Another feature is shared between D. viaemalae and Dicraeosaurus hansemanni (390-0) and indeed found as synapomorphic for the dicraeosaurid clade excluding Dyslocosaurus polyonychius, which shows state 390-1. However, this feature also occurs in Shunosaurus, and is variable within Apatosaurinae. Incompleteness of the type specimen of Dystrophaeus viaemalae (USNM 2364) inhibits the scoring of any character providing synapomorphies of the higher-level clades herein found to include Dystrophaeus (Dicraeosauridae, Flagellicaudata, Diplodocimorpha and Diplodocoidea). A conflicting score occurs in an ambiguous synapomorphy of Diplodocidae (radius has reduced (392-0) instead of well-developed articulation facets for the ulna (392-1)). This implies that USNM 2364 is either not diagnostic due to fragmentary preservation, or is not a diplodocid sauropod.

In order to test these interpretations, constrained tree searches with equal weights were performed forcing USNM 2364 into a position within Dicraeosauridae as found by the implied weight trees, as well as forcing it into different positions outside Diplodocoidea. Imposing a grouping of USNM 2364 with Dicraeosauridae does not increase the tree length, but is unable to recover the exact position of D. viaemalae in the clade. Tree length also remained the same when constraining the position of D. viaemalae into Camarasauridae, where it grouped with Lourinhasaurus alenquerensis. A single synapomorphy supports the grouping with Lourinhasaurus: a beveled distal surface of the radius (393-1)—which is also present in several diplodocid specimens. Forcing USNM 2364 into Titanosauriformes, equal weighting recovers trees with a length of 1978 steps, two more than the most parsimonious unconstrained trees. Constraining USNM 2364 into a camarasaurid position under implied weights yielded a minimal tree length of 194.22685, which is an increase of 0.01082 steps, compared to the most parsimonious trees. As in the constrained equally weighted tree, D. viaemalae groups with Lourinhasaurus. The same result is obtained when excluding Dystrophaeus from Dicraeosauridae. Titanosauriform affinities are supported by minimum tree lengths of 194.3328, 0.11677 steps longer than the shortest trees. Dicraeosaurid and camarasaurid positions of Dystrophaeus are therefore equally supported by the equally weighted analyses. On the other hand, using implied weighting, a grouping of Dystrophaeus with Lourinhasaurus appears the second best interpretation, with a trivial tree length increase of 0.01%. Positions within Dicraeosauridae or Camarasauridae are thus nearly equally supported, whereas an inclusion in Diplodocidae can probably be excluded. More detailed studies are needed, including more representative taxa of basal Macronaria, basal Neosauropoda, and derived, non-neosauropod Eusauropoda, to resolve phylogenetic relationships of Dystrophaeus viaemalae and definitively assess its taxonomic validity.

Amphicoelias altus AMNH 5764

The holotype of Amphicoelias altus is found in the same position within Apatosaurinae in both analyses. However, this finding is in contrast to the positions found by Rauhut et al. (2005), Whitlock (2011a), Mannion et al. (2012) and Tschopp & Mateus (2013b), who recovered it more basal than Dicraeosauridae, and even outside Diplodocimorpha in most analyses (Rauhut et al., 2005; Whitlock, 2011a; Mannion et al., 2012).

Three ambiguous autapomorphies were considered valid for the holotype of Amphicoelias altus (256-1, 275-0, 427-0; Table S111), but are all shared with some diplodocine specimens. Nearly horizontal dorsal postzygapophyses (275-0) are widespread among sauropods, and thus are probably not a meaningful autapomorphy. Furthermore, the orientation of the posterior dorsal postzygapophyses in Amphicoelias contrasts with the state in all other apatosaurines. The possession of a gracile femur (427-0) contributes in part to the “stove-pipe” shape of this element, most often used as the best way to distinguish Amphicoelias from other sauropods (e.g., Wilson & Smith, 1996). In fact, this is the autapomorphy shared with the fewest other taxa in our dataset (Shunosaurus lii, Cetiosauriscus stewartii, Ligabuesaurus leanzai and Diplodocus USNM 10865). Amphicoelias shares the diplodocid synapomorphies of short posterior dorsal transverse processes (263-0), and the presence of a lateral bulge on the femur (428-1), neither of which are present in any other sampled diplodocoid sauropod. A diplodocid affiliation is thus probable. This is also supported by constrained searches, excluding Amphicoelias altus from Apatosaurinae, or forcing it into a close relationships with Supersaurus vivianae, which was found to be the closest fit in a preliminary morphological disparity analysis.

Inhibiting a grouping of Amphicoelias with Apatosaurinae in the equally weighted analysis results in a tree two steps longer than the original (0.1% length increase). Amphicoelias is here found as sister-taxon to Galeamopus hayi within Diplodocinae, but no synapomorphy supports this grouping. When doing the same with implied weighting, tree length increases by 0.01% to 194.24251. Here, Amphicoelias moves into a position basal to Apatosaurinae + Diplodocinae, but still within Diplodocidae. A close relationship with Supersaurus appears substantially less probable, increasing tree length by 0.15% (ew) or 0.11% (iw).

Mean pairwise dissimilarity supports diplodocine affinities of Amphicoelias altus slightly more than a referral to Apatosaurinae: principal coordinates 1 and 2 recover A. altus slightly closer to the diplodocine cluster than to the apatosaurine specimens (Fig. 112). Given the minimal length increase in the constrained analysis with implied weights, the absence of apatosaurine synapomorphies in A. altus, and the fact that previous analyses agreed in a more basal position for this taxon within Diplodocoidea, a position outside Apatosaurinae + Diplodocinae is herein interpreted as more plausible than the apatosaurine affinities recovered in the most parsimonious trees.

Amphicoelias latus AMNH 5765

All our analyses agreed on a position of AMNH 5765 within Camarasauridae. Amphicoelias latus has generally been synonymized with Camarasaurus supremus, following Osborn & Mook (1921).

No autapomorphies are found for Amphicoelias latus. The synapomorphies of Camarasaurus + Turiasauria, not shared with AMNH 5765, are a maximum to minimum mediolateral width of anterior caudal neural spines of 2.0 or greater (327-1), and a fourth trochanter on the femur, which is visible in anterior view (436-1). The first of these synapomorphies has actually been shown to be variable within Camarasaurus by Ikejiri (2004). The second is somewhat dubious, because AMNH 5765 was only scored based on the drawings in Cope (1877b) and Osborn & Mook (1921). Of the four synapomorphies recovered for Camarasaurus (92-0, 333-1, 391-1, 408-0), AMNH 5765 is not scorable for any of these. Furthermore, given that the present analysis is designed to resolve relationships within Diplodocidae, and that AMNH 5765 is highly incomplete (see above), the more basal position compared to the other two Camarasaurus OTUs should not be considered significant. The present result can thus be regarded as corroborating the referral of the holotype material of Amphicoelias latus to Camarasaurus by Osborn & Mook (1921).

Apatosaurus ajax YPM 1860

As type specimen of the type species of Apatosaurus, YPM 1860 has special taxonomic importance. It is herein recovered in two conflicting positions: on the same tree branch as Apatosaurus louisae CM 3018 (ew), or as sister-taxon to the specimen BYU 1252-18531 (iw). Four ambiguous autapomorphies are found for YPM 1860 (52-1; 81-1, 87-0, 292-1; Table S112).

Constrained searches forcing the specimen in the conflicting positions yielded a length increase of four steps, or 0.2% in the equally weighted analysis, and 0.05306 steps or 0.03% in the case of implied weighting. The position recovered by the equally weighted analysis, where Apatosaurus ajax YPM 1860 forms the sister-taxon to a clade with the holotype of A. louisae, is thus better supported than a close relationship with BYU 1252-18531.

Mean pairwise dissimilarity rates corroborate the close relationship of Apatosaurus ajax and A. louisae. The value calculated for an inclusion of A. ajax in Brontosaurus is higher (0.2187) than the one for the clade Apatosaurus (0.1835). An even greater value is found for an inclusion in the species Brontosaurus parvus (0.2406), which is the clade where YPM 1860 was recovered in the analysis under implied weights.

Apatosaurus grandis YPM 1901

The specimen YPM 1901 has long been known not to belong to Apatosaurus, but to typify its own species within Camarasaurus (Marsh, 1878; Osborn & Mook, 1921; McIntosh et al., 1996a; McIntosh et al., 1996b; Ikejiri, 2004). It is herein consistently recovered as sister taxon to the genus-level OTU Camarasaurus, thereby confirming this identification. Apatosaurus grandis is thus referred to Camarasaurus, as Camarasaurus grandis, with the type specimen being YPM 1901.

Amphicoelias fragillimus AMNH 5777

This specimen was the only putative diplodocid holotype specimen not included into the present analysis. Given that it was lost shortly after publication (Carpenter, 2006), and that no other material has yet been referred to the same species, it seems unwise to speculate about its phylogenetic position solely based on the single drawing and inadequate description of this extremely fragmentary specimen. Amphicoelias fragillimus is thus herein considered a nomen dubium.

‘Atlantosaurus’ immanis YPM 1840

Generally considered synonymous to Apatosaurus ajax (McIntosh, 1995; Upchurch, Tomida & Barrett, 2004), the present analyses always find this specimen in a group together with NSMT-PV 20375, as most basal branch within Apatosaurinae. Interestingly, NSMT-PV 20375 was also identified as Apatosaurus ajax in its initial description (Upchurch, Tomida & Barrett, 2004).

‘Atlantosaurus’ immanis YPM 1840 is unambiguously classified as an apatosaurine due to the presence of pcdl and podl in mid- and posterior cervical vertebrae that do not meet anteriorly (186-1), cervical ribs that project well beneath centrum (216-1), and which bear a bump-like anterior process (220-1). However, no recovered autapomorphy for the specimen can be considered valid according to the guidelines established above (Table S113). Also, the sister group arrangement with NSMT-PV 20375 does not yield any synapomorphy not shared with any other apatosaur specimen. The absence of autapomorphies suggests that YPM 1840 has to be treated as undiagnostic, and classified as an indeterminate apatosaurine. ‘Atlantosaurus’ immanis is thus a nomen dubium.

McIntosh (1995) proposed that YPM 1840 and YPM 1861 actually belong to the same individual, and constrained searches were performed to test this hypothesis. Forcing YPM 1840 to group with YPM 1861 in the equally weighted analysis yielded minimal tree lengths of one step more than the most parsimonious trees, or a relative length increase of 0.05%. A constrained search with implied weighting resulted in a minimal tree length of 194.57483, which corresponds to a relative length increase of 0.18%, which is relatively high compared to other differences. Given that no synapomorphies are found to unite these two specimens, it seems more prudent to interpret them as belonging to two different individuals.

Diplodocus longus YPM 1920

YPM 1920 is the type specimen of D. longus, the type species of Diplodocus. Therefore, its anatomical distinctiveness is of particular taxonomic importance. However, results obtained herein raise considerable doubts about the diagnosability of this specimen.

When added to the reduced consensus trees, Diplodocus longus YPM 1920 consistently groups with the other included specimens of Diplodocus in both types of analyses (equal and implied weighting), resulting in a polytomy in the strict consensus, comprising the two specimens of D. carnegii and the D. hallorum clade. Thus, D. longus YPM 1920 can equally parsimoniously occupy a position between the two specimens of D. carnegii, or a position closer to D. hallorum. No autapomorphy of D. longus can be recovered from the main trees, indicating that it is not diagnosable on its own. YPM 1920 shares a single trait with AMNH 223, which is otherwise unique (338-1, the presence of a transverse ridge posterior to the prezygapophyseal facets in mid-caudal vertebrae). However, given that no tree recovers this as a synapomorphy for a clade uniting YPM 1920 and AMNH 223 to the exclusion of all other Diplodocus specimens, this feature should be interpreted as individual variation. A constrained search uniting these two specimens yielded an equally weighted tree length of 1978 steps, and an implied weights tree length of 194.38745 steps. Relative length increase thus amounts to 0.1% and 0.09%, respectively.

Although it is confidently identifiable as belonging to the same genus as the type specimens of D. carnegii and ‘Seismosaurus’ hallorum (see below), YPM 1920 does not appear to be diagnosable to the species level. Therefore, Diplodocus longus is considered to be nomen dubium herein. This creates the taxonomically unsatisfying situation that the otherwise well-known genus Diplodocus is typified by a dubious species. A case to ICZN is therefore being prepared, suggesting the suppression of D. longus as type species of Diplodocus, and its replacement by D. carnegii. D. carnegii is typified by the nearly complete, and articulated type specimen CM 84, which includes a complete vertebral column from the second cervical to the twelfth caudal vertebra, as well as articulated fore- and hindlimb material. A more detailed argumentation for such a substitution will be developed in the case. Pending a decision on the ICZN case, it is hereby suggested to use D. carnegii as the type species of Diplodocus. YPM 1920 is considered not diagnostic at species level, and Diplodocus longus has therefore to be regarded a nomen dubium. A similar case was announced by Upchurch & Martin (2003) for the substitution of Cetiosaurus medius by C. oxoniensis as type species, and submitted in 2009 (Upchurch, Martin & Taylor, 2009). Their reasoning leading to the case was almost identical to the one presented herein. The Cetiosaurus case was accepted by the ICZN in 2014 (ICZN, 2014).

Brontosaurus excelsus YPM 1980

Differences between YPM 1980 and Apatosaurus ajax YPM 1860 are usually considered insufficient to justify generic distinction (Riggs, 1903), leading to the treatment of Brontosaurus as a junior synonym of Apatosaurus (Riggs, 1903; Gilmore, 1936; McIntosh, 1995; Upchurch, Barrett & Dodson, 2004; Upchurch, Tomida & Barrett, 2004). The specimen YPM 1980 is the genoholotype of Brontosaurus. In all principal trees, it forms a clade with the type specimen of the second proposed species of Brontosaurus, B. amplus YPM 1981.

One ambiguous autapomorphy is found to be reliable for B. excelsus YPM 1980 (355-0; Table S114). This low number is probably due to the incomplete scoring of its sister-taxon Brontosaurus amplus YPM 1981, of which only a very short description and very few figures are published (see above). Five changes separate B. excelsus YPM 1980 from B. amplus YPM 1981, which is not considered sufficient for specific separation.

Apatosaurus laticollis YPM 1861

Based on a single, fragmentary, mid- to posterior cervical vertebra, this specimen is one of the least complete included in the present analysis. McIntosh (1995) suggested it to come from the same individual as YPM 1840, but evidence from two partial femora suggest that more than one individual was present in the quarry (McIntosh, 1995). The fact that no tree of the present analysis shows a sister taxon arrangement of YPM 1840 and 1861 casts further doubts on the proposal of McIntosh (1995). A. laticollis YPM 1861 is herein consistently found as most closely related to A. louisae CM 3018 and CM 3378. If true, and if YPM 1861 is considered diagnosable, this would indicate that the two species would be synonymous, and that A. laticollis would therefore have priority over A. louisae.

One ambiguous autapomorphy is found for Apatosaurus laticollis YPM 1861, which is unique within Apatosaurinae (177-1; Table S115). However, because only two traits distinguish A. laticollis from A. louisae, specific separation cannot be justified, and the two traits are more cautiously interpreted as individual variation, at least in the present species. Of the two shared synapomorphies for A. louisae type + CM 3378 + YPM 1861, only one could be scored in YPM 1861. Given that both traits are shared, the presence of only one of these characters cannot be considered enough to diagnose a species. Therefore, A. laticollis YPM 1861 is not sufficiently diagnostic for the species it forms together with CM 3018 and CM 3378, and A. laticollis should be considered a nomen dubium.

As discussed above, forcing Apatosaurus laticollis YPM 1861 into a close relationship with YPM 1840 (following McIntosh, 1995) yielded rather improbable results. In both analyses, YPM 1861 is pulled into the clade where YPM 1840 was found in the unconstrained searches. The fact that YPM 1861 readily changes position further indicates that it is not diagnosable to species level. Pending further detailed studies of the specimens YPM 1840 and 1861, YPM 1861 is herein referred to A. louisae.

Brontosaurus amplus YPM 1981

Brontosaurus amplus YPM 1981 is often considered synonymous with Brontosaurus excelsus (McIntosh, 1995; Upchurch, Tomida & Barrett, 2004), although most studies have stated that further studies are needed in order to assess the taxonomic affinities of B. amplus. The present study does not allow a much more detailed assessment, mostly because of limited personal observations of the specimen due to time constraints during the collection visit at YPM. However, some conclusions can be drawn from the trees recovered, which all found it as sister-taxon to B. excelsus YPM 1980.

One unambiguous (376-1) and three ambiguous ‘autapomorphies’ (354-0, 374-0, 375-0) were recovered for YPM 1981 (Table S116). However, the five changes separating YPM 1981 from B. excelsus do not allow specific separation (see above). Although no apatosaurine synapomorphies can be positively identified in YPM 1981 to date, the transverse ridge on the third sacral rib (288-1) and the proximodistally thick astragalus (451-1) suggest that an identification of YPM 1981 as Brontosaurus can be stated with some confidence. Based on the numerical approach, and on a low pairwise dissimilarity value between the type specimens of B. excelsus and B. amplus (0.1429), B. amplus is herein considered synonymous to B. excelsus, corroborating earlier studies (McIntosh, 1995; Upchurch, Tomida & Barrett, 2004).

Diplodocus lacustris YPM 1922

Marsh (1884) established this species based on the presence of more slender teeth in YPM 1922 compared to those of USNM 2672. This appears to be true (Table S15); however, the proportions of the teeth in both specimens are within the minimum and maximum values of the teeth of the skull CM 11161, which was only found after Marsh’s death (Holland, 1924). The specimen YPM 1922 was found to be the least stable in both main analyses, being mainly responsible for the large polytomy within Diplodocoidea in the complete strict consensus tree.

Given that no characters are known that would allow an identification of diplodocid teeth at the species level, and that both the premaxilla and maxilla referred to this specimen are not diplodocid (see above), the teeth of YPM 1922 can only be identified as Diplodocidae indet. D. lacustris should thus be regarded as a nomen dubium. It is therefore also not available as type specimen for the substitution of the suppressed D. longus YPM 1920. The choice of D. carnegii and CM 84 to typify Diplodocus is thus further supported.

Barosaurus lentus YPM 429

The genoholotype specimen of Barosaurus is relatively complete and well described and figured (Lull, 1919). It was consistently recovered nested within a clade of specimens generally referred to the same species (AMNH 6341 and CM 11984), and does not show any feature which would distinguish it from these two referred specimens and qualify as species autapomorphies (Table S117). YPM 429 is the only type specimen within this clade.

Barosaurus affinis YPM 419

The species B. affinis was initially named in a short note, only stating that it was smaller than the type species, B. lentus (Marsh, 1899). The material (one complete and one incomplete metatarsal) was described, figured and measurements were given by Lull (1919), who misidentified them as metacarpals, though (McIntosh, 2005). Whereas generally treated as junior synonym or B. lentus (McIntosh, 2005; Remes, 2006), our analysis recovered it consistently as sister taxon to Cetiosauriscus stewarti. No autapomorphies were found for the specimen in any analysis.

Constrained searches forcing Barosaurus affinis in a close relationship with B. lentus yielded trees of a length of 1977 (ew) and 194.31603 steps (iw), corresponding to an increase of 0.05% under both weighting strategies. This minimal tree length increase for such an important jump from a non-neosauropod eusauropod into Diplodocidae as found under implied weighting indicates that YPM 419 is not diagnosable at a low taxonomic level. Given that the presence of a distolateral projection on metatarsal I (as occurs in YPM 419) has been shown to have a wider distribution than just Diplodocidae (Nair & Salisbury, 2012), YPM 419 must be considered an indeterminate eusauropod, and B. affinis a nomen dubium.

Diplodocus carnegii CM 84

The holotype of D. carnegii cannot be confidently distinguished from CM 94, with which it forms a clade (Table S118). All recovered synapomorphies uniting CM 84 and CM 94 are definitively present in both, and therefore no concerns can be raised about the diagnosibility of CM 84 or the validity of D. carnegii.

Elosaurus parvus CM 566

The specimen CM 566 is a very juvenile individual, as exemplified by its small size and the absence of neurocentral fusion (Peterson & Gilmore, 1902; McIntosh, 1995; Upchurch, Tomida & Barrett, 2004; Schwarz et al., 2007). Until recently, CM 566 was generally referred to Brontosaurus excelsus, together with the adult specimen UW 15556, with which it was found (Gilmore, 1936; McIntosh, 1995). By means of a specimen-based phylogenetic analysis, Upchurch, Tomida & Barrett (2004) showed that specific separation of CM 566 and UW 15556 from other apatosaurine species is justifiable. Recovered autapomorphies for the species were also shown in the juvenile specimen CM 566, leading Upchurch, Tomida & Barrett (2004) to propose the new combination Apatosaurus parvus. The present analysis also consistently recovers CM 566 close to UW 15556, although the amount of differences between CM 566 and UW 15556 + BYU 1252-18531 would actually allow specific separation. However, as outlined above, several of the ‘autapomorphies’ found in Elosaurus parvus CM 566 (135-0, 227-0, 252-0, 386-0, 437-1; Table S119) are probably ontogenetically variable features. Also, the ‘synapomorphies’ of UW 15556 + BYU 1252-18531 mostly describe the development of cervical and dorsal lamination, which has already been reported to change throughout ontogeny (Schwarz et al., 2007). Therefore, and because juvenile specimens tend to be recovered more basal to their true phylogenetic position (see e.g., Carballido & Sander, 2014), a referral of UW 15556 to the same species as CM 566 appears most parsimonious.

Constrained searches uniting the specimens CM 566 and UW 15556 resulted in equally weighted trees four steps (0.2%) longer than the MPT, whereas implied weighting of 0.16089 steps or an increase of 0.08%. The relatively high increase in the equally weighted analysis might indicate that also BYU 1252-18531 (the sister-taxon to UW 15556) should be included in the same species as CM 566 and UW 15556.

‘Gigantosaurus’ africanus various specimen numbers

The holotype specimen of ‘Gigantosaurus’ africanus consists of several bones excavated in the first expedition to Tendaguru, Tanzania, now housed at SMNS. More elements from the same individual were found later and brought to the MB.R. (Remes, 2006). The species has a complex taxonomic history (see above). After a thorough redescription and study of all preserved material, Remes (2006) re-established it as the separate genus Tornieria, in the combination Tornieria africana, adapting the latinized species name to the female genus. Its generic distinction from Barosaurus has also been demonstrated using phylogenetic analyses (Remes, 2006; Whitlock, 2011a; Mannion et al., 2012). The current study confirms this separation. Skeleton A, of which the holotype material is a part, consistently clusters with a second specimen referred to the same species by Remes (2006), skeleton k, also from Tendaguru. No valid autapomorphies, which would distinguish skeleton A from skeleton k, are found in the type specimen (Table S120). Both specimens together form a relatively basal clade within Diplodocinae. Four shared synapomorphies unite the two specimens (Table S83), but all of them are shared with other diplodocine specimens. A more detailed assessment of species autapomorphies will follow below.

Apatosaurus louisae CM 3018

The type specimen of A. louisae is the most complete type specimen of the entire clade of Apatosaurinae. It is also one of few diplodocid holotypes which has been adequately described and figured (Gilmore, 1936). CM 3018 is thus probably the best known and most used reference specimen for Apatosaurus, even though it is not its genoholotype. In the recovered main trees, it consistently groups with A. laticollis YPM 1861, CM 3378, and LACM 52844.

Even though it is so complete, only one ambiguous ‘autapomorphy’ was found for CM 3018 (311-1; Table S121). This indicates that the other specimens grouping with CM 3018 (i.e., Apatosaurus ‘laticollis’ YPM 1861 and CM 3378) belong to the same species. Because A. laticollis is herein considered a nomen dubium, the only available species name for this group is A. louisae, of which CM 3018 is the holotype (Holland, 1915a). The specimen CM 3018 shows all nine ‘synapomorphies’ found for the clade with CM 3018, CM 3378, YPM 1861, and LACM 52844 (see above). Of these, five qualify as valid autapomorphies for the species, not shared with any other apatosaurine specimen (see updated diagnosis below). Following our numerical approaches, generic separation from A. ajax is not justified, corroborating previous referrals of CM 3018 to Apatosaurus, as A. louisae.

“Apatosaurus” minimus AMNH 675

“Apatosaurus” minimus was described by Mook (1917), based on the sacrum and pelvic girdle of AMNH 675. The specimen has generally been considered as having been misidentified, and its diplodocoid affinities rejected (McIntosh, 1995; Upchurch, Tomida & Barrett, 2004). Whereas the pubis morphology strongly resembles Camarasaurus, the presence of six sacral vertebrae and widely splayed preacetabular lobes of the ilium are generally considered to be titanosauriform characteristics (McIntosh, 1990a; Upchurch, Barrett & Dodson, 2004; Upchurch, Tomida & Barrett, 2004). Due to its incompleteness, the true identity of AMNH 675 still remains dubious. Other than confirming the non-flagellicaudatan (and probably non-diplodocoid) affinities of AMNH 675, the present study does not help much in resolving this issue. Whereas the equally weighted trees recovered AMNH 675 as one of the four most unstable taxa (thus deleted from the pruned consensus), implied weighting resolves AMNH 675 as a somphospondylian titanosauriform, based on the two characteristics mentioned above. The three autapomorphies found for this specimen (288-1, 424-0, 425-1; Table S122) indicate that AMNH 675 probably shows a unique combination of features. Addition of AMNH 675 to the equally weighted reduced consensus tree results in a polytomy with Cetiosauriscus stewarti, Barosaurus affinis YPM 419, Haplocanthosaurus priscus, ‘Apatosaurus’ grandis YPM 1901, ‘Amphicoelias’ latus AMNH 5765, Camarasaurus, Turiasaurus riodevensis, Losillasaurus giganteus, SMA 0009 + more derived Brachiosauridae, Rebbachisauridae, and Flagellicaudata.

Forcing Apatosaurus minimus AMNH 675 into a titanosauriform position in the equally weighted analysis results in a tree one step longer than the most parsimonious tree. The same tree length was also found when imposing apatosaurine affinities, and results in a sister-taxon arrangement of A. minimus and A. ajax YPM 1860. A single ‘synapomorphy’ is found for this clade, which is the absence of a lateral fossa on the ischial shaft (422-0). However, this character is also present in Diplodocinae, a couple of Apatosaurinae, and Macronaria. Camarasaurid affinities of AMNH 675 are more probable, given that a forcing into this group yields the same tree length as the equally weighted most parsimonious trees (1,976 steps). Furthermore, the presence of six sacral vertebrae has already been reported in camarasaurids (AMNH 690, BYU 17465, GMNH-PV 101; Tidwell, Stadtman & Shaw, 2005) and was interpreted as an ontogenetically variable feature. Tree length of the implied weight trees increases to 194.43407 steps, or by a percentage of 0.11%, when restricting AMNH 675 to Apatosaurinae (where it creates a polytomy of all apatosaurine specimens recovered otherwise as more derived than Amphicoelias altus), and to 194.42454 (0.11%) when forcing it into Camarasauridae (where it creates a polytomy with Lourinhasaurus alenquerensis, and the Camarasaurus + Turiasauria clade). Camarasaurid or titanosauriform affinities are thus the most probable for AMNH 675, but more detailed studies of those clades are needed in order to identify AMNH 675 rigorously.

“Diplodocus” hayi HMNS 175

Described by Holland (1924) as “Diplodocus” hayi, HMNS 175 (initially CM 662) was often thought not to belong to Diplodocus (e.g., McIntosh, 1990b; Foster, 1998; Harris, 2006c), due to its relatively robust forelimbs and the widely diverging basipterygoid processes—both traits that are generally interpreted to diagnose apatosaurines (Berman & McIntosh, 1978; McIntosh, 1990a; Upchurch, Barrett & Dodson, 2004). The specimen HMNS 175 is one of the most complete known diplodocines, but has never been completely described. It preserves cranial material, cervical, dorsal, sacral, and caudal vertebrae, as well as a nearly complete forelimb and hindlimb (McIntosh, 1981; E Tschopp, pers. obs., 2010). The current analysis supports generic separation of HMNS 175 from Diplodocus, as it is consistently recovered in a clade more basal to Diplodocus, together with the specimens AMNH 969 and SMA 0011. The species is therefore herein referred to the new genus Galeamopus, of which HMNS 175 is the genoholotype specimen.

No autapomorphies were found for HMNS 175 (Table S123), but this is because of the incomplete preservation of ML 418, which was recovered as sister-taxon to HMNS 175. Forcing Galeamopus hayi HMNS 175 to group with the classical Diplodocus specimens, equally weighted analysis recovers shortest trees of 1984 steps, a length increase of eight steps or 0.4% compared to the unconstrained most parsimonious trees. Applying implied weights, tree length counts 194.67016 steps, corresponding to a relative increase of 0.23%. In both constrained analyses, G. hayi was not found nested within, but as basal-most member of a clade uniting it with the specimens referred to Diplodocus. A generic separation from Diplodocus is thus well supported.

‘Apatosaurus’ alenquerensis MIGM various numbers (lectotype)

‘Apatosaurus’ alenquerensis has a complicated taxonomic history. After being referred to Camarasaurus (McIntosh, 1990b), Dantas et al. (1998) erected the new genus Lourinhasaurus for a number of specimens thought to belong to the same species. No specific type specimen was attributed to the name (only a skeleton was mentioned without specimen number; Dantas et al., 1998), until Antunes & Mateus (2003) established the first specimen found at Moinho do Carmo, Alenquer, Lourinhã, as the lectotype. In the meantime, the specimen on which Dantas et al. (1998) made most observations of differences between Lourinhasaurus and Camarasaurus was redescribed and referred to a new species and genus, Dinheirosaurus lourinhanensis (Bonaparte & Mateus, 1999). Even so, Lourinhasaurus remained accepted, and its generic separation subsequently justified by means of phylogenetic analyses, which did not recover the lectotype specimen in a position close to Camarasaurus or Apatosaurus (e.g., Upchurch, Barrett & Dodson, 2004; Royo-Torres & Upchurch, 2012).

Six ambiguous autapomorphies are found to diagnose Lourinhasaurus (304-1, 306-1, 370-1, 393-1, 424-0, 426-1; Table S124). The fact that Lourinhasaurus was consistently found on a single branch under implied weights indicates that it also exhibits a unique combination of traits. The lectotype specimen is therefore considered diagnostic, and Lourinhasaurus alenquerensis is accepted as valid. The recovered position within Camarasauridae agrees with the latest reassessment of the osteology of Lourinhasaurus (Mocho, Royo-Torres & Ortega, 2014).

Cetiosauriscus stewarti NHMUK R3078

The phylogenetic position of Cetiosauriscus stewarti has been debated (Charig, 1980; McIntosh, 1990b; Heathcote & Upchurch, 2003; Upchurch, Barrett & Dodson, 2004; Rauhut et al., 2005). Diplodocid affinities were proposed by several authors (Charig, 1980; McIntosh, 1990b; Upchurch, Barrett & Dodson, 2004), mostly based on a second specimen containing a whip-lash tail (NHMUK R1967), which has no overlapping bones with the holotype (Heathcote & Upchurch, 2003; Upchurch, Barrett & Dodson, 2004). Diplodocid affinities of the holotype specimen are thus questionable, and consequently, a closer relationship to Mamenchisaurus or Omeisaurus was found by Heathcote & Upchurch (2003) and Rauhut et al. (2005). The current analysis recovers NHMUK R3078 in two different positions, depending on the weighting strategy applied. Equal weighting yields diplodocimorph affinities, more derived than Rebbachisauridae, whereas implied weighting recovers NHMUK R3078 as a non-neosauropod eusauropod, close to Mamenchisaurus or Omeisaurus, as proposed by Heathcote & Upchurch (2003).

No autapomorphies of Cetiosauriscus stewarti were found by any analysis, probably due to the sister relationship with Barosaurus affinis YPM 419. The incompleteness of YPM 419 inhibited the identification of autapomorphies in its sister taxon Cetiosauriscus, because there is little anatomical overlap between the two specimens. The fact that the clade of C. stewarti + B. affinis was found as a separate branch in all trees indicates that NHMUK R3078 is diagnosable, and Cetiosauriscus stewarti thus valid.

A forced sister arrangement with Omeisaurus + Mamenchisaurus under equally weights produced a tree length of 1980 steps or a length increase of 0.2%. In this case, Cetiosauriscus stewarti + Barosaurus affinis were found as sister-taxon to Omeisaurus. Imposing dicraeosaurid or rebbachisaurid affinities under implied weights results in tree lengths of 194.81613 or 195.15186, corresponding to an increase of 0.31% or 0.48%, respectively. Consequently, changing the position from diplodocoid to non-neosauropod eusauropod in the equally weighted tree (in particular close to Omeisaurus) is easier than imposing a diplodocoid position of Cetiosauriscus close to where it was found under equal weights in the implied weights analysis. C. stewarti is thus herein interpreted as non-diplodocoid eusauropod, possibly closely related to Omeisaurus, as already proposed by Heathcote & Upchurch (2003).

Supersaurus vivianae BYU 12962

The holotype specimen of Supersaurus vivianae is restricted to a scapula (Jensen, 1985), but other elements from the same quarry most probably belong to the same individual (Curtice & Stadtman, 2001; Lovelace, Hartman & Wahl, 2007). A scapula is not preserved in the second specimen referred to Supersaurus vivianae by Lovelace, Hartman & Wahl (2007; WDC DMJ-021), which inhibited the recognition of autapomorphies of the scapula in our analyses. However, the fact that both referred specimens consistently group together in all trees indicates that identification of additional elements as belonging to the same individual as the type specimen (Curtice & Stadtman, 2001; Lovelace, Hartman & Wahl, 2007) was likely correct. Therefore, even though the holotype of S. vivianae might not be diagnostic, further material representing the holotypic individual certainly is.

No valid autapomorphies distinguish the type individual from the second specimen, WDC DMJ-021 (Table S125), indicating that they belong to the same species. Of the eight traits uniting the two specimens (Table S79), only one can be considered a valid autapomorphy for the species (231-0), because the other also occur in other diplodocine specimens.

Dystylosaurus edwini BYU 4503

The holotype specimen of Dystylosaurus edwini was previously proposed to belong to the same individual as the Supersaurus vivianae holotype scapula (Curtice & Stadtman, 2001), a view supported by Lovelace, Hartman & Wahl (2007), as well as by the preliminary analyses of the present study (see above). Therefore, Dystylosaurus edwini is herein considered a junior synonym of Supersaurus vivianae. Its type specimen, BYU 4503, was therefore not included in the final analysis as separate OTU. However, information from this specimen was incorporated into the OTU called Supersaurus vivianae BYU+.

‘Seismosaurus halli’ NMMNH 3690

Gillette (1991) named this new genus based on the specimen NMMNH 3690, and later changed to species name to hallorum, in order to correct it for wrongly applied latin grammar (Gillette, 1994). Seismosaurus was later synonymized with Diplodocus (Lucas et al., 2006; Lovelace, Hartman & Wahl, 2007), with uncertainties as to whether it should be retained as separate species or regarded synonymous to Diplodocus longus (Lovelace, Hartman & Wahl, 2007). The latter statement was most probably based on previous identifications of the more complete specimens AMNH 223 and USNM 10865 as Diplodocus longus (Osborn, 1899; Gilmore, 1932), which was herein showed to be erroneous, or at least questionable. ‘Seismosaurus’ hallorum NMMNH 3690 is consistently recovered in a group with AMNH 223, USNM 10865, and DMNS 1494, which together have been shown to constitute a distinct species herein.

NMMNH 3690 is characterized by three ambiguous “autapomorphies” (240-1, 355-1, 415-1; Table S126). However, even though all three would qualify as species autapomorphies, both the apomorph count and the pairwise dissimilarity argue against specific distinction of NMMNH 3690 and its sister clade. Showing four of the five apomorphic traits of the group, ‘Seismosaurus’ hallorum NMMNH 3690 can be considered diagnostic. Because it is the only type specimen in this cluster, and since the number of changes between this cluster and close phylogenetic relatives does not allow generic separation (see above), Diplodocus hallorum is the only valid, available name for this taxon.

Dyslocosaurus polyonychius AC 663

Based on very fragmentary appendicular material, assessment of the phylogenetic position of D. polyonychius is difficult. Although initially described as diplodocid (McIntosh, Coombs & Russell, 1992), the high number of five probable pedal unguals resembles basal sauropods, because the loss of pedal phalanges and unguals is usually considered typical for Eusauropoda and more derived forms (Wilson, 2002; Upchurch, Barrett & Dodson, 2004). However, almost no complete and articulated pes is known from any diplodocid, and of the included specimens, only a few preserve pedal material for direct comparison. A positive confirmation of the absence of vestigial phalanges or unguals is very difficult, if not impossible. The true distribution of the presence of five pedal unguals can thus not be assessed with the present analysis.

All analyses find Dyslocosaurus as the most basal dicraeosaurid. Four of the five synapomorphies that unite Dyslocosaurus and Dicraeosauridae, are only shared with one other dicraeosaurid taxon (431-1, 443-1, and 452-1 are shared with Dicraeosaurus; 477-1 is shared with Suuwassea; and 461-0 is shared with Dicraeosaurus and Suuwassea). None of these traits could be coded in Amargasaurus or Brachytrachelopan, and all of them are also present in some diplodocid taxa.

Three ambiguous autapomorphies are found for AC 663 when it is included in Dicraeosauridae (442-1, 456-1, 468-1; Table S127). Two of these autapomorphies are shared with apatosaurine specimens (442-1, 468-1), and all also occur in diplodocines. The fact that this specimen appears to unite apatosaurine, diplodocine, and dicraeosaurid traits indicates that AC 663—though highly incomplete—is diagnostic, and Dyslocosaurus polyonychius is therefore a valid taxon.

Forcing Dyslocosaurus into a position within Apatosaurinae produced shortest trees of a length of 1980 (ew) and 194.38399 (iw) steps, an increase of 0.2% and 0.09%, respectively. When imposing diplodocine affinities, tree lengths of 1978 and 194.26722 steps are recovered, corresponding to length increases of 0.1% and 0.03%. Diplodocine affinities are thus more parsimonious than referral to Apatosaurinae, but still less so than inclusion in Dicraeosauridae. Despite the presence of characters shared with both diplodocid clades, an identification of Dyslocosaurus as dicraeosaurid diplodocoid is better supported.

‘Apatosaurus’ yahnahpin Tate-001

Apatosaurus yahnahpin Tate-001 has been renamed Eobrontosaurus yahnahpin (Bakker, 1998), but it was never included in any phylogenetic analysis, and no detailed description has yet been published. Based on purportedly primitive features of the pectoral girdle and the cervical ribs, Bakker (1998) interpreted Eobrontosaurus as the basal-most apatosaurine. Upchurch, Barrett & Dodson (2004) stated that the specimen Tate-001 is practically indistinguishable from Camarasaurus, but personal comments of R. Wilhite (cited in Taylor, Wedel & Cifelli, 2011) and (Mannion et al., 2012) implied that the taxon might be a valid diplodocid. The present analysis confirms this: Tate-001 is consistently recovered as apatosaurine diplodocid, within the clade now interpreted to represent the genus Brontosaurus (see above).

Four ambiguous autapomorphies are considered valid for Tate-001 (245-0, 321-0, 394-0, 399-1; Table S128). All of them are unique within Apatosaurinae. Given that generic distinction from the other members of the clade is not warranted, Tate-001 is herein referred to Brontosaurus, constituting the type specimen of Brontosaurus yahnahpin.

“Dinheirosaurus” lourinhanensis ML 414

ML 414 was first described as Lourinhasaurus alenquerensis (Dantas et al., 1998), but a more detailed redescription showed that it belonged to a distinct genus within Diplodocidae, named Dinheirosaurus (Bonaparte & Mateus, 1999). Such a position was later confirmed by phylogenetic analyses and refined to Diplodocinae (Rauhut et al., 2005; Whitlock, 2011a; Mannion et al., 2012). The present analysis supports this assignment but recovered “Dinheirosaurus” in an even more derived position than Whitlock (2011a) or Mannion et al. (2012). Both analyses found “Dinheirosaurus” as closely related to Supersaurus, and more derived than Tornieria.

Three ambiguous autapomorphies are found for ML 414, and thus for “Dinheirosaurus” lourinhanensis (126-0, 230-1, 305-0; Table S129). As mentioned above, the eleven changes found between “Dinheirosaurus” and Supersaurus are not considered enough to justify generic separation, and also pairwise dissimilarity points to the existence of a single genus including the species “Dinheirosaurus” lourinhanensis and Supersaurus vivianae. Therefore, because Supersaurus was named first, “Dinheirosaurus” should be considered a junior synonym of Supersaurus. Supersaurus is thus the only diplodocid genus including two species from two different continents.

Losillasaurus giganteus MCNV Lo-5

Although the holotype of L. giganteus is restricted to an anterior caudal vertebrae, this material actually belongs to a more complete individual (Casanovas, Santafé & Sanz, 2001) and was included as such in the present analysis. The present study supports the inclusion of L. giganteus in Turiasauria, as found by most recent phylogenetic analyses (e.g., Barco, 2009; Carballido et al., 2012b; Royo-Torres & Upchurch, 2012).

Five ambiguous autapomorphies are found for Losillasaurus giganteus (126-0, 262-1, 269-0, 310-1, 387-0; Table S130). Despite the low number of autapomorphies, the numerical approach is not applied here, as non-diplodocid OTUs have not been sampled sufficiently to apply the same standards as established for Diplodocidae. Losillasaurus is thus considered herein as a valid, non-diplodocoid genus, probably a non-neosauropod eusauropod.

Suuwassea emilieae ANS 21122

Suuwassea emilieae was initially described as an indeterminate flagellicaudatan (Harris & Dodson, 2004). Although some subsequent studies suggested diplodocid affinities (Gallina & Apesteguía, 2005; Rauhut et al., 2005; Remes, 2006; Lovelace, Hartman & Wahl, 2007), the discovery of the dentary of the holotype specimen (Whitlock & Harris, 2010) resulted in identification as dicraeosaurid (Whitlock, 2011a; Mannion et al., 2012; Tschopp & Mateus, 2013b), as was already suggested by Salgado, Carvalho & Garrido (2006). Our analysis supports the latter assignment: Suuwassea emilieae ANS 21122 is consistently found as the basal-most dicraeosaurid sauropod.

Suuwassea emilieae ANS 21122 is herein diagnosed by 20 ambiguous autapomorphies (62-0, 72-1, 90-1, 100-1, 114-2, 156-1, 158-1, 166-1, 190-1, 218-1, 296-1, 309-1, 332-1, 346-0, 362-1, 380-2, 441-1, 445-1, 459-1, 467-0; Table S131). The high number of autapomorphies for Suuwassea emilieae reflects not only its diagnosability, but also the fact that the main dicraeosaurid OTUs included in our analysis were not studied in the same detail as the specimens forming the ingroup (thus including ANS 21122). Given that the majority of these autapomorphies are shared with some diplodocid specimens, difficulties encountered in determining its dicraeosaurid affinities are not surprising. However, forcing Suuwassea into an apatosaurine clade (as found by Lovelace, Hartman & Wahl, 2007) yields trees of 1991 or 195.50286 steps (relative length increases of 0.76% and 0.66%, respectively). Diplodocine relationships are found in shortest trees of 1990 and 196.0041 steps, corresponding to increases in tree length of 0.71% and 0.92%. Apatosaurine or diplodocine affinities are thus much less parsimonious than referral to Dicraeosauridae.

Australodocus bohetii MB.R.2455

Whereas the holotype only includes the single cervical vertebra MB.R.2455, a second, probably adjacent, cervical vertebrae most likely belongs to the same animal (MB.R.2454; Remes, 2007). Australodocus was first described as diplodocid (Remes, 2007), but later found to represent a titanosauriform (Whitlock, 2011a; Whitlock, 2011c; Mannion et al., 2012; Mannion et al., 2013). The present analyses consistently find diplodocine affinities for A. bohetii, although its incompleteness resulted in an a posteriori deletion of the OTU in the pruned and reduced consensus trees under equal weights. When calculating a pruned consensus tree including only the OTUs constituting the equally weighted reduced consensus tree plus Australodocus, the latter forms a polytomy with FMNH P25112, SMA 0087, and WDC-FS001A at the base of Diplodocinae. The incompleteness of the type individual complicates the recovery of a stable position for Australodocus.

Three ambiguous autapomorphies were recovered for Australodocus under implied weights, which would be unique within Diplodocinae (130-2, 171-1, 218-0; Table S132). One of these autapomorphies was found as a synapomorphy of Titanosauriformes in the same analysis (130-2). This indicates that the combination of traits is unique in Australodocus, which is thus regarded as valid.

Australodocus is found as sister-taxon to Supersaurus vivianae in the main implied weight trees. When forcing Supersaurus vivianae into a sister relationship with “Dinheirosaurus” lourinhanensis ML 414 under implied weighting (thus excluding Australodocus from such a close relationship as recovered), Australodocus is recovered at the base of Diplodocinae, where also the equally weighted analysis finds the genus. The latter constrained search produced shortest trees of 194.24954 (a 0.02% length increase). Forcing Australodocus into a sister-taxon relationship with Supersaurus vivianae under equal weighting, resulted in MPTs of 1977 steps, one step or 0.05% longer than the unconstrained, equally weighted MPTs. Titanosauriform affinities are less parsimonious according to our analysis: constrained searches produced tree lengths of 1979 (ew) or 194.39687 steps, a relative increase of 0.15% and 0.09%, respectively. According to these results, a basal position within Diplodocinae is the best supported. However, the low number of titanosauriform OTUs in the present study lowers the capability of the analysis to recover Australodocus as belonging to that taxon, such that convergences found with Diplodocinae tend to become more important. The fact that the Australodocus cervical centra have a somphospondylous internal structure (Whitlock, 2011c; P Mannion, pers. comm., 2013), which otherwise only occurs in titanosauriform sauropods, provides additional support for titanosauriform instead of diplodocine affinities. A position close to Supersaurus vivianae therefore appears the least supported of the ones discussed here. An exclusion of Australodocus from the Supersaurus clade is also supported by the relatively high mean pairwise dissimilarity values when comparing Australodocus with the two specimens of Supersaurus vivianae (0.2188 with the holotypic individual; 0.3571 with WDC DMJ-021), and the type specimen of Supersaurus lourinhanensis (0.6). Addition of titanosauriform specimens preserving cervical vertebrae would help to resolve this problem but is not the scope of this analysis.

Kaatedocus siberi SMA 0004

Kaatedocus siberi was initially described as a diplodocine less derived than Tornieria, Diplodocus, or Barosaurus (Tschopp & Mateus, 2013b). In the present analysis, Kaatedocus is consistently recovered in a more derived position, as sister taxon to Barosaurus lentus.

The type specimen SMA 0004 bears one ambiguous ‘autapomorphy’ (86-1, a transverse ridge on the basal tubera; Table S133). Because no ‘synapomorphy’ was found for the sister clade AMNH 7530 + SMA D16-3, only one change separates SMA 0004 from the latter. The presence of such a transverse ridge is thus better interpreted as individual variation. Six of the 19 ‘synapomorphies’ found for the entire group of Kaatedocus siberi qualify as species autapomorphies, not shared with other diplodocine specimens (27-0, 32-1, 178-1, 202-1, 211-1, and 212-1; Table S64).

Leinkupal laticauda MMCH-Pv 63-1

Leinkupal laticauda is the most recently described diplodocid species (Gallina et al., 2014), and the only diplodocid from South America and from the Cretaceous period. L. laticauda was initially found as the sister-taxon of Tornieria africana (Gallina et al., 2014). Herein, it consistently forms its own branch in a position more derived than Supersaurus but basal to Galeamopus. The reason for this conflict might be the limited osteological information included in our specimen-level cladistic analysis, due to the restriction of the OTU to the holotypic caudal vertebra MMCH-Pv 63-1 (see above).

One ambiguous autapomorphy was found for Leinkupal laticauda (314-0; Table S134), but this is because we did not include potential autapomorphic features proposed by Gallina et al. (2014) as character statements. Gallina et al. (2014) proposed four additional autapomorphies: (1) anterior caudal transverse processes that are at least as wide as the centrum, (2) anterior caudal transverse processes marked by strong dorsal and ventral bars, (3) anterior caudal cprl massive, and (4) anterior caudal postzygapophyses bear a distinct foramen dorsally at their base. While we agree that these features are unique at least within Diplodocinae, it will be important in future to define better the robusticity of the cprl and the dorsal and ventral bars of the transverse process. Adding these autapomorphies to the single trait recovered in our analysis, the sum of changes between Leinkupal and Galeamopus + mdD is raised to six, which at least allows for specific separation. However, as mentioned above, tree topology, as well as spatial and temporal isolation from all other diplodocines indicate that also generic separation is warranted.

Forcing Leinkupal into a sister-taxon relationship with Tornieria results in minimally increased tree lengths: 1977 steps under equal weights, and 194.44603 steps under implied weights. These values correspond to length increases of 0.05% and 0.12%, respectively. Given that some middle caudal vertebrae from the same quarry, and referred to Leinkupal by Gallina et al. (2014) have very similar morphology to middle caudal vertebrae referred to Tornieria by Remes (2006), but that neither were included in our analysis, it is not surprising that the species-level comparison in Gallina et al. (2014) recovered the two taxa as sister-groups. The position of Leinkupal in our analysis should therefore be regarded as provisional.

Taxonomic affinities and identification of diplodocid non-type specimens

The non-type specimens are listed alphabetically. For a summary of the species referrals see Table 5.

AMNH 223

Described as Diplodocus longus (Osborn, 1899), AMNH 223 readily became the mostly used reference specimen for this species (Hatcher, 1901; Gilmore, 1932). However, the present analysis does not recover AMNH 223 together with the holotype specimen YPM 1920, but as most basal OTU of a clade including the holotype of Seismosaurus hallorum.

Two ambiguous ‘autapomorphies’ are found for this specimen (359-1, 369-1; Table S135), which describe scapular morphology. The fact that only one of the other three specimens in the same clade preserves a scapula, and the low number of differences between AMNH 223 and the remaining triplet, indicates that these might represent individual variation, and that AMNH 223 is most parsimoniously identified as belonging to the same species, which would be Diplodocus hallorum.

AMNH 460

The specimen AMNH 460 has never been described, but was included in the specimen-level phylogenetic analysis of Upchurch, Tomida & Barrett (2004). In the latter, it has been identified as Apatosaurus ajax, which is not supported by the most parsimonious trees of the present analysis. In our analysis, AMNH 460 was consistently found on a single branch more derived than YPM 1840 + NSMT-PV 20375, but basal to Apatosaurus + Brontosaurus. Tree topology would imply that AMNH 460 represents a different taxon, but the fact that none of the recovered specimen ‘autapomorphies’ is unique within Apatosaurinae (Table S136) makes such an assignment questionable.

A constrained search forcing AMNH 460 into the Apatosaurus clade yielded trees of a length of 1978 or 194.53329 steps, corresponding to relative length increases of 0.1% or 0.16%. AMNH 460 continued to be found as a single slot, more basal to Apatosaurus ajax YPM 1860. Under equal weights, this constraint furthermore resulted in the inclusion of ‘Atlantosaurus’ immanis YPM 1840 in Apatosaurus, as basal-most member of the clade. YPM 1840, AMNH 460, and YPM 1860 were all found on single branches. On the other hand, implied weighting still recovered YPM 1840 with NSMT-PV 20375, but Amphicoelias altus was found as a diplodocine, closely related to Galeamopus. An imposed inclusion in Brontosaurus for AMNH 460 results in tree lengths of 1979 and 194.56056 steps, or relative increases of 0.15% and 0.18%. Under both weighting strategies, AMNH 460 was found as a single OTU on the basal-most branch within Brontosaurus. Forcing AMNH 460 in the clade NSMT-PV 20375 + ‘Atlantosaurus’ immanis YPM 1840 produces tree lengths of 1981 and 194.37514 steps, relative increases of 0.25% and 0.08%.

Mean pairwise dissimilarity values are also ambiguous. At the species level, most support exists for a referral of AMNH 460 to Brontosaurus excelsus (0.1667), followed by Apatosaurus ajax (0.1774) within Apatosaurinae or the basal-most potential new diplodocine species including SMA 0087 and WDC-FS001A (0.172). At the genus level, the lowest values within Apatosaurinae favor a referral of AMNH 460 to Brontosaurus (0.2), or the third genus (0.2019), whereas an inclusion in Apatosaurus is less supported (0.2263). The value for an inclusion in the new diplodocine genus and species remains the same (0.172). Given that all these results of constrained searches and mean pairwise dissimilarity values are all more or less equally supported, it seems most cautious to treat AMNH 460 as an indeterminate apatosaurine, following tree topology, and awaiting a detailed analysis of the specimen.

AMNH 969

This skull was generally considered to belong to Diplodocus (Holland, 1906; Holland, 1924; Berman & McIntosh, 1978), probably due to strong resemblances with the purported skulls of Diplodocus longus USNM 2672 and 2673. However, the latter two specimens cannot be confidently referred to the type species, as there is no overlap with the type specimen YPM 1920 (McIntosh & Carpenter, 1998). Furthermore, given the few differences in skull morphology between diplodocine and apatosaurine species, even less can be expected within Diplodocinae alone. Indeed, the present analysis recovers AMNH 969 consistently within the genus Galeamopus. Constrained searches support this assignment: a forced inclusion in Diplodocus yields shortest trees of 1980 or 194.37642 steps, a relative increase of 0.2% or 0.08%, respectively. The constrained consensus trees are very different from the unconstrained trees and are largely unresolved, which further supports a referral of AMNH 969 to Galeamopus.

One ambiguous ‘autapomorphy’ is found that distinguishes AMNH 969 from the other Galeamopus specimens (112-1; Table S137), but the sum of differences is not enough to justify erection of a distinct species. Thus, taking into account that there is evidence for the presence of two distinct species within Galeamopus (see above), but that AMNH 969 was found as the sister taxon of all other Galeamopus specimens, we cautiously refer the specimen to Galeamopus sp.

AMNH 6341

AMNH 6341 is the most complete specimen generally referred to Barosaurus lentus. Because it is completely prepared, and appears largely undeformed (in contrast to the type specimen YPM 429), AMNH 6341 has generally been used as reference specimen for the genus (see Whitlock, 2011a). Although it was found early after the discovery of the Carnegie Quarry at what was later to be named Dinosaur National Monument (McIntosh, 2005), it was only described recently by McIntosh (2005), and not in comprehensive detail.

In the present analysis, AMNH 6341 was consistently found as sister taxon to the holotype specimen of Barosaurus lentus, YPM 429. None of the recovered ‘autapomorphies’ of AMNH 6341 can be considered valid (Table S138). Our analysis thus confirms previous assignments of AMNH 6341 to Barosaurus lentus.

AMNH 7530

The specimen AMNH 7530 was never described but is labeled as Barosaurus sp. on display at AMNH. It is herein consistently recovered together with Kaatedocus siberi SMA 0004. No autapomorphies are found for the specimen, possibly due to the fragmentary preservation of the specimen with which it forms a dichotomy (the partial skull SMA D16-3). Differences between AMNH 7530 and SMA 0004 exist in the shape of the dorsal edge of the parietal (C62), in the orientation of the longest axes of the basal tubera (C87), and in the development of the pre-epipophyseal anterior spur (C167). However, the sum of recovered ‘autapomorphies’ between the specimens is too low to justify specific separation. The mentioned differences are thus interpreted as individual variation, contrary to the interpretation in Tschopp & Mateus (2013b), where the anterior spur of the pre-epipophysis was stated as autapomorphic for the species Kaatedocus siberi.

Forcing AMNH 7530 into a clade with the other sampled Barosaurus specimens increased tree length by 0.4% (ew) and 0.36% (iw), from 1976 and 194.21603 to 1984 and 194.91145 steps, respectively. Such an assignment is thus considerably less parsimonious than a referral to Kaatedocus siberi.

AMNH 7535

As for AMNH 7530, AMNH 7535 also was tentatively identified as Barosaurus in the AMNH data base, but never described. In contrast to the specimen AMNH 7530, here identified as Kaatedocus, AMNH 7535 consistently groups with other Barosaurus specimens in the present analysis.

Two autapomorphies were recovered for the specimen (50-0, 158-1; Table S139). As stated above, the sum of differences between AMNH 7535 and its sister clade CM 11984 + mdD would be too low to establish specific separation, but mean pairwise dissimilarity suggests otherwise. Pending a detailed study and description of AMNH 7535, this specimen is thus herein referred to Barosaurus sp.

BYU 1252-18531

This specimen is labeled as Apatosaurus excelsus on display at BYU. Under equal weighting, it was consistently recovered as closely related to the type specimen of Elosaurus parvus. A sister-taxon relationship with Apatosaurus ajax YPM 1860, as found under implied weights, has been shown above to be less parsimonious.

Four ambiguous autapomorphies of BYU 1252-18531 were found, none of which are shared with any other apatosaurine specimen (139-1, 184-2, 214-0, 371-0; Table S140). The eight changes between BYU 1252-18531 and its sister-taxon UW 15556 indicate that specific separation could be warranted. However, given the influence of potential ontogenetically variable characters on the recovery of autapomorphic features in this triplet, which includes the very juvenile holotype of “Elosaurus” parvus, UW 15556, and BYU 1252-18531, more detailed studies are needed to justify the erection of a unique species for BYU 1252-18531. We therefore provisionally refer BYU 1252-18531 to Brontosaurus parvus.

CM 94

This specimen was designated the paratype of Diplodocus carnegii (Hatcher, 1901). It complements knowledge of Diplodocus carnegii in crucial parts such as the mid-caudal vertebrae (thus allowing comparisons with the holotype specimen of D. longus YPM 1920), and appendicular elements. When pruning YPM 1920 from the complete consensus trees, CM 94 is consistently recovered as sister taxon to the holotype specimen of D. carnegii, CM 84.

One ‘autapomorphy’ was found to be reliable for the specimen CM 94 (366-1; Table S141). The sum of differences between CM 94 and CM 84 thus amounts to one (no valid ‘autapomorphies’ were found for CM 84). Referral to a single species, and thus an assignment of CM 94 to Diplodocus carnegii as paratype (Hatcher, 1901) is therefore justified.

CM 3378

The specimen CM 3378 was found together with the holotype of Apatosaurus louisae at Dinosaur National Monument and preserves the most complete vertebral column of any of the specimens included herein (McIntosh, 1981). Nonetheless, it has only been described and figured in parts (Holland, 1915b; Gilmore, 1936). It was included into the specimen-based phylogenetic analysis of Upchurch, Tomida & Barrett (2004), and was recovered as a specimen of Apatosaurus louisae. Because none of the recovered ‘autapomorphies’ for CM 3378 can be considered valid (Table S142), our analysis confirms this interpretation.

CM 3452

The specimen CM 3452 is one of very few preserving an almost complete skull in articulation with the first few cervical vertebrae. It was reported as a juvenile to subadult Diplodocus specimen (Holland, 1924; McIntosh & Berman, 1975; Whitlock, Wilson & Lamanna, 2010), but never described in detail. A referral to Diplodocus is questionable, because almost no overlapping material exists between CM 3452 and any type specimen of a Diplodocus species. Now that generic separation from Diplodocus is confirmed for Galeamopus hayi, the only Diplodocus type specimen preserving anterior cervical vertebrae is CM 84. With the inclusion herein of two specimens preserving articulated skulls and cervical vertebrae (SMA 0004 and 0011), affinities of CM 3452 can be assessed more accurately. The present analysis consistently recovers CM 3452 as the sister taxon of Kaatedocus siberi + Barosaurus lentus.

A single ‘autapomorphy’ was found as valid for CM 3452 (89-0; Table S143). Summed differences between CM 3452 and its sister clade amount to three, not justifying specific separation.

Forcing CM 3452 into Diplodocus, following earlier identifications, equal weighting finds shortest trees of 1977 steps, and implied weighting 194.27861 steps—constituting relative length increases of 0.05% and 0.03%, respectively. Imposed affinities with Kaatedocus yield trees with a length of 1977 and 194.23526 steps, corresponding to an increase in length of 0.05% and 0.01%. A forced inclusion into the Barosaurus clade results in length increases of 0.05% and 0.02%, to 1977 and 194.2542 steps, respectively.

Mean pairwise dissimilarity was impossible to calculate for many clades, due to the lack of anatomical overlap. However, the lowest value was retrieved for a grouping with Diplodocus carnegii (0.1594), followed by Diplodocus hallorum (0.1852). Also at the genus-level, Diplodocus is most similar to CM 3452 (0.1667), indicating that a referral to Diplodocus might be better supported morphologically than an inclusion in Kaatedocus (0.2049) or Barosaurus (0.2171). However, given that this is in conflict with the consistently recovered tree topology, we prefer to identify CM 3452 as Diplodocinae indet. pending a more detailed study of the specimen.

CM 11161

This skull-only specimen is generally referred to Diplodocus (Holland, 1915b; Holland, 1924; McIntosh & Berman, 1975; Berman & McIntosh, 1978; Whitlock, Wilson & Lamanna, 2010; Whitlock & Lamanna, 2012), and has been used in numerous publications as a model for feeding strategies or other ecological or behavioral studies concerning this genus (e.g., Haas, 1963; Barrett & Upchurch, 1994; Calvo, 1994; Upchurch & Barrett, 2000; Whitlock, 2011b; Young et al., 2012). However, because no overlap exists with any of the type specimens of Diplodocus species, referral of CM 11161 to that genus remains controversial. Given that no skull with articulated vertebrae included in our analysis can be confidently referred to Diplodocus (AMNH 969, SMA 0011, and USNM 2673 belong to Galeamopus, SMA 0004 belongs to Kaatedocus, and CM 3452 is an indeterminate diplodocid), only indirect evidence can be used for such an assignment. Indeed, our analysis was not able to resolve the position of CM 11161 due to this lack of sufficient anatomical overlap with other taxonomically relevant specimens.

One ambiguous ‘autapomorphy’ was found for CM 11161 (42-0; Table S144): the short posteroventral process of the jugal (42-0). However, this feature is not preserved in USNM 2672 (E Tschopp, pers. obs., 2011) and CM 11255 (Whitlock & Harris, 2010; but see above).

Constrained searches were performed, forcing CM 11161 to group with diplodocine taxa preserving articulated skull material. Imposed relationships with Galeamopus produced trees 0.05% and 0.04% longer than the most parsimonious trees, with lengths of 1977 and 194.28745 steps, respectively. A forced assignment to Kaatedocus yielded shortest trees of 1978 and 194.33526 steps, a relative increase in length of 0.1% and 0.06%. In all constrained searches, CM 11161 was found in a clade with USNM 2672. Given that these alternative assignments do not increase tree length considerably, a referral to Diplodocus—although still the most parsimonious interpretation—remains uncertain. Given that nearly complete specimens including articulated skulls, vertebrae from anterior cervical to distal caudal elements, as well as appendicular elements including manual and pedal material are known from Galeamopus, the latter genus appears more appropriate as representative of the diplodocine clade in phylogenetic analyses.

CM 11255

This skull was described and figured as a juvenile of Diplodocus by Holland (1924), McIntosh & Berman (1975) and Whitlock, Wilson & Lamanna (2010). In our analysis, it is consistently found as being within the clade comprising Kaatedocus + Barosaurus.

Four ambiguous ‘autapomorphies’ were recovered for CM 11255 (18-0, 77-0, 91-0, 111-0; Table S145). Two of them might be influenced by ontogeny: (1) the small antorbital fenestra compared to the orbit—indicating the presence of a proportionally larger orbit than in other diplodocines (18-0); and (2) the relatively round snout (111-0), compared to a more rectangular one in large diplodocid skulls. When calculating a pruned consensus tree only with the OTUs represented in the reduced consensus trees plus CM 11255, the latter specimen forms a polytomy with the specimens referred to Barosaurus lentus, and the clade Kaatedocus.

Constrained searches forcing CM 11255 into the genus Diplodocus yielded trees of 1977 (ew) and 194.26603 (iw) steps, a length increase of 0.05% and 0.03%, respectively. The insignificant increase in tree length to find CM 11255 within Diplodocus, coupled with the inability to find it within a defined genus-level clade in the most parsimonious trees, indicates that CM 11255 is best identified as Diplodocinae indet. at present.

The same conclusion follows from the mean pairwise dissimilarity values. Of the five species, with which direct morphological comparison is possible (Barosaurus lentus, Tornieria africana, Kaatedocus siberi, Galeamopus hayi, and the probable second species of Galeamopus), CM 11255 shows most smiliarity with the second species of Galeamopus (0.1952). Also at the genus level, a referral of CM 11255 to Galeamopus (0.2125) is better supported than an inclusion in the genera CM 11255 was found with in the phylogenetic trees (Kaatedocus, 0.2336; Barosaurus, 0.2222). This ambiguity, and the relatively high values for all of these groupings indicate that CM 11255 probably does not belong to any of the five species with which direct comparison is possible.

CM 11984

The specimen CM 11984 was partly described, and referred to Barosaurus lentus by McIntosh (2005), but remains largely unprepared. The present analysis finds CM 11984 in all most parsimonious trees as sister taxon to Barosaurus lentus YPM 429 + AMNH 6341.

‘Autapomorphies’ recovered for CM 11984 were all shared with other diplodocine specimens, and thus not considered reliable (Table S146). Although the four ‘synapomorphies’ found for its sister clade Barosaurus lentus YPM 429 + AMNH 6341 (Table S60) would thus not suffice to erect a second species within Barosaurus, mean pairwise dissimilarity indicates the presence of multiple species within the clade CM 11984 + (YPM 429 + AMNH 6341) (see above). Given that the ‘synapomorphies’ found for the triplet do not qualify as species autapomorphies (see above), whereas two ‘synapomorphic’ features of YPM 429 + AMNH 6341 would, we accept McIntosh’s (2005) referral of this specimen to Barosaurus, but not to the species Barosaurus lentus. Complete preparation and a detailed study of CM 11984 is needed to establish its exact taxonomic referral, and to see if an identification as Barosaurus lentus is warranted. CM 11984 is thus herein treated as Barosaurus sp.

DMNS 1494

Although undescribed, DMNS 1494 is often considered to be a specimen of Diplodocus longus (McIntosh, 1981; Gillette, 1991), probably based on similarities with AMNH 223, the specimen described as D. longus by Osborn (1899). Because the identification of AMNH 223 as D. longus was rejected by our analysis, the referral of DMNS 1494 to D. longus also appears questionable. In the present analysis DMNS 1494 is consistently found as the sister taxon of USNM 10865.

A single ambiguous ‘autapomorphy’ was found for the specimen (422-1; Table S147), but only in the analyses recovering FMNH P25112 as an apatosaurine. Because this is the only valid difference between DMNS 1494 and USNM 10865 (Table S67), the two specimens can be confidently referred to Diplodocus hallorum.

FMNH P25112

This specimen is one of the few non-type specimens that has been adequately described (Riggs, 1903). Riggs (1903) referred FMNH P25112 to Apatosaurus excelsus (herein reinterpreted as Brontosaurus excelsus), an identification that was accepted by Gilmore (1936). Upchurch, Tomida & Barrett (2004) recovered FMNH P25112 as a unique clade, proposing that it might belong to its own species within Apatosaurus. In our analysis, FMNH P25112 changes between a position as basal-most diplodocine (under equal weights) and as Brontosaurus (under implied weights). In the morphospace comparing principal coordinates 1 and 2, FMNH P25112 clusters with the apatosaurine specimens (Fig. 112). Three ‘autapomorphies’ for the specimen were found when placed within Diplodocinae (350-1, 427-2, 430-2), whereas four are considered valid when placed within Apatosaurinae (309-1, 324-1, 416-0, 430-2; Table S148).

Forcing FMNH P25112 into Apatosaurinae under equal weights, tree lengths increase by 0.05% to 1977 steps. Imposing an inclusion in Brontosaurus, the shortest trees measure 1979 steps, an increase of 0.15%. A grouping with Diplodocinae under implied weights (as proposed by the equally weighted analysis) increases tree lengths by 0.03%, to 194.26889 steps.

Mean pairwise dissimilarity values were calculated for referrals of FMNH P25112 to any diplodocid species and genus. At the species level, attribution of FMNH P25112 within Apatosaurinae to Brontosaurus yahnahpin is best supported (0.1528), followed by the new, basal-most species (0.1848). At the generic level, referral to the third genus is most probable (0.1848), followed by an inclusion in Brontosaurus (0.2061). Within Diplodocinae, the two most probable genera and species are Leinkupal (0.1765), and the most basal, possibly new species and genus including SMA 0087 and WDC-FS001A (0.1842). Whereas these values are higher than the most probable referral to an apatosaurine species, they are lower than the genus-level values within Apatosaurinae. However, given the very limited information about Leinkupal, a lower value comparing FMNH P25112 with this genus is expected. Taking all these results of constrained searches and pairwise dissimilarity into account, it is clear that a more detailed study of FMNH P25112 is needed, and that it is most parsimonious to treat the specimen as a Diplodocidae indet. at this stage.

LACM 52844

Tree topology partly confirms earlier referrals of LACM 52844 to Apatosaurus louisae (McIntosh, 1981). The specimen was recovered in a polytomy with the specimens referred to A. louisae under equal weighting, whereas implied weighting found LACM 52844 more basal to the three specimens of A. louisae. However, apomorphy count indicates that LACM 52844 might belong to a third species of Apatosaurus: seven (iw) or eight (ew) changes separate it from other candidate specimens of A. louisae.

A referral of LACM 52844 to a species distinct from Apatosaurus louisae is also supported by the fact that mean pairwise dissimilarity rates are lower for an inclusion of LACM 52844 in A. ajax (0.16) or Brontosaurus excelsus (0.1647) than in A. louisae (0.1944). At generic level, the most probable referral of LACM 52844 is to Apatosaurus (0.1835), followed by Brontosaurus (0.2086).

Five ambiguous autapomorphies were found for LACM 52844 (134-0, 158-1, 304-0, 332-1, 382-1; Table S149), all of which are not present in other apatosaurines, and would thus qualify as species autapomorphies. However, because LACM 52844 was found closely associated with the holotype skeleton of A. louisae, CM 3018 Gilmore, 1936; McIntosh, 1981, and because a number of elements mentioned in McIntosh (1981) were not located and therefore could not be studied during a collection visit to LACM (E Tschopp, pers. obs., 2013), we herein refrain from naming a separate species for LACM 52844, which would be based on incomplete information. Given the conflicting results from tree topology and pairwise dissimilarity, we refer LACM 52844 to Apatosaurus sp.

MB.R. skeleton k

Skeleton k is the second individual referred to Tornieria africana by Remes (2006). The individual includes a braincase (MB.R.2386), which was interpreted as not belonging to that taxon by Harris (2006a). However, based on preserved quarry maps, referral of this material to a single individual appears justified (Heinrich, 1999; Remes, 2006). The present analysis consistently recovers skeleton k with the holotype individual of Tornieria africana. Because no autapomorphy was found distinguishing skeleton k from skeleton A, Remes’ (2006) referral of the two specimens to a single species is corroborated herein.

ML 418

Although consisting of very fragmentary material, ML 418 was always found as the sister taxon of Galeamopus hayi in our analyses. However, the only ‘synapomorphy’ recovered for this group is shared with other specimens within Diplodocinae (165-0, the absence of distinct subfossae in the sdf of anterior and mid-cervical neural spines). No valid autapomorphy was found for ML 418 (Table S150).

ML 418 was referred to Dinheirosaurus by Antunes & Mateus (2003), and later assigned to Apatosaurus sp. by Mateus (2005). However, Mannion et al. (2012) noted that it cannot be confidently identified as either of these two taxa, as it lacks their autapomorphic traits, and identified it as an indeterminate diplodocid. When imposing a monophyletic Galeamopus excluding ML 418, tree length was minimally increased by 0.05% to 1977 steps (ew) or by 0.0001% to 194.21613 steps (iw). Under equal weights, ML 418 was found in a basal polytomy within Neosauropoda, with unclear relationships to most of its subclades, whereas implied weighting recovered it in a polytomy within Diplodocinae, with Leinkupal and Galeamopus + mdD. Constrained searches forcing ML 418 into the Supersaurus clade produce equally weighted trees with lengths of 1978 steps, whereas implied weighting finds shortest trees of 194.25338 steps, corresponding to length increases of 0.1% and 0.02%, respectively. In both cases, ML 418 was not found as the sister-taxon to the “Dinheirosaurus” lourinhanensis holotype specimen. A forced close relationships with Apatosaurus results in tree length increases of 0.2% (ew) or 0.3% (iw) to 1980 or 194.79149 steps. These results imply that Mannion et al. (2012) were correct in considering ML 418 to be a possible second Portuguese diplodocid taxon, although the specimen is not diagnosable based on preserved material. Because inclusion of ML 418 in Apatosaurinae is much less parsimonious than identification as a diplodocine, the specimen is herein considered indeterminate Diplodocinae.

NSMT-PV 20375

The specimen NSMT-PV 20375 was described by Upchurch, Tomida & Barrett (2004) and identified as Apatosaurus ajax, by means of a specimen-based phylogenetic analysis. However, it was not found in close relationship with the holotype specimen of Apatosaurus ajax in any of the analyses reported herein. In fact, NSMT-PV 20375 consistently occupies the most basal position within Apatosaurinae, together with YPM 1840. No valid ‘autapomorphies’ were recovered for NSMT-PV 20375 (Table S151).

Imposing a grouping of NSMT-PV 20375 with Apatosaurus ajax, as found by Upchurch, Tomida & Barrett (2004) produced trees of 1979 and 194.52068 steps, a relative increase of 0.15% and 0.16%. In both cases, the position of NSMT-PV 20375 remained stable, but the type specimen of A. ajax was transferred into the basal-most clade within Apatosaurinae. Under equal weighting, the type specimen of Apatosaurus laticollis (YPM 1861) was also transferred into the same clade, whereas implied weighting still found YPM 1861 with A. louisae. The most parsimonious interpretation thus seems the arrangement found by the implied weights trees, with NSMT-PV 20375 and YPM 1840 forming the basal-most taxon within Apatosaurinae. Thus, it seems that one more previously unrecognized taxon occurs within Apatosaurinae. However, support for such a separation is low, and more detailed studies are needed to confirm such a hypothesis. We thus refer NSMT-PV 20375 to Apatosaurinae indet. pending further studies.

SMA 0009

This small juvenile specimen was initially described as a diplodocid (Schwarz et al., 2007), but subsequent studies after further preparation suggested brachiosaurid affinities (Carballido et al., 2012a). Our analyses recovered SMA 0009 consistently outside Diplodocoidea, either in a clade with Brachiosaurus and Giraffatitan (under equal weights), or in a position basal to Titanosauriformes + Diplodocoidea (under implied weights). Brachiosaurid affinities therefore seem better supported herein than a referral to Diplodocidae.

Constrained searches for apatosaurine affinities resulted in tree length increases of 0.86% (ew) and 0.66% (iw), to 1993 and 195.50453 steps, respectively. A forced inclusion in Diplodocinae yielded trees 1.67% (ew) and 1.16% (iw) longer than the unconstrained MPTs. Under implied weights, brachiosaurid affinities were found in constrained trees of a length of 194.32227, an increase of 0.05%. Given the highest increase in constrained tree lengths including SMA 0009 in Diplodocidae, among all constrained searches performed in this study, non-diplodocid affinities are much less probable than a referral to Brachiosauridae. Such a referral is also supported by the minimal increase when forcing SMA 0009 into Brachiosauridae under implied weighting, in which SMA 0009 was initially found in a more basal position. Furthermore, such a constraint also results in the recovery of a monophyletic Macronaria, which is instead polyphyletic in all MPTs of our four main analyses. SMA 0009 is therefore referred to Brachiosauridae herein, and because Brachiosaurus altithorax is the only known brachiosaurid from the Morrison Formation, we tentatively include SMA 0009 in this species.

SMA 0011

SMA 0011 consistently groups with the holotype of Galeamopus hayi, HMNS 175, ML 418, and two skulls previously identified as Diplodocus, AMNH 969 and USNM 2673 (Holland, 1906; McIntosh & Berman, 1975). The specimen SMA 0011 has one unambiguous (191-1) and seven ambiguous autapomorphies (154-2, 186-1, 226-0, 279-0, 380-2, 386-0, 391-1), which would justify specific separation from Galeamopus hayi (Table S152). However, given the unclear positions of ML 418 and the two skulls within the Galeamopus clade, we refrain from naming a new species herein without a detailed description. SMA 0011 is thus referred to Galeamopus sp.

SMA 0087

The specimen SMA 0087, yet unreported but from the same quarry as SMA 0011, forms a clade together with WDC-FS001A, which is located at the base of Diplodocinae. One valid ‘autapomorphy’ is found by the present analysis (469-0; Table S153), but the number of changes between SMA 0087 and WDC-FS001A is too low to establish distinct species (Table S85). Two of the four shared synapomorphies between SMA 0087 and WDC-FS001A would qualify as species autapomorphies (324-1, 455-1), given that they are not shared with any other diplodocine specimen. Diplodocine affinities are indicated for SMA 0087 by the presence of seven ambiguous synapomorphies (267-1, 273-1, 293-1, 300-3, 355-0, 422-0, 459-1) of the clade.

SMA D16-3

This partial skull has not been described in detail thus far. It is herein consistently found as belonging to Kaatedocus siberi. No autapomorphies were found in any of the trees. A referral to Kaatedocus siberi is thus warranted.

SMA O25-8

The second isolated partial skull (besides SMA D16-3) from Howe Quarry exhibits both internal and external differences in braincase morphology, compared with specimens of Kaatedocus siberi (Schmitt et al., 2013). It was excluded from all reduced consensus trees. In the pruned consensus trees, it consistently forms a polytomy within the Kaatedocus + Barosaurus clade, including CM 11255, the specimens referred to Barosaurus lentus, and the clade Kaatedocus siberi.

The specimen SMA O25-8 can be confidently identified as Diplodocidae due to the hook-shaped posterior process of the prefrontal and the slightly concave posterior surfaces of the basal tubera, and as Diplodocinae given the box-like basal tubera and the presence of a basipterygoid recess. It is included in the Kaatedocus + Barosaurus clade based on the distinct nuchal fossae on the parietal, and the ridge on the posterior surface of the paroccipital process.

Forcing SMA O25-8 into a clade with Barosaurus lentus does not increase tree length, but a confident assignment to this taxon is hampered by the lack of overlap with definitive Barosaurus lentus specimens. Constraining it to group with Kaatedocus siberi also does not increase tree length, but no synapomorphies are found for an inclusion of SMA O25-8 into Kaatedocus siberi.

Mean pairwise dissimilarity values for referrals to directly comparable diplodocine species are nearly all higher than 0.222, the threshold established for generic distinction within Diplodocinae. The only lower value was found for an inclusion in Kaatedocus (0.2162), the values for the other species all exceed 0.3. These values thus corroborate the position found in the equally weighted pruned consensus tree, showing a closer relationship of SMA O25-8 with Kaatedocus than with Galeamopus. However, they also contradict a referral of SMA O25-8 to the species Kaatedocus siberi.

Taking all the information into account, SMA O25-8 can be confidently identified as a derived diplodocine, most closely related to either Kaatedocus or Barosaurus. Pending further studies, and given the differences found between SMA O25-8 and the known Kaatedocus braincases, SMA O25-8 is herein referred to Diplodocinae indet.

USNM 2672

The specimen USNM 2672 is another skull usually identified as Diplodocus, which was included in the study. It also preserves a partial atlas. Unfortunately, because no definitive Diplodocus specimen is known with either an atlas or a skull, confident identification of USNM 2672 is not possible, as is also the case for CM 11161 (see above).

No ‘autapomorphy’ was found in the equally weighted pruned consensus tree, the only tree to include USNM 2672. Nonetheless, the specimen can be confidently identified as a diplodocid due to the broad contact between maxilla and quadratojugal, the large preantorbital fenestra, the concave dorsal margin of the antorbital fenestra, the medially curving anteromedial corner of the prefrontal, the hook-shaped posterior process of the prefrontal, the slightly concave posterior face of the basal tubera, the absence of a coronoid eminence, and the absence of direct crown-to-crown occlusion in the teeth. Diplodocine affinities are confirmed by the box-like basal tubera.

The same constrained searches were performed as for CM 11161, in order to test affinities with species for which cranial material is known. Affinities with Galeamopus are found in constraint searches, resulting in trees of length 1977 or 194.28745 steps (an increase of 0.05% or 0.04%). Forcing an inclusion into the Kaatedocus clade yields trees of a length of 1978 and 194.33526 steps, corresponding to a 0.1% and 0.06% length increase. As with CM 11161, a referral of USNM 2672 to any diplodocine species seems premature, and both specimens are thus best treated as Diplodocinae indet.

USNM 2673

This partial skull has generally been referred to Diplodocus (McIntosh & Berman, 1975; Whitlock, Wilson & Lamanna, 2010). In our analysis, however, it was consistently found within the new genus Galeamopus.

Two ambiguous ‘autapomorphies’ were found for USNM 2673 (26-1, 73-0; Table S154), resulting in eight changes between the specimen and its sister-clade within Galeamopus. This would justify specific separation, but because detailed description is lacking for all the specimens in this clade, it seems most cautious to recognize just a single species at present.

Forcing USNM 2673 into the Diplodocus clade found trees of 1978 (ew) and 194.42079 (iw) steps, a length increase of 0.1% and 0.11%, respectively. In both cases, such a referral results in large polytomies. A referral to Galeamopus therefore appears to be much better supported, even though the two skulls USNM 2672 and 2673 were apparently found in the same quarry (McIntosh & Carpenter, 1998).

USNM 10865

On display at USNM, the specimen USNM 10865 is the second relatively complete skeleton referred to Diplodocus longus after AMNH 223 (Osborn, 1899; Gilmore, 1932), and has been partially described by Gilmore (1932). In the present analysis, USNM 10865 consistently forms a clade with DMNS 1494.

No valid ‘autapomorphy’ is found for USNM 10865 (Table S155), and as stated above, specific distinction from DMNS 1494, AMNH 223, and most importantly the holotype of Seismosaurus hallorum, NMMNH 3690, is not warranted. Because Seismosaurus was synonymized with Diplodocus, the specimen USNM 10865 is herein referred to the species Diplodocus hallorum.

UW 15556

Described in detail by Hatcher (1902) and Gilmore (1936), the specimen UW 15556 (previously CM 563) is one of the best known apatosaurine specimens. It was often referred to Apatosaurus excelsus (Hatcher, 1902; Gilmore, 1936; McIntosh, 1981; McIntosh, 1995), but was recently found to constitute its own species within Apatosaurus, together with the holotype of Elosaurus parvus, CM 566 (Upchurch, Tomida & Barrett, 2004). Upchurch, Tomida & Barrett (2004) thus proposed the new combination Apatosaurus parvus. However, as shown above, Elosaurus parvus is included in the Brontosaurus clade, resulting in the combination Brontosaurus parvus.

Our analysis found UW 15556 in a sister-taxon relationship with BYU 1252-18531, and together they formed the sister-clade to the type specimen of “Elosaurus” parvus, CM 566. Four ambiguous autapomorphies were recovered for UW 15556 (202-1, 242-1, 305-0, 389-0; Table S156). However, as discussed above, even though differences would be numerous enough to justify two distinct species, we prefer to unite all three specimens in a single species. The specimen UW 15556 is thus herein referred to Brontosaurus parvus.

WDC DMJ-021

WDC DMJ-021 was described by Lovelace, Hartman & Wahl (2007), and identified as Supersaurus vivianae. Herein, it was always found in a clade with the BYU specimen of Supersaurus vivianae, thus confirming the assignment of Lovelace, Hartman & Wahl (2007).

No valid autapomorphies of WDC DMJ-021 were found in any of our analyses (Table S157), but eight shared synapomorphies unite the two specimens of Supersaurus (Table S79). One of them is unique within Diplodocinae and can be considered an autapomorphy of the species (231-0).

WDC-FS001A

Only the manus of the present specimen has been described in detail (Bedell & Trexler, 2005). The specimen was identified as Diplodocus cf. carnegii, based on morphology of a caudal vertebra, which was different from the specimens generally considered to represent ‘Diplodocus longus,’ and the general slenderness of the appendicular bones (Bedell & Trexler, 2005). Our analyses consistently found WDC-FS001A together with SMA 0087, as a basal clade within Diplodocinae.

No valid ‘autapomorphies’ were found for WDC-FS001A (Table S158). Inclusion into Diplodocinae is supported by the occurrence of one shared (421-1) and one ambiguous diplodocine synapomorphy (442-1).

A forced clustering with Diplodocus (as proposed by Bedell & Trexler, 2005) produces tree lengths of 1982 and 194.58597 steps, representing increases of 0.3% and 0.19%. Given this large increase, referral of WDC-FS001A to Diplodocus is improbable. It therefore seems that WDC-FS001A and SMA 0087 represent a distinct diplodocine genus is present, but the two specimens should be prepared and described in detail before establishing a new name. WDC-FS001A is thus referred to Diplodocinae indet.

Systematic Paleontology

Updated diagnoses of the main diplodocoid subclades

The following lists of synapomorphies only includes the named nodes and stems in the recovered phylogenetic tree, which directly lead to Diplodocidae, as well as its sister clade Dicraeosauridae. Synapomorphies are divided into their qualitative states as defined above, and ordered based on anatomical regions. Where conflicting interpretations exist between the analyses using equal or implied weighting, the synapomorphy is attributed to the less inclusive clade. Additional synapomorphies are added to the diagnoses following earlier studies, if supported by our dataset. Where our analysis did not support the recognition of previously proposed synapomorphies, we have explained why. References for the synapomorphies credit the first recognition of the respective trait as synapomorphic for the taxon in question.

Diplodocoidea Marsh, 1884.

Definition: All taxa more closely related to Diplodocus than to Saltasaurus (stem-based; Wilson & Sereno, 1998).

Unambiguous synapomorphies (i.e., features unique to the clade under question, and shared by all members of the clade, see definition above):

1. Premaxilla is a single elongate unit with nearly no distinction between the body and the nasal process (3-1 (i.e., state 1 of character 3); Upchurch, Barrett & Dodson, 2004).

2. Posteroventral edge of the ascending process of the premaxilla is straight in lateral view, and directed posterodorsally (5-2; Upchurch, 1995).

3. The dorsal process of the maxilla extends posterior to the posterior process (13-1; Wilson, 2002).

4. The external nares are retracted to a position between the orbits, facing dorsally or dorsolaterally (21-1; Marsh, 1898).

5. A large contribution of the jugal to the antorbital fenestra, bordering approximately one-third of its perimeter (40-1; Upchurch, 1995).

6. The anterior terminus of the quadratojugal lies below the anterior margin of the orbit or beyond (45-1; Rauhut et al., 2005).

7. Angle between anterior and dorsal processes of the quadratojugal is greater than 90°, approaching 130°, so that the quadrate shaft slants posterodorsally (46-1; McIntosh, 1990b).

8. The basipterygoid processes are angled less than 75° to the skull roof (normally approximately 45°) (93-1; Calvo & Salgado, 1995).

9. The transverse flange (i.e., ectopterygoid process) of the pterygoid lies anterior to the antorbital fenestra (102-1; Upchurch, 1998).

10. Four or more replacement teeth per alveolus (115-1; Wilson, 2002).

11. Planar wear facets of the teeth (117-1).

12. Cylindrical cross-sectional shape of the teeth at midcrown (121-1; Marsh, 1884).

13. The fibular facet of the astragalus faces posterolaterally, such that the anterior margin is visible in posterior view (454-1).

Exclusive synapomorphies (i.e., features unique to the clade under question, but not necessarily shared by all members of the clade, see definition above):

14. External surface of the premaxilla is marked by vascular grooves (2-1).

15. The anterior maxillary foramen lies on the medial edge of the maxilla, opening medially into the premaxillary-maxillary boundary (11-1).

16. Maximum diameter of the antorbital fenestra is subequal (greater than 90%) to the orbital maximum diameter (18-1; Wilson, 2002).

17. The articular surface of the quadrate is roughly triangular in shape (49-1).

18. SI values for tooth crowns are 3.4 or greater (119-1; McIntosh, 1990b).

19. Short cervical ribs, not reaching the posterior end of the centrum (214-1; Berman & McIntosh, 1978).

Shared synapomorphies (i.e., features shared with species outside the clade under question, but shared by all members of the clade, see definition above):

20. The posterolateral process of the premaxilla and the lateral process of the maxillary are without any midline contact (6-0; Wilson, 2002).

21. Maximum diameter of the external nares is shorter than the orbital maximum diameter (22-0).

22. Cervical ribs overlap no more than the next cervical vertebra in sequence (215-1).

23. The proximal expansion of the humerus is more or less symmetrical (384-0).

Ambiguous synapomorphies (i.e., features shared with species outside the clade under question, and not necessarily shared by all members of the clade, see definition above):

24. The distal end of the occipital process of the parietal curves laterally, such that the dorsolateral edge becomes concave distally (64-1).

25. The articular surface of the occipital condyle is continuously grading into the condylar neck (77-1).

26. A humerus-to-femur ratio of less than 0.7 (379-0; Huene, 1927).

27. The participation of the pubis in the acetabulum is significantly smaller, compared to the one from the ischium (416-1).

28. The absence of a lateral bulge in the femur (428-0).

29. The presence of a laterally directed ventral shelf on the astragalus, which underlies the distal end of the fibula (455-0).

Previously suggested synapomorphies:

A very acute angle between medial and lateral margins of the premaxilla (Upchurch, Barrett & Dodson, 2004). The character describing the angle between medial and lateral borders of the premaxilla was redefined herein (C4), and the numeric boundary changed in order to be able to distinguish between Dicraeosauridae and Diplodocidae. An angle lower than 17° is synapomorphic for both Rebbachisauridae and Diplodocidae, but not for Dicraeosauridae (Table S2). The same character was further found by Whitlock (2011a) to diagnose Diplodocimorpha.

An elongate subnarial foramen (Upchurch, Barrett & Dodson, 2004). A character describing the elongation of the subnarial foramen was not included in the present analysis, as it is impossible to code in most specimens. Even when rostral skull elements are preserved, the fossa containing the subnarial and the anterior maxillary foramina is often obliterated with matrix (e.g., USNM 2672), and only CT scanning would reveal the true shape.

A strongly reduced anteroposterior diameter of the supratemporal fenestra (Upchurch, Barrett & Dodson, 2004). The relation of anteroposterior diameter of the supratemporal to occipital width was not included in the present analysis, because it was not well explained what was measured for obtaining a value for the occiput width (Upchurch, Barrett & Dodson, 2004). Furthermore, the anteroposterior diameter of supratemporal fenestrae seems to be more variable within diplodocids than previously recognized, and is frequently deformed by taphonomy (compare the two diplodocid skulls CM 11161 and 11255; Holland, 1924). It is therefore difficult to score accurately and was not used.

Elongate basipterygoid processes (McIntosh, 1990b; Upchurch, 1998). This trait was recovered as a diplodocimorph instead of as a diplodocoid synapomorphy by Wilson (2002) and Whitlock (2011a). In fact, however, Diplodocimorpha as found by Wilson (2002) and Whitlock (2011a) includes the same taxa as the Diplodocoidea of McIntosh (1990b) and Upchurch (1998). Whitlock (2011a) resolved this as a diplodocimorph synapomorphy only due to the use of the DELTRAN optimization strategy, combined with a recovered basal position of Haplocanthosaurus (for which the cranium is unknown) outside his Diplodocimorpha. In the present analysis, definition of the character (C94) was slightly changed, to encompass variation observed within Diplodocidae. It can thus not be considered a synapomorphy for any named clade herein.

A rectangular snout (Upchurch, Barrett & Dodson, 2004). The rectangular snout outline seen in dorsal view was herein included as diagnosing Diplodocimorpha (111-1 and 111-2), following Whitlock (2011a).

Dentary with ventrally projecting ‘chin’ (Wilson & Sereno, 1998). At the time Wilson & Sereno’s (1998) monograph was published, no dentary was known from diplodocoids more basal than Flagellicaudata. The recovery of Nigersaurus and Demandasaurus dentaries showed that such a ‘chin’ was absent in rebbachisaurids (Sereno et al., 1999; Sereno & Wilson, 2005; Torcida Fernández-Baldor et al., 2011). Consequently, its presence was later found as a synapomorphy for Flagellicaudata (Whitlock, 2011a; 104-1 in this study).

The anterior restriction of the tooth row (McIntosh, 1990a). The length of the tooth row is recovered as diplodocimorph synapomorphy by Whitlock (2011a), applying DELTRAN. In the present analysis (C113), the number of states has been increased, compared to the definition of Whitlock (2011a), due to the recognition of a higher diversity within diplodocids. Also, brachiosaurid skulls have anteriorly restricted tooth rows (Janensch, 1935; Wilson & Sereno, 1998), which shows that this feature is present in diplodocoid outgroups as well.

Atlantal intercentrum with anteroventral lip (Wilson & Sereno, 1998). The same doubts apply here as for the presence of a ‘chin’ in the dentary (see above). The question is furthermore complicated because no rebbachisaurid atlas has been described to date. With the present dataset it is thus more cautious to add this trait (144-1) as synapomorphy of Flagellicaudata.

Cervical and anterior dorsal vertebrae opisthocoelous (McIntosh, 1990a). Opisthocoelous cervical and anterior dorsal vertebrae are actually widespread among sauropod dinosaurs, and represent the plesiomorphic condition. No phylogenetic analysis was thus able to support this trait as a synapomorphy of Diplodocoidea.

Deeply divided V-shaped posterior cervical and anterior dorsal neural spines (McIntosh, 1990b). Subdivided cervical and dorsal neural spines are known from a variety of sauropod dinosaurs from different clades (Upchurch, Barrett & Dodson, 2004; Wedel & Taylor, 2013). Furthermore, given that basal diplodocoids have undivided neural spines, the bifurcation cannot be considered diagnostic for the entire clade. Instead, it is a synapomorphy of Flagellicaudata (126-1). The shape of the subdivision was proposed as distinguishing feature between diplodocids and Camarasaurus (V- versus U-shaped; McIntosh, 1990a), but has rarely been used in phylogenetic analyses. In the present analysis, a character is used to describe the base of the notch between the metapophyses (C244). The occurrence of U-shaped notches by our definition is not restricted to Camarasaurus, but is also present in some diplodocoids (e.g., Amargasaurus cazaui, Apatosaurus ajax YPM 1860). Therefore, the presence of V-shaped notches was not recovered as a synapomorphy of any named clade.

The left and right spinoprezygapophyseal laminae of dorsal vertebra unite towards the spine summit (Whitlock, 2011a). Here, this feature (231-1) is recovered as diagnosing a more inclusive clade, SMA 0009 + md eusauropods, in the equally weighted reduced consensus tree, as well as in both main implied weights trees. The difference is a result of the addition of the titanosauriform species Giraffatitan brancai, Ligabuesaurus leanzai, and Isisaurus colberti, where spinoprezygapophyseal and prespinal laminae join dorsally (Janensch, 1950; Jain & Bandyopadhyay, 1997; Bonaparte, González Riga & Apesteguía, 2006).

Posterior dorsal centra are amphicoelous (McIntosh, 1990a). Detailed study of diplodocine posterior dorsal centra showed that most of them are actually slightly opisthocoelous (e.g., Diplodocus carnegii CM 84) to distinctly so (Supersaurus vivianae). The amphicoelous condition (270-0) was herein recovered as synapomorphic for Brontosaurus.

Arches of dorsal and caudal vertebrae tall (more than two and one-half times dorsoventral centrum height) (Wilson & Sereno, 1998). This synapomorphy actually includes two characters as used by Whitlock (2011a) and in our study (C254, C302). Both were interpreted to diagnose Diplodocimorpha by Whitlock (2011a). In our study, state boundaries for the dorsal neural arch height were changed to distinguish between diplodocids, which actually show variable ratios of neural arch height to posterior centrum height (Table S32). A detailed study of the proportional height of diplodocid caudal neural spines showed that many specimens have neural spines that are actually less than 1.5 times taller than the posterior articular surface of the centrum (Table S38). Therefore, neither of the two characters was recovered as diplodocoid or diplodocimorph synapomorphy.

Proximal caudal vertebrae with procoelous centra (McIntosh, 1990b). Procoelous centra have been shown to have a much wider distribution outside Diplodocoidea (Carballido et al., 2012b; D’Emic, 2012; Mannion et al., 2013). Herein, the character describing caudal articular surface shape (C304), is subdivided into four states, including slight and strong procoely (following Carballido et al., 2012b). Whereas most diplodocines have slightly procoelous anterior caudal centra, many other diplodocid specimens actually have flat posterior articular surfaces. To state that all diplodocoid taxa have procoelous centra would thus over simplify the variety of morphologies seen within the clade.

Caudal vertebrae with wing-like transverse processes (McIntosh, 1990b). This trait was found to diagnose Diplodocimorpha by Whitlock (2011a). Many non-diplodocid sauropod species do have first caudal vertebrae with transverse processes that expand onto the neural arch and that have a distinct shoulder on their dorsal edge on their first caudal vertebra. These are herein interpreted as having wing-like transverse processes (299-1), although their processes are more triangular than the subrectangular processes of diplodocoid taxa, which have typically been described as wing-like. The problem is best exemplified by a putative diplodocid anterior caudal from the Cretaceous of China (PMU R263; Upchurch & Mannion, 2009), which was later reidentified as somphospondylan titanosauriform (Whitlock, D’Emic & Wilson, 2011). A more precise definition of wing-like would be beneficial for future analyses.

Presence of a “whip-lash” tail (at least 30 elongate, biconvex posterior caudal vertebrae) (McIntosh, 1990a; Wilson & Sereno, 1998). The present analysis is not able to identify this feature as synapomorphic for any clade, due to the incompleteness of the included specimens. Only the two specimens of Apatosaurus louisae (CM 3018 and 3378) preserve a tail complete enough to confidently score them for this character. The trait was thus not included into any clade diagnosis. However, it is possible that this feature is a valid synapomorphy of Diplodocoidea or a lower-level taxon within this clade, because the elongate distal caudal vertebrae typical for a “whip-lash” tail occur in several genera within all major diplodocoid clades (e.g., Barosaurus lentus YPM 429, Dicraeosaurus hansemanni MB.R.4886, and potentially Limaysaurus tessonei MUCPv-205; Lull, 1919; Janensch, 1929a; Calvo & Salgado, 1995).

Presence of forked chevrons (McIntosh, 1990b; C353 herein). Although named for this peculiar morphology, Diplodocus (meaning “double-beam”; Marsh, 1878) and higher-level clades based on Diplodocus (e.g., Diplodocidae) are not the only taxa to have forked chevrons. In fact, recent studies show that this might actually be a retained plesiomorphy that is already present in basal sauropods like Shunosaurus and Spinophorosaurus (Zhang, 1988; Remes et al., 2009), and got subsequently lost in macronarians.

Short metacarpals (McIntosh, 1990a). This character (C399) has a similar distribution to that of forked chevrons: relatively short metacarpals are plesiomorphic for Sauropoda, whereas the elongate metacarpals diagnose macronarian taxa (Wilson, 2002; Upchurch, Barrett & Dodson, 2004; Apesteguía, 2005; Tschopp et al., in press).

Ischia have expanded distal ends (McIntosh, 1990b). Expanded distal ends of the ischia were present in all diplodocoid specimens preserving ischia known in 1990. However, more recently, rebbachisaurids have been found to have distally unexpanded ischia, rendering this trait a synapomorphy of Flagellicaudata.

Diplodocimorpha Calvo & Salgado, 1995

Definition: Diplodocus and Rebbachisaurus, their most recent ancestor and all of its descendents (node-based; Taylor & Naish, 2005).

Unambiguous synapomorphies:

1. The anterior margin of the premaxilla does not have a step (1-0; Wilson, 2002. This synapomorphy was not found by the present analysis, but recovered as such by Wilson (2002) and Whitlock (2011a). Because the data matrix indeed supports an identification of this trait as unambiguous synapomorphy for Diplocimorpha, it has been included in the present list. The reason why it was not recovered as synapomorphy by TNT is probably the fact that only a minority of specimens could be scored for this character).

2. The sprl extend onto lateral aspect of anterior caudal neural spines (318-1).

Exclusive synapomorphies:

3. Squared (111-2) or blunted snout (111-1; Berman & McIntosh, 1978. As the absence of a premaxillary step, also the squared or blunted snout was found as a synapomorphy by Whitlock (2011a), but not directly confirmed by the present analysis, although supported by the scores in our data matrix).

4. Transition from ‘fan’-shaped to ‘normal’ caudal ribs occurs between Cd 4 and Cd 5 (300-1).

Ambiguous synapomorphies:

5. Biconvex distal-most caudal centra (346-1; Upchurch, 1998. This character state only occurs in Diplodocimorpha in the present analysis (but absent in Suuwassea, Harris, 2006a), and was recovered as a diplodocimorph synapomorphy by Whitlock (2011a) as well. However, biconvex caudal vertebrae also occur in titanosauriforms (Wilson, Martinez & Alcober, 1999). Therefore, this character state only qualifies for an ambiguous synapomorphy of Diplodocimorpha).

Previously suggested synapomorphies:

The analysis of Whitlock (2011a) produced a high number of synapomorphies for Diplodocimorpha (“Rebbachisauridae + Flagellicaudata” in Whitlock, 2011a). Several of these are herein recovered as synapomorphic for Diplodocoidea (see above): the straight, and posterodorsally directed nasal process of the premaxilla, the absence of a sharp distinction between the premaxillary main body and the nasal process, the lack of a midline contact of the posterolateral process of the premaxilla and the lateral process of the maxilla, the dorsal process of the maxilla extends posterior to the posterior process, the diameters of the antorbital and orbital fenestra are approximately equal, the external nares are retracted, the jugal contribution to the antorbital fenestra is large, the anterior ramus of the quadratojugal extends anterior to the orbit, the angle between the anterior and the dorsal process of the quadratojugal is wide, the angle between basipterygoid processes and skull roof is low, the transverse flange of the pterygoid extends anterior to the antorbital fenestra, and four or more replacement teeth are present per alveolus. Because no skull is known for Haplocanthosaurus, which is the outgroup of Diplodocimorpha, and currently constitutes the most basal diplodocoid, the recovery of these synapomorphies for Diplodocidea or Diplodocimorpha depends on the optimization method used. With ACCTRAN, they are synapomorphies of Diplodocoidea, whereas DELTRAN recovers them as synapomorphies of Diplodocimorpha. Additional synapomorphies previously recovered for Diplodocimorpha are the following:

Parietal is excluded from the margin of the posttemporal foramen (Calvo & Salgado, 1995; Upchurch, 1998; Wilson, 2002). Because rebbachisaurid parietals participate in the posttemporal foramen, the exclusion of the parietal from the posttemporal foramen (59-1) is recovered as a synapomorphy of Flagellicaudata herein, as already proposed by Whitlock (2011a).

Squamosal extends anteriorly past the posterior margin of the orbit (Whitlock, 2011a). The anterior extension of the squamosal is restricted in Kaatedocus (Tschopp & Mateus, 2013b), which inhibits an identification of the anteriorly reaching squamosal (55-1, and 55-2) as a diplodocimorph synapomorphy in the present analysis.

Tooth crowns aligned along jaw axis, not overlapping (Wilson, 2002). The absence of overlap between tooth crowns (120-1) is not restricted to Diplodocoidea, but also occurs in Giraffatitan brancai, for example (Janensch, 1935; Wilson & Sereno, 1998). It was thus not recovered as a synapomorphy of any clade in our analysis.

Mid-caudal vertebral centra length at least twice its height (Upchurch, Barrett & Dodson, 2004). The mid-caudal centra are generally more elongate in diplodocoids, compared to other taxa. However, they only reach ratios of two times centrum height in advanced diplodocines, as a more detailed assessment of this character shows (Table S39). Therefore, state boundaries were changed to 1.7 (C332). The higher ratio of 2 can thus not be regarded synapomorphic for Diplodocimorpha.

Distal-most caudal centra at least five times longer than tall (Wilson, Martinez & Alcober, 1999). The elongation of these distal caudal vertebrae was coded differently in Whitlock (2011a) and herein, which resulted in Apatosaurus specimens (which have proportionally shorter centra) being scored differently to Diplodocus specimens. The ratio of greater than 5.0, as proposed by Whitlock (2011a) might thus still be valid, but cannot be recovered as synapomorphic with our analysis due to the use of different state boundaries.

Proximal margin of humerus expanded, lateral margin concave in anterior/posterior view (Janensch, 1961). The last diplodocimorph synapomorphy recovered by Whitlock (2011a) describes the concave lateral border of the humerus. This feature (385-0) is actually also present in most of the basal sauropods used as outgroups herein. It is thus a plesiomorphic trait and cannot be used as synapomorphy of Diplodocimorpha.

Flagellicaudata Harris & Dodson, 2004

Definition: Dicraeosaurus and Diplodocus, their most recent ancestor and all of its descendents (node-based; Harris & Dodson, 2004).

Unambiguous synapomorphies:

1. Subnarial foramen and anterior maxillary foramen are separated by a narrow bony isthmus (8-1; Wilson, 2002).

2. Presence of a preantorbital fossa (15-1).

3. An elongate and slender posterior end of the quadrate (posterior to posterior-most extension of pterygoid ramus) (54-1).

4. The absence of any squamosal-quadratojugal contact (56-1).

5. The absence of a parietal contribution to the post-temporal fenestra (59-1; Whitlock, 2011a).

6. Vomer articulates with maxilla (103-1; Wilson, 2002. The recovery of this trait as synapomorphy for Flagellicaudata is supported by our analysis but not recovered as such, probably due to the very low percentage of specimens scorable for the character).

7. The anteroventral margin of the dentary bears a sharply projecting triangular process or ‘chin’ (104-1; Wilson & Smith, 1996).

8. Anteriorly oriented, procumbent teeth (122-1).

9. Atlantal intercentrum bears an anteroventral lip (144-1. Recovered as diplodocoid synapomorphy by Wilson & Sereno (1998), the presence of the anteroventral lip can actually only be confirmed for Flagellicaudata, because no rebbachisaurid atlas has yet been reported. The data matrix supports an identification of the derived as diagnostic for Flagellicaudata, even though it was not recovered as such).

10. The distal shaft of the ischium is triangular, with its depth increasing medially (423-1).

Exclusive synapomorphies:

11. The longest axes of the basal tubera are oriented in an angle to each other, pointing towards the occipital condyle (87-1).

12. The lateral spinal lamina of anterior-most caudal neural spines expands anteroposteriorly towards its distal end, and becomes rugose (303-1).

Shared synapomorphies:

13. A shallow quadrate fossa (51-0).

14. Absence of longitudinal grooves on the lingual aspect of the teeth (123-0).

15. The hyposphene-hypantrum system is well developed in posterior dorsal vertebrae, having a rhomboid shape up to last element (276-0).

16. Anterior diapophyseal laminae (acdl, prdl) are well defined in in anterior caudal vertebrae (313-1).

17. A ‘crus’ bridging the haemal canal is present in some chevrons (352-0; Wilson, 2002).

18. Pubis with a prominent ambiens process (414-1 and 414-2; McIntosh, 1990b; this synapomorphy was not recovered by our analysis, even though the data matrix supports its inclusion as a shared synapomorphy. In our analysis we made a distinction between the hook-like ambiens process as present in Diplodocus and Dicraeosaurus (Hatcher, 1901; Janensch, 1961), for example, and the less developed, but still prominent process of apatosaurines (Ostrom & McIntosh, 1966). A prominent ambiens process can thus still be confirmed as synapomorphic for Flagellicaudata, but because a somewhat prominent ambiens process also occurs in Omeisaurus (He, Li & Cai, 1988), it could only be treated as a shared synapomorphy).

19. The cross-sectional shape of ischial distal shafts is V-shaped, forming an angle of nearly 50° with each other (424-0; Upchurch, 1998).

20. The ischial shaft is transversely expanded distally (425-1; Upchurch, 1998).

21. The distal condyle of metatarsal I bears a posterolateral projection (463-1; Berman & McIntosh, 1978).

Ambiguous synapomorphies:

22. Absence of a squamosal-quadratojugal contact (58-1).

23. Presacral neural spine bifurcation present (126-1; McIntosh, 1990b).

24. Mid- and posterior cervical centra have longitudinal flanges on the lateroventral edge on the posterior part of the centrum (179-1).

25. Transversely compressed posterior cervical epipophyses (202-0).

26. Mid-dorsal neural spines are bifid, inclusive of at least the fifth dorsal vertebra (250-1).

27. Mid- and posterior dorsal neural arches have divided centropostzygapophyseal lamina, with the lateral branch connecting to the pcdl (261-1).

28. Anterior caudal centra (excluding the first) are procoelous/distoplatyan (304-1).

29. The ventral surface is marked by irregular foramina on some anterior caudal centra (305-1).

30. The posterior edge of the distal blade of anterior chevrons is posteriorly expanded in a step-like fashion (355-1).

31. The expansion of the distal end of the scapular blade is less than two times the narrowest width of the shaft (371-1).

32. Metatarsals I-III are marked by rugosities on the dorsolateral margins near their distal ends (465-1).

Previously suggested synapomorphies:

Quadrate articular surface roughly triangular in shape (Whitlock, 2011a). The triangular articular surface of the quadrate (49-1) was recovered as an exclusive diplodocoid synapomorphy herein, with rebbachisaurids developing crescent-shaped surfaces. This is most probably due to the fact that the character was herein treated as ordered, thus assuming that a common ancestor of rebbachisaurs and flagellicaudatans must have had triangular articular surfaces. If the character states would instead be treated as unordered, the triangular shape might be found as a synapomorphy of Flagellicaudata, as in Whitlock (2011a).

Distance between supratemporal fenestrae twice the length of the longest axis of the supratemporal fenestrae (Salgado & Calvo, 1992). A detailed assessment of this ratio showed that most diplodocids have in fact a ratio of less than 2.0 (Table S7). Even after redefining the state boundaries (C61), variation between diplodocid specimens results in differential scorings. A high ratio, and thus wide distance between the supratemporal fenestrae can thus not be regarded synapomorphic for Flagellicaudata.

Ventrally directed occipital condyle (Upchurch, 1998). The orientation of the occipital condyle was not included in the present analysis, because it was found to be very difficult to define a character in an unambiguous way. However, the occipital condyle of flagellicaudatans does project more ventrally compared to its orientation in other taxa, when orienting the skull such that the frontals are horizontal. A more detailed study of this character might thus show that these different orientations of the occipital condyle are indeed synapomorphic for certain clades.

Single planar occlusal facet on teeth (Wilson, 2002). This synapomorphy includes two characters as used in the present analysis, the distinction between V-shaped and planar facets (C117), and the double versus single occlusal facets (C118). The planar facets were found herein as synapomorphy for Diplodocoidea, whereas the single facets are not found to be typical for any clade.

Seventeen dentary teeth or fewer (Wilson, 2002). Whereas it is true that flagellicaudatans have fewer than 17 teeth, the same is true for basal macronarian dinosaurs (e.g., Camarasaurus or Giraffatitan; Gilmore, 1925; Janensch, 1935), as well as for the rebbachisaurid Demandasaurus. It thus seems more parsimonious to interpret the fewer than 17 dentary teeth state as ancestral to all neosauropods, with subsequent reversal to a higher number of teeth in Nigersaurus (Sereno & Wilson, 2005).

Low-angled, planar wear facets on the teeth (Calvo, 1994). The angulation of the wear facets was not included as a character in the present analysis, as an acute angle only characterizes rebbachisaurids, and enough characters were already used to resolve the position and relationship of that clade. Low angles are not restricted to diplodocids either, being also present as late stages in the wear of camarasaur teeth (e.g., SMA 0002; Wiersma, 2013).

Anterior cervical neural spines bifid (McIntosh, 1990b). Anterior neural spines are rarely preserved in cervical vertebrae, even in nearly complete specimens such as the holotypes of Apatosaurus louisae or Diplodocus carnegii (CM 3018 and 84, respectively; Wedel & Taylor, 2013). Diplodocid specimens preserving anterior neural spines actually all show the bifurcation to initiate posterior to CV 5 or 6, and thus not in the anterior elements. This variation was captured by our character 140. The only diplodocoid genera in which bifid neural spines definitely occur in anterior cervical vertebrae are Dicraeosaurus and Amargasaurus (Janensch, 1929a; Janensch, 1929b; Salgado & Bonaparte, 1991).

Presence of a median tubercle in bifurcated cervical and dorsal neural spines (Wilson, 2002). Although generally present in Flagellicaudata, some specimens do not show such a tubercle (e.g., Amargasaurus cazaui, or UW 15556). Also, the probable non-diplodocoid Australodocus does have a median tubercle, such that its presence could at most be interpreted as an ambiguous synapomorphy. Since it was not recovered as such by the present analysis, it was not included in the diagnosis.

Anterior dorsal vertebrae with divided centropostzygapophyseal laminae (Wilson, 2002). A divided centropostzygapophyseal lamina was only positively identified in mid- and posterior dorsal vertebrae, but not in anterior ones. Therefore, the character was restricted to mid- and posterior elements.

Height of sacral neural spines nearly four times length of centrum (Wilson, 2002). This ratio was redefined, and posterior dorsal vertebrae were included into the description (C282). The derived state was found as a synapomorphy of Diplodocimorpha under implied weights, but found to be invalid because it also occurs in other taxa within Neosauropoda (Table S108).

Anterior caudal neural arches with spinoprezygapophyseal lamina (sprl) on lateral surface of neural spine (Wilson, 2002). The extension of the caudal spinoprezygapophyseal lamina onto the lateral surface of the neural spine (318-1) is actually a diplodocimorph synapomorphy, because it is also present in rebbachisaurids (Sereno et al., 2007), but absent in Haplocanthosaurus (Hatcher, 1903).

Procoelous first caudal centrum (Wilson, 2002). The first caudal centrum is actually flat posteriorly in many flagellicaudatan specimens (e.g., CM 84, E Tschopp, pers. obs., 2011), and only more posterior elements develop a slight convexity, if at all. This trait (295-1) is thus not included as synapomorphic for any clade herein.

Dicraeosauridae Huene, 1927

Definition: All taxa more closely related to Dicraeosaurus than to Diplodocus (stem-based; Sereno, 1998).

Unambiguous synapomorphies:

1. The crista prootica is expanded laterally, forming a dorsolateral process (76-1; Salgado & Calvo, 1992; although not recovered by the present analysis, the only OTUs scored for the apomorphic state are Dicraeosaurus hansemanni and Amargasaurus cazaui. In Suuwassea emilieae, the crista prootica is broken, such that it could not be scored for this character. Therefore, a treatment of the expanded crista prootica as unambiguous synapomorphy of Dicraeosauridae is supported by our analysis).

2. Basal tubera narrower than occipital condyle (C83; Wilson, 2002. This synapomorphy was not found by our analysis because the state boundaries used herein (C83) do not allow identification of the lowest ratio (<1.3) as synapomorphic for Dicraeosauridae. However, the actual distribution of these ratios (Table S9) shows that a ratio of <1.0 only occurs in dicraeosaurids. An inclusion of this trait as a synapomorphy of Dicraeosauridae is thus supported).

3. Basipterygoid processes are narrowly diverging (<31°) (92-2; Wilson, 2002; also this apomorphy was not found as diagnostic for Dicraeosauridae, but the same accounts as for C76-1 above).

4. The area between the basipterygoid processes and parasphenoid rostrum forms a deep slot-like cavity that passes posteriorly between the bases of the basipterygoid processes (95-1; Upchurch, Barrett & Dodson, 2004; as above, although not found herein, the data matrix supports an inclusion in the list).

5. Subtriangular cross-sectional shape of the symphysis of the dentary, tapering sharply towards its ventral extreme (105-1; Whitlock & Harris, 2010; also this synapomorphy lacks among the recovered ones, but Suuwassea and Dicraeosaurus are the only OTUs positively scored for this state, thus supporting an addition to this list).

6. Presence of a tuberosity on the labial surface of the dentary, near the symphysis (106-1; Whitlock & Harris, 2010; the same accounts here as in C105-1 above).

Shared synapomorphies:

7. The width to height ratio of cervical vertebrae is less than 0.5 (128-0; Upchurch, Barrett & Dodson, 2004; although not found as synapomorphy by our analyses, the only taxa with whom this state is shared are outside Diplodocoidea. The phylogenetic distance is thus herein considered large enough for an inclusion of this state as a shared synapomorphy of Dicraeosauridae).

8. Mid-cervical neural spines are anteriorly inclined (169-1; Rauhut et al., 2005; also this trait was not recovered as synapomorphic for Dicraeosauridae in our analysis, although four OTUs in the clade were scored positively for it. The trait is shared with some apatosaurine specimens, but given that all dicraeosaurids, which could be scored for this character share the derived state, we include it in this list of shared synapomorphies).

9. Posterior cervical and anterior dorsal bifid neural spines are parallel to converging (211-1; Rauhut et al., 2005; as in the trait above, the analysis did not recover this synapomorphy even though all dicraeosaurid OTUs preserving bifurcate vertebrae were scored for the derived state. This state is shared with Kaatedocus, but phylogenetic distance is considered large enough for a treatment of this trait as a shared synapomorphy).

10. The height of posterior dorsal and/or sacral neural spines (not including arch) is more than 3 times centrum length (282-2; McIntosh, 1990a; this synapomorphy was not found by our analyses but still included because all dicraeosaurid OTUs scorable for this character show the derived state and the only non-dicraeosaurid with whom this features is shared is the derived rebbachisaurid Demandasaurus).

11. The position of the highest point of the femoral head is laterally shifted in anterior view, and lies above the main portion of the shaft (431-1).

12. Presence of a short transverse ridge on the anteromedial surface of the distal end of the tibia (443-1).

13. A ratio of mediolateral width of the astragalus to maximum anteroposterior length of less than 1.6 (452-1).

14. Metatarsal I is relatively gracile, proximal transverse width to greatest length is less than 0.8 (461-0).

15. Pedal phalanges III-1 and IV-1 are equally long to longer than wide (476-0).

16. The groove on the lateral surface of pedal unguals extends straight horizontally (477-1).

Previously suggested synapomorphies:

Premaxilla with anteroventrally orientated vascular grooves originating from an opening in the maxillary contact (Wilson, 2002). These grooves (2-1) are also present in some diplodocid specimens (see comments on C2). The identification of this trait as dicraeosaurid synapomorphy is thus questionable.

Frontal symphysis is fused in adult individuals (26-1; Salgado & Calvo, 1992). This feature is difficult to assess, because the ontogenetic sequence in the fusion of skull bones is not yet entirely understood. For example, Kaatedocus siberi has unfused frontals as well as parietals, but completely fused cervical vertebrae, including fusion of the ribs to the centrum (Tschopp & Mateus, 2013b). Herein, only clearly adult specimens were scored for this character, and dicraeosaurids do not appear to be the only taxa where left and right frontals fuse during ontogeny: also the potential Brachiosaurus skull USNM 5730 (E Tschopp, pers. obs., 2014) and Spinophorosaurus nigerensis (Knoll et al., 2012) are scored as possessing the derived state.

Frontal contributes to the margin of the supratemporal fenestra (reversal; Wilson & Sereno, 1998). Although this reversal (C34-0) occurs in Dicraeosaurus and Amargasaurus (Janensch, 1935; Salgado & Calvo, 1992), Suuwassea does not show any participation of the frontal in the supratemporal fenestra (ANS 21122, E Tschopp, pers. obs., 2011). Therefore, the present analysis was not able to recover this as synapomorphic for the entire clade Dicraeosauridae.

Presence of a postparietal foramen (66-1; Salgado & Calvo, 1992). A postparietal foramen occurs in a wide variety of sauropods, including some diplodocids (e.g., Kaatedocus siberi, Tschopp & Mateus, 2013b), and should thus not be regarded synapomorphic for Dicraeosauridae.

Supratemporal fenestra smaller than foramen magnum (71-1; Salgado & Calvo, 1992). This feature is also present in the rebbachisaurid Limaysaurus (Whitlock, 2011a). On the other hand, a large supratemporal fenestra occurs in the basal dicraeosaurid Suuwassea (ANS 21122, E Tschopp, pers. obs., 2011). Therefore, it remains unclear how to interpret the reduction in size (either as a diplodocoid or diplodocimorph synapomorphy with reversals, or as convergently acquired traits of Rebbachisauridae and Dicraeosauridae).

Ventrally directed prong on squamosal (Whitlock, 2011a). A ventrally directed process of the squamosal (58-1) is also present in some diplodocids (e.g., Apatosaurus louisae CM 11162, E Tschopp, pers. obs., 2011), and has a very similar morphology as in Dicraeosaurus (see comments on C58). An enlarged prong-like structure is only present in Amargasaurus (Salgado & Calvo, 1992), which does not allow an identification of this feature as synapomorphic for Dicraeosauridae.

The anterolateral corner of the tooth row is displaced labially (112-1; Whitlock & Harris, 2010). Originally described as potential dicraeosaurid synapomorphy, this condition is actually also present in the rebbachisaurid Nigersaurus (Sereno et al., 2007) and, in a more weakly developed form, in the skull AMNH 969, herein referred to Galeamopus sp.

‘Petal’ shaped posterior dorsal neural spines (294-1; Wilson, 2002). The peculiar ‘petal’ shape of dorsal, and sacral neural spines of dicraeosaurids is also present in rebbachisaurids, and could not be scored in the basal-most dicraeosaurids herein. This led to an identification of this feature as a rebbachisaurid synapomorphy under equal weights. If all dicraeosaurids could also be scored for the derived state, this might then be more parsimoniously interpreted as a diplodocimorph synapomorphy. It is therefore excluded from the list of dicraeosaurid synapomorphies.

Cervical vertebrae with longitudinal ridge on ventral surface (Sereno et al., 2007). The presence of a longitudinal ridge on the ventral surface of cervical centra is a plesiomorphic feature within sauropods, and also occurs in some diplodocid specimens (e.g., SMA 0004, YPM 429; Lull, 1919; Tschopp & Mateus, 2013b). Dicraeosaurids have well-developed keels in anterior cervical centra (159-0), shared with Shunosaurus (Zhang, 1988), but also with Galeamopus SMA 0011 (E Tschopp, pers. obs., 2012). The presence or absence of ventral ridges and keels is therefore too homoplastic to be used as a synapomorphic of any clade.

Lateral pleurocoels (i.e., deep, well-delimited, lateral fossae) are absent in mid- and posterior dorsal centra (252-0; Janensch, 1929a). Under implied weights, this trait was recovered as a synapomorphy of a clade within Dicraeosauridae, which includes the taxa Brachytrachelopan, Dicraeosaurus, and Amargasaurus. Given the basal dicraeosaurid position of Suuwassea and potentially Dystrophaeus, for which indications for the presence of pleurocoels in mid- and posterior dorsal centra are relatively strong, the absence of these pleurocoels cannot be interpreted as a synapomorphy of the entire Dicraeosauridae.

Anterior caudal centra with irregularly placed foramina on ventral surface (305-1; Harris, 2007). The presence of ventral foramina in anterior caudal vertebrae is herein recovered as a flagellacaudatan synapomorphy, because it also occurs in numerous diplodocids (e.g., Tornieria africana, Barosaurus lentus; Remes, 2006; YPM 429, E Tschopp, pers. obs., 2011).

Mid-caudal vertebral centra with longitudinal ridge located at mid-height of the lateral surface, centra hexagonal in anterior/posterior view (Whitlock, 2011a). Similar longitudinal ridges are also present in the mid-caudal vertebrae of Camarasaurus, as well as many apatosaurine specimens (Gilmore, 1925; Gilmore, 1936). Their presence (333-1) could thus only be interpreted as shared synapomorphy for Dicraeosauridae. Since it was not recovered as such, it is not included in the diagnosis herein.

Humerus with pronounced proximolateral corner (383-1; Wilson, 2002). This trait was recovered as neosauropod synapomorphy under implied weights, with a reversal in Diplodocidae. Because the definition of ‘pronounced’ is somewhat vague, Wilson’s (2002) interpretation of this character might have been different than ours. Our definition is explained and figured above (see comment on C383; Fig. 91).

Diplodocidae Marsh, 1884

Definition: All taxa more closely related to Diplodocus than to Dicraeosaurus (stem-based; Sereno, 1998).

Unambiguous synapomorphies:

1. Maxilla-quadratojugal contact broad (14-1; Rauhut et al., 2005; not recovered by the present analysis, it is still supported by the data matrix. The reason why it was not recovered is probably the low percentage of specimens preserving these two bones).

2. Antorbital fenestra with concave dorsal margin (20-1; Wilson, 2002; this trait was also not recovered as diplodocid synapomorphy herein, although supported by the specimens for which a scoring was possible. The reason is probably the same as that for the previous synapomorphy).

3. Posterior process of the prefrontal is hooked (25-1; Berman & McIntosh, 1978).

4. Mandible without strong coronoid eminence (108-1; Whitlock, 2011a; as in the previous characters, the low number of specimens preserving the mandible probably precluded an identification of this character as synapomorphy for Diplodocidae, although supported by the dataset).

5. Direct crown-to-crown occlusion absent (116-1; Wilson, 2002; yet another trait not found as synapomorphic, probably due to low percentage of preservation, but supported by the dataset).

6. The 14 to 15 cervical vertebrae (127-1; Huene, 1929).

Exclusive synapomorphies:

7. Preantorbital fenestra occupies at least 50% of the preantorbital fossa (17-1).

8. Medial margin of the prefrontal is curving distinctly medially at its anterior end to embrace the anterolateral corner of the frontal (23-1).

9. Ten dorsal vertebrae (224-2; Huene, 1929).

10. Presence of an accessory laminae in the region between posterior centrodiapophyseal lamina and posterior centroparapophyseal lamina of mid- and posterior dorsal vertebrae (259-1).

11. Anterior centrodiapophyseal lamina (acdl) of anterior caudal vertebrae is divided (314-1; Wilson, 2002).

12. Anterior and mid-caudal vertebrae bear ventrolateral ridges (329-1).

Shared synapomorphies:

13. Short mid- and posterior dorsal transverse processes (263-0).

14. Anterior caudal transverse processes with anteroposteriorly expanded lateral extremities (316-1).

15. The sprl and spol contact each other on anterior caudal neural spines (319-1; Wilson, 1999).

16. Absence of a pronounced proximolateral corner of the humerus (383-0).

17. Presence of a lateral bulge on the femur (428-1).

Ambiguous synapomorphies:

18. Presence of a short transverse ridge medially on the posterior side of the ventral ramus of the quadrate, close to the articular surface with the lower jaw (50-1).

19. Presence of an accessory, subvertical lamina in the pocdf of posterior cervical vertebrae, with a posteriorly facing free edge (199-1).

20. Roughened lateral aspect of the prdl of posterior cervical and anterior dorsal vertebrae (208-1).

21. Mid- and posterior dorsal parapophyses are located above the centrum, posterior to the anterior edge of the centrum (256-0).

22. Posterior dorsal postzygapophyses have oblique articular facets, which include an angle of almost 90° (275-1).

23. Anterior-most caudal centra bear large pneumatic fossae (297-1).

24. The prespinal lamina of anterior caudal neural spines has a thickened anterior rim (321-1).

25. The distal articular surface of the radius for the ulna is well developed, and bears one or two distinct longitudinal ridges (392-1).

26. Metatarsal I has a proximal transverse width to greatest length ratio of 0.8 or more (461-1).

Previously suggested synapomorphies:

Antorbital fenestra subequal to orbital maximum diameter (18-1; Wilson, 2002). The large antorbital fenestrae are recovered as diplodocoid synapomorphy herein (with a reversal in the possibly juvenile diplodocine CM 11255), because they also occur in Nigersaurus (Sereno et al., 1999; Sereno et al., 2007).

Prefrontal posterior process elongate (Wilson, 2002). Determination of the length of the posterior process of the prefrontal is highly influenced by the orientation of the skull roof, as shown previously. Taking this into account, an elongated posterior process of the prefrontal (24-1) is not present in all diplodocid specimens (e.g., it is absent in Kaatedocus siberi; Tschopp & Mateus, 2013b). This trait was thus excluded from the diagnosis of Diplodocidae.

No internarial bar (Upchurch, Barrett & Dodson, 2004). An internarial bar also appears to be absent in dicraeosaurids (Janensch, 1935; Harris, 2006b). It would thus more appropriately be interpreted as a flagellicaudatan synapomorphy. However, this character was not included in the present analysis, because the absence of an internarial bar is difficult to distinguish from incomplete preservation in most specimens.

Frontal contribution to dorsal margin of orbit roughly equal to contribution of the prefrontal (Whitlock, 2011a). Re-measuring the contribution of the frontal and prefrontal in various diplodocid skulls showed that variation occurs both within but also outside Diplodocidae (Table S5). Neither state can thus be confidently considered as synapomorphic for any clade.

Jugal forms substantial part of caudoventral margin of antorbital fenestra (40-1; Upchurch, 1998). The large contribution of the jugal to the antorbital fenestra was recovered as a diplodocoid synapomorphy herein, because Nigersaurus shows the same morphology seen in diplodocids (Sereno & Wilson, 2005).

An angle between the rostra1 and dorsal quadratojugal processes of 130° (46-1; Upchurch, Barrett & Dodson, 2004). A wide angle between rostral and dorsal processes of the quadratojugal also occurs in Nigersaurus (Sereno & Wilson, 2005), resulting in a recovery of this feature as diplodocoid synapomorphy herein.

Quadrate fossa shallow (51-0; Wilson, 2002). A shallow quadrate fossa was later found in the dicraeosaurid Suuwassea (Harris, 2006a), showing that this trait is not restricted to Diplodocidae. Consequently, it has here been found as a flagellicaudatan synapomorphy.

Squamosal-quadratojugal contact absent (56-1; Wilson, 2002). Tschopp & Mateus (2013b) showed that a contact between the squamosal and the quadratojugal was also absent in Suuwassea (contrary to Harris, 2006a). Therefore, the present trait was herein recovered as flagellicaudatan synapomorphy.

The distal end of the paroccipital process rounded and tongue-like (Upchurch, Barrett & Dodson, 2004). This character was not used in the present analysis because it was unclear what “tongue-like” precisely means. It was substituted by a character describing dorsoventral expansion towards the distal ends of the paroccipital processes (C68), which varies within Diplodocidae and thus does not qualify as a reliable synapomorphy.

The parasphenoid rostrum is a laterally compressed, thin spike lacking the longitudinal dorsal groove (Upchurch, Barrett & Dodson, 2004). A dorsal groove is actually present on many diplodocid parasphenoid rostra (e.g., CM 11161, E Tschopp, pers. obs., 2011). Transverse compression of the parasphenoid rostrum is also apparent in Camarasaurus (Madsen, McIntosh & Berman, 1995). Generally, diplodocid parasphenoid rostra are more spike-like, or dorsoventrally compressed, compared to Giraffatitan or Camarasaurus (Janensch, 1935; Madsen, McIntosh & Berman, 1995), but that is difficult to translate into a valid phylogenetic character, and was thus not used as such herein.

The ectopterygoid process of the pterygoid located below the antorbital fenestra (102-1; Upchurch, Barrett & Dodson, 2004). Such an anterior position of the ectopterygoid process is shared with rebbachisaurids (Whitlock, 2011a), and was thus recovered as a diplodocoid synapomorphy herein.

The ectopterygoid process of the pterygoid reduced, so that it cannot be seen below the ventral margin of the skull in lateral view (Upchurch, Barrett & Dodson, 2004). No such character was included in the present analysis. However, given the rareness of palatal complexes preserved in their true position, it remains doubtful if the analysis would have been capable to confidently resolve the distribution of this character state.

The breadth of the main body of the pterygoid at least 33% of pterygoid length (Upchurch, Barrett & Dodson, 2004). Given that only one disarticulated diplodocid pterygoid was available for direct study (SMA 0011), no character was included in the present analysis to test the distribution of this trait. Generally, diplodocid pterygoids do appear more elongate compared to non-diplodocid taxa, but only rarely measurements can be taken directly from the specimen. This condition was therefore not included in the diagnosis herein.

Cervical vertebrae with longitudinal sulcus on ventral surface (133-1; Upchurch, 1998). A ventral longitudinal sulcus covering the entire anteroposterior length of the cervical centrum rarely occurs in apatosaurines. Consequently, the sulcus was not recovered as a diplodocid synapomorphy herein.

Bifurcated centroprezygapophyseal lamina in cervical vertebrae, with a medial and a lateral ramus connecting to the zygapophysis (185-2; Wilson, 2002). Possibly because Supersaurus does not seem to have divided cprl, the current analysis did not recover this trait as a synapomorphy of Diplodocidae.

Posterior centroparapophyseal lamina of mid- and posterior dorsal neural arches present as single lamina (258-1; Wilson, 2002). Although recovered as a synapomorphy for Diplodocidae, this feature is herein treated as invalid because its distribution within the clade is ambiguous. In fact, all three states occur within Diplodocidae, and some dicraeosaurids also show state 1.

Posterior dorsal, sacral and anterior caudal neural spines rectangular through most of their length (294-0; Whitlock, 2011a). This state would represent a reversal to the plesiomorphic condition, but only if the derived state was recovered as diplodocimorph synapomorphy due to the shared derived condition in rebbachisaurids and dicraeosaurids. Because this was not the case in our analyses, also the reversal could not be found as synapomorphic for Diplodocidae.

A count of 70-80 caudal vertebrae (Upchurch, Barrett & Dodson, 2004). This character is difficult to score in a specimen-based phylogenetic analysis, because only very few specimens preserve reasonably complete caudal series. In the present analysis, for example, only the Apatosaurus louisae specimens CM 3018 and 3378 would positively confirm the presence of high counts of caudal vertebrae in diplodocids. Furthermore, indirect evidence for an elongated tail also comes from the rod-like distal caudal vertebrae in some dicraeosaurid specimens (e.g., ANS 21122, MB.R.4886, E Tschopp, pers. obs., 2011), as well as in Limaysaurus tessonei (Calvo & Salgado, 1995). The number of caudal vertebrae is thus not included in the diagnosis here.

Presence of diapophyseal laminae on anterior caudal vertebrae (Upchurch, 1998). This character has been divided in the present analysis, distinguishing between anterior (C313) and posterior diapophyseal laminae (C315). Apatosaurines, as well as Supersaurus, tend to have much broader posterior diapophyseal laminae compared to diplodocines, thus not qualifying to be scored as ‘distinct.’ On the other hand, well-developed anterior diapophyseal laminae also occur in dicraeosaurs. Therefore, the latter were recovered as flagellicaudatan synapomorphy, whereas distinct posterior diapophyseal laminae were found to diagnose Leinkupal + mdD.

Insertion of the M. iliofibularis on the fibula located above midshaft (448-1; Wilson & Sereno, 1998). In fact, insertion of this muscle on the fibula is located further distally in apatosaurines and Tornieria than in more derived diplodocines, as a detailed assessment showed (see above). The proximal location of the insertion is thus recovered as synapomorphic for Supersaurus + mdD herein.

An absence of a calcaneum (McIntosh, 1990b). The absence of a calcaneum as diplodocid synapomorphy is most probably a preservational artifact. As shown by Bonnan (2000), at least one pes of Diplodocus preserves a calcaneum (CM 30767), and personal observations in two putative apatosaurine pedes (CM 30766 and NHMUK R3215) reveal the probable presence of such an element in apatosaurines. Its absence is thus not included in the diagnosis of any clade.

Pedal phalanx I-1 having a proximoventral margin drawn out into a thin plate or heel that underlies the distal end of metatarsal I (Upchurch, Barrett & Dodson, 2004). The distribution of this trait is more complicated: it is also present in the non-diplodocid Turiasaurus and Cetiosauriscus stewarti (E Tschopp, pers. obs., 2011–2012), and absent in Apatosaurus louisae CM 3018 (Gilmore, 1936). Its presence would thus only qualify for an ambiguous synapomorphy, but was not recovered as such by the present analysis.

Pedal phalanx II-2 reduced in craniocaudal length and having an irregular shape (Upchurch, Barrett & Dodson, 2004). Whereas all included diplodocid specimens preserving this element show a reduced craniocaudal length in php II-2, the same is also present in Mamenchisaurus (Ouyang & Ye, 2002). Because no complete pes is known from any dicraeosaurid or rebbachisaurid, the true distribution of this trait cannot currently be assessed, and it is thus excluded from the updated diagnosis of Diplodocidae.

Apatosaurinae Huene, 1927

Definition: All taxa more closely related to Apatosaurus than to Diplodocus (stem-based; Taylor & Naish, 2005).

Unambiguous synapomorphies:

1. Cervical ribs projecting well beneath centrum, such that the length of the posterior process is subequal in length to the fused diapophysis/tuberculum (216-1, recovered as shared synapomorphy under equal weights, due to the diplodocine position of FMNH P25112).

Shared synapomorphies:

2. Absence of paired pneumatic fossae on the ventral surface of anterior cervical vertebrae (160-0).

3. Posterior centrodiapophyseal lamina (pcdl) and postzygodiapophyseal laminae (podl) of mid- and posterior cervical transverse processes do not meet anteriorly, such that the postzygapophyseal centrodiapophyseal fossa extends onto the posterior face of the transverse process (186-1).

4. Anterior process of posterior cervical ribs is reduced to a short bump-like process or absent (220-1).

5. Postspinal lamina or rugosity of anterior caudal neural spines terminates at or beneath the dorsal margin of the neural spine (324-0).

Ambiguous synapomorphies:

6. Rectangular coracoid outline (372-1; McIntosh, 1995).

Previously suggested synapomorphies:

To our knowledge, only one phylogenetic study is published recognizing an apatosaurine clade including more than just the genus Apatosaurus: Lovelace, Hartman & Wahl (2007) also recovered Supersaurus and Suuwassea as apatosaurine diplodocids, but did not provide a diagnosis for the clade. The current diagnosis is thus the first for Apatosaurinae based on a cladistic analysis.

Diplodocinae Marsh, 1884

Definition: All taxa more closely related to Diplodocus than to Apatosaurus (stem-based; Taylor & Naish, 2005).

Unambiguous synapomorphies:

1. Box-like basal tubera (82-1; although not recovered as synapomorphic, the only OTUs scored for the derived character are diplodocines. The reason why TNT was not able to recognize this feature as synapomorphic was probably the lack of skulls in the basal-most diplodocines).

2. Lateral surfaces of the posterior cervical neural spines are marked by a dorsoventrally elongate coel (204-1; also this synapomorphy was not recovered as such by TNT, probably due to the lack of posterior cervical vertebrae in basal diplodocines. However, the datamatrix supports an inclusion of this feature at least for Supersaurus + mdD, if not Diplodocinae).

Shared synapomorphies:

3. Articular surfaces of mid- and posterior cervical prezygapophyses are flat (180-0, even though not recovered as synapomorphy, the datamatrix supports an addition of this feature as shared synapomorphy of Diplodocinae, shared with Spinophorosaurus and Australodocus, if the latter is a titanosauriform).

4. Transition from ‘fan’-shaped to ‘normal’ caudal ribs occurs between Cd 6 and Cd 7, or more posteriorly (300-3 and 300-4).

5. The scapular acromial process that lies nearly at midpoint of the scapular body (364-1; although not found as a synapomorphy of Diplodocinae, the only diplodocid taxa with which this trait is shared are the Limaysaurinae. We therefore consider this trait as a shared synapomorphy of Diplodocinae).

6. A subtriangular proximal articular surface of the tibia (442-1; Harris, 2007).

Ambiguous synapomorphies:

7. Presence of triangular aliform processes on mid- and posterior dorsal neural spines, which do not project as far laterally as postzygapophyses (267-1).

8. A deeply excavated, triangular parapophyseal centrodiapophyseal fossa in posterior dorsal neural arches (273-1).

9. Caudal neural spines with triangular lateral processes (293-1).

10. Participation of the pubis in the acetabulum is subequal to larger than the one of the ischium (416-0).

11. Presence of an elongate muscle scar on the proximal end of the ischial shaft (421-1).

12. The dorsal/anterior surface of the metatarsal I is marked by several foramina (459-1).

Previously suggested synapomorphies:

Elongation index of mid-cervical vertebrae greater than 4.0 (Upchurch, 1998). State boundaries were changed herein in comparison to Upchurch (1998). However, a mean value of less than four occurs in several diplodocine specimens, and a value of 4.0 or greater was convergently acquired by various outgroup taxa (Table S21). The EI value of greater than 4.0 is thus excluded from the diagnosis of Diplodocinae.

Quadrangular anterior articular surface of anterior caudal centra (Wilson, 2002). There is a wide range of articular surface shapes in these elements, and it is difficult to describe them qualitatively or divide them into only two categories, as was done by Wilson (2002: circular versus quadrangular). Most diplodocine anterior caudal centra have a flat ventral edge of the anterior articular surface (e.g., Barosaurus lentus YPM 429; Lull, 1919), but this is accounted for in other characters (e.g., C296). The shape becomes gradually more quadrangular towards middle caudal vertebrae in Diplodocus (e.g., AMNH 223; Osborn, 1899), but not in Barosaurus, which retains its rounded lateral edges (e.g., AMNH 6341; E Tschopp, pers. obs., 2011). Although anterior caudal centra with flat ventral border can still be confidently assigned to Diplodocinae, more rounded centra cannot be excluded just based on this morphology. A quadrangular shape of the anterior face as proposed by Wilson (2002) should thus not be regarded a true synapomorphy of Diplodocinae.

Centrum length doubles over the first 20 caudal vertebrae (Wilson, 2002). The presence of caudal centra that almost double in length within the first 20 tail elements is not restricted to Diplodocinae. It is shared by the non-diplodocoid Cetiosauriscus stewarti (NHMUK R3078, E Tschopp, pers. obs., 2011), the rebbachisaurid Zapalasaurus bonapartei (Salgado, Carvalho & Garrido, 2006), the dicraeosaurid Suuwassea emilieae (Harris, 2006a) and the probable apatosaurine FMNH P25112 (Gilmore, 1936). It is therefore not considered to be a diplodocine synapomorphy herein.

Presence of a ventral longitudinal hollow in anterior and mid-caudal centra (330-1; Marsh, 1895). The sheer presence of such a hollow cannot be considered a diplodocine synapomorphy anymore, because it also occurs in a very shallow manner in some apatosaurines, and as a deep hollow in the rebbachisaurid Demandasaurus (Torcida Fernández-Baldor et al., 2011).

Middle caudal neural spines vertical (Wilson, 2002). Actually, the majority of diplodocine specimens preserving mid-caudal vertebrae have slightly posterodorsally directed neural spines (e.g., Diplodocus longus YPM 1920; Marsh, 1878). The only species with vertical mid-caudal neural spines is Diplodocus hallorum.

Updated diagnoses of valid diplodocid genera and species

The following diagnoses include autapomorphies found by the analysis as well as additional traits found to be unique at least within the respective higher-level clade (Apatosaurinae or Diplodocinae). Autapomorphies found only in one specimen, but not preserved in others, are marked by an asterisk. Referred specimens as well as localities and horizons only include information from the present analysis. Specific or generic identification of other specimens is often not done with enough detail (i.e., without phylogenetic analysis or accurate description of the material), such that earlier referrals require a reappraisal before definitely including them in the species lists. Geographical and temporal distribution of the genera and species proposed herein have thus to be regarded as smallest possible ranges.

Diplodocidae Marsh, 1884

Amphicoelias Cope, 1877a

Type and only referred species: Amphicoelias altus Cope, 1877a.

Invalid proposed species: Amphicoelias latus Cope, 1877a (= Camarasaurus); Amphicoelias fragillimus Cope, 1878 (nomen dubium).

Revised diagnosis: Amphicoelias cannot be diagnosed based on unambiguous autapomorphies at present. However, it can be distinguished from nearly all diplodocids by the very slender femur (RI <0.22; 427-0*, only shared with USNM 10865 within Diplodocidae). Furthermore, Amphicoelias is distinct from the majority of apatosaurines due to the presence of the the following local autapomorphies: (1) anteriorly displaced parapophyses in mid- and posterior dorsal vertebrae (256-0*); (2) posterior dorsal neural spines taper towards the summit (265-1*, only shared with the holotype specimen of Brontosaurus yahnahpin, Tate-001, among apatosaurines); and (3) posterior dorsal postzygapophyses almost horizontal, such that the two articular facets include a wide angle (275-0*). Amphicoelias can be excluded from Diplodocinae due to a mediolateral width of the femur that is subequal to the anteroposterior diameter (430-0*, only shared with CM 566 and Dicraeosaurus within Diplodocoidea). Finally, three more traits are shared with only a small number of diplodocine specimens: (1) amphicoelous posterior dorsal centra (270-0*, shared with SMA 0087); (2) a ventrally open, relatively shallow parapophyseal centrodiapophyseal fossa in posterior dorsal neural arches (273-0*, shared with Galeamopus SMA 0011); and (3) longer than wide bases of posterior dorsal neural arches (279-0*, shared with Galeamopus SMA 0011).

Comments: The characters initially used by Cope (1877a) to diagnose the genus are now known to be more widespread among sauropods, such as the amphicoelous dorsal centra (which still serve to distinguish Amphicoelias from most diplodocines), or the weak development of the greater trochanter on the femur. Osborn & Mook (1921) first recognized the extreme slenderness of the femur of Amphicoelias, compared to other sauropods. Wilson & Smith (1996) reported two autapomorphies for the skull, based on a second specimen referred to the genus. However, no detailed description nor figures of the material have yet been published, such that the validity of these traits as autapomorphic features for Amphicoelias are herein regarded questionable. The assignment of the specimen to Amphicoelias was mainly based on the circular cross section of the femur midshaft (Wilson & Smith, 1996), which has been recovered as autapomorphic herein as well. Upchurch, Barrett & Dodson (2004) proposed the unusual, slightly posterodorsal orientation of the posterior dorsal neural spine as an autapomorphy of the genus. Although characters were included in the present analysis to code for this morphology (C265 and 280), only one of them was found potentially useful to distinguish Amphicoelias from apatosaurines, because both are shared with specimens from both Apatosaurinae and Diplodocinae.

Locality and horizon: Cope Quarry 12, Garden Park Area, Fremont County, Colorado. Upper-most Brushy Basin Member, Morrison Formation (probably Tithonian). Dinosaur zone 4 (Turner & Peterson, 1999), Zone 6 (Foster, 2003).

Amphicoelias altus Cope, 1877a

Type specimen: AMNH 5764.

Referred specimens:-

Diagnosis, locality, and horizon as for genus.

Apatosaurinae Huene, 1927.

ApatosaurusMarsh, 1877a.

Type species: Apatosaurus ajax Marsh, 1877a.

Referred species: Apatosaurus louisae Holland, 1915a.

Invalid proposed species: Apatosaurus grandis Marsh, 1877a (= Camarasaurus grandis), A. laticollis Marsh, 1879 (nomen dubium; = A. louisae), A. minimus Mook, 1917 (non-diplodocoid neosauropod), A. alenquerensis Lapparent & Zbyszewski, 1957 (= Lourinhasaurus alenquerensis), A. yahnahpin Filla & Redman, 1994 (= Brontosaurus yahnahpin).

Revised diagnosis: Apatosaurus is diagnosed by the following autapomorphies: (1) presence of an accessory horizontal lamina in the spinodiapophyseal fossa of mid- and posterior cervical vertebrae, not connected to any surrounding lamina (187-1, unique within Apatosaurinae), (2) absence of a roughened lateral aspect of the prezygodiapophyseal lamina in posterior cervical and anterior dorsal vertebrae (208-0, unique within Diplodocidae), and (3) a straight scapular blade in lateral view (368-0, unique within Diplodocidae).

Comments: Berman & McIntosh (1978) proposed the relative positions of ectopterygoid and pterygoid as distinguishing character between the skulls CM 11161 and 11162. It was used as a phylogenetic character by Wilson (2002). However, there are only very few diplodocid skulls available, with the palatal complex articulated and complete. One of these is the juvenile probable Diplodocus skull CM 11255, which was interpreted to have a morphology more similar to the state in Apatosaurus than to Diplodocus (Whitlock, Wilson & Lamanna, 2010). However, recent studies appear to show that actually Apatosaurus CM 11162 has the same arrangement as Diplodocus CM 11161 (Whitlock & Lamanna, 2012). The distribution of this character thus seems very difficult to interpret. The fact that there are so few specimens preserving this area also decreases the phylogenetic value of this character. Therefore, until a more numerous sample of diplodocid skulls with articulated palatal complex is found, this feature should not be used in diagnoses. In general, autapomorphies previously proposed for the genus Apatosaurus most often describe a more inclusive clade in the present analysis, because two taxa previously included in the genus are actually forming their own genera (Brontosaurus and a third, new genus). These traits are thus not further discussed here.

Locality and horizon: Lakes’ quarry 10, near Morrison, Colorado and Dinosaur National Monument, Carnegie Quarry, Utah. Middle to upper part of the Upper Jurassic Morrison Formation, Late Kimmeridgian to Early Tithonian. Apatosaurine intervals 2 and 3 (Bakker, 1998); Dinosaur zone 3B upper (Turner & Peterson, 1999); Zone 5 (Foster, 2003).

Apatosaurus ajax Marsh, 1877a

Type specimen: YPM 1860.

Referred specimens: none

Revised diagnosis: A. ajax is diagnosed by the following autapomorphies: (1) a shallow, second fossa marks the quadrate shaft medially to the pterygoid flange (not the quadrate fossa) (52-1*, unique within Apatosaurinae), (2) a pit on the basioccipital, between the occipital condyle and the basal tubera (81-1*, unique within Apatosaurinae), (3) the longest axes of the basal tubera being oriented parallel to each other (87-0*, unique within Apatosaurinae), and (4) an elliptical depression between the lateral spinal lamina of caudal neural spines and the postspinal lamina (292-1*, unique within Apatosaurinae).

Comments: In the most recent revised diagnosis of the species, Upchurch, Tomida & Barrett (2004) proposed four more autapomorphies of the species, which are not found in the present analysis, due to the differing set of referred specimens to the species. Upchurch, Tomida & Barrett (2004) also recovered the specimens AMNH 460, NSMT-PV 20375, YPM 1840, and 1861 within A. ajax, whereas our analysis found the first three specimens more basally within Apatosaurinae, and YPM 1861 as Apatosaurus louisae. Wide cervical vertebrae, and low cervical neural spines, autapomorphies found by Upchurch, Tomida & Barrett (2004) to characterize A. ajax, are thus variable within Apatosaurinae. The dorsolateral process of the distal condyle of mt I, as well as the flange-like proximoventral process of php II-1might diagnose NSMT-PV 20375 instead.

Locality and horizon: Lakes’ Quarry 10, Morrison, Gunnison County, Colorado (YPM 1860). Upper-most Morrison Formation, Late Kimmeridgian to Early Tithonian. Apatosaurine interval 3 (Bakker, 1998); Dinosaur zone 3B upper (Turner & Peterson, 1999); Zone 5 (Foster, 2003).

Apatosaurus louisae Holland, 1915a

Syn. Apatosaurus laticollis Marsh, 1879

Type specimen: CM 3018.

Referred specimens: CM 3378, CM 11162, YPM 1861.

Revised diagnosis: A. louisae can be diagnosed by the following autapomorphies: (1) the prenantorbital fossa has indistinct margins (16-0*, unique within Diplodocoidea), (2) the lateral side of the dorsal portion of the lacrimal is flat (48-0*, unique within Flagellicaudata), (3) the distal end of the occipital process of the parietal curves laterally, such that the dorsolateral edge becomes concave distally (65-1*, unique within Diplodocidae), (4) the dorsal extension of the supraoccipital is high and vaulted, such that the dorsolateral edges are strongly sinuous (73-0*, unique within Apatosaurinae), (5) short basipterygoid processes with a ratio of length/basal transverse diameter of <4 (94-0*, unique witin Flagellicaudata), (6) the posterior wing of the atlantal neurapophyses is marked by a foramen (149-1*, unambiguous), (7) length increases considerably from vervical vertebrae 2 to 3, CV 3 is at least 1.3 times the length of CV 2 (155-1*, unique within Apatosaurinae), (8) pleurocoels of anterior and mid-cervical centra are pierced by one or two large, rounded foramina around centrum midlength (162-1*, unique within Apatosaurinae), (9) presence of a dorsoventrally elongate coel on anterior and mid-cervical neural spines (165-1*, unique within Apatosauridae), (10) posterior cervical prezygapophyses terminate well behind anterior ball (194-1, unique within Flagellicaudata), (11) absence of a subvertical lamina in the postzygapophyseal centrodiapophyseal fossa of posterior cervical vertebrae, with the free edge facing posteriorly (199-0, unique within Apatosaurinae), (12) presence of a rounded, subtriangular process on posterior cervical ribs, below the tuberculum (222-1, unambiguous), (13) an abrupt transition from bifurcate to single dorsal neural spines (234-1*, unique within Apatosaurinae), (14) DV 2 is longer than DV 1 (239-1, unique within Diplodocoidea), (15) pleurocoel on the first dorsal centra located posteriorly (240-1, unique within Apatosaurinae), (16) parapophysis of DV 3 lies mid-way between centrum and prezygapophyses (246-1, unique among Diplodocidae), (17) pleurocoels of anterior and mid-dorsal centra invade the neural arch pedicels (247-1*, unique within Apatosaurinae), (18) presence of an oblique ridge on the rib head of some dorsal ribs (283-1, unique within Apatosaurinae), (19) the transition from ‘fan’-shaped to ‘normal’ caudal ribs is between Cd 6 and Cd 7 (300-3, unique within Apatosaurinae), (20) anterior caudal neural spines are longer than wide (317-0*, unique within Apatosaurinae), (21) slightly bifid anterior caudal neural spines (326-1*, unique within Apatosaurinae), (22) last caudal ribs occur on Cd 14 (349-2, unique within Neosauropoda), (23) lateral surface of anterior chevrons is smooth (356-0*, unique within Apatosaurinae), (24) dorsoventral height to mediolateral width ratio of the proximal end of the metacarpal I is 1.8 or greater (401-1*, unique within Apatosaurinae), (25) the proximal articular surface of metacarpal V is significantly larger than the proximal articular surface of mc III and IV (403-1*, unique within Apatosaurinae), (26) metatarsal II bears a posterolateral process at the distal articular surface (469-1*, unique within Apatosaurinae), (27) the proximal articular surface of metatarsal IV is L- to V-shaped (470-0*, unique within Apatosaurinae), and (28) the proximal and ventral surfaces of pedal phalanx I-1 meet at approximately 90° (473-0*, unique within Diplodocoidea).

Comments: The list of autapomorphies is very long, but one has to keep in mind that many of these features are only present in the skull CM 11162 or the associated postcranial skeleton CM 3018. Furthermore, the skull CM 11162 is the only relatively complete apatosaurine skull in our analysis, and therefore, many of the proposed skull autapomorphies could actually also characterize the genus Apatosaurus, or all apatosaurines. Herein, we preferred a DELTRAN approach, resulting in an identification of these features as autapomorphies of the species A. louisae. In their revised diagnosis, Upchurch, Tomida & Barrett (2004) also proposed the presence of pneumatopores in the dorsal ribs as autapomorphic for A. louisae. However, pneumatized dorsal ribs were already figured by Marsh (1896) from the holotype of Brontosaurus excelsus, YPM 1980, and are also present in YPM 1981 (E Tschopp, pers. obs., 2011). The anterior restriction of the sacral ribs as interpreted to be present in the holotype specimen by Upchurch, Tomida & Barrett (2004) is herein regarded a questionable autapomorphy, because original matrix was left filling the space between the sacral ribs, which might thus be partly obliterated. Two more autapomorphies put forward by Upchurch, Tomida & Barrett (2004) are actually also present in other apatosaurine specimens: the heart-shaped anterior caudal centra, and the medially beveled glenoid surface of the scapula.

Locality and horizon: Dinosaur National Monument, Jensen, Uintah County, Utah (CM 3018, 3378, and 11162), and Lakes’ Quarry 10, Morrison, Gunnison County, Colorado (YPM 1861). Upper middle to upper-most Morrison Formation, Late Kimmeridgian to Early Tithonian. Apatosaurine intervals 2 and 3 (Bakker, 1998); Dinosaur zone 3B upper (Turner & Peterson, 1999); Zone 5 (Foster, 2003).

Brontosaurus Marsh, 1879

Syn.: Elosaurus Peterson & Gilmore, 1902, Eobrontosaurus Bakker, 1998.

Type species: Brontosaurus excelsus Marsh, 1879.

Referred species: Brontosaurus parvus (Peterson & Gilmore, 1902), Brontosaurus yahnahpin (Filla & Redman, 1994).

Invalid proposed species: Brontosaurus amplus Marsh, 1881 (= Brontosaurus excelsus).

Revised diagnosis: Brontosaurus can be diagnosed by the following autapomorphies: (1) a longer than wide base of posterior dorsal neural spines (279-0, unique among Apatosaurinae), (2) the area on the scapula posterior to the acromial ridge and the distal blade is excavated (365-0, unique among Apatosaurinae), (3) the acromial edge of the scapular blade bears a rounded expansion at its distal end (367-1, unique among Apatosaurinae), (4) the ratio of the proximodistal length/transverse breadth of the astragalus is 0.55 or greater (451-1, unique among Apatosaurinae).

Locality and horizon: various sites in Utah and Wyoming, USA. Middle to Upper Morrison Formation, Late Kimmeridgian to Early Tithonian. Dinosaur zone 3B upper (Turner & Peterson, 1999), Zone 5 (Foster, 2003).

Brontosaurus excelsus Marsh, 1879

Syn. Brontosaurus amplus Marsh, 1881.

Type specimen: YPM 1980.

Referred specimens: YPM 1981.

Revised diagnosis: Brontosaurus excelsus can be diagnosed by the following autapomorphies: (1) absence of a median tubercle in posterior cervical and anterior dorsal, bifid neural spines (210-0*, unique among Diplodocidae), (2) orientation of the tuberculum of mid-dorsal ribs follows the straight direction of the rib shaft (285-1*, unique among Apatosaurinae), (3) the posterior end of mid- and posterior caudal neural spine summits lies more or less straight above the postzygapophyses (343-1*, unique among Apatosaurinae), (4) the ratio of iliac blade height above the pubic peduncle to its anteroposterior length is 0.40 or greater (405-1*, unique among Apatosaurinae), (5) the highest point on dorsal margin of the ilium lies anterior to the base of the pubic process (410-1*, unique among Apatosaurinae), (6) presence of a large nutrient foramen opening on midshaft anteriorly on the femur (434-1*, unique among Apatosaurinae), (7) absence of a laterally directed ventral shelf on the astragalus, which underlies the distal end of the fibula (455-1*, unique among Apatosaurinae).

Comments: The autapomorphies proposed for ‘Apatosaurus’ excelsus by Upchurch, Tomida & Barrett (2004) are questionable. Cervical ribs that terminate in front of the posterior end of the centrum are widespread among Diplodocoidea, and are recovered as synapomorphic for that clade herein. The ventromedially projecting process on the anterior end of the cervical ribs is here reinterpreted as shortened anterior process of the cervical rib. The spine summits in anterior dorsal vertebrae are actually longer than wide (Ostrom & McIntosh, 1966: plates 17 and 18), and the slight medial widening is due to the presence of a medial ridge on the metapophyses, which is also present on other apatosaurine specimens (e.g., CM 3018, UW 15556; Gilmore, 1936).

Locality and horizon: Reed’s Quarries 10 and 11, Como Bluff, Albany County, Wyoming. Middle (Bakker, 1998) to upper (Foster, 1998) Morrison Formation, Late Kimmeridgian to ?Early Tithonian.

Brontosaurus parvus (Peterson & Gilmore, 1902)

Syn. Elosaurus parvus (Peterson & Gilmore, 1902).

Type specimen: CM 566.

Referred specimens: UW 15556 (previously CM 563), BYU 1252-18531 (provisionally).

Revised diagnosis: Brontosaurus parvus is diagnosed by the following autapomorphies: (1) unbifurcated cervical neural spines expanded laterally towards their summit in anterior/posterior view (141-1, unique among Apatosaurinae), (2) the axial neural spine is restricted anterior to the postzygapophyseal facets (153-2*, unique among Apatosaurinae), (3) posterior cervical vertebrae have an accessory lateral lamina connecting the postzygodiapophyseal and spinoprezygapophyseal laminae (197-1, unique among Apatosaurinae), (4) the base of the notch between the metapophyses of anterior, bifid dorsal vertebrae is narrow and V-shaped (244-1, unique among Apatosaurinae), (5) the height above the postzygapophyses of mid-dorsal neural arches to the height below (pedicel) is less than 2.1 (249-1, unique among Apatosaurinae), (6) mid- and posterior dorsal transverse processes develop a distinct dorsal bump or spur (264-1, unique among Apatosaurinae, not developed in the small juvenile CM 566), (7) greatly reduced spinoprezygapophyseal laminae in posterior dorsal vertebrae (274-0, unique within Diplodocoidea), (8) the ventral surface of anterior caudal centra is without irregularly placed foramina (305-0*, unique among Apatosaurinae), (9) and cross-sectional shape of the femur is subround (430-0, unique among Apatosaurinae).

Comments: In their revised diagnosis of ‘Apatosaurus’ parvus, Upchurch, Tomida & Barrett (2004) further mentioned wider than high posterior dorsal centra, a right angle between acromial ridge and scapular blade, differences in length of the ulnar proximal branches, a constriction in the distal half of mc III, and subequal width and depth of the distal articular surface of mc V. Wider than high dorsal centra are also present in NSMT-PV 20375 (Upchurch, Tomida & Barrett, 2004), an almost right angle between acromial ridge and distal blade occur in A. ajax as well as in “Eobrontosaurus” yahnahpin (Filla & Redman, 1994), and different lengths of the ulnar branches also mark A. ajax (Table S45). The characters from the manus could not have been positively identified in the specimens included, and were thus omitted from the revised diagnosis.

Locality and horizon: Sheep Creep Quarry E, Albany County, Wyoming, and possibly Mill Canyon Quarry, Moab quarry, Utah. Middle Morrison Formation, probably Late Kimmeridgian. Dinosaur zone 3B lower (Turner & Peterson, 1999), Zone 4 (Foster, 2003).

Brontosaurus yahnahpin (Filla & Redman, 1994)

Syn. Apatosaurus yahnahpin Filla & Redman, 1994; Eobrontosaurus yahnahpin (Filla & Redman, 1994).

Type specimen: Tate-001.

Referred specimens:-

Revised diagnosis: Brontosaurus yahnahpin can be diagnosed by the following autapomorphies: (1) the medial surface of anterior dorsal, bifid neural spines is gently rounded transversely (245-0*, unique within Apatosaurinae), (2) mid- and posterior dorsal neural spines narrow dorsally to form a triangular shape in lateral view, with the base approximately twice the width of the dorsal tip (265-1*, unique among Apatosaurinae), (3) absence of a thickened anterior rim of anterior caudal prespinal lamina (321-0*, unique among Apatosaurinae), (4) a rounded anteroventral margin of the coracoid (372-0*, unique among Apatosaurinae), (5) the distal breadth of the radius is less than 1.8 times larger than midshaft breadth (394-0*, unique among Apatosaurinae), (6) a ratio of the longest metacarpal to radius length of 0.40 or greater (399-1*, unique among Diplodocoidea), (7) metatarsal I is as long or longer than metatarsal V (458-0*, unique among Apatosaurinae), and (8) the distal articular surface of the metatarsal I being perpendicular to the axis of the shaft (462-1*, unique among Flagellicaudata).

Comments: Bakker (1998) mentioned three more diagnosing features: long cervical ribs, distal scapular blade expanded, and coracoid suture at right angle with the long axis of the scapular blade. The presence of long cervical ribs could not be confirmed based on the available pictures of the type specimen. The distally expanded scapular blade is actually shared with many apatosaur specimens (e.g., CM 3018, UW 15556, Gilmore, 1936). The unexpanded state is primarily based on the type specimen of Apatosaurus ajax, YPM 1860, but personal observations showed that the edges of the distal end are broken, and that the true expansion can therefore not be assessed in its entirety. The angle between the coracoid articulation and the distal blade, measured from photographs, is 74° (Table S40). Even if that should be wrong, the specimen described by Upchurch, Tomida & Barrett (2004), NSMT-PV 20375 shows an almost right angle, which would thus impede an interpretation as autapomorphy for Brontosaurus yahnahpin.

Locality and horizon: Bertha Quarry, Como Bluff, Albany County, Wyoming. Lower Morrison Formation, Kimmeridgian. Apatosaurine interval 1 (Bakker, 1998), Dinosaur zone 2 (Turner & Peterson, 1999), Zone 2 (Foster, 2003).

Diplodocinae Marsh, 1884

DiplodocusMarsh, 1878.

Syn. Seismosaurus Gillette, 1991

Type species: Diplodocus carnegii Hatcher, 1901 (suppressing the D. longus Marsh, 1878, see above).

Referred species: Diplodocus hallorum (Gillette, 1991).

Invalid proposed species: Diplodocus longus Marsh, 1878 (nomen dubium, previous type species, case to ICZN in preparation to propose D. carnegii as substitute), D. lacustris Marsh, 1884 (nomen dubium), D. hayi Holland, 1924 (= Galeamopus hayi).

Revised diagnosis: Diplodocus can be diagnosed by the following autapomorphies: (1) base of posterior dorsal neural spines anteriorly inclined (280-1, unique within Diplodocinae), (2) transition from ‘fan’-shaped to ‘normal’ caudal ribs occurs between Cd 7 and Cd 8 or more posteriorly (300-4, unique among Diplodocidae), (3) pneumatopores of anterior caudal centra persist until caudal 16 or more posteriorly (308-1, unique among Diplodocoidea), (4) trapezoidal articular surfaces in mid-caudal centra (334-2, unique among Flagellicaudata), (5) the last caudal ribs occur on Cd 18 or more posteriorly (349-4, unambiguous), (6) the ratio of iliac blade height above the pubic peduncle to its anteroposterior length is 0.40 or greater (405-1, unique among Diplodocinae), and (7) the proximal end of the fibula bears an anteromedially directed crest, which extends into a notch behind the cnemial crest of the tibia (447-1, unique among Diplodocinae).

Comments: Whitlock (2011a) proposed three cranial traits as autapomorphies of Diplodocus: a well-defined preantorbital fossa, the pterygoid that lies medial to the ectopterygoid, and the anteriorly inclined, procumbent teeth. Because no skull can be definitely attributed to Diplodocus, these suggestions are questionable. Furthermore, distinct preantorbital fossae, and procumbent teeth are also present on other diplodocine taxa (e.g., Galeamopus, Kaatedocus), and the relative positions of the pterygoid and ectopterygoid are not established with enough certainty to use it as diagnostic character (see above). Upchurch, Barrett & Dodson (2004) also defined Diplodocus solely based on cranial traits, most of which are actually shared with other diplodocine species that were not described or recognized at the time (Galeamopus, Kaatedocus). Wilson (2002) proposed the anteriorly expanded femoral distal condyles as autapomorphic for Diplodocus, as shared characteristic with advanced titanosauriforms. However, although the distal condyles are accompanied anteriorly by two distinct vertical ridges, the articular surface does not extend onto them as in Rapetosaurus krausei FMNH PR 2209, for example (Curry Rogers, 2009).

Locality and horizon: various sites in Colorado, New Mexico, Utah, and Wyoming. Middle Morrison Formation, probably Late Kimmeridgian. Apatosaurine interval 2 (Bakker, 1998), Dinosaur zones 3A to 3B upper (Turner & Peterson, 1999), Zones 3 to 5 (Foster, 2003).

Diplodocus carnegii (Hatcher, 1901)

Syn. Diplodocus carnegiei (misspelling)

Type specimen: CM 84.

Paratype: CM 94.

Referred specimens:-

Revised diagnosis: Diplodocus carnegii is diagnosed by the following autapomorphies: (1) axis has a postspinal lamina (152-1*, unique within Diplodocidae), (2) absence of a prespinal lamina in anterior cervical vertebrae (161-0*, unique within Diplodocinae), (3) spinopostzygapophyseal laminae (spol) of posterior dorsal neural arches divided near the postzygapophyses (277-1, unique among Flagellicaudata), (4) presence of a large nutrient foramen opening at midshaft anteriorly on femur (434-1*, unique among Diplodocinae), (5) metatarsal I to metatarsal V proximodistal length ratio of 1.0 or greater (458-0*, unique among Diplodocinae), and (6) slender metatarsal II (mean proximal and distal transverse breadth/maximum length <0.53) (466-0*, unique among Diplodocoidea).

Comments: Hatcher (1901) proposed two different characters to distinguish D. carnegii from D. longus: shorter cervical ribs, and more posteriorly directed caudal neural spines. However, comparisons were not based on the holotype of D. longus, but on two referred specimens (USNM 4712 and AMNH 223), which are now known not to belong to D. longus: the cervical vertebra Hatcher (1901) mentions (USNM 4712) actually has apatosaurine affinities (Hatcher, 1903), whereas the specimen AMNH 223, on which Hatcher (1901) based his comparisons, is herein interpreted to belong to Diplodocus hallorum. The short cervical ribs are widespread among Diplodocinae, and do thus not qualify as species autapomorphy. Caudal neural spine orientation is one of the main features distinguishing D. carnegii from D. hallorum, but the vertical spines from the latter species are herein found to be the derived state, such that the more posteriorly inclined spines in D. carnegii cannot be used to diagnose the species.

Locality and horizon: Sheep Creek Quarries D (CM 94) and D(3) (CM 84), Albany County, Wyoming. Middle Morrison Formation, Late Kimmeridgian. Dinosaur zone 3B lower (Turner & Peterson, 1999), Zone 4 (Foster, 2003).

Diplodocus hallorum (Gillette, 1991)

Syn. Seismosaurus hallorum, Seismosaurus halli.

Type specimen: NMMNH 3690.

Referred specimens: AMNH 223, DMNS 1494, USNM 10865.

Revised diagnosis: Diplodocus hallorum can be diagnosed by the following autapomorphies: (1) dorsal end of the postspinal lamina of single dorsal neural spines concave transversely (234-1, unique among Diplodocoidea), (2) mid-caudal neural arches are situated on the anterior half of the centrum (337-1, unique among Diplodocoidea), (3) vertical mid-caudal neural spines (340-1, unambiguous), (4) posterior end of mid- and posterior caudal neural spine summits lies more or less straight above the postzygapophyses (343-1, unique among Diplodocinae), (5) posterior caudal prezygapophyses project beyond the anterior edge of the centrum (345-1*, unique among Flagellicaudata), (6) presence of distinct fossae on the medial surfaces of the proximal branches of middle chevrons (357-1, unique among Diplodocinae), (7) a gracile femur (robustness index (sensu Wilson & Upchurch, 2003) <0.22) (427-0*, unique among Diplodocinae), and (8) the groove on the lateral surface of pedal unguals extends straight horizontally (477-1*, unique among Diplodocinae).

Comments: Lucas et al. (2006) in their taxonomic reappraisal of Seismosaurus hallorum proposed two more characters that distinguish the type specimen of D. hallorum from other species of Diplodocus: a more robust pubis, and paddle-shaped distal blades of the chevrons. Whereas the first is difficult to quantify and is thus provisionally omitted from the present diagnosis, the paddle shape of the chevrons is partly included in the character coding the posterior expansion of the chevron blade (C355), which is not present in the other specimens referred to D. hallorum. The specific chevron shape of NMMNH 3690 is thus herein regarded as individual variation.

Locality and horizon: Seismosaurus Quarry, Sandoval County, New Mexico (NMMNH 3690), Dinosaur National Monument Quarry, Uintah County, Utah (DMNS 1494, USNM 10865), and AMNH 223 Quarry, Como Bluff, Albany County, Wyoming (AMNH 223). Middle Morrison Formation, Late Kimmeridgian. Apatosaurine interval 2 (Bakker, 1998), Dinosaur zones 3B lower to upper (Turner & Peterson, 1999), Zones 4 to 5 (Foster, 2003).

Barosaurus (Marsh, 1890)

Type and only species: Barosaurus lentus (Marsh, 1890).

Invalid proposed species: Barosaurus affinis (Marsh, 1899) (nomen dubium), Barosaurus gracilis (Russell, Béland & McIntosh, 1980) (nomen nudum).

Revised diagnosis: Barosaurus can be diagnosed by the following autapomorphies: (1) absence of a short transverse ridge medially on the posterior side of the ventral ramus of the quadrate, close to the articular surface with the lower jaw (50-0*, unique among Diplodocidae), (2) pleurocoel not extending onto parapophysis in anterior cervical vertebrae (158-1*, unique among Diplodocidae), (3) elongation index of posterior cervical vertebrae (without anterior condyle) greater than 2.6 (192-2*, unique among Diplodocoidea), (4) an anterior projection on the prdl of posterior cervical, or anterior and mid-dorsal vertebrae, right lateral to the prezygapophysis (213-1, unique among Diplodocoidea), and (5) anterior dorsal centra without a ventral keel (242-0, unique among Diplodocinae).

Comments: Whitlock (2011a) does not list any autapomorphies for Barosaurus. McIntosh (2005) states four more diagnosing features for the genus: bifurcation of cervical neural spines restricted to the posterior half of the neck, summits of caudal neural spines undivided, a proportionally shorter tail, and a less prominent ventral hollow in anterior and mid-caudal centra. However, all of these traits represent the basal diplodocid morphology and are shared, e.g., with Kaatedocus or Supersaurus (Lovelace, Hartman & Wahl, 2007; Tschopp & Mateus, 2013b). Upchurch, Barrett & Dodson (2004) suggested an additional autapomorphy: the parapophysis of DV 2 is situated at the bottom of the centrum. Such a low position of the parapophysis is also present in DV 2 of Galeamopus SMA 0011, and can thus not be regarded diagnostic for Barosaurus.

Locality and horizon: various sites in South Dakota, Utah, and Wyoming. Lower to middle Morrison Formation, Kimmeridgian. Apatosaurine intervals ?1 to 2 (Bakker, 1998), Dinosaur zones 2 to 3B upper (Turner & Peterson, 1999), Zones 2 to 5 (Foster, 2003).

Barosaurus lentus Marsh, 1890

Type specimen: YPM 429.

Referred specimen: AMNH 6341.

Revised diagnosis: Barosaurus lentus can be diagnosed by the following autapomorphies: (1) cervical vertebrae pierced by a foramen on the dorsal side of the postzygodiapophyseal lamina, just anterior to the base of the neural spine process (137-1, unique among Diplodocoidea, when assuming titanosauriform affinities of Australodocus), (2) EI (cervical centrum length, excluding condyle, divided by posterior centrum height) of posterior cervical vertebrae is higher than 2.6 (192-2, unique among Diplodocoidea), (3) posterior cervical postzygapophyses terminate in front of the posterior edge of the centrum (200-1, unique within Diplodocinae), (4) nine dorsal vertebrae (224-3*, unambiguous), (5) the anterior-most caudal neural spine height (not including the arch) is 1.5 times the centrum height or more (302-1, unique among Diplodocidae), (6) anterior caudal neural spines without a thickened anterior rim of the prespinal lamina (321-0*, unique among Diplodocinae), (7) the articular surface of mid-caudal centra has a flat ventral margin but rounded lateral edges (334-3, unique among Diplodocidae), (8) last caudal ribs occur on Cd 15-17 (349-3, unique among Diplodocoidea), (9) position of the highest point of the femoral head is laterally shifted, above the main portion of the shaft in anterior view (431-1, unique among Diplodocinae), (10) mediolateral width of the astragalus to its maximum anteroposterior length ratio is less than 1.6 (452-1*, unique among Diplodocinae), and (11) the depth of the ventral hollow increases from anterior to posterior caudal centra (the present trait could not be assessed in the current analysis, but is provisionally included in the diagnosis of Barosaurus lentus following Upchurch, Barrett & Dodson, 2004).

Comments: This diagnosis also includes features that are developed differently in the other two specimens referred to Barosaurus (AMNH 7535, CM 11984). Therefore, some of the proposed diagnostic traits for B. lentus might not stand once more detailed studies of these or other potential B. lentus specimens are published, and more specimens are definitely referred to the species.

Locality and horizon: Piedmont Butte, Meade County, South Dakota (YPM 429), Dinosaur National Monument Quarry, Uintah County, Utah (AMNH 6341). Middle to Upper Morrison Formation, late Kimmeridgian to early Tithonian. Apatosaurine interval 2 (Bakker, 1998), Dinosaur zone 3B upper (Turner & Peterson, 1999), Zone 5 (Foster, 2003).

Tornieria Sternfeld, 1911

Type and only species: Tornieria africana Fraas, 1908. The species was originally assigned to Gigantosaurus africanus Fraas, 1908.

Invalid proposed species: Tornieria robustus Fraas, 1908 (=Janenschia robusta).

Revised diagnosis: Tornieria is diagnosed by the following autapomorphies: (1) mid- and posterior cervical neural arches have centroprezygapophyseal lamina that are dorsally divided, resulting in a lateral and medial lamina, the medial lamina being linked with the interprezygapophyseal lamina and not with the prezygapophysis (185-1*, unique within Diplodocidae), (2) the base of the notch between the metapophyses of anterior, bifid dorsal vertebrae is wide and rounded (244-0*, unique among Diplodocinae), (3) a straight posterior border of the sternal plate (377-1*, unique among Neosauropoda), (4) ratio of the pubic articulation of the ischium to the anteroposterior length of the pubic pedicel of the ischium is 1.5 or greater (420-1, unique among Diplodocinae), and (5) distal femoral condyles expand onto the anterior portion of the femoral shaft (439-1*, unique among Diplodocidae).

Comments: Whitlock (2011a) listed a single autapomorphy for the genus: the absence of a ventral hollow in anterior and mid-caudal centra. Contrary to Whitlock (2011a) ventral hollow is present in the preserved caudal vertebrae of both specimens included herein (Remes, 2006). In his revision of Tornieria, Remes (2006) proposed additional autapomorphies: frontal forms the entire dorsal margin of the orbit, prefrontal with a short posterior process, elongate cervical vertebrae, relatively long anterior caudal vertebrae, pleurocoel located on the upper third of the caudal centra, caudal transverse processes situated high on the centrum, caudal neural spines single, and lacking lateral processes, the distal blade of the scapula is only slightly expanded, unequal lengths of the proximal ulnar processes, robust ischial shaft, and a low tibia to femur length ratio. The traits of the frontal and prefrontal were later shown to be present in Kaatedocus as well (Tschopp & Mateus, 2013b). Elongate cervical vertebrae developed several times within Diplodocinae (e.g., Barosaurus, Supersaurus; McIntosh, 2005; Lovelace, Hartman & Wahl, 2007). Centrum length increases from anterior-most towards middle caudal vertebrae in all diplodocines, making relative length a serially variable character. It was thus not included in the present analysis, and a detailed assessment of the relative position of the anterior caudal vertebrae in the Tornieria specimens would be needed before including relative centrum length as diagnosing trait for the genus. The position of the pleurocoel in the preserved anterior-most caudal vertebra of the holotype individual (SMNS 12141a) does not appear to be restricted to the upper third (Remes, 2006: Fig. 4C). Pneumatic foramina are dorsally located in the referred caudal vertebrae from trench dd (MB.R.2956 to MB.R.2958; Remes, 2006), but since this trait appears different in the holotype, it should not be used in a diagnosis. The same accounts for the dorsal location of the transverse processes, which is most probably influenced by the position of the pleurocoel. Single caudal neural spines without lateral processes can only be observed in the referred caudal vertebrae, which were not included in the present analysis. However, these traits also occur in other diplodocine species, and are thus not reliable characters to distinguish Tornieria. A slight expansion of the scapular blade as well as robustness of the ischial shaft are difficult to quantify, but ratios do not appear to be significantly different from other diplodocine taxa. Unequally long ulnar proximal processes are shared with Galeamopus SMA 0011 (Table S45), as is the low tibia to femur ratio (Table S53).

Locality and horizon: localities A and k, Upper Saurian Beds, Tendaguru, District of Lindi, Tanzania. Tithonian.

Tornieria africana (Fraas, 1908)

Type specimen: SMNS 12141a, 12145a, 12143, 12140, and 12142. The individual also contains the specimens SMNS 12145c, MB.R.2672, 2713, and 2728 (Remes, 2006).

Referred specimens: MB.R.2386, 2572, 2586, 2669, 2673, 2726, 2730, 2733, 2913, and 3816 (all belonging to a single individual; Heinrich, 1999; Remes, 2006).

Diagnosis, locality, and horizon as for the genus.

Supersaurus Jensen, 1985

Syn. Dystylosaurus Jensen, 1985; Ultrasauros Olshevsky, 1991; Dinheirosaurus Bonaparte & Mateus, 1999.

Type species: Supersaurus vivianae Jensen, 1985.

Referred species: Supersaurus lourinhanensis (Bonaparte & Mateus, 1999).

Revised diagnosis: Supersaurus can be diagnosed by the following autapomorphies: (1) the ventral surface of mid- and posterior cervical vertebrae bears paired pneumatic fossae, separated by a ventral midline keel (176-1, unique among Diplodocinae), (2) the lateral edge of mid- and posterior cervical vertebrae, posterior to the parapophysis is marked by a deep groove extending anteroposteriorly along the edge (177-1, unique among Diplodocinae), (3) mid-dorsal neural spines bear an oblique accessory lamina that connects the postspinal lamina with the spinopostzygapophyseal lamina (251-1, unambiguous), and (4) dorsal ribs have pneumatopores (284-1, unique among Diplodocinae).

Comments: Lovelace, Hartman & Wahl (2007) listed several additional diagnosing traits for Supersaurus: elongate cervical vertebrae, an extreme narrowing of the ventral surface of cervical centra, well-developed parallel keels that mark the ventral surface of cervical centra, lateral pneumatopores on cervical centra small, located within a shallow coel, anterior dorsal vertebrae with a ventral keel, tall posterior dorsal neural spines, relatively low posterior dorsal neural arch, and a dorsally expanded scapular blade. Most of these traits are actually shared with other diplodocine species: the elongate cervical vertebrae (e.g., Tornieria), the well-developed parallel keels (herein called posteroventral flanges), the restricted and small lateral pneumatic foramina of cervical vertebrae (e.g., Galeamopus SMA 0011), the ventral keel in anterior dorsal centra, the low dorsal neural arches, the tall dorsal neural spines (typical for diplodocids in general), as well as the dorsally expanded scapular blade (e.g., Galeamopus). The extreme narrowing of the ventral surface of cervical centra is herein interpreted as a consequence of the centrum elongation, because a narrowing is generally seen relative to the centrum length.

Locality and horizon: Colorado and Wyoming, USA, and Lourinhã, Portugal. Middle Morrison Formation, and Amoreira-Porto Novo Member, Lourinhã Formation, Late Kimmeridgian to ?Early Tithonian. Dinosaur zone 3B lower (Turner & Peterson, 1999), Zone 4 (Foster, 2003).

Supersaurus vivianae Jensen, 1985

Syn. Dystylosaurus edwini Jensen, 1985; Ultrasauros macintoshi (Jensen, 1985).

Type specimen: BYU 12962. The holotypic individual probably also includes the specimens BYU 4503, 4839, 9024-25, 9044-45, 9085, 10612, 12424, 12555, 12639, 12819, 12861, 12946, 13016, 13018, 13981, 16679, and 17462 (Lovelace, Hartman & Wahl, 2007).

Referred specimens: WDC DMJ-021.

Revised diagnosis: Supersaurus vivianae can be diagnosed by the following autapomorphies: (1) cervical epipophyses reduced to absent (138-0, unique among Diplodocinae), (2) the spinoprezygapophyseal lamina of anterior and mid-cervical vertebrae is continuous as a lamina (163-0*, unique among Diplodocinae), (3) spinoprezygapophyseal laminae in single dorsal neural spines separate along their entire length (231-0, unique among Diplodocoidea), (4) presence of an infradiapophyseal pneumatopore between the acdl and the pcdl of mid- and posterior dorsal neural arches (262-1*, unique among Diplodocinae), (5) opisthocoelous posterior dorsal centra (270-2, unique among Diplodocoidea), (6) a ‘crus’ bridging the haemal canal is present in all chevrons (351-1*, unique among Neosauropoda), (7) an angle between the acromial ridge and the distal blade greater than 81° (362-2*, unique among Diplodocinae), (8) a widely expanded distal end of the scapular blade (at least 2 times the narrowest width of the shaft in lateral view; 371-0*, unique among Diplodocinae), and (9) the highest point on dorsal margin of the iliac blade lies anterior to the base of the pubic process (410-1*, unique among Diplodocinae).

Locality and horizon: Dry Mesa Quarry, Montrose County, Colorado, and Jimbo Quarry, Converse County, Wyoming. Middle Morrison Formation, Late Kimmeridgian to ?Early Tithonian. Dinosaur zone 3B lower (Turner & Peterson, 1999), Zone 4 (Foster, 2003).

Supersaurus lourinhanensis (Bonaparte & Mateus, 1999)

Syn.: Dinheirosaurus lourinhanensis Bonaparte & Mateus, 1999.

Type specimen: ML 414.

Referred specimens: None.

Revised diagnosis: Supersaurus lourinhanensis can be diagnosed by the following autapomorphies: (1) single posterior cervical and anterior dorsal neural spines (126-0*, unique among Flagellicaudata), (2) the ventral keel is restricted to the posterior portion of the posterior cervical centrum (193-1*, unique within Flagellicaudata), (3) three small fossae on the lateral face of the posterior cervical neural spine, posterior to the elongated coel (unambiguous; this trait was not included as character, but in the diagnosis following Mannion et al., 2012), (4) dorsal centrum length (excluding articular ‘ball’) remains approximately the same along the sequence (225-0*, unique among Diplodocinae), (5) dorsal transverse processes are more than 30° inclined dorsally from the horizontal (230-1*, unique among Diplodocidae), and (6) the ventral surface of anterior caudal centra is without irregularly placed foramina (305-0*, unique within Diplodocinae).

Comments: In their redescription of the species, Mannion et al. (2012) mention two additional autapomorphies: an accessory, subvertical lamina in the postzygapophyseal centrodiapophyseal fossa, and an accessory lamina linking the hyposphene to the posterior centrodiapophyseal lamina in mid- and posterior dorsal neural arches. A subvertical accessory lamina actually subdivides the pocdf in a variety of diplodocid and diplodocine taxa (e.g., Galeamopus hayi), whereas a lamina connecting hyposphene and pcdl is also present in posterior dorsal neural arches of Supersaurus vivianae.

Locality and horizon: Praia de Porto Dinheiro, Lourinhã, Portugal. Amoreira-Porto Novo Member, Lourinhã Formation, Late Kimmeridgian.

Kaatedocus Tschopp & Mateus, 2012

Type and only species: Kaatedocus siberi Tschopp & Mateus, 2012.

Revised diagnosis: Kaatedocus can be diagnosed by the following autapomorphies: (1) the dorsoventral depth of the anterior portion of the premaxilla remains the same as posteriorly, or widens gradually (7-0, unique among Diplodocidae), (2) the anterior maxillary foramen lies detached from the maxillary-premaxillary boundary, facing dorsally (11-0*, unique among Diplodocoidea), (3) the medial margin of the prefrontal is without any distinct anteromedial projection (23-0*, unique among Diplodocidae), (4) the anteroposterior length of the frontal is at least 1.4 times longer than the minimum transverse width (27-0, unique among Flagellicaudata), (5) the contribution of the frontal to the dorsal margin of the orbit is at least 1.5 times the contribution of prefrontal (32-1, unique among Flagellicaudata), (6) basal tubera breadth is more than 1.85 times occipital condyle width (83-2*, unambiguous), (7) a rugosity on the anterodorsal corner of the lateral side of mid- and posterior cervical centra (178-1, unique among Diplodocidae), (8) posterior cervical prezygapophyseal facets are posteriorly followed by a transverse sulcus (195-1*, unambiguous), (9) posterior cervical epipophyses are dorsoventrally compressed (202-1, unique among Diplodocinae), (10) posterior cervical neural spines parallel to converging (211-1, unique among Diplodocidae), and (11) the distance between the bifid posterior cervical neural spine summits is subequal to neural canal width (212-1, unique among Diplodocidae).

Comments: The species and genus reference given above (‘Tschopp & Mateus, 2012’) does not refer to the publication listed in the references as Tschopp & Mateus (2012), but to Tschopp & Mateus (2013b). This is because the online version of the description of K. siberi was published in 2012, and thus the name is valid since that year. The printed version of the paper, however, was only published in 2013.

Tschopp & Mateus (2013b) list several other autapomorphies as well: a U-shaped notch between the frontals, presence of a post-parietal foramen, a sharp, narrow sagittal nuchal crest, a straight anterior edge of the basal tubera, and the cervical pre-epipophysis that forms a distinct anterior spur. The notch is herein shown to be shared with Galeamopus SMA 0011. The presence of a post-parietal foramen is difficult to interpret in most diplodocid skulls, due to often fractured surfaces in this area of the skull. Moreover, it is present as well in another diplodocine braincase from the Howe Quarry, SMA O25-8. A relatively sharp sagittal nuchal crest also occurs in the skull of Galeamopus hayi HMNS 175 (Holland, 1906). Straight to convex anterior margins of the basal tubera are shared with CM 3452 and SMA 0011. The development of the cervical pre-epipophysis is actually different in the holotype and the referred specimen AMNH 7530, where no distinct anterior spur is present. The presence or absence of a spur is thus better interpreted as individually variable within Kaatedocus, and thus not diagnostic for the present genus.

Locality and horizon: Howe Quarry, Shell, Bighorn County, Wyoming. Lower Morrison Formation, Kimmeridgian. Dinosaur zone 2 (Turner & Peterson, 1999), Zone 2 (Foster, 2003).

Kaatedocus siberi Tschopp & Mateus, 2012

Type specimen: SMA 0004.

Referred specimens: AMNH 7530, SMA D16-3.

Diagnosis, locality, and horizon as genus.

Leinkupal Gallina et al., 2014

Syn. Leikupal Gallina et al., 2014 (misspelling).

Type species: Leinkupal laticauda Gallina et al., 2014.

Revised diagnosis: Leinkupal can be diagnosed by the following autapomorphies: (1) anterior caudal transverse processes have a single anterior centrodiapophyseal lamina (314-0*, unique among Diplodocidae). The following autapomorphies of the genus are included provisionally, following Gallina et al. (2014, p. 2): (2) “anterior caudal transverse process extremely developed (about equal or wider to centrum width) with lateroventral expansions reinforced by robust dorsal and ventral bars”; (3) “very robust centroprezygapophyseal lamina in anterior caudal vertebra”; (4) “ paired pneumatic fossae located on the base of the postzygapophysis, opposite to the articular side, in anterior-most caudal vertebra.”

Comments: Because we only included the holotype specimen, and did not add any autapomorphy proposed by Gallina et al. (2014) as a phylogenetic character, most autapomorphies could not be tested in this analysis, and are thus included directly from Gallina et al. (2014) in our revised diagnosis.

Locality and horizon: 40 km south of Picún Leufú town, Neuquen, Argentina. Bajada Colorada Formation, late Berriasian to Valanginian.

Leinkupal laticauda Gallina et al., 2014

Syn. Leikupal laticauda (misspelling).

Type specimen: MMCH-Pv 63-1.

Paratypes: MMCH-Pv 63-2 to 63-8.

Diagnosis, locality, and horizon as for the genus.

Galeamopus gen. nov

Type species: Galeamopus hayi comb. nov. (Holland, 1924). The type species was originally assigned to “Diplodocus” hayi.

Diagnosis: Galeamopus is diagnosed by the following autapomorphies: (1) the distal end of the paroccipital process is curved in lateral view (69-1, unique among Diplodocinae), (2) teeth with paired wear facets (118-0, unique among Flagellicaudata), (3) well-developed anteromedial processes on the atlantal neurapophyses, which are distinct from the posterior wing (146-1, unique among Diplodocoidea), (4) the atlantal neural arch bears a small subtriangular, laterally projecting spur at its base (147-1, unique among Diplodocidae), (5) the posterior wing of atlantal neurapophyses remains of subequal width along most of its length (148-1, unambiguous), (6) the axial prespinal lamina develops a transversely expanded, knob-like tuberosity at its anterior end (151-1, unambiguous), and (7) the interpostzygapophyseal lamina of mid- and posterior cervical neural arches does not project beyond the posterior margin of the neural arch (190-0, unique among Diplodocinae).

Etymology: ‘Galeam’ means helmet, and ‘opus’ need, necessity in Latin, which literally translates to the German name Wilhelm (meaning “want helmet, protection”) and its English translation William. Galeamopus remembers and honors the two ‘Williams’ intimately connected with the genoholotype specimen HMNS 175: William H. Utterback and William J. Holland. Utterback found HMNS 175 in 1902 and Holland described its braincase in 1906, and named the holotype species G. hayi as Diplodocus hayi in 1924—although already stating that the morphological differences between G. hayi and Diplodocus might be enough to allow the erection of a new genus in future. Galeamopus is also an allusion to the fact that the fragile braincase is the only described part of the holotype skeleton to date.

Locality and horizon: Various sites in Colorado and Wyoming. Lower to Middle Morrison Formation, Kimmeridgian. Apatosaurine interval 1 (Bakker, 1998), Dinosaur zone 2 to possibly 3 (Turner & Peterson, 1999), Zones 2 to possibly 3 or 4 (Foster, 2003).

Galeamopus hayi (Holland, 1924)

Syn.: Diplodocus hayi Holland, 1924.

Type specimen: HMNS 175 (previously CM 662).

Referred specimens:-

Diagnosis: Galeamopus hayi is diagnosed by the following autapomorphies: (1) dorsoventral height of the parietal occipital process is low, subequal to less than the diameter of the foramen magnum (63-0*, unique among Diplodocinae), (2) basipterygoid processes widely diverging (>60°; 92-0*, unique among Diplodocinae), (3) an ulna to humerus length of more than 0.76 (387-2*, unique within Diplodocoidea), (4) distal articular surface for the ulna on the radius is reduced and relatively smooth (392-0*, unique within Diplodocidae), (5) the distal condyle of the radius is beveled at least 15° to the long axis of the shaft (393-1*, unique within Diplodocinae), (6) and the lateral edge of the proximal end of the tibia forms a pinched out projection, posterior to the cnemial crest (446-0*, unique among Diplodocidae).

Comment: Given the possible occurrence of a second species within Galeamopus, the diagnosis of G. hayi is here restricted to its holotype, which is the only specimen definitely referrable to this species.

Locality and horizon: Quarry A, Red Fork of the Powder River, Johnson County, Wyoming. Lower Morrison Formation, Kimmeridgian. Apatosaurine interval 1 (Bakker, 1998).

Discussion

The phylogenetic history of Diplodocidae

Most earlier phylogenetic studies of sauropods just included the three diplodocid genera: Apatosaurus, Diplodocus, and Barosaurus (e.g., Upchurch, 1998; Wilson, 2002; Upchurch, Barrett & Dodson, 2004). More recent analyses with a narrower focus on diplodocoid intrarelationships included more diplodocid species (Upchurch, Tomida & Barrett, 2004; Rauhut et al., 2005; Remes, 2006; Salgado, Carvalho & Garrido, 2006; Lovelace, Hartman & Wahl, 2007; Sereno et al., 2007; Whitlock, 2011a; Carballido et al., 2012b; Mannion et al., 2012; Tschopp & Mateus, 2013b). However, other than Upchurch, Tomida & Barrett (2004), all of them included the genera Apatosaurus and Diplodocus as single OTUs, rather than their component species, and no analysis was ever done with all proposed diplodocid species as separate OTUs (Fig. 119). Basic relationships between diplodocid taxa generally remained the same among these studies, probably as a consequence of the fact that until the publication of a study focusing on intrarelationships of Diplodocidea (Whitlock, 2011a), most were based on Wilson (2002), with only minor changes (Rauhut et al., 2005; Remes, 2006; Salgado, Carvalho & Garrido, 2006; Lovelace, Hartman & Wahl, 2007; Sereno et al., 2007). The greatest changes between the analyses of Rauhut et al. (2005), Remes (2006), Salgado, Carvalho & Garrido (2006), Lovelace, Hartman & Wahl (2007) and Sereno et al. (2007) occur in the position of Suuwassea, which was recovered as a dicraeosaur (Salgado, Carvalho & Garrido, 2006), within Apatosaurinae (Lovelace, Hartman & Wahl, 2007), in a polytomy with Apatosaurus and Diplodocinae (Remes, 2006), just outside Apatosaurinae + Diplodocinae (Rauhut et al., 2005), or in a trichotomy with Diplodocidae and Dicraeosauridae (Sereno et al., 2007). Other than Apatosaurus, Diplodocus and Barosaurus, only Tornieria was included in more than one of these four analyses, and was found within Diplodocinae (Rauhut et al., 2005; Remes, 2006).

Figure 119 Strict consensus trees of previous phylogenetic analyses with special focus on diplodocoid intrarelationships, with the number of taxa (T) and characters (C) indicated.

In brackets the number of diplodocid taxa and newly proposed characters. Taxon names were changed according to more recent publications, and diplodocid OTU highlighted with the red box.

Given the strong focus on interspecific relationships of Apatosaurus, Upchurch, Tomida & Barrett (2004) had a very reduced dataset, with only 16 OTUs and 32 characters. The character list was assembled based on earlier descriptions and diagnoses of the different species (mostly Riggs, 1903; Holland, 1915a; Gilmore, 1936), with some original characters added (Upchurch, Tomida & Barrett, 2004). The study of Whitlock (2011a), although based in part on that of Wilson (2002), can be considered as a new analysis as well, given the large number of modifications and added characters (total: 169 parsimony-informative characters), and the greatly increased number of taxa (26 taxa) included in order to resolve diplodocoid intrarelationships. Subsequent analyses (Mannion et al., 2012; Tschopp & Mateus, 2013b; Gallina et al., 2014) represent modifications of Whitlock (2011a).

The present analysis further increases both the taxon and character lists of Whitlock (2011a), by about 300% and 250%, respectively (81 versus 26 OTUs, 477 versus 189 characters), and can thus be considered largely independent as well. Nonetheless, the positions of most common genera included in the analyses remain the same. Analyses of diplodocoid phylogeny so far therefore generally corroborate each other.

Combined cladogram

Although generally corroborating the results of previous studies, our analysis proposes three major taxonomic changes within Diplodocidae: (1) the resuscitation of Brontosaurus as a distinct genus from Apatosaurus; (2) the discovery of an additional genus within Diplodocinae, herein named Galeamopus and typified by the species G. hayi, which was previously referred to Diplodocus; and (3) the treatment of “Dinheirosaurus” as junior synonym of Supersaurus, creating the new combination Supersaurus lourinhanensis. Other differing interpretations are the inclusion of Amphicoelias altus in Diplodocidae, the recognition of an additional, potentially new species in both Diplodocinae and Apatosaurinae (not named herein), and the referral of the species Eobrontosaurus yahnahpin and Elosaurus parvus to the genus Brontosaurus, as Brontosaurus yahnahpin and Brontosaurus parvus, respectively. Based on the identifications discussed above (Table 5), a combined species-level cladogram was created to summarize our results (Fig. 120). This cladogram represents the most up-to-date species-level taxonomy of Diplodocidae. Outgroup taxa are pruned considerably compared to the trees recovered by the main analyses, in order to increase the intended focus on Diplodocidae.

Figure 120 Speciel-level cladogram of Diplodocidae.

Combined cladogram of diplodocid species-level intrarelationships, summarizing the results of the present thesis. Stem-based higher-level taxa are marked by an arrowhead, node-based taxa by a dot.

Biostratigraphic and paleobiogeographical implications

Our analysis rejects diplodocid affinities of the probable Middle Jurassic taxa Cetiosauriscus stewarti and Dystrophaeus viaemalae, and the potentially Cretaceous species Losillasaurus giganteus and Dyslocosaurus polyonychius. The first representative of the clade is therefore a caudal vertebra from the Oxfordian of Georgia (Gabunia et al., 1998; Mannion et al., 2012). Until recently, no definite Cretaceous diplodocid material was recognized: a single anterior caudal vertebra previously identified as Cretaceous diplodocid (Upchurch & Mannion, 2009) was subsequently shown to belong to Titanosauriformes (Whitlock, D’Emic & Wilson, 2011). However, at least Diplodocinae continued into the Cretaceous, as demonstrated by the recent discovery of Leinkupal laticauda (Gallina et al., 2014).

The highest diversity of Diplodocidae is known from the Morrison Formation, which is interpreted as representing a time span of about seven (Swierc & Johnson, 1996; Kowallis et al., 1998) to eleven million years (Platt & Hasiotis, 2006). Simply dividing the number of the at least 14 diplodocid species that existed during this period by the duration of the Morrison Formation, it appears that more than one diplodocid lived contemporaneously at any time throughout the entire duration of the sedimentation of the Morrison Formation, in addition to non-diplodocid sauropods such as Suuwassea, Camarasaurus, Haplocanthosaurus, Brachiosaurus, and others. If precise stratigraphical levels and geological ages were known for all the sites where diplodocids were found, the present analysis would provide a good phylogenetic foundation on which hypotheses of speciation, standing diversity, and niche partitioning within diplodocids from the Morrison Formation could be based. However, exact geological dating has not been widely conducted, and has provided controversial results (in particular for the Howe Ranch sites, Tschopp & Mateus, 2013b). Furthermore, no reliable marker beds appear to be present throughout the entire extent of the Morrison Formation (Trujillo, 2006; contra Turner & Peterson, 1999). Therefore, long distance stratigraphic correlation between Morrison Formation quarries is nearly impossible at present. Proposed biostratigraphical zones within the formation (Bakker, 1998; Turner & Peterson, 1999; Foster, 2003; Ikejiri, 2004) have thus to be regarded questionable and provisional. Their validity is furthermore debatable because they heavily rely on species and genus referrals that have not been tested by means of phylogenetic analyses, and often only include the classic diplodocid genera Diplodocus, Apatosaurus, and Barosaurus. Given that the diversity of diplodocids appears to have been underestimated, as indicated by our analysis, these referrals will have to be reconsidered.

Diplodocidae is most diverse in the Late Jurassic of North America, but the earliest find from Georgia suggests that the origin of the clade lies in Europe (Mannion et al., 2012). All non-North American diplodocid OTUs included herein (ML 418, “Dinheirosaurus” lourinhanensis, Leinkupal laticauda, Tornieria africana) can also be confidently referred to Diplodocinae (Whitlock, 2011a; Mannion et al., 2012; Gallina et al., 2014; this study). The fact that these non-North American species lie at the base of the diplodocine radiation (Fig. 120) furthermore corroborates a hypothesis of an extra-North American origin of this clade. Interestingly, apatosaurine specimens have only been recovered from North America to date, indicating that they entirely evolved on that continent.

Conclusions

The present paper increases knowledge about the phylogenetic relationships of diplodocid sauropods. In order to resolve relationships within Diplodocidae, a specimen-based phylogenetic analysis was performed, which included all holotypes that have been identified as belonging to a diplodocid sauropod at some point in history.

By doing so, one of the main challenges was where to decide if specific or generic separation of the included specimens is warranted. Given that the only applicable species concept in paleontology is based on morphological differences, summing of differences can be the only way to approach this issue. Based on the assumption that rates of evolution were similar in the temporally and spatially coexisting taxa Diplodocinae and Apatosaurinae, accumulation of individually varying traits is assumed to lead to speciation at the same speed in both taxa. Thus, two numerical approaches were used to make taxonomic decisions. One of them, pairwise dissimilarity, is based on morphological disparity and includes all the morphologial evidence. The second approach is restricted to apomorphic features recovered as such by the software TNT, and thus excludes morphological features considered taxonomically insignificant by the software. In combination, these approaches are able to account for the influence of individual variation, and provided a useful tool to assess the validity of the included taxa in a more objective way.

The numerical approaches established in the present analysis allowed a reassessment of the validity of the numerous taxonomic names proposed within Diplodocidae. Thereby, it was found that apatosaurine diversity was particularly underestimated in the past. One genus previously synonymized with Apatosaurus is considered to be valid based on our quantitative approaches: Brontosaurus forms the sister clade to Apatosaurus in the present analysis. On the other hand, Elosaurus and Eobrontosaurus were found to be junior synonyms of Brontosaurus, and one more cluster of specimens was recovered at the base of Apatosaurinae, which might even represent a further, new apatosaurine genus. Apatosaurus was found to include only the two species A. ajax and A. louisae. This results in three genera and six species belonging to Apatosaurinae. In a less inclusive and less detailed specimen-based analysis of Apatosaurus, Upchurch, Tomida & Barrett (2004) found five species as probably valid, but did not include Eobrontosaurus yahnahpin. The species count recovered by our analysis is comparable to that proposed by Upchurch, Tomida & Barrett (2004).

The intrarelationships of Diplodocinae were already well established in previous work (Whitlock, 2011a; Mannion et al., 2012; Tschopp & Mateus, 2013b). However, by including single specimens, we were able to further assess the validity of the various species proposed in Diplodocus. Thereby, the type species D. longus was considered a nomen dubium, given the undiagnostic, fragmentary holotype specimen. In order to avoid the unsatisfying situation, where Diplodocus would be typified by an invalid species, a case is being prepared for submission to ICZN proposing D. carnegii as the new type species, and suppressing D. longus. Furthermore, the holotype specimen of ‘Diplodocus’ hayi, often mentioned to probably not belong to Diplodocus (Holland, 1924; McIntosh, 1990a; Curtice, 1996; Foster, 2003), was found to form its own genus (herein named Galeamopus), together with specimen SMA 0011, and the diplodocine skulls AMNH 969 and USNM 2673 – the latter two also having previously been identified as Diplodocus (Holland, 1906; Holland, 1924; McIntosh & Berman, 1975). Interestingly, no diplodocine specimen preserving articulated cranial and postcranial elements was herein found to group with Diplodocus: AMNH 969 and ‘Diplodocus’ hayi are referred to Galeamopus, and CM 3452, on which Holland (1924), McIntosh & Berman (1975) and Berman & McIntosh (1978) based their identification of the skull-only specimens as Diplodocus, is recovered as more closely related to Barosaurus and Kaatedocus. Therefore, although they are essentially complete and well preserved, skulls such as CM 11161 or USNM 2672 cannot be definitely referred to Diplodocus. However, their recovered intermediate position between Galeamopus and Kaatedocus + Barosaurus indicates that a referral to Diplodocus might be justifiable, even though direct evidence is lacking. In any case, given the completeness and articulation of the two Galeamopus specimens HMNS 175 and SMA 0011, as well as the presence of at least two additional, referred skulls, the morphology of Galeamopus can be considered better known than that of Diplodocus, for which information on the skull, forelimb, and distal tail morphology is not available from type specimens.

In total, nine to eleven different species in seven or eight genera are recognized within Diplodocinae and six to seven species in three genera within Apatosaurinae. Together with the probable non-apatosaurine, non-diplodocine diplodocid Amphicoelias altus, this totals 15–18 valid diplodocid species, 12–15 of which are from the Morrison Formation of the western United States.

Supplemental Information

Supplemental Information 1 Definitions of positional terms for vertebrae

Click here for additional data file.

Supplemental Information 2 Taxa and specimens included as OTUs in the phylogenetic analysis, with personal observations, included bones, and additional references

Click here for additional data file.

Supplemental Information 3 Tables with ratios corresponding to the numerical characters used in the phylogenetic analysis

Includes supplementary tables 2-58

Click here for additional data file.

Supplemental Information 4 Tables with the synapomorphies of the clades found and the autapomorphies of the OTUs recovered by TNT

Includes supplementary tables 59-158, with the qualitative assessment of the validity of apomorphies for the apomorphy count.

Click here for additional data file.

Supplemental Information 5 Mean pairwise dissimilarity values between diplodocid species and genera

Click here for additional data file.

Supplemental Information 6 Used constraints for constrained tree searches in TNT

Click here for additional data file.

Supplemental Information 7 Apomorphies recovered from TNT under equal weighting

Click here for additional data file.

Supplemental Information 8 Apomorphies recovered by TNT under implied weighting

Click here for additional data file.

Supplemental Information 9 Phylogenetic matrix

file for TNT

Click here for additional data file.

With this paper forming part of ET’s dissertation, first thanks go to the members of the doctoral committee, who are Octávio Mateus, Martin Sander (Bonn, Germany), João Pais, Rogerio de Rocha (both GeoBioTec), and Louis Jacobs (Dallas, USA).

A specimen-based phylogenetic analysis as performed herein is highly dependent on personal observations of the specimens included. Although it was not possible to see all of them, many collection visits were possible thanks to the help and hospitality from the following people: Kate Wellspring (AC), Carl Mehling, Mark Norell, and Alana Gishlick (AMNH), Ted Daeschler and Ned Gilmore (ANS), Brooks Britt and Rodney Scheetz (BYU), Amy Henrici, Matthew Lamanna, and Dan Pickering (CM), Rafael Royo-Torres and Edoardo Espilez (CPT), Virginia Tidwell and Logan Ivy (DMNS), David Temple (HMNS), Luis Chiappe and Maureen Walsh (LACM), Daniela Schwarz (MB.R.), Margarita Belinchón (MCNV), Miguel Ramalho (MIGM), Paul Barrett and Sandra Chapman (NHMUK), Hans-Jakob Siber and Thomas Bolliger (SMA), Ralf Kosma, Achim Ritter, and Ulrich Joger (Staatl. Naturhist. Mus. Braunschweig), Matt Carrano and Mike Brett-Surman (USNM), Bill Wahl and Malcolm Bedell (WDC), and Dan Brinkman and Marilyn Fox (YPM). Several people shared numerous pictures of specimens we were not able to see ourself: Christophe Hendrickx (GeoBioTec), Carl Mehling, Dave Lovelace (Univ. of Wisconsin), Heinrich Mallison (MB.R.), José Carballido (MPEF), Jay Nair (Univ. of Queensland, Australia), John Whitlock (Mount Aloysius College), Kelli Trujillo (Univ. of Wyoming), Larry Witmer (Ohio Univ.), Malcolm Bedell, Mike Brett-Surman, Matt Wedel (West. Univ. of Health Sciences, Pomona), Phil Mannion (Imperial College London), Ralf Kosma, Spencer Lucas (NMMNH), Takehito Ikejiri (Univ. of Alabama), Virginia Tidwell, William Gearty (YPM), and William Simpson (FMNH). Toru Sekiya (Fukui Prefectural Dinosaur Musem, Japan) helped in figuring out the current location and specimen numbers of Mamenchisaurus specimens mentioned in the supplemental tables.

We thank the Willi Hennig Society for making the phylogeny software TNT freely accessible, and Andrea Cau (Museo Geologico “Cappellini,” Bologna), Jay Nair, and especially José Carballido for invaluable help with phylogenetic techniques.

Christophe Hendrickx (GeoBioTec), Jay Nair (Univ. of Queensland), and Mike Taylor (Univ. of Bristol) reviewed an earlier version of portions of this paper and provided corrections for the English. Phil Mannion and John Whitlock are greatly acknowledged for their detailed reviews, and Andrew Farke for his editorial advice. Finally, we want to thank the editorial staff of PeerJ for their technical help and advice about handling such an extensive manuscript.

Institutional abbreviations

AC Beneski Museum of Natural History, Amherst College, Amherst, Massachusetts, USA

AL Memorial Museum of Jiushao QIN, Yunju Mountain, Anyue County, China

AMNH American Museum of Natural History, New York City, New York, USA

ANS Academy of Natural Sciences, Philadelphia, Pennsylvania, USA

BYU Brigham Young University, Museum of Paleontology, Provo, Utah, USA

CCG Chengdu College of Geology, Sichuan, China

CM Carnegie Museum of Natural History, Pittsburgh, Pennsylvania, USA

CMC Cincinnati Museum Center, Cincinnati, Ohio, USA

CMNH Cleveland Museum of Natural History, Cleveland, Ohio, USA

CPT Conjunto Paleontológico de Teruel, Dinópolis, Teruel, Spain

DMNS Denver Museum of Nature and Science, Denver, Colorado, USA

DNM Dinosaur National Monument, Jensen, Utah, USA

FMNH Field Museum of Natural History, Chicago, Illinois, USA

GCP Grupo Cultural Paleontológico de Elche, Museo Paleontológico de Elche, Elche, Spain

GMNH Gunma Museum of Natural History, Gunma, Japan

HMNS Houston Museum of Nature and Science, Houston, TX, USA

ISIR Paleontological Collection, Geology Museum, Indian Statistical Institute, Calcutta, India

IVPP Institute of Vertebrate Paleontology and Paleoanthropology, Chinese Academy of Sciences, Beijing, China

KUVP Kansas University Natural History Museum, Lawrence, Kansas, USA

LACM Los Angeles County Museum of Natural History, Los Angeles, USA

MACN Museo Argentino de Ciencias Naturales, Neuquén, Argentina

MB.R. Museum für Naturkunde, Berlin, Germany

MCF Museo Carmen Funes, Plaza Huincul, Neuquén, Argentina

MCNV Museo de Ciencias Naturales, Valencia, Spain

MCUT Museum of the Chengdu University of Technology, Chengdu, China

MDS Museo de Dinosaurios de Salas de los Infantes, Salas de los Infantes, Burgos, Spain

MIGM Museu Geológico do Instituto Geológico e Mineiro de Portugal, Lisboa, Portugal

ML Museu da Lourinhã, Lourinhã, Portugal

MMCH Museo Municipal “Ernesto Bachmann,” Villa El Chocón, Neuquén province, Argentina

MNN Musée National du Niger, Niamey, Republic of Niger

MOZ Museo Provincial de Ciencias Naturales ‘Prof. Dr. Juan A. Olsacher,’ Zapala, Neuquén, Argentina

MPCA Museo Provincial Carlos Ameghino, Cipolletti, Río Negro, Argentina

MPEF Museo Paleontológico Egidio Feruglio, Trelew, Argentina

MUCPv Museum of the University of Comahue-Patagonia, Argentina

NHMUK Natural History Museum, London, United Kingdom

NMB Staatliches Naturhistorisches Museum Braunschweig, Germany

NMMNH New Mexico Museum of Natural History and Science, Albuquerque, New Mexico, USA

NSMT National Museum if Nature and Science, Tokyo, Japan

OMNH Sam Noble Oklahoma Museum of Natural History, Norman, Oklahoma, USA

PMU Evolutionsmuseet Paleontologi, University of Uppsala, Uppsala, Sweden

SMA Sauriermuseum Aathal, Aathal, Switzerland

SMNS Staatliches Museum für Naturkunde, Stuttgart, Germany

Tate Tate Geological Museum, Casper College, Casper, Wyoming, USA

UMNH Utah Museum of Natural History, Salt Lake City, Utah, USA

USNM United States National Museum, Smithsonian Institution, Washington DC, USA

UUVP University of Utah, Salt Lake City, Utah, USA

UW University of Wyoming Geological Museum, Laramie, Wyoming, USA

WDC Wyoming Dinosaur Center, Thermopolis, Wyoming, USA

YPM Yale Peabody Museum, New Haven, Connecticut, USA

ZDM Zigong Dinosaur Museum of Sichuan Province, China

Anatomical abbreviations

aal acetabular articulation surface length

ac acetabular surface

aCd anterior caudal vertebrae

acdl anterior centrodiapophyseal lamina

acl acromion length

acm acromion

acpl anterior centroparapophyseal lamina

acr acromial ridge

aCV anterior cervical vertebrae

aDV anterior dorsal vertebrae

amc amphicoelous

amCd anterior-most caudal vertebrae

amp amphiplatyan

an angular

anp antotic process

aof antorbital fenestra

ap anterior process

apd anteroposterior depth

apf anterior pneumatic fossa

apl anteroposterior length

as astragalus

at atlas

ato anterior tooth

ax axis

Bc braincase

bic biconvex

bns bifid neural spine

bo basioccipital

bph basipterygoid hook

bpr basipterygoid process

bs basisphenoid

bt basal tuber

c carpal

ca coracoid articulation

cal calcaneum

can crista antotica

cap capitulum

cc cnemial crest

Cd caudal vertebra

CF coracoid foramen

Ch chevrons

cl centrum length

co coracoid

cph centrum posterior height

cpol centropostzygapophyseal lamina

cpr crista prootica

cprl centroprezygapophyseal lamina

CR cervical ribs

cth centrum height

CV cervical vertebra

cw centrum width

d dentary

dapd distal anteroposterior depth

db distal blade

dCd distal caudal vertebrae

de dentin

di diapophysis

dip distal process

dpc deltopectoral crest

DR dorsal ribs

dt denticles

dtw distal transverse width

DV dorsal vertebra

emf external mandibular fenestra

en enamel

ep ectopterygoid

epi epipophysis

er ectopterygoid ramus

ex exoccipital

f frontal

fe femur

fi fibula

Fl forelimb

fm foramen magnum

FS facial skull

h humerus

hc haemal canal

hca anterior centrum height

hcd height condyle

hct height cotyle

Hl hindlimb

hns height neural spine

hys hyposphene

il ilium

ip iliac peduncle

is ischium

isa ischial articular surface

j jugal

la lacrimal

LJ lower jaw

lr lateral ridge

ls laterosphenoid

lspol lateral spinopostzygapophyseal lamina

ltf laterotemporal fenestra

m maxilla

Ma manus

maxW maximum transverse width

mc metacarpal

mCd mid-caudal vertebrae

mCV mid-cervical vertebrae

mDV mid-dorsal vertebrae

minW minimum transverse width

mspol medial spinopostzygapophyseal lamina

mt median tubercle

mts metatarsal

na nasal

naf neural arch foramen

nc neural canal

ncs neurocentral synostosis

ns neural spine

o orbit

oc occipital condyle

of obturator foramen

olf olfactory foramen

opc opisthocoelous

opf optic foramen

os orbitosphenoid

p parietal

pa palate

pabh preacetabular blade height

pap parapophysis

pas proximal articular surface

pCd posterior caudal vertebrae

pcdl posterior centrodiapophyseal lamina

PcG pectoral girdle

pcpl posterior centroparapophyseal lamina

pCV posterior cervical vertebrae

pd proximal depth

pdl proximodistal length

pDV posterior dorsal vertebrae

Pe pes

pf prefrontal

phm manual phalanx

php pedal phalanx

pl pleurocoel

pm premaxilla

pnf pneumatic foramina

po postorbital

pocdf postzygapophyseal centrodiapophyseal fossa

podl postzygodiapophyseal lamina

popr paroccipital process

posl postspinal lamina

poz postzygapophysis

ppapd pubic peduncle anteroposterior depth

ppf posterior pneumatic fossa

ppfo postparietal foramen

ppl pneumatopore length

ppw pubic peduncle transverse width

pra proatlas

prap preacetabular process

prc procoelous

prcdf prezygapophyseal centrodiapophyseal fossa

prdl prezygodiapophyseal lamina

pre pre-epipophysis

pro prootic

prpl prezygoparapophyseal lamina

prsl prespinal lamina

prz prezygapophysis

psr parasphenoid rostrum

pt pterygoid

ptc platycoelous

ptf posttemporal fenestra

pto posterior tooth

ptr vertical distance from proximal articular surface to trochanter

ptw proximal transverse width

pu pubis

pup pubic peduncle

pvf posteroventral flanges

pvfo posteroventral fossa

PvG pelvic girdle

pvlp posterior ventrolateral process

q quadrate

qj quadratojugal

qr quadrate ramus

r radius

sa surangular

saf surangular foramen

sc scapula

sdf spinodiapophyseal fossa

snc sagittal nuchal crest

so supraoccipital

SP sternal plates

spdl spinodiapophyseal lamina

spof spinopostzygapophyseal fossa

spol spinopostzygapophyseal lamina

sprl spinoprezygapophyseal lamina

sq squamosal

stf supratemporal fenestra

SV sacral vertebrae

sw shaft width

sy sacricostal yoke

sym symphysis

T teeth

tb tibia

tc tooth crown

tp transverse process

tpol interpostzygapophyseal lamina

tprl interprezygapophyseal lamina

tr tooth root

tub tuberculum

u ulna

ucp ulnar condylar processes

v vomer

vlh ventral longitudinal hollow

wct width cotyle

wf wear facet

Other Abbreviations

AmAl Amphicoelias altus

AtIm Atlantosaurus immanis

AuBo Australodocus bohetii

C23-1 state 1 of character 23

CeSt Cetiosauriscus stewarti

EI elongation index

ew equal weighting

HaPr Haplocanthosaurus priscus

HOS histological ontogenetic stage

iw implied weighting

mdA more derived Apatosaurines

mdD more derived Diplodocoidea

mdE more derived Eusauropoda

OTU operational taxonomic unit

PMI premaxilla-maxilla index

RI robustness index

SI slenderness index

SuVi Supersaurus vivianae

ToAf Tornieria africana.

Additional Information and Declarations

Competing Interests

Author Contributions

Data Deposition

New Genus Registration

The authors declare there are no competing interests.

Emanuel Tschopp and Roger B.J. Benson conceived and designed the experiments, performed the experiments, analyzed the data, contributed reagents/materials/analysis tools, wrote the paper, prepared figures and/or tables, reviewed drafts of the paper.

Octávio Mateus conceived and designed the experiments, contributed reagents/materials/analysis tools, reviewed drafts of the paper.

The following information was supplied regarding the deposition of related data:

MorphoBank: http://morphobank.org/permalink/?P2124.

The following information was supplied regarding the registration of a newly described genus:

The new genus is registered on ZooBank.

urn:lsid:zoobank.org:pub:3AEF1593-FD4F-45A2-80AD-05DDA8DB93B4.

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
