# Peer review of "A specimen-level phylogenetic analysis and taxonomic revision of Diplodocidae (Dinosauria, Sauropoda)"

_PeerJ, doi:10.7717/peerj.857_

## Round 0.1 · original submission · Major Revisions

Dear Dr. Tschopp:

As the handling editor for your manuscript, I have given it an initial examination prior to sending it out for review. Although my preliminary pass shows the article to have considerable merit (and I particularly applaud the thorough description of the new specimen), I am concerned that the sheer size of the manuscript will prevent a fair and thorough review. With over 270 pages of manuscript alone (not counting tables or figures!), in my opinion it will be quite difficult to solicit reviewers. Perhaps more importantly, I worry that some of the more pertinent and important details of the research may be lost to the reader in a sea of text, tables, and figures. In my opinion, aspects of the paper (particularly apomorphy lists) could be moved to supplementary information or presented more succinctly than given here, without harm to the paper. These and other comments are summarized below.

I should emphasize that the paper has not been sent for formal review (this decision is by way of a 'pre-review'). If you can make the changes suggested, then I believe I will be in a better situation to seek reviewers (and the manuscript will be more readable as a result). Once you return the manuscript I should warn you, as I am sure you suspect, that it may take longer than normal for the reviews to come in.

1) The status of the holotype specimen for Galeamopus [REDACTED] at the Sauriermuseum Aathal must be addressed in the manuscript. Is the specimen appropriately and permanently reposited in the public trust? This point is absolutely necessary to verify before we can proceed with review.

2) Could some of the tables be transferred into supplementary information? The primary specimen measurements for Galeamopus [REDACTED] certainly make sense to keep in the main body of the paper, but it seems as if many of the tables of ratios for the phylogenetic analysis could be moved to supplementary information.

3) Many areas of the manuscript could be condensed drastically. For instance, pages 143-231 are perhaps most amenable to shortening. Lists of synapomorphies are repetitious and not particularly accessible. As an alternative, you may wish to create a supplementary table listing the synapomorphies for each clade, or move this to supplementary text.

4) Along the same lines, is it necessary to diagnose clades above the species level in the text? More "traditional" node- or stem-based taxonomic statements are more useful and easier to present for the reader. For all of these diagnoses, there isn't much reason to list why previous characters are now invalid...it just adds too much text that is not necessary (in my opinion). If a character is more broadly distributed than previously thought, that should speak for itself.

5) Give strong consideration to combining some figures. For instance, Figures 35-37 could certainly be incorporated into one, and it would seem logical to do so (and it probably wouldn't require rescaling for most, if any). Could some of the vertebral figures be combined (e.g., Figures 18 and 19)? This would also help to reduce redundancy in figure caption abbreviations and make the reading smoother for everyone.

6) As a more extreme alternative, you may consider splitting this into two papers: one dealing with Galeamopus and the second dealing with the specimen-based phylogeny of Diplodocidae. I do not consider this absolutely necessary, but wanted to mention it as a possibility.

I realize that these suggestions create some additional work for you, but in my opinion they are necessary to make the paper accessible to readers and reviewers. Thank you for your attention to the suggestions, and please let me know if you have any questions.

---

## Round 0.2 · Minor Revisions

This is a tremendously detailed and well-documented paper that sets a high standard for descriptive and phylogenetic work within dinosaurs. The reviewers provide some comments for your consideration during revision; these are summarized below and outlined in detail elsewhere in this email and in documents uploaded to the PeerJ manuscript submission system.

- Both reviewers provide detailed comments requesting clarification of certain points; these are highlighted in the marked-up manuscript as well as the full review. Please address these in your revised manuscript or rebuttal letter.

- Reviewer 1 has a concern with the way in which genera are recognized on the basis of the cladistic analysis. Although there is of course some degree of subjectivity in splitting genera, the reviewer's comments on the utility of "distance" and the splitting of monophyletic groups should be addressed in revision.

- Reviewer 1 also raises some questions about the way juveniles are treated within the analysis, which must be addressed in revision. As you know, the best way to deal with juvenile specimens in a phylogenetic analysis is a debated topic. However, weighting characters is, in my opinion (and that of the reviewer), a dicey approach. As an alternative method, it may be worth 1) more discretely identifying ontogenetically variable characters; and 2) running a version of the analysis with ontogenetically variable characters states coded as unknown for distinctly juvenile or subadult specimens (rather than completely tossing out the characters). In any case, other analyses have shown that inclusion of juveniles can be quite problematic, and this must be accounted for in your work. See, for example, Campione NE, Brink KS, Freedman EA, McGarrity CT, Evans DC (2013) “Glishades ericksoni”, an indeterminate juvenile hadrosaurid from the Two Medicine Formation of Montana: implications for hadrosauroid diversity in the latest Cretaceous (Campanian-Maastrichtian) of western North America. Palaeobiodiversity and Palaeoenvironments 93: 65–75. doi:10.1007/s12549-012-0097-1.

- Among other suggestions, Reviewer 2 provides some guidance on a few ways in which particular sections of text can be reorganized for clarity and flow. Please incorporate these as you are able, because they will improve the readbility of the paper. Note that a marked up .doc file uploaded to the PeerJ manuscript submission website contains my own comments and requests for rephrasing or clarification.

·

Basic reporting

No Comments

Experimental design

No Comments

Validity of the findings

No Comments

Additional comments

This is an excellent and extremely thorough attempt to revise the taxonomy of an important and iconic group of dinosaurs, as well as providing a detailed description of a new species. This paper will provide a benchmark for future studies examining all aspects of diplodocid evolution, but will also play an important role in discussions of genus versus species determination in paleontology, and teasing out whether differences are taxonomically significant or represent individual variation. The detailed discussion of each phylogenetic character is also extremely welcome, as is the illustration of each state – on both accounts this raises the bar for papers focused on phylogenetic relationships of fossil groups.

Although I have made numerous comments throughout the MS (annotated on the PDF), these are predominantly of a minor nature (including typos and grammar), but I have summarized several broader issues below.

1. The systematic section at the start of your paper, where you name Galeamopus, is a little confusing as presently written. You need to make it clear that you're providing a new generic name for Diplodocus hayi in this section: this isn't remotely clear at present. Also, the type species of your new genus, G. hayi, needs a diagnosis, and again should be provided in this section. I realise that it is provided in your Discussion, but it makes much more sense to go here.

2. There are several references that have accidentally been left in from this work’s origin as a thesis (e.g. “see next chapter” on line 298). I think I’ve flagged them all up, but it’s possible that I’ve missed some.

3. The opening section of the Discussion (lines 4455–4544) feels much more like it belongs either in the Background/Materials and Methods section, or that it should come after the subsequent sections whereby you go through all of the different clade supports. Also – are these support sections not really Results? In general I think it would be good to try and make the sections that would more normally end in an appendix separate from true discussion, which will be the parts that most readers will want to read.

4. I think it would be extremely useful if the affinities of each specimen were summarized in a table, so that a reader can easily see what belongs to what, without having to wade through a huge amount of text.

5. The section ‘Biostratigraphic and paleobiogeographical implications” should be moved to the end of the Discussion – bigger picture information should follow on from all the diagnoses, not be buried amongst them. This should basically be the final part of the paper prior to your Conclusions. This also seems a disproportionately short section of the paper – perhaps there are other aspects worth discussing of diplodocid evolution? Or you could explicitly outline what work still needs to be done? The lack of actual discussion almost seems to sell short the enormous amount of work undertaken.

Best wishes,

Phil Mannion

·

Basic reporting

Article meets basic reporting criteria.

Experimental design

Article meets basic standards, but is lacking in certain aspects (see attached).

Validity of the findings

Basic results are valid and reproducible, but issues with experimental design color interpretation of the results (see attached).

---

## Round 0.3 · Minor Revisions

Thank you for your careful revision of the manuscript - it is greatly improved as a result. The same reviewers who saw the first version of the paper have given a few final, minor comments; once these are addressed (either by incorporation into the manuscript or rebuttal), we should be able to move forward in very short order.

·

Basic reporting

No Comments

Experimental design

No Comments

Validity of the findings

No comments

Additional comments

Dear authors,

As I expressed in my review of the original submission, this is an excellent and welcome contribution. The loss of the description of the new taxon is also probably a good thing in terms of focusing just on the phylogeny and taxonomy.

I have made numerous minor comments throughout the MS, many of which are just spotting typos (please see the attached annotated version of the PDF), but the only moderately substantial issue I would raise concerns your decision of where to separate species/genera and relates to specimen completeness. If a specimen is quite incomplete, then it's likely to be difficult to find enough autapomorphies to reach the required species/genus quota, whereas lots of autapomorphies might have been present in other (non-preserved) parts of the skeleton. In the case of Dinheirosaurus (and it's your proposed synonymization of that taxon with Supersaurus that got me thinking about this issue), there's only a couple of cervical vertebrae and a dorsal series, and yet it "almost" passes the genus test. Perhaps you need to also consider weighting the number of differences by the completeness of the specimen when determining your species/genus determinations too? I realise that this adds a whole extra level of complexity because you're making comparisons between increasing numbers of specimens/taxa, but it's definitely something to at least think about and should probably be discussed, if not enacted here.

I also still think that you might be better served moving the character list to the end of the text, as an Appendix, but included in the main paper. However, if you're determined to keep it in the main text, I think it would be better to have a combined Materials and Methods section, with the character list at the end of this.

I am also happy with the responses to comments from my previous review.

I look forward to seeing the final version of this published.

Best wishes,

Phil Mannion

·

Basic reporting

Meets all of the appropriate requirements

Experimental design

Meets all of the appropriate requirements

Validity of the findings

Meets all of the appropriate requirements

Additional comments

In light of the revision of the MS and the removal of the description of the SMA material, I will restate my original concerns with the MS as they apply to the current MS:

– “I disagree wholeheartedly with the notion that one can split genera based on the results of a phylogenetic analysis that maintains monophyly of the original genus. (see the case of Brontosaurus vs. Apatosaurus for a primary case example). “
I remain unconvinced by the author’s arguments, but there have been exceptional efforts to address this concern in this revision. I consider this concern settled, at least as far as the publication of this paper.
– “Finally, I don’t know that a phylogenetic analysis including juvenile specimens alongside adult specimens is going to give you a particularly trustworthy result. “
As before, I remain unconvinced but feel that the authors have made a satisfactory effort to address it.
I think the revised structure of the paper is a great improvement. I think it also is improved by removing the SMA description, which indeed seemingly is its own paper.
All in all, I think it remains a challenging and interesting paper - for all that I might disagree with the methods, it nonetheless is a useful attempt to approach some interesting and important questions, and I think it will end up highly cited as a result. I can see no further barrier to publication.

---

## Round 0.4 · accepted · Accept

Thank you for your attention to the comments from the reviewers; in my opinion, the manuscript is now ready for publication.